# Low planktic foraminiferal diversity and abundance observed in a spring 2013 West-East Mediterranean Sea plankton tow transect

**Miguel Mallo[1], Patrizia Ziveri[1,2], P. Graham Mortyn[1,3], Ralf Schiebel[4] and Michael Grelaud[1]**

1. Institute of Environmental Science and Technology (ICTA), Autonomous University of Barcelona (UAB), Bellaterra 08193, Spain

2. Catalan Institution for Research and Advanced Studies (ICREA), Pg. Lluís Companys 23, 08010 Barcelona, Spain

3. Geography Department, UAB, Bellaterra 08193, Spain

4. Climate Geochemistry, Max Planck Institute for Chemistry, Hahn-Meitner-Weg 1, 55128 Mainz, Germany

## Abstract

Planktic foraminifera were collected with 150 µm BONGO nets from the upper 200 m water depth at 20 stations across the Mediterranean Sea between 02 May and 02 June, 2013. The main aim is to characterize the species distribution and test the covariance between foraminiferal area density ($\rho_A$) and seawater carbonate chemistry in a biogeochemical gradient including ultraoligotrophic conditions. Average foraminifera abundances are 1.42 ±1.43 ind. 10 m$^{-3}$ (ranging from 0.11 to 5.20 ind. 10 m$^{-3}$), including twelve morphospecies. Large differences in species assemblages and total abundances are observed between the different Mediterranean sub-basins, with an overall dominance of spinose, symbiont-bearing species indicating oligotrophic conditions. The highest values in absolute abundance are found in the Strait of Gibraltar and the Alboran Sea. The western basin is dominated by *Globorotalia inflata* and *Globigerina bulloides* at slightly lower standing stocks than in the eastern basin. In contrast, the planktic foraminiferal assemblage in the warmer, saltier and more nutrient-limited eastern basin is dominated by *Globigerinoides ruber* (white). These new results when combined with previous findings, suggest that temperature-induced surface water stratification, and food availability are the main factors controlling foraminiferal distribution. In the oligotrophic and highly alkaline and supersaturated with respect to calcite and aragonite Mediterranean surface water, standing stocks and $\rho_A$ of *G. ruber* (white) and *G. bulloides* are affected by both food availability and seawater carbonate chemistry. Rapid warming increased surface ocean stratification impacting food availability and changes in trophic conditions could be the causes of reduced foraminiferal abundance, diversity, and species-specific changes in planktic foraminiferal calcification.

## 1. Introduction

The single-celled foraminifera comprise the most diverse group of calcareous zooplankton of the modern
ocean. The majority of foraminifer species are benthic. About 50 morphospecies are planktic, which have
a calcareous test organized in chambers (e.g., d'Orbigny, 1826; Hemleben et al., 1989; Goldstein, 1999).
The species from different environments can be characterized by differences in wall structure, pore size
and spatial density, spines and test shape, which are partly related to adaptation. The distribution of
foraminifera is thought to be influenced by food availability, temperature, salinity, turbidity, sunlight, and
predation; these factors provoke an overall water depth preference, which shifts during ontogeny, and
seasonal preference for each species (e.g., Rebotim et al., 2017; Schiebel and Hemleben, 2005; Hemleben
et al., 1989). Some species are found only in the photic zone because they are symbiont-bearing and
depend on light for photosynthesis. After reproduction, the empty shells sink to the seafloor, where the
fossils are useful for paleoceanographic studies (e.g., Shackleton, 1968; Rohling et al., 2004; Mojtahid et
al., 2015). Ecological tolerance limits of modern foraminifera are not completely defined, but progressive
reduction in abundance (caused by worsening of their organic functions like food uptake, growth and
reproduction, until death) is related with their departure from optimum conditions (Bé, 1977; Arnold and
Parker, 1999). The absolute abundance of foraminifera is also affected by a predictable and distinct
seasonal cycle for each species driven by the food content and temperature of the water mass (Hemleben,
1989; Bé and Tolderlund, 1971; Jonkers and Kučera, 2015; Žarić et al., 2005; for Mediterranean
examples see: Pujol and Vergnaud Grazzini, 1995; Bárcena et al., 2004; Hernández-Almeida et al., 2011;
Rigual-Hernández et al., 2012; de Castro Coppa et al., 1980).
A vast majority of studies on planktic foraminifera are based on samples from bottom sediments and
sediment cores, mainly for paleoceanographic purposes, with few studies considering the modern
population in the water column, including the Mediterranean Sea. The first modern study of planktic
foraminifera in this specific area was based on surface sediment samples collected by the Swedish Deep-
Sea expedition of 1947-48 (Pettersson, 1953). A subsequent study found different species assemblages
between the western basin, the eastern basin, and the Aegean Sea (Parker, 1955). The pioneering study of
foraminifera population variability in the water column of the Mediterranean was conducted by Glaçon et
al. (1971) in the Ligurian Sea, showing large seasonal variations of the relative abundances of the
different species. Such variations of planktic foraminiferal assemblages in the water column were also
reported for the Bay of Naples (de Castro Coppa et al., 1980). Cifelli (1974) was the first to cover the
broader Mediterranean, with plankton tows of the upper 250m of the water column from west Madeira in
the Atlantic Ocean to the Isle of Rhodes in June 1969; they identified different relative abundances of
subtropical and subpolar species in different parts of the Mediterranean.
Thunell (1978) studied the upper 2 cm of sediment cores retrieved from different sites of the
Mediterranean Sea and concluded that the distribution of planktic foraminifera was closely related to the
distribution of the different surface water masses. Each water mass has a characteristic range of
temperature and salinity (Brown et al., 2001) and a partial isolation effect in the different basins and sub-
basins of the Mediterranean. Those hydrographic differences result in different species assemblages in
each region. This contradicts somewhat with Pujol and Vergnaud Grazzini (1995), who gained
quantitative data with flow-metered plankton tows in the upper 350 m of the water column, through a
NW-SE Mediterranean transect from September-October 1986 and February 1988, and the Alboran Sea
in April 1990. They concluded that despite the W-E temperature and salinity gradients observed, those
were not large enough and no close correlation was found to justify the extremely variable foraminifera
assemblages, with high seasonal and geographical variations in absolute and relative abundances. They
suggested that food availability is the main factor controlling their seasonal and geographical distribution
and abundance. Hydrographic structures like eddies and fronts exert control on the distribution of species
in case food is present in ample amounts.
Despite no recent plankton tow study being carried out in the entire Mediterranean Sea, three regional
studies based on sediment traps were realized in the Alboran Sea (Bárcena et al., 2004; Hernández-
Almeida et al., 2011) and the Gulf of Lion (Rigual-Hernández et al., 2012). The one-year time-series
study of the Alboran Sea sediment traps (July 1997 – May 1998) shows big differences in the main
species distribution and daily export production, driven by food availability (related with water
mixing/stratification periods) and temperature (Bárcena et al., 2004; Hernández-Almeida et al., 2011).
The 12-year sediment trap foraminifera flux record in the Gulf of Lion (October 1993 – January 2006)
shows a strong seasonal pattern, with more than 80% of the annual export production recorded from
winter to spring related to higher food supply and mixing state of the water column (Rigual-Hernández et
al., 2012).
The calcification of foraminifera is affected by the chemical state of ambient seawater. Theoretically,
their shell mass is positively related to temperature, $p$H, $[Ca^{2+}]$, alkalinity, and $[CO_3^{2-}]$, and negatively
related to the $[CO_2]$ of ambient seawater (Schiebel and Hemleben, 2005). Different studies conducted on
water column foraminifera show differential results, as their shell mass can either be positively (Aldridge
et al., 2012; Beer et al., 2010a; Marshall et al., 2013; Moy et al., 2009) or negatively related to $[CO_2]$
(Beer et al. 2010a). Also, other studies report a positive effect of the temperature on foraminifera shell
mass (Mohan et al. 2015; Aldridge et al., 2012; Marshall et al., 2013; Weinkauf et al., 2016). Beer et al.
(2010a) suggest a species-specific relation between shell mass and $[CO_3^{2-}]$, depending on the presence or
absence of symbionts. Some authors suggest that other factors like ecological stress do not affect the
calcification intensity (Weinkauf et al., 2013). For further studies that relate foraminiferal calcification
with environmental parameters see Weinkauf et al. (2016); Table 7. From the onset of the industrial era,
anthropogenic emissions of $CO_2$ have led to ocean acidification, decreasing seawater $p$H and $[CO_3^{2-}]$,
which provokes reduced stability of $CaCO_3$ that may reduce the formation of foraminiferal test calcite
(Zeebe, 2012; de Moel et al., 2009; Moy et al., 2009).
Studies of the ecology of foraminifera in the Mediterranean waters remain scarce. Few studies exist
covering the entire Mediterranean Sea. Most studies are focused on specific regions, e.g., the Gulf of
Naples (de Castro Coppa et al., 1980) or the Alboran Sea plus the southwestern Mediterranean (van
Raden et al., 2011). Data on living planktic foraminiferal abundances are provided by Cifelli (1974;
spring only) and more recently by Pujol and Vergnaud Grazzini (1995). In addition, few size-normalized
weight (SNW) and area density ($\rho_A$) studies to infer the calcification intensity of water column
foraminifera are available in the literature (see Schiebel et al., 2007; Beer et al., 2010a; Aldridge et al.,
2012; Marshall et al., 2013; Mohan et al., 2015; Marshall et al., 2015; Weinkauf et al., 2016). New data
are needed, since environmental conditions of the water column and associated foraminiferal assemblages
might have changed over the past 20 years.
In this study, new quantitative and qualitative data are presented on living planktic foraminifera across the
Mediterranean Sea during spring 2013. Comparisons are made with previous studies from Pujol and
Vergnaud Grazzini (1995), Cifelli (1974), de Castro Coppa et al. (1980), Bárcena et al. (2004),
Hernández-Almeida et al. (2011), Rigual-Hernández et al. (2012) and Thunell (1978). The study by
Thunell (1978) is based on surface sediments, but might be biased by differential transportation and
dissolution of tests (Thunell, 1978; Caromel et. al., 2014; Schiebel et al., 2007). Although core top
samples (0-2 cm) are suitable to infer variability of modern conditions (Thunell; 1978), they can cover the
last few decades to few centuries, depending on the sedimentation rate, while our plankton tow sampling
represents a "snap shot" of the modern water column (Mortyn and Charles, 2003), in this case the
Mediterranean. Correlated results between plankton tows (Pujol and Vergnaud Grazzini, 1995) and
surface sediments (Vergnaud Grazzini et al., 1986) at coincident places in the Mediterranean confirm the
results obtained by Thunell (1978).
The objectives here are to (1) delineate new absolute abundances of planktic foraminifera within the
different regions of the Mediterranean Sea during spring, (2) characterize ecological demands at the
species level by comparison with previous studies, and (3) provide new $\rho_A$ data for comparisons between
sub-basins of the Mediterranean Sea and with other studies, in the context of ocean warming and
acidification over the past 20 to 40 years.

## 2. Oceanographic Setting

The Mediterranean Sea, with a strong thermohaline and wind-driven circulation, and a surface of
approximately 2,500,000 km$^2$, is divided into two main basins near the Strait of Sicily: the western and
eastern basins. These basins are composed of different sub-basins due to partial isolation caused by sills
that influence the water circulation, and by different water properties (Rohling et al., 2015; Rohling et al.,
2009). Natural connection with the ocean is through the narrow Strait of Gibraltar, where nutrient-rich
Atlantic surface waters enter the Mediterranean and experience an eastward increase of temperature and
salinity (Fig. 1) driven by insolation and evaporation, having a negative hydrological balance
(evaporation exceeding precipitation). The Mediterranean becomes increasingly oligotrophic towards the
east (Fig. 1; Fig. 2). In addition, the incoming Atlantic waters enter the Algero–Provençal Basin as far as
the Tyrrhenian Sea, and contribute to deep water formation in the Gulf of Lion in cold winters (Rohling et
al., 2015; Rohling et al., 2009).
In the eastern basin, two main sources of deep-water formation are active mainly during winter in the
Adriatic and the Aegean Seas. Cold dry winds cause evaporation and cooling forming denser and more
saline water masses that sink to depth (Rohling et al., 2015; Rohling et al., 2009; Hassoun et al., 2015b).
The same process is active in the Levantine basin, forming an intermediate water mass, which becomes
progressively cooler and fresher toward the western basin. Some waters reach the Tyrrhenian Sea. Waters
returning to the Atlantic through the Strait of Gibraltar at depth are cooler and saltier than the inbound
waters, and compensate for the inflow from the Atlantic. The Mediterranean Sea has a large
physicochemical gradient for such a small marginal sea (Rohling et al., 2015; Rohling et al., 2009; Fig.
152 1).

## 3. Methodology

### 3. 1. Study Area

Plankton tow samples were collected during the MedSeA (Mediterranean Sea Acidification in a Changing
Climate) cruise from 02 May to 02 June 2013 on board the Spanish R/V *Ángeles Alvariño*. The transect
was divided into two legs (Fig. 2). The first leg ranged from the Atlantic Ocean near the Gibraltar Strait
(adjacent to Cadiz Harbour, Spain) as far as the Levantine sub-basin in the Eastern Mediterranean (3879
km long, 11 sampling sites). The second leg started from Heraklion (Crete, Greece) into the Ionian Sea,
passed south of the Adriatic and Tyrrhenian Seas, and ended in the North Algero-Provençal basin,
adjacent to Barcelona, Spain (3232 km long, 9 sampling sites, Fig. 2).

### 3. 2. Material and methods

Twenty samples were collected down to 200 m water depth with BONGO nets (Table 1), mesh size 150
μm, and 40 cm diameter (for further details see Posgay, 1980). The sampling device was equipped with a
flow-meter allowing the estimation of the volume filtered in each tow. The data for temperature, salinity,
oxygen, and fluorescence were integrated over the upper 200 m from the nearest CTD stations retrieved
during the same cruise (for the complete dataset see Ziveri and Grelaud, 2015). Seawater carbonate data
(total alkalinity (AT), and dissolved inorganic carbon (DIC)) were obtained from water samples retrieved
at various depths during the CTD casts (see Goyet et al., 2015). These data were used to calculate $p$H,
$pCO_2$, and $[CO_3^{2-}]$ using the software CO2Sys (Lewis and Wallace, 1998) with the equilibrium constants
of Mehrbach (1973) refitted by Dickson and Millero (1987). These three parameters of the carbonate
system were then integrated for the upper 200 m water depth. The nutrient concentrations ($[PO_4]$ and
$[NO_3]$) were measured by OGS (Italaian National Institute of Oceanography and Experimental
Geophysics). The water samples were filtered on glass fiber filters (Whatman GF/F; 0.7 μm) and then
kept at -20ºC onboard. The samples were then analyzed in the laboratory with a Bran+Luebbe3
AutoAnalyzer (see Grasshoff et al., 1999). Surface chlorophyll *a* concentration was obtained from
MODIS Aqua L2 satellite data (NASA Goddard Space Flight Center: http://oceandata.sci.gsfc.nasa.gov/).
Foraminiferal samples were collected either at daytime or nighttime. Plankton samples were preserved by
adding a 4 % formaldehyde solution buffered with hexamethyltetramine at $p$H = 8.2 on board. Individuals
were not necessarily alive when collected and no distinction was made between cytoplasm-bearing tests:
as alive or dead but still containing cytoplasm (see also Boltovskoy and Lena, 1970), and empty tests
(dead) were considered for this study. From each sampling station, the foraminifera were isolated and
identified at the species level. When necessary, samples were split into aliquots of 1/4 and 1/6. For each
sample, each species was counted and isolated according to 3 size fractions (150–350 μm, 350–500 μm,
and >500 μm) to determine the absolute and relative abundances. Foraminifera smaller than 150 μm
and/or with tests partially broken, making them unrecognizable or unmeasurable, were discarded.
We classified the different foraminifera species by visual identification using incident light microscopy.
Following the morphometric guidelines and taxonomic nomenclature proposed by Aurahs et al. (2011)
for *Globigerinoides ruber* (white), *Globigerinoides ruber* (pink) and *Globigerinoides elongatus*. For
*Trilobatus sacculifer* (with sac) and *T. sacculifer* (without sac) we followed Spezzaferri et al. (2015). The
taxonomy of Hemleben et al. (1989) was applied to classify *Globigerina bulloides, Orbulina universa,*
*Globorotalia inflata, Globorotalia menardii*, and *Hastigerina pelagica. Trilobatus sacculifer* morphotype
*quadrilobatus* was inferred from Spezzaferri et al. (2015) after André et al. (2013); this morphotype is
referred as *T. quadrilobatus* in this study and is treated separately from *T. sacculifer* (without sac). The
*Globigerinella siphonifera/ G. calida/ G. radians* plexus (see Weiner et al., 2015) is treated as *G.*
*siphonifera* in our study.
For the area density ($\rho_A$) study, we selected three main species: *G. ruber* (white), *G. bulloides*, and *O.*
*universa*. All specimens, without partially broken tests and/or with organic matter attached, of these three
species were photographed with a *Canon EOS 650 D* camera device attached to a *Leica Z16 AP0*
microscope to measure their long axis and silhouette area using the software ImageJ (Schneider et al.,
2012). For each station and each of the three selected species, the individuals were weighed together by
triplicate with a Mettler Toledo XS3DU microbalance (±1 μg of nominal precision) within 50 μm size
fraction increments (150-200 μm, 200-250 μm, etc.). Cytoplasm-filled or empty dry-weighed
foraminifera tests were weighted together since dry cytoplasm has no statistically significant effect on the
weight of tests >150 μm (Schiebel et al., 2007). Specimens containing notable organic matter attached to
the outside of the test were discarded. The maximum number of individuals weighed together was five.
At some stations, individuals were measured individually in case more than one specimen was not
available. In all cases, the mean weight per specimen of the three weightings was applied (cf. Beer et al.
2010b; Movellan et al. 2012). The silhouette area obtained was then used to calculate the $\rho_A$ (Weinkauf et
al., 2013; Marshall et al., 2013; Marshall et al., 2015).
**3.3. Statistical methods**

Principal component analysis (PCA; Varimax rotation) of the environmental parameters (temperature,
salinity, oxygen, fluorescence, $NO_3$, $PO_4$, $p$H, $pCO_2$ and $CO_3^{2-}$) characterizing the 20 stations was
extracted using SPSS Statistic 23 software. The two first PCA factors explain about 77 % of the total
variance in environmental parameters (Fig. 3). The first factor exhibited positive loadings on the nutrients
and the fluorescence and negative loadings on temperature and salinity (and to a lesser degree on $[CO_3^{2-}]$;
Table 2). The first factor explains 56.99% of the total variance and depicts well the general trend
observed in the Mediterranean Sea with in general colder and more productive waters in the western basin
and warmer and less productive waters in the eastern one. The second factor explains about 20.02% of the
total variance and is characterized by positive loadings of $p$H and oxygen concentrations (and to a lesser
degree on $[CO_3^{2-}]$) and negative loading of the $pCO_2$ (Table 2). It is interpreted as variations of the
carbonate system in the Mediterranean Sea with in general lower $p$H/$[CO_3^{2-}]$ in the western basin
compared to the eastern basin. The sample scores of the first two factors with an overlay of absolute
abundances of foraminifera species (*G. ruber* (white), *G. bulloides*, *G. inflata*, *O. universa* and *T.*
*sacculifer* (without sac)) and area density (*G. ruber* (white), *G. bulloides* and *O. universa*) are shown in
Figure 3.

**4. Results**

**4. 1. Absolute and relative abundance**
The absolute abundance of planktic foraminifera collected with BONGO nets has a mean value of 1.42
±1.43(SD) individuals 10 m$^{-3}$. A maximum value of 5.2 ind. 10 m$^{-3}$ in the Strait of Gibraltar is followed
by 4.14 ind. 10 m$^{-3}$ in the Alboran Sea, 3.61 ind. 10 m$^{-3}$ in the Tyrrhenian Sea, and 3.00 ind. 10 m$^{-3}$ off
southern Crete (Fig. 4; Fig. 3a). With the exception of these four regions, a standing stock of 1.7 ind. 10
m$^{-3}$ is not surpassed at any other station. A minimum standing stock occurs in the Adriatic Sea (0.11
ind. 10 m$^{-3}$). The westernmost stations 2 and 3, with the highest Atlantic influence, have the highest
abundance values (4.67 ind. 10 m$^{-3}$ on average), followed by the eastern Mediterranean Stations 9 to 13
(1.31 ind. 10 m$^{-3}$; Fig. 4; Fig. 3a; Appendix A). Pervasively, the most common size fraction of
foraminifera is 150–350 µm (65.57%; Fig. 5), especially due to the presence of *G. ruber* (white) and *G.*
*bulloides*. The 350-500-µm size fraction in the first leg dominates in the western Mediterranean and is
progressively reduced eastwards (Fig. 5). Higher percentages of individuals >500 µm in the first leg are
found in the western part of the Mediterranean compared to the eastern part (Fig. 5). The highest
percentages of >500 µm tests are found at the Strait of Sicily and the Northern Ionian Sea (St. 7a, 16-18;
Fig. 5; Fig. S1; Appendix A). In concordance with Pujol and Vergnaud Grazzini (1995), no differences
are observed between samples collected during day and night. However, due to the extremely low
standing stocks the above observations are mere snapshots, and may not be generalized.
The most abundant species is *G. ruber* (white) (with an average of 0.30 ind. 10 m$^{-3}$, representing 21.49%
of the total assemblage); its highest abundances are found in the Tyrrhenian Sea (St. 19, 1.69 ind. 10 m$^{-3}$)
and in the eastern Mediterranean (Stations 10 and 13). *Globigerinoides ruber* (white) is not present in the
Adriatic Sea, at Station 16–18, and in the northwestern Mediterranean. It is found in low numbers in the
southwestern Mediterranean, Atlantic, and Strait of Gibraltar stations (Fig. 4; Fig. 3d). Individuals >350
µm in long test axis are rare (Appendix A). *G. inflata* is the second most abundant species (0.29 ind. 10
m$^{-3}$; 20.19%), mainly due to its high abundance in the Alboran Sea (3.5 ind. 10 m$^{-3}$; 61.08% of the
sample). It is mainly present in the western Mediterranean (Fig. 4; Fig. 3b). The dominant size fraction is
350-500 µm (Appendix A). *G. bulloides* has an average abundance of 0.24 ind. 10 m$^{-3}$ (17.20 %), mainly
due to its abundance in the Strait of Gibraltar (2.31 ind. 10 m$^{-3}$; 47.34 %). It is slightly more abundant in
the southwestern Mediterranean and the Tyrrhenian Sea than in the eastern Mediterranean. It is a quite
ubiquitous species being absent at four stations (Fig. 4; Fig. 3e). It rarely appears in the >350-μm test-size
fraction (Appendix A).
*Trilobatus sacculifer* (without sac on average 0.13 ind. 10 m$^{-3}$; 9.16 %), is especially notable at the Strait
of Gibraltar (50.91 %; Fig. 4; Fig. 3c). *O. universa* is ubiquitous in the whole Mediterranean Sea with the
exception of the three Stations 6, 9, and 14 (Fig. 4; Fig. 3f). Its average abundance is 0.12 ind. 10 m$^{-3}$
(8.70 %). Its dominant size fractions are >350 μm (Appendix A; Fig. 5). *G. elongatus* (0.09 ind. 10 m$^{-3}$;
6.41 %) is found mostly at the same stations as *G. ruber* (white), but is usually less abundant (Fig. 4). It is
most frequent in the 350-500-μm test-size fraction, and some individuals >500 μm are found in the
Atlantic (Appendix A). The other species and morphotypes appear in very low numbers: *T. quadrilobatus*
(0.07 ind. 10 m$^{-3}$), *G. siphonifera* (0.03 ind. 10 m$^{-3}$), *G. ruber* (pink) (0.02 ind. 10 m$^{-3}$), *H. pelagica* (0.008
ind. 10 m$^{-3}$), *G. menardii* (0.001 ind. 10 m$^{-3}$) and *T. sacculifer* (with sac) (0.001 ind. 10 m$^{-3}$; Fig. 4;
Appendix A).
The PCA performed on the environmental parameters and the sample scores of the two first components
show clear separation, between the western and eastern Mediterranean stations in Factor 1 (Fig. 3). The
western basin is characterized by higher food availability to the foraminifera, lower temperatures, lower
salinities, and highest absolute planktic foraminifera abundances (Fig. 3a). In the eastern basin, station 10
is an exception with a considerable contribution of *G. ruber* (white) to the absolute abundances (Fig. 3a).
In PCA Factor 2, the stations influenced by the incoming waters from the Atlantic and lowest [CO$_3^{2-}$]
values score highest. The stations where absolute abundances show some affinity for higher [CO$_3^{2-}$]
values conditions are in the NW Mediterranean, the Tyrrhenian Sea, and in the northern Ionian Sea
(stations 14, 15 and 16). Overall, highest absolute abundances of the total planktic foraminifera
assemblage seems to be related to food availability, and only secondarily to the carbonate system (Fig.
3a).
With the exception of the Tyrrhenian Sea (St. 19), *G. ruber* (white) abundance is related with warmer and
saltier waters, and lower *p*H (St. 9, 10, 11, 12, 13, 14, 15; Fig. 3d). The opposite is observed for *G.
bulloides*, and higher abundances occur where more food is available and at stations where *p*H is lower
(Fig. 3e). *O. universa* shows a ubiquitous distribution with no remarkable trends within the two PCA
factors (Fig. 3f). The more patchy distribution of *T. sacculifer* (without sac) does not follow any trend
(Fig. 3c). *G. inflata* positively correlates with food availability, and the regional distribution follows the
path of Atlantic waters (Fig. 3b).
To show the relative abundance of the various species, some stations were grouped together to achieve a
minimum number of foraminifera (>95 tests); the grouping was set by location proximity in which
foraminiferal assemblages were similar. The stations at the Strait of Sicily and the western Mediterranean
(Stations 20, 21, 22) are not shown due to low numbers of individuals (< 90; Fig. 6). The Tyrrhenian Sea
and the eastern Mediterranean stations were dominated by *G. ruber* (white), the Alboran Sea by *G.
inflata*. The dominance of a single species in the southwestern Mediterranean is less clear, which might
be due to low numbers of individuals (*G.inflata* being the main species followed by *G. bulloides* as in the
Alboran Sea). *T. sacculifer* (without sac) has a high relative abundance in the Atlantic Ocean and in the
Strait of Gibraltar, being the main and the second most abundant species, respectively. At all other
stations analyzed, *T. sacculifer* (without sac) is less abundant. *G. bulloides* is most frequent in the entire
western Basin and the Atlantic Ocean, being the main species in the Strait of Gibraltar. It is less frequent
in the Tyrrhenian Sea, and in the eastern Basin and its sub-basins. *G. bulloides* contrasts with *G. ruber*
(white), which always represents a small percentage of the assemblage in the western Mediterranean but
dominates the Tyrrhenian Sea and the eastern Basin (Fig. 6; Appendix A).

**4. 2. Area density ($\rho_A$)**
Due to their high abundance, *G. ruber* (white), *G. bulloides*, and *O. universa* were analyzed for their area
density ($\rho_A$; Fig. 7 including their Coefficient of Variation (CV); Fig. 3g-i). The two-dimensional
(silhouette) area-to-long axis correlation is best fitted by a power regression (Fig. S2). Similar allometric
developments can be seen in *G. ruber* (white), *G. bulloides*, and *O. universa* with that correlation,
graphically represented by the shape of a power function (Fig. S2). The allometric developments of
species result from increasing size of tests when adding chambers during the successive ontogenetic
stages from juvenile to adult: planktic foraminifera grow "faster" when they are younger and smaller
(steepest in the lower left part of the regression line) and "slower" when they are older and bigger (less
steep in the upper right part of the regression line; Fig. S2). The specimens of *G. ruber* (white) from the
Atlantic have a significantly larger area than those from the Tyrrhenian Sea ($p \leq 0.003$), which in turn
have significantly larger area than those from the East Ionian Sea grouping ($p \leq 0.001$). In the other two
species *G. bulloides* and *O. universa*, a similar trend is observed regarding the two basins, with the
eastern Mediterranean hosting the smallest individuals, while the largest individuals occurred in the
Atlantic and the northwestern Mediterranean (Fig. S2). The different locations were grouped using the
same criteria as in Fig. 6.
The long axis-to-weight relation of *G. ruber* (white) specimens yielded an $r^2 = 0.841$ (linear regression
throughout this paragraph; Fig. S3), followed by *O. universa* ($r^2 = 0.63$), and *G. bulloides* ($r^2 = 0.516$; Fig.
S3). *O. universa* was finally discarded for comparisons between $\rho_A$ at different locations due to a low
area-weight correlation and no remarkable trend observable between locations (Fig. S4c; Fig. 3i); while
data from *G. ruber* (white) correlate well (Fig. S4a). The eastern Mediterranean specimens are the lightest
in both species (*G. ruber* (white), *G. bulloides*), with more extreme W-E differences in *G. ruber* (white)
than in *G. bulloides* (Fig. S4d-e).
The $\rho_A$ of *G. ruber* (white) specimens from six locations were compared (Fig. 7). The data of all the
locations show a similar CV value. The eastern Mediterranean individuals have the lowest median $\rho_A$
(approximately between $7.5 \cdot 10^{-5}$ and $9 \cdot 10^{-5}$ µg µm$^{-2}$), with lower values eastward, and a small
interquartile range (IQR = $Q_3 - Q_1$). The Atlantic individuals of *G. ruber* (white) show the highest median
value ($1.55 \cdot 10^{-4}$ µg µm$^{-2}$) and IQR. The $\rho_A$ of Tyrrhenian individuals ranges between those from the
eastern Mediterranean and Atlantic Ocean ($1.2 \cdot 10^{-4}$ µg µm$^{-2}$). The $\rho_A$ of *G. ruber* (white) for each station
was compared with the two PCA factors; higher $\rho_A$ are related to slightly lower *p*H and to higher food
availability in the western Mediterranean and Atlantic stations (Fig. 3g).
For *G. bulloides* specimens, seven locations were compared (Fig. 7). The data from these locations show
similar CV values. Specimens from the Atlantic have the lowest median $\rho_A$ (8.75 $10^{-5}$ µg µm$^{-2}$) and the
smallest IQR, showing an opposite trend than *G. ruber* (white). Also contrary to *G. ruber* (white), *G.*
*bulloides* from the eastern Mediterranean tend to have a higher median $\rho_A$ (9.75 $10^{-5}$ µg µm$^{-2}$) and a larger
IQR. The differences in $\rho_A$ between the eastern and western Mediterranean are smaller in *G. bulloides*
than in *G. ruber* (white).  The $\rho_A$ of *G. bulloides* at each station was compared with the two PCA factors.
Results show a less clear overall trend for *G. bulloides* than for *G. ruber* (white), with higher $\rho_A$
associated with slightly higher *p*H in the eastern Mediterranean (Fig. 3h).

## 5. Discussion


### 5. 1. Abundance and diversity patterns

Absolute abundance values of 4.2 individuals per 10 m$^{-3}$ (>150 µm) on average are low in comparison
with earlier studies, even in oligotrophic regions. For example, in the oligotrophic northern Red Sea, less
than 100 ind. 10 m$^{-3}$ (>125 µm) were reported from surface waters, and standing stocks were much higher
than 100 ind. 10 m$^{-3}$ at most of the sites sampled in 1984 and 1985 (Auras-Schudnagies et al., 1989).  In
the oligotrophic to mesotrophic Caribbean and Sargasso Seas, standing stocks were up to 786 ind. 10 m$^{-3}$
(>100 µm) and 907 ind. 10 m$^{-3}$ (>202 µm), respectively (Schmuker and Schiebel, 2002, and references
therein). In the Atlantic, south of the Azores Islands, Schiebel et al. (2002) counted an average of 66.15
ind. 10 m$^{-3}$ for the upper 100 m in August 1997, and 422.97 ind. 10 m$^{-3}$ in January 1999 (>100 µm).
Similar studies show higher abundances of one or two orders of magnitude (e.g. Sousa et al., 2014;
Boltovskoy et al., 2000; Kuroyanagi and Kawahata, 2004; Rao et al., 1991; Ottens, 1992; Schiebel et al.,
1995). At higher latitudes, in the Fram Strait (Arctic Ocean), Pados and Spielhagen (2014) obtained
approximate values of 117 ind. 10 m$^{-3}$ from the upper 500 m in late June-early July of 2011. Mortyn and
Charles (2003), in February-March 1996, at 200 m depth range in the Atlantic sector of the Southern
Ocean, found as a minimum value 0.1 ind. 10 m$^{-3}$, with an approximate mean of 73 ind. 10 m$^{-3}$.
Within the Mediterranean, a previous study with results comparable to the data presented here, sampled
the upper 350 m of the water column (Pujol and Vergnaud Grazzini, 1995). In the Alboran Sea, samples
were obtained during a similar period of the year (April 1990) with values around 16, 6 and 9 ind. 10 m$^{-3}$,
greater than in the Station 3 (4.14 ind. 10 m$^{-3}$). Samples from different seasons have higher abundances,
with highest values in February (Pujol and Vergnaud Grazzini, 1995), and a high annual average of 9.3
ind. 10 m$^{-3}$. Regarding Pujol and Vergnaud Grazzini (1995), western Mediterranean abundances are
higher than the eastern ones, due to more oligotrophic conditions and higher temperature and salinities in
the east that limit foraminiferal production during winter and late summer.
Comparing with previous studies that covered the Mediterranean, we notice that Thunell (1978, surface
sediments) and Pujol and Vergnaud Grazzini (1995, water column) did not find *G. menardii*, while the

species was reported by Cifelli (1974) in very low abundances. The fact that *G. menardii*, which has a preference for tropical waters, is not found in the surface sediments suggests that it is a new species in the Mediterranean Sea (Cifelli, 1974). Its recent presence in the Mediterranean Sea could be related to the warming of surface waters. All other species found in our study were also found in the past studies covering the Mediterranean Sea (Cifelli, 1974; Thunell, 1978; Pujol and Vergnaud Grazzini, 1995). It remains unclear whether Thunell (1978) found *G. elongatus* and *T. sacculifer* (without sac) and classified them as *G. ruber* and *G. sacculifer*, respectively. Also, it is not certain if Cifelli (1974) found *G. calida* and classified it with *G. aequilateralis* (older synonym of *G. siphonifera*). From the figures in Cifelli (1974), we suspect that *G. elongatus* was classified with *G. ruber*. In the same way, we do not find any evidence of *T. sacculifer* (with sac) from the figures presented by Cifelli (1974), but we cannot discard the possibility that this species was classified as *Globigerinoides trilobus*.

*Trilobatus quadrilobatus* was not found in any previous plankton tow studies in the Mediterranean, but is abundant in sedimentary cores (e.g. Margaritelli et al., 2016; Lirer et al., 2013; Cramp et al., 1988; Rio et al., 1990); there exists the possibility to classify it with *T. sacculifer* or *T. trilobus* in previous studies as suggested by Hemleben et al. (1989). Some species, which are absent from our samples, reached high frequencies in the aforementioned studies, e.g., *Turborotalita quinqueloba*, *Neogloboquadrina pachyderma*, and *Globorotalia truncatulinoides*. The fact that these species were not sampled in the present study may be due to their absence or presence at extremely low abundances of adult specimens at the sampled stations in May, as they present generally low abundances in spring according to a 12-year sediment trap record in the Gulf of Lion (Rigual-Hernández et al., 2012). Another possibility is their presence in test sizes smaller than 150 µm (our BONGO nets). For example, Pujol and Vergnaud Grazzini (1995) used a mesh size of 120 µm for sampling, which included *T. quinqueloba*.

To propose a quantitative comparison of the number of species found in previous studies in the Mediterranean, we used the morphospecies identified in them by the authors of each study. We identified 12 morphospecies, which is clearly less than Cifelli (1974), Thunell (1978) and Pujol and Vergnaud Grazzini (1995), reporting 18 morphospecies in total. The lower absolute abundance of individuals in our study compared to Pujol and Vergnaud Grazzini (1995), together with low species diversity in this study, may indicate a trend of changing conditions over the last decades, as it has been reported for temperature and salinity (Yáñez et al., 2010), alkalinity (Cossarini et al., 2015; Hassoun et al., 2015a), and water mass mixing (Hassoun et al., 2015b). These changing conditions could also imply changes in environmental conditions and distribution of planktic foraminifera, as discussed below; see also Field et al. (2006). Note that our mesh size is larger than that of Pujol and Vergnaud Grazzini (1995), but similar to that of Cifelli (1974) who used a mesh size of 158 µm. A larger mesh size would explain the lower numbers in absolute abundance and reduced diversity. In contrast, the higher diversity observed by Cifelli (1974) using a wider mesh for sampling in June supports our idea of changing ecological conditions.

The western part of the first transect (from the Atlantic to the Strait of Sicily) has a higher percentage of larger size fractions than the eastern part. The main cause of the increase in test size is a change in species

composition. For example, large sized *G. inflata* (especially in the 350-500 µm fraction) are present with
higher abundances in the west than in the east. The same is true for the presence of large *O. universa*
(especially in the >500 µm size fraction), plus the contribution of *G. siphonifera*, which is larger at
stations where it is more frequent (Appendix A; Fig. 5).

**5. 2. Factors controlling the abundance of the main species**
Abundance patterns of the five most frequent species in our samples possibly result from a combination
of environmental conditions as, for example, food and temperature (Fig. 3; Table 2). The spinose and
symbiont-bearing species *G. ruber* (white), *O. universa*, and *T. sacculifer* (without sac), which mainly
inhabit tropical and subtropical waters. *G. ruber* (white) is the main species in the Atlantic. *O. universa* is
rather ubiquitous, also present in warm transitional Atlantic waters (Bé and Tolderlund, 1971). The
spinose and symbiont-barren species *G. bulloides* tolerates a wide temperature range and is typical of
subpolar and transitional regions as well as upwelling areas, it is also found in subtropical and tropical
waters at lower abundances (Thunell, 1978; Bé and Tolderlund, 1971). The non-spinose species *G. inflata*
is typical of the temperate Atlantic Ocean (Bé and Tolderlund, 1971).
5. 2. 1. *Globigerinoides ruber* (white)
In our study and the one by Cifelli (1974), *G. ruber* (white) occurs with higher abundances in the eastern
compared to the western Mediterranean Basin, being the most abundant species in the Levantine Basin
and the South Ionian Sea. Also like Cifelli (1974), in our study, *G. ruber* (white) from the Atlantic station
is found with slightly higher relative abundances than in the western Mediterranean Basin. Temperature-
related factors may be the main cause, e.g.: warmer Atlantic waters (16.1 ºC) compared to the western
Mediterranean (14.3 ºC in the SW, 14.0 ºC in the NW; Fig. 1a). In the South Ionian Sea and the Levantine
Basin it seems that *G. ruber* (white) occurs independent of seasons, winter included, which is also true for
the pink variety (see also Thunell, 1978; Pujol and Vergnaud Grazzini, 1995). The increasing dominance
of *G. ruber* (white) from the western to the eastern Mediterranean Basin coincides with eastward
increasing salinity and temperature (Fig. 3d; Table 2). Its higher relative abundance in the eastern basin
results from the ability of *G. ruber* to thrive in food-depleted conditions (Hemleben et al., 1989).
*G. ruber* (white) remains scarce (St. 9, 14, 15) or absent (St. 16-18) in the Ionian Sea stations (Fig. 4),
increasing its abundance towards the Tyrrhenian Sea. On the other hand, in the Ionian Sea, it exhibits
relative abundances around 40 % to more than 60 % in the surface sediments (Thunell, 1978), and
decreases towards the Tyrrhenian Sea. This situation could be due to higher food availability in the
Tyrrhenian Sea in comparison to the Ionian Sea observed during May 2013 (Fig. 1c; Fig. 3d) plus a small
difference in temperature between both seas (Fig. 1a; Fig. 3d).  Also, we note that in May 1979, a scarce
presence of *G. ruber* was reported in the Bay of Naples (de Castro Coppa et al., 1980), whereas in our
study *G. ruber* is present at 47 % in the Tyrrhenian Sea, being the dominant species.
The dominance of *G. ruber* (white) and abundance peaks in May in the eastern Mediterranean (this
study), coincides with the positive temperature gradient between Station 9 and Station 13 (16.2–17.3 ºC;
Fig. 1). In late summer, *G. ruber* experiences its highest abundance at warmer temperatures and more
oligotrophic conditions, clearly being the main species from the north of Algeria to the Levantine Basin
(Pujol and Vergnaud Grazzini, 1995). *G. ruber* (pink) is the dominant species at the Strait of Sicily and
eastwards (Pujol and Vergnaud Grazzini, 1995), whereas in May 2013 it was rare at some locations,
especially around Crete. In February, at low sea surface temperature, *G. ruber* (pink) almost disappears
from the Mediterranean (Pujol and Vergnaud Grazzini, 1995; Rigual-Hernández et al., 2012).
Presumably, *G. ruber* (white) is better adapted to lower temperatures than the pink variety. To conclude,
food availability seems to be the limiting factor for the abundance of *G. ruber* once it has reached its
optimum temperature range (Table 2).
5. 2. 2. *Globorotalia inflata*
The presence of *G. inflata* is related to cold waters and high food availability (Pujol and Vergnaud
Grazzini, 1995; Rigual-Hernández et al., 2012), following high nutrient concentrations (Ottens, 1992).
This explains its higher abundance in the cooler nutrient-rich western basin, and its progressive scarcity
toward the warmer oligotrophic eastern Mediterranean (Fig. 1; Cifelli, 1974; Thunell, 1978). The same
pattern is observed in late summer. From spring to late summer, *G. inflata* shows a displacement from the
eastern Alboran Sea to the northwestern Mediterranean, decreasing frequency in the Algero–Provençal
Basin and the southwestern Mediterranean Basin, maintaining a residual presence in the eastern basin
(Pujol and Vergnaud Grazzini, 1995). In winter, at lower temperatures, the opposite process happens, and
*G. inflata* becomes the dominant species in the Alboran Sea (Bárcena et al., 2004) and the southwestern
basin, with high frequencies in the Strait of Sicily and toward the Ionian Sea. Eastwards its presence is
maintained at only residual levels (Pujol and Vergnaud Grazzini, 1995). Its distribution along the seasons
shows that *G. inflata* is less frequent or absent in warmer, stratified and nutrient-depleted regions of the
Mediterranean than in more productive waters.
*G. inflata* is absent in the Tyrrhenian Sea, despite temperature ranges being comparable to those observed
in the southwestern Mediterranean, where this species is abundant (this study). In contrast, in May 1979,
*G. inflata* was reported in the Tyrrhenian Sea as the main species, and practically absent in the warmer
summer months (de Castro Coppa et al., 1980). *G. inflata* is reported in sediment trap data in the Gulf of
Lion (Rigual-Hernández et al. (2012), close to our northwestern Mediterranean stations (St. 20, 21, 22) at
which *G. inflata* is absent. In addition, the absolute abundances of *G. inflata* are closely related to the
PCA Factor 1, suggesting a certain affinity with food availability inferred from nutrient concentrations
and fluorescence data (see sample scores in Fig. 3b; Table 2). Consequently, food depletion may play a
more important role in limiting the distribution of *G. inflata* than temperature.
The distribution of *G. inflata* during spring, with *G. bulloides* as a secondary species in the Alboran Sea
confirm the findings of other studies (Pujol and Vergnaud Grazzini, 1995; van Raden et al., 2011). *G.*
*inflata* peak abundances appear more to the west than those reported by Cifelli (1974) to the east of the
Balearic Islands. Those peaks can be associated with nutrient-rich upwelling areas rich in foraminifer
prey within the temperature range of *G. inflata* (Fig. 1; Fig. 2).
5. 2. 3. *Globigerina bulloides*
Following Cifelli (1974), *G. bulloides* is the dominant species in the Atlantic close to the Strait of
Gibraltar, whereas in our study it shares dominance with other species (Station 1; Fig. 4). The *G.*
*bulloides* dominance in the Strait of Gibraltar during late spring–early summer confirms the findings of
Cifelli (1974). The abundance peak of *G. bulloides* in the Strait of Gibraltar (this study), coincides with
high nutrient concentration and upwelling (Figs. 1, 2, and 4), with station 2 holding highest standing
stocks of planktic foraminifera of the whole transect analyzed here. This confirms its association with
upwelling, and the production of phytoplankton as the major food source of this opportunistic species
(Pujol and Vergnaud Grazzini, 1995; Sousa et al., 2014; Bárcena et al., 2004; Hernández-Almeida et al.,
2011; Rigual-Hernández et al., 2012). Consequently, higher standing stocks of *G. bulloides* are related
with higher nutrient concentration (e.g., Mortyn and Charles, 2003; Fig. 1; Fig. 3e; Table 2).
In April (Pujol and Vergnaud Grazzini, 1995; van Raden et al., 2011) and May (this study), *G. bulloides*
is the second most abundant species, surpassed by *G. inflata*, in the westernmost Alboran Sea. High
temperature anomalies could provoke an inverse situation, thanks to more suitable environmental
conditions for *G. bulloides*, which profits from more successful reproduction than *G. inflata*, which
instead stays further from its optimum temperature (Bárcena et al., 2004). One month later, *G. bulloides*
is found to be the dominant species replacing *G. inflata*, which is still dominant in the eastern Alboran
Sea (Cifelli, 1974). Its ubiquity and larger abundance in the western basin with respect to the east is
supported by previous studies (e.g., Cifelli, 1974; Thunell, 1978), with a higher difference in abundance
in February than in September–October (Pujol and Vergnaud Grazzini, 1995; Rigual-Hernández et al.,
2012). In late summer, it decreases in numbers, with abundance peaks only around the Strait of Sicily and
south of Sardinia. In winter, *G. bulloides* occurs at maximum relative but lower absolute abundance peaks
in the Gulf of Lion, as well as in the Strait of Sicily and south of Sardinia (Pujol and Vergnaud Grazzini,
1995; Rigual-Hernández et al., 2012).
*G. bulloides* decreases in abundance due to food depletion in the eastern Mediterranean, where it is
always less abundant than in the western basin, and more oligotrophic conditions due to water column
stratification (Rigual-Hernández et al., 2012). During spring to late summer in the eastern basin, *G.*
*bulloides* is less frequent, and is more abundant just east of the Strait of Sicily (Cifelli, 1974; Pujol and
Vergnaud Grazzini, 1995). During winter its abundance increases and it becomes the second most
abundant species in the Levantine Basin preceded by *G. ruber* (white), and it is also one of the main
species in the Ionian Sea. Permanent eddies in the Levantine Basin sustain phytoplankton blooms,
explaining the presence of *G. bulloides* in winter (Pujol and Vergnaud Grazzini, 1995). In the northern
Levantine Basin and in the Aegean Sea its abundances are comparable to those in the western basin
regarding surface sediment data from Thunell (1978).
*G. bulloides* has more affinity for cooler upwelled waters than warmer more stratified waters (Sousa et
al., 2014; Thunell, 1978), being present in subtropical waters only during the colder months (Ottens,
1992). The coldest station of the first leg of this study (Strait of Gibraltar, 14.2 ºC) coincides with an
abundance peak of *G. bulloides*, and it is absent from the warmest station (off the Nile Delta, 17.6 ºC;
Fig. 1a), which is also one of the most depleted stations in foraminiferal prey (Fig. 1c; Fig. 2). . To
conclude, the distribution of *G. bulloides* seems to be limited by food availability, caused by stratification
and consequent nutrient depletion of the surface water column, and increased sea surface temperatures
(Table 2).
5. 2. 4. *Orbulina universa*
*Orbulina universa* was found to be ubiquitous by Pujol and Vergnaud Grazzini (1995), being present in
all the stations and seasons, reaching peak abundances in the southwestern Mediterranean both in late-
summer and winter. Regarding our data, it follows the same pattern during spring, being absent from only
three stations (St. 6, 9, and 14; Fig. 4; Fig. 3f). No abundance peak occurs in spring (Cifelli, 1974, and
this paper) but abundances are slightly higher in the western basin than in the east. These small
differences can be caused by more nutrient-rich upwelling areas (cf. Sousa et al., 2014; Morard et al.,
2013) in the western basin or by higher salinities in the eastern than western basin.
5. 2. 5. *Trilobatus sacculifer* (without sac)
In June, *T. sacculifer* (without sac) has a wide distribution and represents 5 % of the assemblage in the
Strait of Gibraltar (Cifelli, 1974). *T. sacculifer* constituted up to 25 % of the assemblages in May 2013,
and was absent from seven stations (St. 5, 7a, 14, 15, 16-18, 20, 22). Low relative abundance occurred in
April in the Alboran Sea (Pujol and Vergnaud Grazzini, 1995). In September–October *T. sacculifer* shows
high abundances and is one of the main species from north of Minorca to the southwestern
Mediterranean, and rare near the Strait of Sicily (Pujol and Vergnaud Grazzini, 1995). In late summer, it
progressively decreases in numbers to the east, where *G. ruber* dominates assemblages (Pujol and
Vergnaud Grazzini, 1995), probably due to slightly higher temperature and salinities (see also Bijma et
al., 1990). On the other hand, in February *T. sacculifer* (without sac) disappears from the north Levantine
Basin and its abundance considerably decreases (Pujol and Vergnaud Grazzini, 1995).

**5. 3. Factors controlling planktic foraminiferal test weight**
The area density ($\rho_A$) of tests of both *G. ruber* (white) and *G. bulloides* follow a systematic change from
the Atlantic towards the eastern Mediterranean (Fig. 7). Therefore, the $\rho_A$ of these two species is
interpreted and discussed for possible environmental effects and biological prerequisites in the following.
In contrast, the $\rho_A$ of *O. universa* does not show any change between the western and eastern basins (Fig.
3i), and cannot be interpreted for any particular environmental effects. Unfortunately, we cannot address
the effects of reproduction (e.g. Bijma et al., 1994), and ontogenetic development on the distribution
patterns and test calcite mass of species, because a lack data at the species level do not allow any such
statistics.

5.3.1 Unknown control of the $\rho_A$ of *O. universa*
Since environmental and biological factors may affect individuals of the different genotypes of O.
universa to varying degrees, we could not detect any systematic change in $\rho_A$ in the data presented here.
Only one out of three genotypes of *O. universa* (e.g. Type III, after Darling and Wade, 2008) occurs in
the Mediterranean Sea (Mediterranean species, after de Vargas et al., 1999), The Mediterranean Type III
has been found to include two sub-types, Type IIIa and Type IIIb (André et al., 2014). The different
genotypes and morphotypes of *O. universa* tolerate wide ranges of salinity and temperature in surface
waters (e.g., de Vargas et al., 1999). Whereas the various types of *O. universa* differ in the pore-size (de
Vargas et al., 1999; Morard et al., 2009; Marshall et al., 2015), their pore-size is also affected by
environmental conditions including water temperature (e.g., Bé et al., 1973). Likewise, thickness of the
test wall has been described to vary between types (de Vargas et al., 1999; Morard et al., 2009; Marshall
et al., 2015), and is as well affected by environmental conditions and ontogenetic stage of specimens.
Adult *O. universa* have been shown to continuously add calcite layers to the proximal surface of the same
sphere (Spero, 1988; Spero et al., 2015).
The reason why the $\rho_A$ of *O. universa* is particularly low and highly variable in the Mediterranean despite
high carbonate ion concentration ($[CO_3^{2-}]$) and *p*H (Fig. 1) might be sought in factors other than, and in
addition to, chemical and physical conditions along the transect from the Atlantic Ocean to the Levantine
Basin.
5.3.2 Factors affecting the $\rho_A$ of *G. ruber* (white) and *G. bulloides*
The $\rho_A$ of *G. ruber* (white) is only partly controlled by carbonate chemistry, being instead affected by
other factors like food availability, similar to *O. universa*. In contrast to *O. universa*, the $\rho_A$ data of *G.*
*ruber* and *G. bulloides* follow systematic correlations. High $\rho_A$ of *G. ruber* in the Atlantic and Tyrrhenian
Sea correlates with enhanced primary production (enhanced fluorescence, Fig. 1d; Fig. 3g; Table 2), and
presumably enhanced food availability (Fig. 3g; Fig. 7; Fig. 2, also noticeable in Fig. S2d and Fig. S4d).
Under more oligotrophic conditions, low $\rho_A$ of *G. ruber* (white) might be caused by limited food
availability. An opposite trend is reported for *G. ruber* (white) from sediment trap samples in the Madeira
Basin, in which, apart from showing a negative significant correlation between calcification intensity and
productivity, $\rho_A$ shows positive correlation with temperature (Weinkauf et al., 2016).
The relationship between food availability and $\rho_A$ in *G. bulloides* is opposite to *G. ruber* (white) (Fig. 3g-
h; Fig. 7; Table 2). The $\rho_A$ of *G. bulloides* tests increases from the Atlantic toward the eastern
Mediterranean. In both species, larger IQRs are found toward higher absolute $\rho_A$ (Fig. 7).
An opposite trend in $\rho_A$ of the two species *G. ruber* (white) and *G. bulloides* had earlier been described
from the Arabian Sea, and could neither be assigned to changes in $[CO_3^{2-}]$ of ambient seawater nor
growth conditions (Beer et al., 2010a). Due to its symbionts, *G. ruber* would rather have an advantage
over symbiont-barren *G. bulloides* in oligotrophic waters, and support formation of test calcite through
$CO_2$ consumption and increasing $[CO_3^{2-}]$ and *p*H (see also Köhler-Rink and Kühl, 2005). Those findings
may still point toward differences in growth conditions: Reproduction of both *G. ruber* and *G. bulloides*
might be hampered under less optimal conditions, and additional calcite layers might be added to the
proximal test before reproduction, similar to the process described for *O. universa* (see above). Therefore,
tests may grow heavier under less than optimal food availability, given that carbonate chemistry of
ambient seawater does not seem to limit the formation of test calcite in our samples.
Comparing weight-to-long axis relations, *G. ruber* (255–350 μm size fraction) from plankton tows of the
western Arabian Sea have an average weight of 11.5 ±0.69 μg (de Moel et al., 2009), which is heavier
than the individuals from our study (5.9 ±0.31 μg; Fig. S3a; Appendix A). The difference in weight-to-
long axis relation may indicate that *G. ruber* is produced under more suitable conditions for shell calcite
formation in the Arabian Sea especially during non-upwelling periods and still higher overall primary
productivity and food availability. However, the comparison might be biased by the fact that *G. ruber*
(white) and *G. elongatus* were not separately analyzed by de Moel et al. (2009).
Data for supra-regional comparison of the weight-to-long axis relation of *G. bulloides* from the water
column possible for the 200–250 μm size fraction: In the north Atlantic (56-63 °N), in June 2009,
Aldridge et al. (2012) report a range of 1.75–2.92 μg ($r^2 = 0.52$). In the same size fraction, our results (36
°N) show heavier tests in the Alboran Sea (3.46 ±0.15 μg), and similar weights at the Strait of Gibraltar
(2.57 ±0.00 μg; Fig. S3b). For the same water depth as in our samples, Schiebel et al. (2007) found
heavier average weight-to-long axis relations in fall (5.19 ±0.25 μg) than in spring (4.21 ±0.2 μg) in the
eastern North Atlantic, and 5.51 ±0.31 μg during the SW monsoon in the Arabian Sea. In general, higher
$\rho_A$ occurs at lower latitudes and lower $\rho_A$ at higher latitudes (see also Schmidt et al., 2004). For *G.
bulloides* and *G. ruber*, increased longevity and ongoing production of additional calcite layers at the
proximal side of shells may result in an increased $\rho_A$, given that seawater carbonate chemistry is only
partially affecting the calcite formation in our samples.


## 6. Conclusions
Absolute and relative abundances of planktic foraminifera were studied from plankton tow samples across
the Mediterranean, collected in May 2013. The samples show large differences in species abundance and
assemblages between the different basins and sub-basins of the Mediterranean Sea. Absolute abundance
and diversity of planktic foraminifer assemblages are low in comparison to other regions of the world
ocean. Average standing stocks in the upper 200 m of the water column are 1.42 ±1.43 ind. 10 m$^{-3}$,
including twelve morphospecies in total. Planktic foraminifer assemblages are indicative of changing
temperatures and salinities, as well as trophic conditions, between the eastern and the western
Mediterranean Sea. Highest standing stocks of total planktic foraminifera occurred in the Strait of
Gibraltar and the Alboran Sea. Overall, the largest foraminiferal tests occurred in the western
Mediterranean, driven by the assemblage composition, and the presence of large *G. inflata*.
*Globigerinoides ruber* was the most abundant species; its dominance in the east compared to the west is
likely caused by stratification of the surface water column, enhanced SST, and trophic conditions. *G.
ruber* is a symbiont-bearing species, which might be an advantage over symbiont-barren species like *G.
bulloides* under oligotrophic and food-limited conditions as in the Levantine Basin. *G. bulloides* was
more abundant in upwelled waters in the Strait of Gibraltar, in the Alboran Sea, and in the western
Mediterranean. *O. universa* was present at balanced standing stocks along the entire transect from the
west to the east. In general, distribution patterns of the main planktic foraminiferal species in the
Mediterranean seem to be mainly related to a combination of food availability, controlled by sea surface
temperature and stratification.
In the Mediterranean surface waters are supersaturated with respect to calcite and aragonite (Schneider et
al., 2007; Gemayer et al., 2015). Calcification and $\rho_A$ of the most frequent planktic foraminifera species,
*G. ruber* (white) and *G. bulloides*, are largely affected by food availability. *G. ruber* is more affine to
oligotrophic conditions, and grows heaviest tests in less food-limited waters in the western basin near
Gibraltar and in the Tyrrhenian Sea. In contrast, *G. bulloides* grows heaviest tests under more food-
limited conditions in the eastern Mediterranean Sea. We speculate that reproduction is hindered when the
species-specific food sources are limited, while individuals continue adding calcite to the outer shell, and
grow heavier tests than individuals that reproduced earlier in ontogeny.
These observations highlight the need for more interdisciplinary studies on the causes of changing
foraminiferal assemblages and decreasing shell production, especially in the Mediterranean as a marginal
basin, which is assumed particularly sensitive to changes of the environment and global climate.
**Appendices**
**Appendix A.** Planktic foraminifera data from BONGO nets: relative and absolute abundances, and weight and size parameters. The
nomenclature *G. bulloides* represents the *G. bulloides/G. falconensis* plexus, and *G. siphonifera* represents the *G. siphonifera/ G.*
*calida/ G. radians plexus*.

| Location | Atlantic | Gibraltar | Alboran Sea | South-Central Western Med. | Strait of Sardinia | Strait of Sicily | South of Ionian Sea | Off Southern Crete | Eastern Basin | Off Nile Delta | Off Lebanon | Antikythera Strait | Eastern Ionian Sea | Adriatic Sea | Otranto Strait | Northern Ionian Sea | Tyrrhenian Sea | North-Central Western Med. | Central Western Med. | Catalano-Balear |
|---|---|---|---|---|---|---|---|---|---|---|---|---|---|---|---|---|---|---|---|---|
| Station | 1 | 2 | 3 | 5 | 6 | 7a | 9 | 10 | 11 | 12 | 13 | 14 | 15 | 17 | 16 | 16-18 | 19 | 20 | 21 | 22 |
| **Absolute abundance (individuals\*10 m$^{-3}$)** | | | | | | | | | | | | | | | | | | | | |
| Total numbers | | | | | | | | | | | | | | | | | | | | |
| G. ruber (white) | 0.079 | 0.037 | 0.007 | 0.022 | 0 | 0 | 0.212 | 1.314 | 0.403 | 0.247 | 1.260 | 0.389 | 0.102 | 0 | 0.338 | 0 | 1.688 | 0 | 0 | 0 |
| G. elongatus | 0.118 | 0.019 | 0.007 | 0 | 0.024 | 0 | 0 | 0.282 | 0.054 | 0.027 | 0.202 | 0.269 | 0 | 0 | 0.182 | 0.070 | 0.537 | 0 | 0.025 | 0 |
| T. sacculifer (without sac) | 0.236 | 1.323 | 0.028 | 0 | 0.047 | 0 | 0.047 | 0.219 | 0.027 | 0.082 | 0.050 | 0 | 0 | 0.023 | 0.234 | 0 | 0.256 | 0 | 0.025 | 0 |
| G. bulloides | 0.148 | 2.311 | 0.456 | 0.501 | 0.142 | 0 | 0.165 | 0.094 | 0.054 | 0 | 0.076 | 0 | 0.102 | 0 | 0.052 | 0.023 | 0.307 | 0.197 | 0.102 | 0.147 |
| G. inflata | 0.118 | 0.503 | 3.514 | 0.545 | 0.449 | 0.358 | 0.071 | 0.125 | 0.027 | 0 | 0 | 0 | 0 | 0.023 | 0 | 0 | 0 | 0 | 0 | 0 |
| O. universa | 0.128 | 0.093 | 0.014 | 0.218 | 0 | 0.291 | 0 | 0.219 | 0.054 | 0.027 | 0.050 | 0 | 0.077 | 0.023 | 0.468 | 0.141 | 0.281 | 0.028 | 0.179 | 0.177 |
| G. siphonifera | 0.029 | 0.056 | 0.043 | 0.022 | 0 | 0.313 | 0 | 0.063 | 0 | 0 | 0.025 | 0 | 0 | 0 | 0 | 0 | 0 | 0 | 0.102 | 0 |
| T. quadrilobatus | 0.010 | 0.335 | 0.007 | 0.087 | 0 | 0.045 | 0.118 | 0.063 | 0.027 | 0 | 0 | 0 | 0 | 0.023 | 0 | 0 | 0.230 | 0.112 | 0.204 | 0.236 |
| H. pelagica | 0 | 0 | 0 | 0 | 0 | 0 | 0 | 0.125 | 0 | 0.027 | 0 | 0 | 0 | 0 | 0 | 0 | 0 | 0 | 0 | 0 |
| T. sacculifer (with sac) | 0 | 0 | 0 | 0 | 0 | 0 | 0 | 0 | 0 | 0 | 0 | 0 | 0 | 0 | 0 | 0 | 0.026 | 0 | 0 | 0 |
| G. ruber (pink) | 0 | 0.075 | 0 | 0 | 0.024 | 0 | 0.024 | 0.125 | 0 | 0.027 | 0 | 0.120 | 0 | 0 | 0 | 0 | 0 | 0 | 0 | 0 |
| G. menardii | 0 | 0 | 0 | 0 | 0 | 0 | 0 | 0 | 0 | 0 | 0 | 0 | 0 | 0 | 0 | 0 | 0 | 0 | 0 | 0.029 |
| Unknowns | 0.118 | 0.447 | 0.064 | 0.065 | 0.024 | 0 | 0.047 | 0.375 | 0.108 | 0 | 0.025 | 0.120 | 0.026 | 0.023 | 0.208 | 0.023 | 0.281 | 0.028 | 0 | 0.088 |
| Total | 0.985 | 5.120 | 4.141 | 1.460 | 0.709 | 1.006 | 0.683 | 3.003 | 0.753 | 0.439 | 1.689 | 0.898 | 0.307 | 0.114 | 1.482 | 0.258 | 3.607 | 0.365 | 0.638 | 0.678 |
| 150-350 μm size fraction | | | | | | | | | | | | | | | | | | | | |
| G. ruber (white) | 0.030 | 0.037 | 0.007 | 0.022 | 0 | 0 | 0.212 | 1.314 | 0.403 | 0.247 | 1.109 | 0.389 | 0.102 | 0 | 0.338 | 0 | 1.560 | 0 | 0 | 0 |
| G. elongatus | 0.020 | 0 | 0 | 0 | 0.024 | 0 | 0 | 0.282 | 0.054 | 0.027 | 0.202 | 0.269 | 0 | 0 | 0.182 | 0.047 | 0.460 | 0 | 0.026 | 0 |
| T. sacculifer (without sac) | 0.148 | 1.174 | 0.029 | 0 | 0.047 | 0 | 0 | 0.188 | 0.027 | 0.082 | 0.050 | 0 | 0 | 0.023 | 0.234 | 0 | 0.230 | 0 | 0.026 | 0 |
| G. bulloides | 0.128 | 2.199 | 0.449 | 0.415 | 0.142 | 0 | 0.165 | 0.094 | 0.054 | 0 | 0.076 | 0 | 0.102 | 0 | 0.052 | 0.023 | 0.307 | 0.197 | 0.077 | 0.118 |
| G. inflata | 0.069 | 0.335 | 1.176 | 0.109 | 0.095 | 0.022 | 0 | 0.063 | 0 | 0 | 0 | 0 | 0 | 0.023 | 0 | 0 | 0 | 0 | 0 | 0 |
| O. universa | 0 | 0.075 | 0.007 | 0.087 | 0 | 0 | 0 | 0.094 | 0 | 0 | 0 | 0 | 0 | 0 | 0.208 | 0 | 0.026 | 0 | 0.026 | 0 |
| G. siphonifera | 0 | 0.019 | 0.029 | 0 | 0 | 0.022 | 0 | 0 | 0 | 0 | 0.025 | 0 | 0 | 0 | 0 | 0 | 0 | 0 | 0.102 | 0 |
| T. quadrilobatus | 0.010 | 0.280 | 0.007 | 0.087 | 0 | 0 | 0.071 | 0.063 | 0.027 | 0 | 0 | 0 | 0 | 0.023 | 0 | 0 | 0.230 | 0.112 | 0.204 | 0.236 |
| H. pelagica | 0 | 0 | 0 | 0 | 0 | 0 | 0 | 0.063 | 0 | 0 | 0 | 0 | 0 | 0 | 0 | 0 | 0 | 0 | 0 | 0 |
| G. ruber (pink) | 0 | 0.075 | 0 | 0 | 0.024 | 0 | 0.024 | 0.125 | 0 | 0.027 | 0 | 0.120 | 0 | 0 | 0 | 0 | 0 | 0 | 0 | 0 |
| Total | 0.404 | 4.193 | 1.703 | 0.719 | 0.331 | 0.045 | 0.471 | 2.284 | 0.564 | 0.384 | 1.462 | 0.778 | 0.205 | 0.068 | 1.014 | 0.070 | 2.814 | 0.309 | 0.459 | 0.354 |
| 350-500 μm size fraction | | | | | | | | | | | | | | | | | | | | |
| G. ruber (white) | 0.049 | 0 | 0 | 0 | 0 | 0 | 0 | 0 | 0 | 0 | 0 | 0 | 0 | 0 | 0 | 0 | 0.051 | 0 | 0 | 0 |
| G. elongatus | 0.088 | 0.019 | 0.007 | 0 | 0 | 0 | 0 | 0 | 0 | 0 | 0 | 0 | 0 | 0 | 0 | 0.023 | 0.077 | 0 | 0 | 0 |
| T. sacculifer (without sac) | 0.079 | 0.130 | 0 | 0 | 0 | 0 | 0.047 | 0.031 | 0 | 0 | 0 | 0 | 0 | 0 | 0 | 0 | 0.026 | 0 | 0 | 0 |
| G. bulloides | 0.020 | 0.112 | 0.029 | 0.022 | 0 | 0 | 0 | 0 | 0 | 0 | 0 | 0 | 0 | 0 | 0 | 0 | 0 | 0 | 0.026 | 0.029 |
| G. inflata | 0.049 | 0.149 | 2.138 | 0.414 | 0.307 | 0.313 | 0.071 | 0.031 | 0.027 | 0 | 0 | 0 | 0 | 0 | 0 | 0 | 0 | 0 | 0 | 0 |
| O. universa | 0.049 | 0.019 | 0.007 | 0.109 | 0 | 0.067 | 0 | 0.125 | 0.027 | 0 | 0 | 0 | 0 | 0.023 | 0.130 | 0.023 | 0.153 | 0.028 | 0.051 | 0.118 |
| G. siphonifera | 0.020 | 0.019 | 0.007 | 0.022 | 0 | 0.201 | 0 | 0.031 | 0 | 0 | 0 | 0 | 0 | 0 | 0 | 0 | 0 | 0 | 0 | 0 |
| T. quadrilobatus | 0 | 0 | 0 | 0 | 0 | 0.022 | 0.047 | 0 | 0 | 0 | 0 | 0 | 0 | 0 | 0 | 0 | 0 | 0 | 0 | 0 |
| H. pelagica | 0 | 0 | 0 | 0 | 0 | 0 | 0 | 0.063 | 0 | 0.027 | 0 | 0 | 0 | 0 | 0 | 0 | 0 | 0 | 0 | 0 |
| T. sacculifer (with sac) | 0 | 0 | 0 | 0 | 0 | 0 | 0 | 0 | 0 | 0 | 0 | 0 | 0 | 0 | 0 | 0 | 0.026 | 0 | 0 | 0 |
| G. menardii | 0 | 0 | 0 | 0 | 0 | 0 | 0 | 0 | 0 | 0 | 0 | 0 | 0 | 0 | 0 | 0 | 0 | 0 | 0 | 0.029 |
| Total | 0.354 | 0.447 | 2.188 | 0.567 | 0.307 | 0.604 | 0.165 | 0.282 | 0.054 | 0.027 | 0 | 0 | 0 | 0.023 | 0.130 | 0.047 | 0.333 | 0.028 | 0.077 | 0.177 |


(**Appendix A**, cont.).

| Location | Atlantic | Gibraltar | Alboran Sea | South-Central Western Med. | Strait of Sardinia | Strait of Sicily | South of Ionian Sea | Off Southern Crete | Eastern Basin | Off Nile Delta | Off Lebanon | Antikythera Strait | Eastern Ionian Sea | Adriatic Sea | Otranto Strait | Northern Ionian Sea | Tyrrhenian Sea | North-Central Western Med. | Central Western Med. | Catalano-Balear |
|---|---|---|---|---|---|---|---|---|---|---|---|---|---|---|---|---|---|---|---|---|
| Station | 1 | 2 | 3 | 5 | 6 | 7a | 9 | 10 | 11 | 12 | 13 | 14 | 15 | 17 | 16 | 16-18 | 19 | 20 | 21 | 22 |
| **>500 μm size fraction** | | | | | | | | | | | | | | | | | | | | |
| *G. ruber s.l.* | 0.010 | 0 | 0 | 0 | 0 | 0 | 0 | 0 | 0 | 0 | 0 | 0 | 0 | 0 | 0 | 0 | 0 | 0 | 0 | 0 |
| *T. sacculifer* (without sac) | 0.001 | 0.019 | 0 | 0 | 0 | 0 | 0 | 0 | 0 | 0 | 0 | 0 | 0 | 0 | 0 | 0 | 0 | 0 | 0 | 0 |
| *G. inflata* | 0 | 0.019 | 0.135 | 0.022 | 0.047 | 0.022 | 0 | 0.031 | 0 | 0 | 0 | 0 | 0 | 0 | 0 | 0 | 0 | 0 | 0 | 0 |
| *O. universa* | 0.079 | 0 | 0 | 0.022 | 0 | 0.224 | 0 | 0 | 0.027 | 0.028 | 0.050 | 0 | 0.077 | 0 | 0.130 | 0.117 | 0.102 | 0 | 0.102 | 0.059 |
| *G. siphonifera* | 0.010 | 0.019 | 0.007 | 0 | 0 | 0.089 | 0 | 0.031 | 0 | 0 | 0 | 0 | 0 | 0 | 0 | 0 | 0 | 0 | 0 | 0 |
| *T. quadrilobatus* | 0 | 0 | 0 | 0 | 0 | 0.022 | 0 | 0 | 0 | 0 | 0 | 0 | 0 | 0 | 0 | 0 | 0 | 0 | 0 | 0 |
| Total | 0.108 | 0.056 | 0.143 | 0.044 | 0.047 | 0.358 | 0 | 0.063 | 0.027 | 0.027 | 0.050 | 0 | 0.077 | 0 | 0.130 | 0.117 | 0.102 | 0 | 0.102 | 0.059 |
| **Relative abundance (%)** | | | | | | | | | | | | | | | | | | | | |
| *G. ruber* (white) | 8.00 | 0.72 | 0.17 | 1.49 | 0 | 0 | 31.03 | 43.75 | 53.57 | 56.25 | 74.63 | 43.33 | 33.33 | 0 | 22.81 | 0 | 46.81 | 0 | 0 | 0 |
| *G. elongatus* | 12.00 | 0.36 | 0.17 | 0 | 3.33 | 0 | 0 | 9.38 | 7.14 | 6.25 | 11.94 | 30.00 | 0 | 0 | 12.28 | 27.27 | 14.89 | 0 | 4.00 | 0 |
| *T. sacculifer* (without sac) | 24.00 | 25.45 | 0.69 | 0 | 6.67 | 0 | 6.90 | 7.29 | 3.57 | 18.75 | 2.99 | 0 | 0 | 20.00 | 15.79 | 0.00 | 7.09 | 0 | 4.00 | 0 |
| *G. bulloides* | 15.00 | 44.44 | 11.02 | 34.33 | 20.00 | 0 | 24.14 | 3.13 | 7.14 | 0 | 4.48 | 0 | 33.33 | 0 | 3.51 | 9.09 | 8.51 | 53.85 | 16.00 | 21.74 |
| *G. inflata* | 12.00 | 9.68 | 84.85 | 37.31 | 63.33 | 35.56 | 10.34 | 4.17 | 3.57 | 0 | 0 | 0 | 0 | 20.00 | 0 | 0 | 0 | 0 | 0 | 0 |
| *O. universa* | 13.00 | 1.79 | 0.34 | 14.93 | 0 | 28.89 | 0 | 7.29 | 7.14 | 6.25 | 2.99 | 0 | 25.00 | 20.00 | 31.58 | 54.55 | 7.80 | 7.69 | 28.00 | 26.09 |
| *G. siphonifera* | 3.00 | 1.08 | 1.03 | 1.49 | 0 | 31.11 | 0 | 2.08 | 0 | 0 | 1.49 | 0 | 0 | 0 | 0 | 0 | 0 | 0.00 | 16.00 | 0 |
| *T. quadrilobatus* | 1.00 | 6.45 | 0.17 | 5.97 | 0 | 4.44 | 17.24 | 2.08 | 3.57 | 0 | 0 | 0 | 0 | 20.00 | 0 | 0 | 6.38 | 30.77 | 32.00 | 34.78 |
| *H. pelagica* | 0 | 0 | 0 | 0 | 0 | 0 | 0 | 4.17 | 0 | 6.25 | 0 | 0 | 0 | 0 | 0 | 0 | 0 | 0 | 0 | 0 |
| *T. sacculifer* (with sac) | 0 | 0 | 0 | 0 | 0 | 0 | 0 | 0 | 0 | 0 | 0 | 0 | 0 | 0 | 0 | 0 | 0.71 | 0 | 0 | 0 |
| *G. ruber* (pink) | 0 | 1.43 | 0 | 0 | 3.33 | 0 | 3.45 | 4.17 | 0 | 6.25 | 0 | 13.33 | 0 | 0 | 0 | 0 | 0 | 0 | 0 | 0 |
| *G. menardii* | 0 | 0 | 0 | 0 | 0 | 0 | 0 | 0 | 0 | 0 | 0 | 0 | 0 | 0 | 0 | 0 | 0 | 0 | 0 | 4.35 |
| Unknowns | 12.00 | 8.60 | 1.55 | 4.48 | 3.33 | 0 | 6.90 | 12.50 | 14.29 | 0 | 1.49 | 13.33 | 8.33 | 20.00 | 14.04 | 9.09 | 7.80 | 7.69 | 0 | 13.04 |
| **Weight and size** | | | | | | | | | | | | | | | | | | | | |
| *G. ruber* (white) | | | | | | | | | | | | | | | | | | | | |
| size fraction (μm) | 250-300 | | | | | | | 200-250 | 200-250 | | 200-250 | 250-300 | | | 250-300 | | 200-250 | | | |
| n° of individuals | 1 | | | | | | | 4 | 4 | | 4 | 2 | | | 4 | | 4 | | | |
| average size (μm) | 285 | | | | | | | 221 | 215.25 | | 221.5 | 281 | | | 268 | | 218.5 | | | |
| average weight (μg) | 4.667 | | | | | | | 1.583 | 2.417 | | 2 | 3.167 | | | 5.5 | | 2.083 | | | |
| SD (μg) | 0.577 | | | | | | | 0.144 | 0.289 | | 0 | 0.577 | | | 0 | | 0.144 | | | |
| size fraction (μm) | 350-400 | | | | | | | 250-350 | 250-300 | | 250-300 | 300-350 | | | | | 250-300 | | | |
| n° of individuals | 4 | | | | | | | 5 | 1 | | 3 | 1 | | | | | 5 | | | |
| average size (μm) | 390 | | | | | | | 267 | 261 | | 264 | 317 | | | | | 280.6 | | | |
| average weight (μg) | 14.333 | | | | | | | 3.867 | 2.667 | | 5.111 | 6.667 | | | | | 4.8 | | | |
| SD (μg) | 0.289 | | | | | | | 0.115 | 0.577 | | 0.192 | 0.577 | | | | | 0.2 | | | |
| size fraction (μm) | 400-450 | | | | | | | 300-350 | 350-400 | | 300-350 | | | | | | 300-350 | | | |
| n° of individuals | 1 | | | | | | | 3 | 1 | | 2 | | | | | | 5 | | | |
| average size (μm) | 412 | | | | | | | 313.333 | 356 | | 323.5 | | | | | | 343.4 | | | |
| average weight (μg) | 14.667 | | | | | | | 7.444 | 5.667 | | 11 | | | | | | 9.867 | | | |
| SD (μg) | 1.155 | | | | | | | 0.385 | 1.155 | | 0 | | | | | | 0.231 | | | |
| size fraction (μm) | | | | | | | | 350-400 | | | | | | | | | 350-400 | | | |
| n° of individuals | | | | | | | | 2 | | | | | | | | | 4 | | | |
| average size (μm) | | | | | | | | 374 | | | | | | | | | 366 | | | |
| average weight (μg) | | | | | | | | 8.833 | | | | | | | | | 9.083 | | | |
| SD (μg) | | | | | | | | 0.764 | | | | | | | | | 0.144 | | | |

(**Appendix A**, cont.).

| Location / Station | Atlantic 1 | Gibraltar 2 | Alboran Sea 3 | South-Central Western Med. 5 | Strait of Sardinia 6 | Strait of Sicily 7a | South of Ionian Sea 9 | Off Southern Crete 10 | Eastern Basin 11 | Off Nile Delta 12 | Off Lebanon 13 | Antikythera Strait 14 | Eastern Ionian Sea 15 | Adriatic Sea 17 | Otranto Strait 16 | Northern Ionian Sea 16-18 | Tyrrhenian Sea 19 | North-Central Western Med. 20 | Central Western Med. 21 | Catalano-Balear 22 |
|---|---|---|---|---|---|---|---|---|---|---|---|---|---|---|---|---|---|---|---|---|
| size fraction (µm) | | | | | | | | | | | | | | | | 400-450 | | | | |
| n° of individuals | | | | | | | | | | | | | | | | 2 | | | | |
| average size (µm) | | | | | | | | | | | | | | | | 413 | | | | |
| average weight (µg) | | | | | | | | | | | | | | | | 16.167 | | | | |
| SD (µg) | | | | | | | | | | | | | | | | 1.258 | | | | |

*G. bulloides*

| Location / Station | Atlantic 1 | Gibraltar 2 | Alboran Sea 3 | South-Central Western Med. 5 | Strait of Sardinia 6 | Strait of Sicily 7a | South of Ionian Sea 9 | Off Southern Crete 10 | Eastern Basin 11 | Off Nile Delta 12 | Off Lebanon 13 | Antikythera Strait 14 | Eastern Ionian Sea 15 | Adriatic Sea 17 | Otranto Strait 16 | Northern Ionian Sea 16-18 | Tyrrhenian Sea 19 | North-Central Western Med. 20 | Central Western Med. 21 | Catalano-Balear 22 |
|---|---|---|---|---|---|---|---|---|---|---|---|---|---|---|---|---|---|---|---|---|
| size fraction (µm) | 300-350 | 200-250 | 200-250 | 350-400 | 300-350 | | | | | | | | | | | | | | 400-450 | 300-350 |
| n° of individuals | 2 | 7 | 8 | 1 | 1 | | | | | | | | | | | | | | 1 | 3 |
| average size (µm) | 326.5 | 228.143 | 227.875 | 364 | 337 | | | | | | | | | | | | | | 414 | 318.333 |
| average weight (µg) | 4.5 | 2.571 | 3.458 | 4.667 | 4 | | | | | | | | | | | | | | 11.667 | 8.222 |
| SD (µg) | 0.5 | 0 | 0.144 | 0.577 | 1 | | | | | | | | | | | | | | 0.577 | 0.385 |
| size fraction (µm) | | 250-300 | 250-300 | | | | | | | | | | | | | | | | | 400-450 |
| n° of individuals | | 12 | 2 | | | | | | | | | | | | | | | | | 1 |
| average size (µm) | | 263.75 | 270 | | | | | | | | | | | | | | | | | 441 |
| average weight (µg) | | 2.833 | 2.833 | | | | | | | | | | | | | | | | | 20.333 |
| SD (µg) | | 0 | 0.289 | | | | | | | | | | | | | | | | | 1.155 |
| size fraction (µm) | | 300-350 | 350-400 | | | | | | | | | | | | | | | | | |
| n° of individuals | | 2 | 4 | | | | | | | | | | | | | | | | | |
| average size (µm) | | 310.5 | 386.5 | | | | | | | | | | | | | | | | | |
| average weight (µg) | | 4.5 | 9.667 | | | | | | | | | | | | | | | | | |
| SD (µg) | | 0.5 | 0.144 | | | | | | | | | | | | | | | | | |
| size fraction (µm) | | 350-400 | 400-450 | | | | | | | | | | | | | | | | | |
| n° of individuals | | 2 | 2 | | | | | | | | | | | | | | | | | |
| average size (µm) | | 375.5 | 429 | | | | | | | | | | | | | | | | | |
| average weight (µg) | | 5.833 | 11 | | | | | | | | | | | | | | | | | |
| SD (µg) | | 0.289 | 0 | | | | | | | | | | | | | | | | | |
| size fraction (µm) | | 400-450 | 450-500 | | | | | | | | | | | | | | | | | |
| n° of individuals | | 1 | 1 | | | | | | | | | | | | | | | | | |
| average size (µm) | | 447 | 477 | | | | | | | | | | | | | | | | | |
| average weight (µg) | | 9.333 | 7.333 | | | | | | | | | | | | | | | | | |
| SD (µg) | | 0.577 | 0.577 | | | | | | | | | | | | | | | | | |

*O. universa*

| Location / Station | Atlantic 1 | Gibraltar 2 | Alboran Sea 3 | South-Central Western Med. 5 | Strait of Sardinia 6 | Strait of Sicily 7a | South of Ionian Sea 9 | Off Southern Crete 10 | Eastern Basin 11 | Off Nile Delta 12 | Off Lebanon 13 | Antikythera Strait 14 | Eastern Ionian Sea 15 | Adriatic Sea 17 | Otranto Strait 16 | Northern Ionian Sea 16-18 | Tyrrhenian Sea 19 | North-Central Western Med. 20 | Central Western Med. 21 | Catalano-Balear 22 |
|---|---|---|---|---|---|---|---|---|---|---|---|---|---|---|---|---|---|---|---|---|
| size fraction (µm) | 350-400 | 250-300 | 500-550 | 400-450 | | 450-500 | | 300-350 | 350-400 | 700-750 | 650-700 | | 700-750 | 450-500 | 300-350 | 400-450 | 400-450 | 400-450 | 450-500 | 350-400 |
| n° of individuals | 3 | 1 | 1 | 2 | | 1 | | 1 | 1 | 1 | 1 | | 2 | 1 | 1 | 1 | 1 | 1 | 2 | 1 |
| average size (µm) | 390 | 286 | 501 | 445 | | 479 | | 342 | 398 | 719 | 687 | | 722.5 | 452 | 347 | 444 | 441 | 441 | 479.5 | 377 |
| average weight (µg) | 17.667 | 7 | 20.667 | 11.667 | | 31 | | 3 | 6.333 | 47 | 43 | | 24.167 | 14.333 | 5.333 | 18.667 | 24.333 | 22.667 | 31 | 20 |
| SD (µg) | 0.333 | 0 | 0.577 | 0.289 | | 1 | | 0 | 0.577 | 1 | 0 | | 0.289 | 0.577 | 0.577 | 0.577 | 0.577 | 0.577 | 0.5 | 1 |
| size fraction (µm) | 400-450 | | | 450-500 | | 500-550 | | 350-400 | 500-550 | | 750-800 | | 750-800 | | 350-400 | 550-600 | 450-500 | | 550-600 | 400-450 |
| n° of individuals | 1 | | | 3 | | 2 | | 3 | 1 | | 1 | | 1 | | 1 | 1 | 1 | | 1 | 2 |
| average size (µm) | 444 | | | 479 | | 539.5 | | 373.667 | 539 | | 781 | | 785 | | 369 | 559 | 455 | | 571 | 425.5 |
| average weight (µg) | 28.667 | | | 22.889 | | 33.833 | | 6.556 | 25.667 | | 54.667 | | 53.667 | | 6.667 | 34.333 | 23.667 | | 45 | 24.167 |
| SD (µg) | 1.155 | | | 0.192 | | 0.289 | | 0.385 | 0.577 | | 0.577 | | 0.577 | | 0.577 | 0.577 | 0.577 | | 1 | 0.577 |
| size fraction (µm) | 500-550 | | | 650-700 | | 600-650 | | 400-450 | | | | | | | 400-450 | 600-650 | 500-550 | | 650-700 | 450-500 |
| n° of individuals | 1 | | | 1 | | 1 | | 1 | | | | | | | 1 | 2 | 6 | | 2 | 1 |
| average size (µm) | 527 | | | 656 | | 603 | | 439 | | | | | | | 412 | 640 | 534.5 | | 676 | 482 |
| average weight (µg) | 36.667 | | | 25.667 | | 50.667 | | 13.667 | | | | | | | 13 | 54.833 | 30.278 | | 84.333 | 35 |
| SD (µg) | 0.577 | | | 1.155 | | 0.577 | | 1.155 | | | | | | | 0 | 0.289 | 0.096 | | 0.289 | 1 |
| size fraction (µm) | 550-600 | | | | | 650-700 | | 450-500 | | | | | | | 450-500 | 650-700 | | | 750-800 | 500-550 |
| n° of individuals | 6 | | | | | 6 | | 1 | | | | | | | 1 | 2 | | | 1 | 1 |
| average size (µm) | 578.667 | | | | | 674.333 | | 460 | | | | | | | 476 | 656.5 | | | 762 | 509 |
| average weight (µg) | 45.389 | | | | | 47.889 | | 17.333 | | | | | | | 24 | 63.333 | | | 136 | 42 |
| SD (µg) | 0.096 | | | | | 0.096 | | 1.155 | | | | | | | 1 | 0.289 | | | 0 | 0 |
| size fraction (µm) | 600-650 | | | | | 700-750 | | | | | | | | | 500-550 | | | | | |
| n° of individuals | 1 | | | | | 2 | | | | | | | | | 3 | | | | | |
| average size (µm) | 605 | | | | | 720 | | | | | | | | | 527.333 | | | | | |
| average weight (µg) | 48.667 | | | | | 34 | | | | | | | | | 21.778 | | | | | |
| SD (µg) | 0.577 | | | | | 0 | | | | | | | | | 0.192 | | | | | |


(**Appendix A**, cont.).

| Location Station | Atlantic 1 | Gibraltar 2 | Alboran Sea 3 | South-Central Western Med. 5 | Strait of Sardinia 6 | Strait of Sicily 7a | South of Ionian Sea 9 | Off Southern Crete 10 | Eastern Basin 11 | Off Nile Delta 12 | Off Lebanon 13 | Antikythera Strait 14 | Eastern Ionian Sea 15 | Adriatic Sea 17 | Otranto Strait 16 | Northern Ionian Sea 16-18 | Tyrrhenian Sea 19 | North-Central Western Med. 20 | Central Western Med. 21 | Catalano-Balear 22 |
|---|---|---|---|---|---|---|---|---|---|---|---|---|---|---|---|---|---|---|---|---|
| size fraction (µm) | 650-700 | | | | | 750-800 | | | | | | | | | 550-600 | | | | | |
| n° of individuals | 1 | | | | | 1 | | | | | | | | | 1 | | | | | |
| average size (µm) | 651 | | | | | 772 | | | | | | | | | 570 | | | | | |
| average weight (µg) | 50.667 | | | | | 48 | | | | | | | | | 17.333 | | | | | |
| SD (µg) | 0.577 | | | | | 1 | | | | | | | | | 1.528 | | | | | |
| size fraction (µm) | | | | | | | | | | | | | | | 600-650 | | | | | |
| n° of individuals | | | | | | | | | | | | | | | 1 | | | | | |
| average size (µm) | | | | | | | | | | | | | | | 625 | | | | | |
| average weight (µg) | | | | | | | | | | | | | | | 23 | | | | | |
| SD (µg) | | | | | | | | | | | | | | | 0 | | | | | |
| size fraction (µm) | | | | | | | | | | | | | | | 650-700 | | | | | |
| n° of individuals | | | | | | | | | | | | | | | 2 | | | | | |
| average size (µm) | | | | | | | | | | | | | | | 654.5 | | | | | |
| average weight (µg) | | | | | | | | | | | | | | | 31.167 | | | | | |
| SD (µg) | | | | | | | | | | | | | | | 0.289 | | | | | |


## Acknowledgments

We thank the captain and crew of the Spanish research vessel R/V Ángeles Alvariño. B. d'Amario is thanked for her software guidance and overall advice as well. The work was funded by the EC FP7 'Mediterranean Sea Acidification in a changing climate' project (MedSeA; grant agreement 265103).

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

**Tables**
**Table 1.** Date, time, location, volume filtered and environmental parameters of the sampled stations. Sea
surface temperature (SST) and sea surface salinity (SSS) measured at 5 m depth. The remaining
parameters are averaged from 5 to 200 depth with their respective SDs in parenthesis.
**Table 2.** Loadings of the environmental parameters in the PCA and additional Pearson correlation
coefficients (*r*) for relationships between other variables and PCA factors (n=20) and the results for the
abundances of *G. ruber* (white) (n=13), *T. sacculifer* (whitout sac) (n=13), *G. bulloides* (n=16), *G. inflata*
(n=10), *O. universa* (n=17) and the total abundances (n=20); and the area density of *G. bulloides* (n=16),
*G. ruber* (white) (n=13) and *O. universa* (n=17). *r*-values in bold are significant at $p<0.05$, *$p<0.1$.

**Table 1.**

| Leg | Station Code | Station Name | Day (DD/MM/YYYY) | Time | Latitude | Longitude | Volume (m³) | Temperature (ºC) | SST (ºC) | Salinity (PSU) | SSS (PSU) | Fluorescense (µg/l) | pH | $[CO_3^{2-}]$ (mmol/kg) |
|---|---|---|---|---|---|---|---|---|---|---|---|---|---|---|
| 1 | 1 | Atlantic | 03/05/2013 | 0:03 | 36º03' | -6º65' | 1016 | 16.08 (0.84) | 17.88 | 36.27 (0.10) | 35.95 | 0.36 (0.32) | 8.06 (0.05) | 178.89 (22.25) |
| | 2 | Gibraltar | 03/05/2013 | 12:47 | 35º94' | -5º56' | 537 | 14.22 (1.05) | 17.11 | 37.51 (0.81) | 36.35 | 0.11 (0.06) | 8.06 (0.02) | 179.90 (6.15) |
| | 3 | Alboran Sea | 05/05/2013 | 20:55 | 36º12' | -4º19' | 1403 | 15.06 (1.17) | 16.87 | 37.13 (0.68) | 36.37 | 0.45 (0.44) | 8.09 (0.03) | 191.50 (13.84) |
| | 5 | South-Central Western Mediterranean | 08/05/2013 | 10:44 | 38º54' | 5º56' | 459 | 14.33 (1.19) | 16.99 | 37.95 (0.23) | 37.65 | 0.18 (0.22) | 8.10 (0.02) | 200.36 (10.06) |
| | 6 | Strait of Sardinia | 09/05/2015 | 20:34 | 38º27' | 8º69' | 423 | 14.34 (1.16) | 17.50 | 38.23 (0.19) | 37.77 | 0.19 (0.26) | 8.08 (0.03) | 199.89 (15.38) |
| | 7a | Strait of Sicily | 11/05/2013 | 0:20 | 37º04' | 13º18' | 447 | 15.12 (0.86) | 17.27 | 38.16 (0.52) | 37.43 | 0.23 (0.23) | 8.09 (0.01) | 207.14 (3.38) |
| | 9 | South of Ionian Sea | 12/05/2013 | 11:31 | 35º12' | 18º29' | 425 | 16.17 (1.01) | 19.53 | 38.78 (0.10) | 38.64 | 0.13 (0.14) | 8.12 (0.02) | 232.36 (3.30) |
| | 10 | Off Southern Crete | 14/05/2013 | 14:40 | 33º81' | 24º27' | 320 | 16.51 (1.44) | 19.58 | 39.00 (0.39) | 36.60 | 0.12 (0.19) | 8.11 (0.01) | 232.38 (8.43) |
| | 11 | Eastern Basin | 15/05/2013 | 13:01 | 33º50' | 28º00' | 372 | 17.21 (1.30) | 20.59 | 38.80 (0.44) | 36.19 | 0.10 (0.07) | 8.12 (0.02) | 243.57 (10.26) |
| | 12 | Off Nile Delta | 17/05/2013 | 3:14 | 33º22' | 32º00' | 364 | 17.59 (1.46) | 21.82 | 38.99 (0.25) | 37.45 | 0.15 (0.12) | 8.11 (0.02) | 239.99 (9.93) |
| | 13 | Off Lebanon | 17/05/2013 | 16:15 | 34º23' | 33º23' | 397 | 17.35 (1.33) | 21.58 | 38.73 (1.48) | no data | 0.16 (0.13) | 8.11 (0.02) | 238.28 (7.52) |
| 2 | 14 | Antikythera Strait | 20/05/2013 | 6:06 | 36º70' | 23º42' | 334 | 16.66 (1.21) | 20.00 | 39.07 (0.03) | 39.15 | 0.12 (0.08) | 8.13 (0.01) | 241.84 (6.26) |
| | 15 | Eastern Ionian Sea | 21/05/2013 | 21:25 | 36º40' | 20º81' | 391 | 16.52 (1.31) | 20.27 | 39.05 (0.01) | 39.10 | 0.15 (0.15) | no data | no data |
| | 17 | Adriatic Sea | 23/05/2013 | 21:09 | 41º84' | 17º25' | 440 | 14.67 (1.30) | 18.76 | 38.82 (0.05) | 39.12 | 0.20 (0.21) | 8.10 (0.02) | 218.53 (14.65) |
| | 16 | Otranto Strait | 24/05/2013 | 23:49 | 40º23' | 18º84' | 385 | 15.67 (1.15) | 19.49 | 38.70 (1.34) | 30.47 | 0.16 (0.15) | 8.13 (0.01) | 236.93 (12.88) |
| | 16-18 | Northern Ionian Sea | 25/05/2013 | 9:30 | 39º07' | 18º70' | 426 | no data | no data | no data | no data | no data | no data | no data |
| | 19 | Tyrrhenian Sea | 27/05/2013 | 12:40 | 39º83' | 12º52' | 391 | 14.74 (1.47) | 18.60 | 38.30 (0.20) | 37.97 | 0.18 (0.24) | 8.12 (0.02) | 216.97 (11.27) |
| | 20 | North-Central Western Mediterranean | 29/05/2013 | 20:00 | 41º32' | 5º66' | 356 | 13.88 (0.94) | 15.52 | 38.29 (0.20) | 33.75 | 0.36 (0.24) | 8.14 (0.02) | 219.89 (11.27) |
| | 21 | Central Western Mediterranean | 30/05/2013 | 10:30 | 40º07' | 5º95' | 392 | 13.98 (0.95) | 16.78 | 37.66 (1.74) | 37.37 | 0.17 (0.21) | 8.11 (0.01) | 204.41 (7.70) |
| | 22 | Catalano-Balear | 31/05/2013 | 13:55 | 40º95' | 3º32' | 339 | 14.08 (1.33) | 16.81 | 38.43 (0.08) | 38.34 | 0.25 (0.39) | 8.13 (0.02) | 218.43 (13.11) |


**Table 2.**

| | PCA results | | Abundances | | | | | | Area density | | |
|---|---|---|---|---|---|---|---|---|---|---|---|
| | Factor 1 | Factor 2 | G. ruber (white) | T. sacculifer (without sac) | G. bulloides | G. inflata | O. universa | TOTAL | G. bulloides | G. ruber (white) | O. universa |
| Factor 1 | 1 | 0 | -0.297 | 0.353 | 0.511 | 0.242 | 0.009 | 0.309 | -0.369 | 0.324 | -0.449 |
| Factor 2 | 0 | 1 | 0.121 | -0.549 | -0.470 | -0.209 | -0.127 | -0.406 | 0.279 | -0.296 | 0.133 |
| Environmental factor loadings: | | | | | | | | | | | |
| Temperature | -0.825 | -0.030 | 0.346* | -0.158 | -0.333* | -0.154 | -0.198 | -0.154 | 0.294 | -0.324* | 0.464 |
| Salinity | -0.777 | 0.532 | 0.296 | -0.353* | -0.425* | -0.479 | -0.005 | -0.393* | 0.346* | -0.721 | 0.355* |
| Oxygen | -0.084 | 0.602 | -0.149 | -0.675 | -0.684 | -0.241 | 0.042 | -0.682 | 0.050 | 0.072 | 0.509 |
| Fluorescence | 0.721 | -0.185 | -0.378* | -0.101 | -0.020 | 0.459 | -0.063 | 0.028 | -0.275 | 0.738 | -0.246 |
| [NO3] | 0.912 | -0.113 | -0.344* | 0.460 | 0.567 | 0.166 | -0.063 | 0.290 | -0.295 | 0.156 | -0.548 |
| [PO4] | 0.893 | -0.272 | -0.361* | 0.461 | 0.579 | 0.293 | -0.168 | 0.340* | -0.264 | 0.252 | -0.538 |
| pH | -0.189 | 0.969 | 0.215 | -0.559 | -0.563 | -0.351* | 0.117 | -0.448 | 0.263 | -0.381* | 0.236 |
| $pCO_2$ | 0.086 | -0.941 | -0.170 | 0.589* | 0.554 | 0.196 | -0.160 | 0.378* | -0.167 | 0.154 | -0.177 |
| $[CO_3^{2-}]$ | -0.594 | 0.729 | 0.352* | -0.451 | -0.566 | -0.452 | -0.016 | -0.447 | 0.406* | -0.614 | 0.434 |
| | n=20 | n=20 | n=13 | n=13 | n=16 | n=10 | n=17 | n=20 | n=16 | n=13 | n=17 |


**Figures**
**Fig. 1**. **(a)** Temperature ($^{o}$C), **(b)** salinity, **(c)** fluorescence ($\mu$g l$^{-1}$), **(d)** pH, and **(e)** [CO$_3^{2-}$] ($\mu$mol kg$^{-1}$)
values of the water column of the transect. Values follow a color scale (under every graph), also values
shown in the isometric lines. X axis: water depth. Y axis: longitude (degrees). Measurement locations
indicated with white dots, with the coinciding stations numbered at top. The station number and the map
section are shown on the map (f). For station code names see Table 1. Note reversed color scale at (d) and
(e). Software used: Ocean Data View (Schlitzer, 2016).
**Fig. 2**. Sampled stations with BONGO nets (dots). The numbers in the picture represent the station codes:
First transect: 1 to 13, second transect: 14 to 22. For station code names see Table 1. Color scale at right
represents the values of surface chlorophyll concentration (in $\mu$g/l), retrieved from *MODIS Aqua (L2),*
from the closest day as possible, specified in the upper part, of the first transect.
**Fig. 3.** Sample scores on the two PCA factors with the loadings of the environmental parameters on each
factor represented by the red axis. The black axis represents the overlay of the absolute abundance values
(individuals·10 m$^{-3}$) according to every station scores of **(a)** all the foraminifera sample, **(b)** *G. inflata*, **(c)**
*T. sacculifer* (without sac), **(d)** *G. ruber* (white), **(e)** *G. bulloides*, and **(f)** *O. universa*. Overlay of the Area
density ($\rho_A$) values ($\mu$g·$\mu$m$^{-2}$) of **(g)** *G. ruber* (white), **(h)** *G. bulloides*, and **(i)** *O. universa*. In blue colour
western Mediterranean stations (incl. Atlantic and Strait of Gibraltar), in red color the eastern
Mediterranean stations.
**Fig. 4.** Absolute abundance of planktic foraminifera from BONGO nets during leg 1 (stations 1 to 13) and
leg 2 (stations 22 to 14). Category 'Others' is comprised of *G. siphonifera/G. calida/ G. radians* plexus,
*T. quadrilobatus*, *H. pelagica*, *G. ruber* (pink), *G. menardii* and *T. sacculifer* (with sac).
**Fig. 5.** Percentage of each planktic foraminifera size fraction in each station from leg 1 (stations 1 to 13)
and leg 2 (stations 22 to 14). Sample size is indicated in italics at the top of each station bar.
**Fig. 6.** Relative abundance of planktic foraminifera (%). Category 'Others' is comprised of *G.*
*siphonifera/G. calida/ G. radians* plexus, *T. quadrilobatus*, *H. pelagica*, *G. ruber* (pink), *G. menardii* and
*T. sacculifer* (with sac). Less than 1% values are not shown. Number in parenthesis indicates the total
individuals of each location.
**Fig. 7.** Area density of *G. ruber* (white) and *G. bulloides* in box-and-whisker plots representation for the
different location groupings in the Mediterranean. Box extends from the lower (Q$_1$) to upper (Q$_3$)
quartiles values of the data, with a line at the median (Q$_2$). Whiskers extend from the quartiles to values
comprised within a 1.5 interquartile range (IQR = Q$_3$ – Q$_1$) distance: Q$_1$ - 1.5·IQR; Q$_3$ + 1.5 IQR. The
Coefficient of Variation (CV) of each location grouping is represented as a black dot.
**Figure 1**

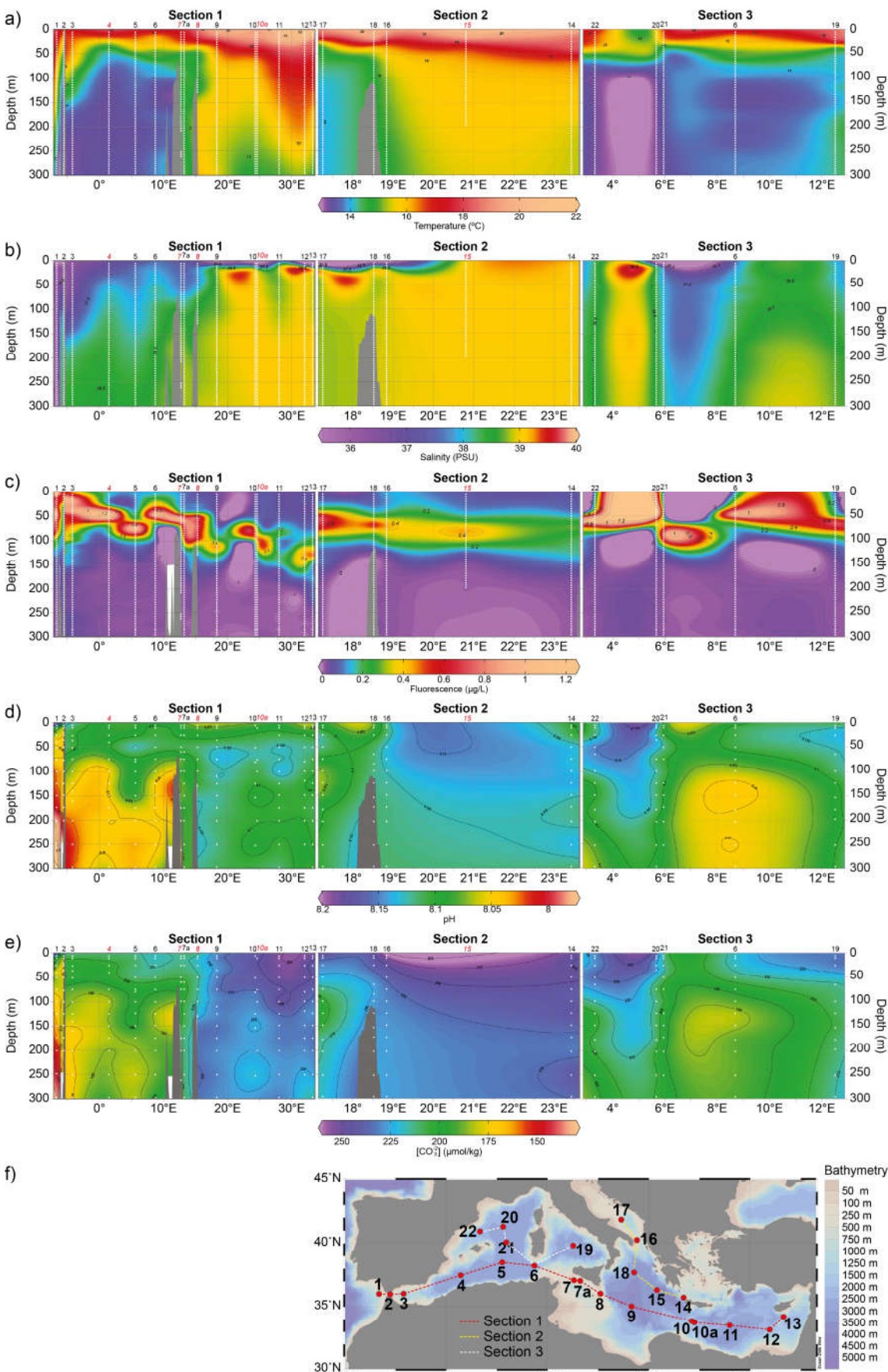


**Figure 2**

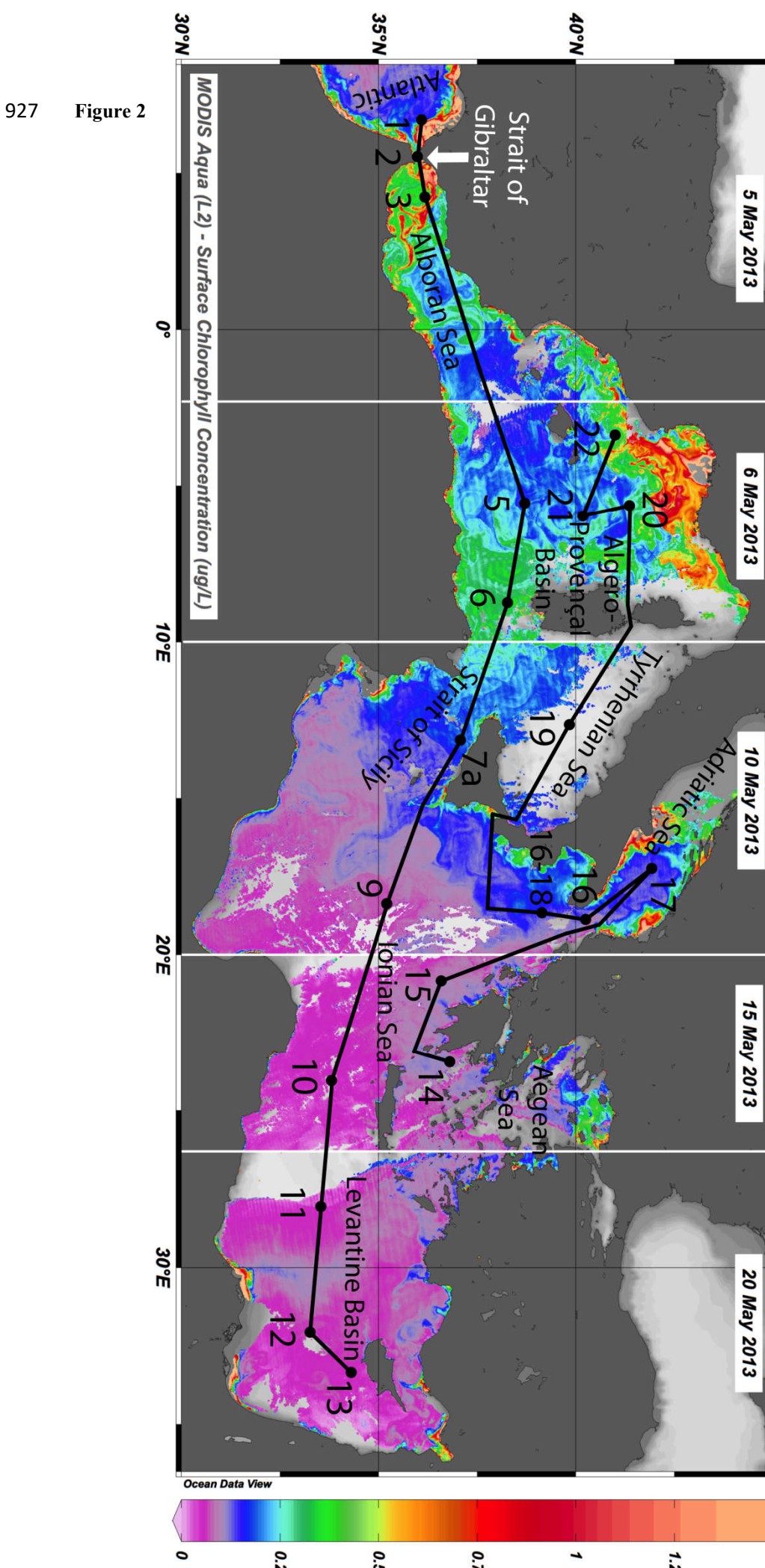

**Figure 3**

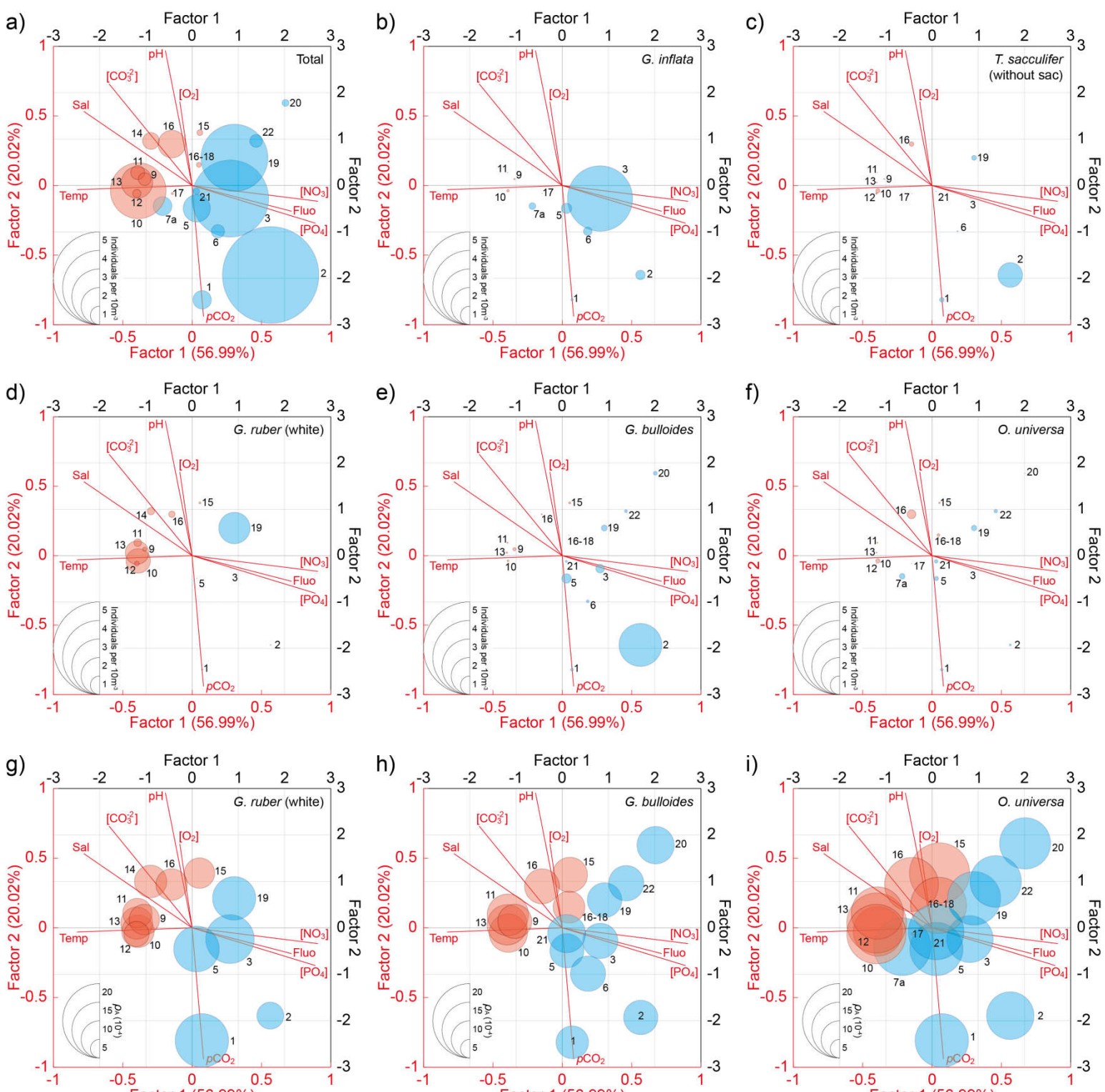

**Figure 4**

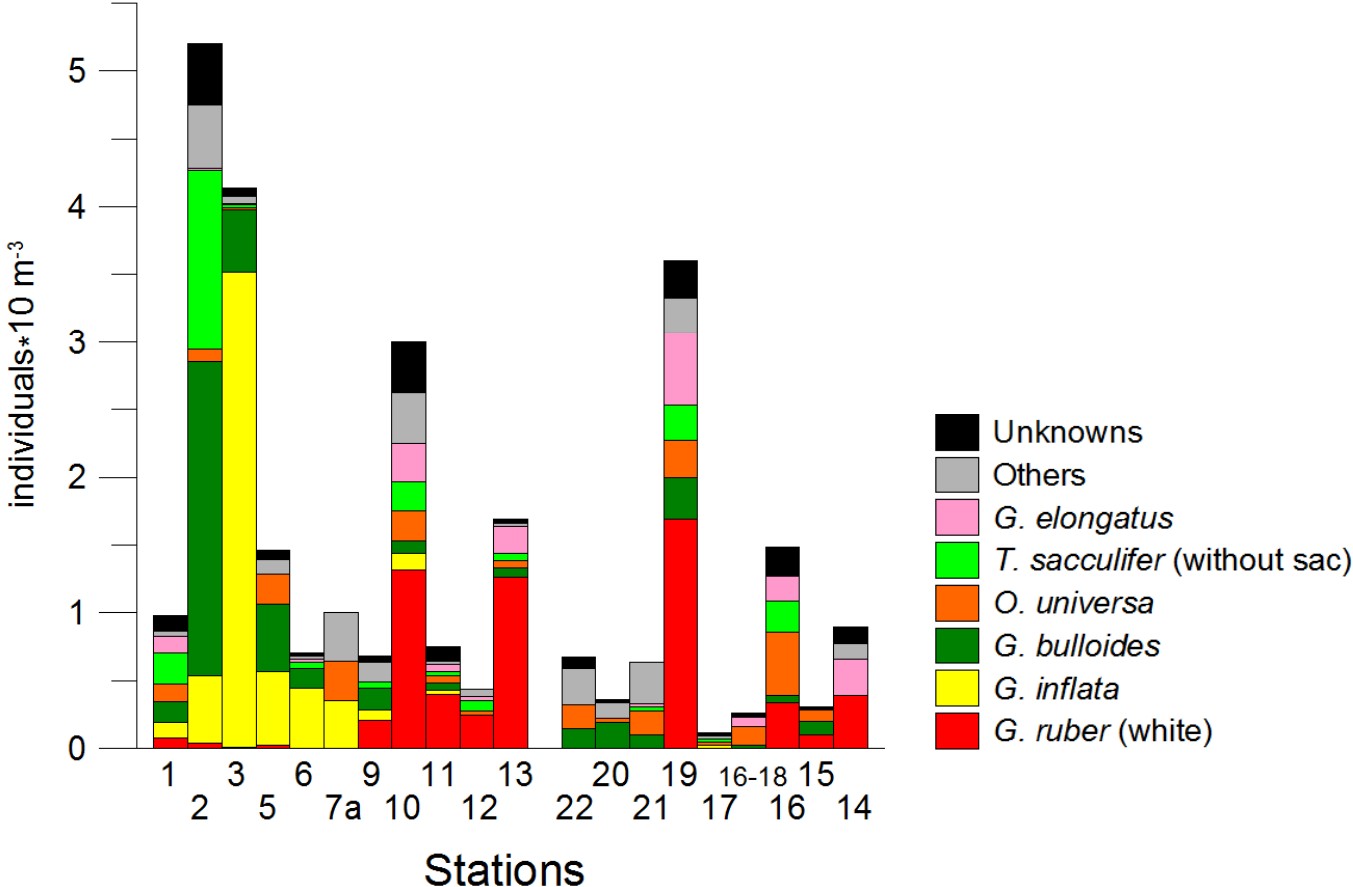

**Figure 5**

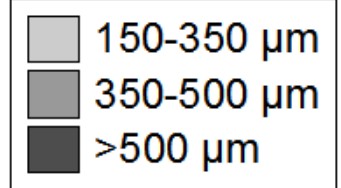

Figure 5

Stations


**Figure 6**

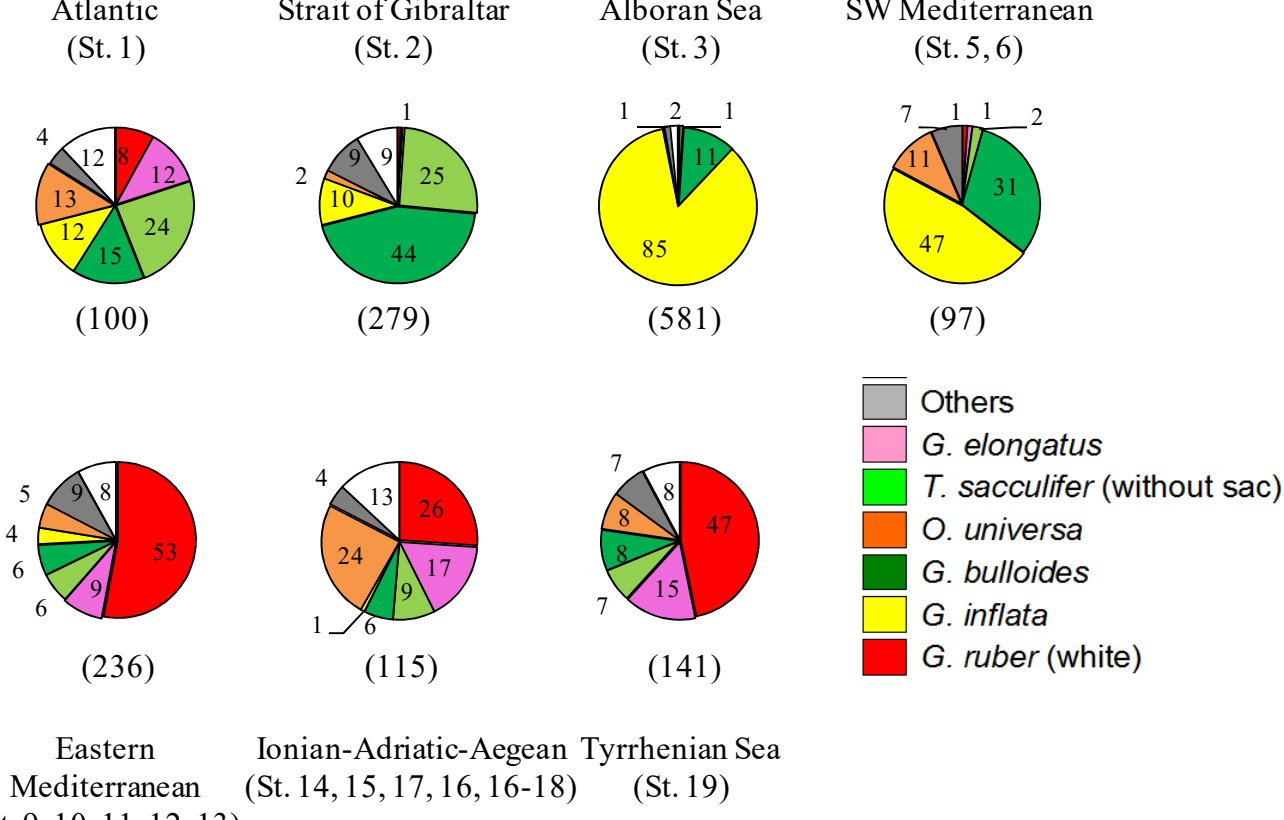

**Figure 7**

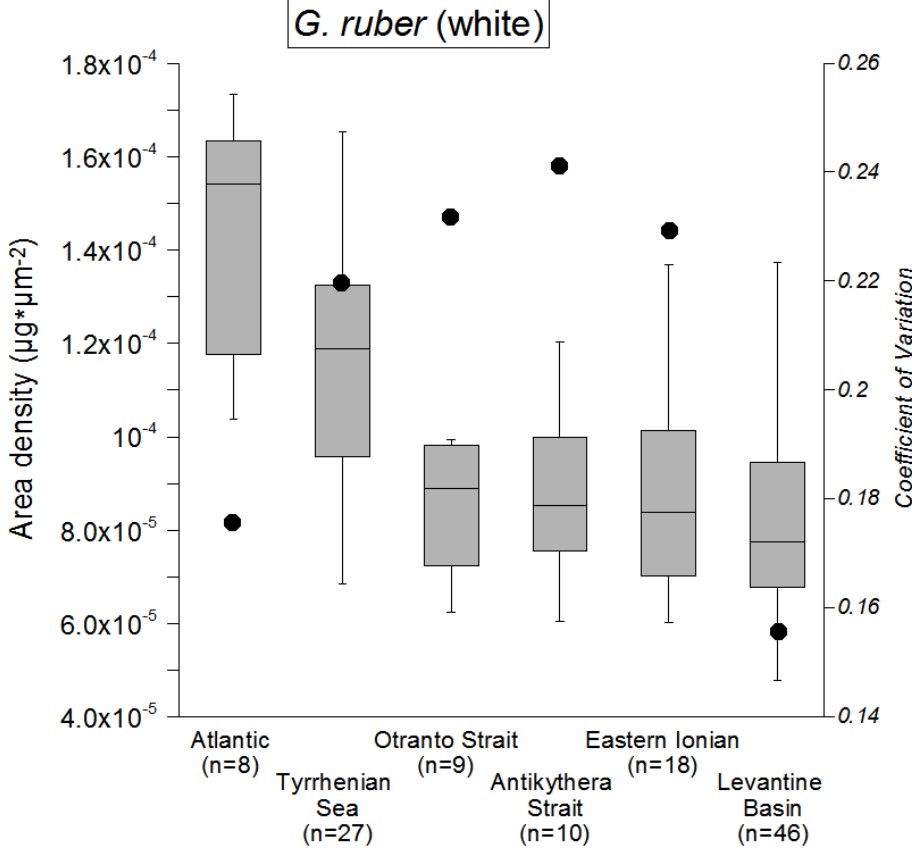


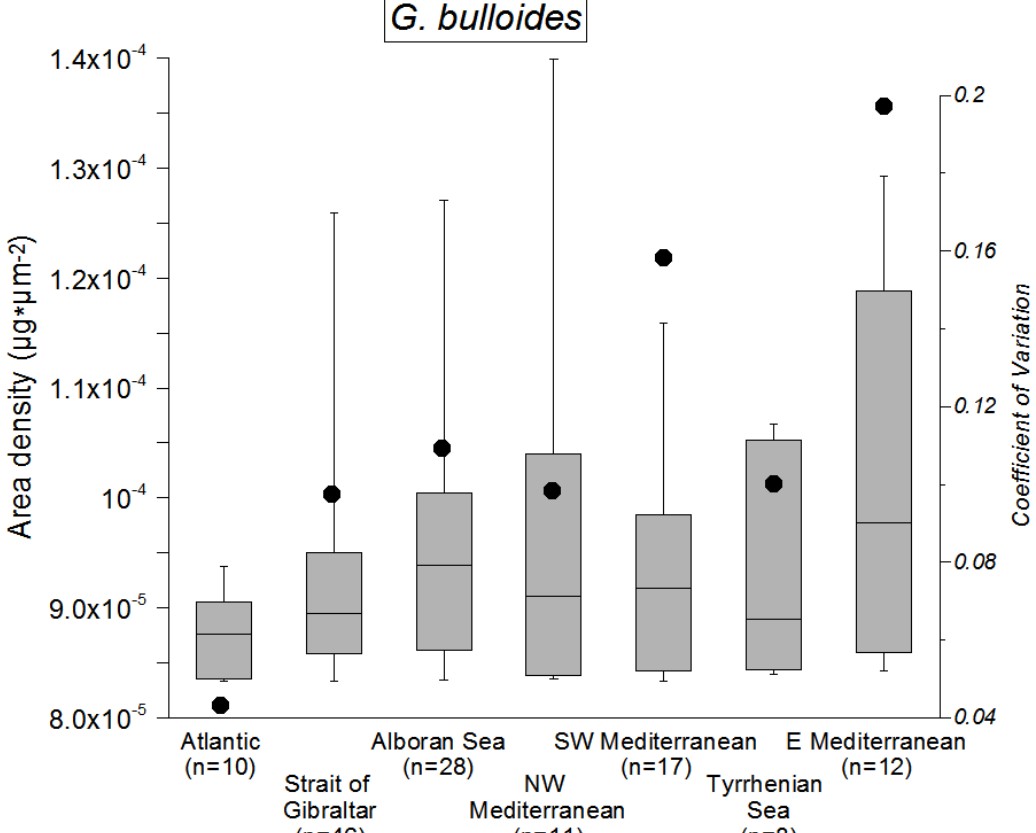
