# Peer review of "Low planktic foraminiferal diversity and abundance observed in a"

_Biogeosciences, 2016_

## Referee Comment (RC1) · Anonymous Referee #1 · 18 Jul 2016

Mallo et al. present new data on planktonic foraminifera abundance and shell weight from the upper water column in the Mediterranean Sea. They observe large differences in the species composition across the basin and suggest a suite of parameters (stratification, temperature, food availability etc) that could be responsible for the observed patterns. They also investigated the size-normalised weight (SNW) of two species and suggest that food availability, rather than carbonate chemistry of the seawater, may be the dominant control on shell weight of these species.

The data appear of good quality and manuscript is well written and most of the figures are clear. It is also good to see that the data are made available. I think this paper could be a valuable addition, but the interpretation of the data is not well thought through

and clearly needs more work/analysis. At present it is not really clear what is new about this study, or – and that wouldn't be a disadvantage – how existing concepts on foraminifera abundance and shell weight variability are confirmed. In addition, I have numerous small questions and some major doubts about the (statistical) treatment of the data and hence the interpretation.

Major comments

Unclear methodology for SNW: what diameter was measured? Neither ruber or bulloides approximates a sphere, so this is important to mention. It is unclear to me if the same specimens were used to measure diameter + area and weight, please explain. And I also understand – but I am not sure - that at each station only specimens were measured within a certain 50 m size range, is that right? If so, could that not bias the trends because samples weren't selected randomly and trends in shell weight may be affected by trends in shell size?

Analysis of species assemblages: I have a major problem with the fact that to analyse the species distribution the authors use relative abundances. This does not make sense if one looks at individual species (closed sum effect: the % of species A will because % B changes; see for instance station 10 and 13 where the absolute abundance of G. ruber ss is similar, yet the relative very different) and if one wants to investigate species assemblage variability other techniques (PCA, cluster analysis) that look at the entire assemblage are more appropriate. I don't see what bias large variability in absolute abundance could cause (L222), why would this be bias? The authors should decide what they want to do: investigate the assemblage and use a different technique or investigate individual species abundance and use absolute abundances. The discussion and conclusions will then need to be rewritten.

SNW regressions: first of all, the rationale behind the regressions isn't clear to me. Why investigate the relation between area and diameter? Why is diameter of interest if one normalises to area in SNW? It is entirely unclear to me what we learn from this and

hence how to these analyses can be used to exclude O. universa from further analysis. This really needs more explanation. Moreover, in this respect it may also be better to use the term area density (e.g. Marshall et al., 2013) rather than SNW to distinguish from sieve-based size measurements.

Then on to the actual regressions. What is the rationale/bio-physical reason that area and diameter should be linearly related in log-log space? First of all, this power regression implies that neither area nor diameter can actually be zero, which cannot be correct, the regression line should go through the origin. Secondly, why wouldn't a simple power regression model suffice (I'd expect the equation for O. universa to be close to pi*r^2). Something similar holds for the regression of area and weight and area and diameter (again why diameter?). Why, in the case for area, a different model for each species, are there any reasons for these differences and for the choice of any model in particular? The problem with the present equations is immediately clear when looking at Fig S4c: the predicted weight of shells with an area of 10^5 m^2 is 0 g, which is physically impossible. Since weight is linearly related to volume (through density) wouldn't one expect y to be related to x^3?

Moreover, and this applies to both the analysis of the species abundance and the SNW, looking at Fig.1 it appears that with the exception of fluorescence, all parameters are strongly correlated; so how do the authors determine which of these parameters is really important for the prediction of the species assemblage or SNW? The actual correlations between water column characteristics and foraminifera abundance or SNW are not shown, yet this is the most important of the study. This should be amended and the predictive power of the proposed models should be shown.

Influence of wall thickness on shell weight: the authors mention this briefly in the discussion about O. universa. I'm surprised that the study by Marshall et al (2015) on exactly the same topic is not mentioned.

The conclusion that SNW and therefore calcite formation is not limited by carbonate

chemistry in my opinion not supported by the data. Both pH and [CO32-] are high in the Med, so how can one exclude the possibility that both parameters limit calcification? If anything, but the authors need to firmly demonstrate this, it may be that at high pH and [CO32-] other parameters may be more important. But see also the comment above about the fact that in seawater everything appears correlated with everything making it very difficult to isolate the influence of a single parameter. Fig. 6 is not very revealing in this respect: it shows a spatial trend (based on unclear grouping of the data) that may or may not be statistically significant and that may or may not be related to carbonate chemistry or food availability. The authors have a unique dataset including ancillary data and could do better to explain the variability in SNW.

Lunar and seasonal abundance variations, diel migration: the possible effect of a lunar reproductive cycle on the abundance should be mentioned in the introduction and discussed (Bijma and Hemleben, 1994; Jonkers et al., 2015). And even though the authors discuss the effect of seasonality in the discussion, it would be good to mention it also in the introduction. There at least two long-term sediment trap studies from the Western Med that could be used to place these new observations in perspective (Bárcena et al., 2004; Rigual-Hernández et al., 2012). Moreover, the authors also allude to diel migration, yet don't really make anything out of this (I wonder if it's possible with nets that integrate over the upper 200 m).

Suggested trend in abundance and diversity (L28-30; 84-85; 300-305): while such a trend would be very interesting I really don't think that this is anything else than speculation. Two cruises almost 20 years apart are not enough to constrain intra- and interannual variability in foram abundance and diversity, so there is simply not enough data to support this statement. There are also important sampling differences between the present study and the one by Pujol and Grazzini: 1) the maximum depth of observations (350 m vs 200 m) and the spatial distribution, which both affect the observed abundance and diversity and a simple comparison of mean abundance or total diversity compromised. I suggest that the authors remove this speculative remark

from the paper.

Minor/technical comments

Frequent use of locations that are not indicated on a map. Please make sure that each locality is indicated.

Fig.1: use same x axis scale for each panel.

Fig. 3: use same y axis scale (perhaps log based?). A better representation of the data (Figs 3-6) may be to plot them on a map, or add a small map inset to the figures.

Fig. 4: what is 'n' below the graphs (16-18 > 16?)?

L38: perhaps replace radiation with sunlight.

L39: not only depth habitat, also seasonality.

L50: provide a reference for the expedition.

L52: large not high abundance variations.

L57-69: From this §it seems that the controls on the sedimentary assemblage are different from those on the water column assemblage. The main difference of course is that fact that water column observations are mere snap shots in time, whereas the sediment integrates centuries to millennia. Could the controls really be different, could the sedimentary signal integrate enough to obscure intra- and interannual variability in food availability? I'd find it interesting if the authors could spend a bit of time on this.

L66: correlation with what? The authors appear use correlation and statistically significant quite often without referring to what was tested, how and with what confidence interval.

L70: Consider changing 'Its weight..' by 'Their shell mass...' to be more consistent.

L73: What's the conclusion, implication of mentioning of the De Beer study?

[Figure]

L78: Studies of the water column in the Mediterranean (or similar); not Mediterranean studies.

L88-95: while all of this holds true of course, the most important difference between the living (water) assemblages and the dead (sedimentary) is the time integrated in the sample (see also above) and (post)depositional changes to the assemblage. This needs to be mentioned. Living specimens are of course also advected; in fact, advection during life is probably more important than during sinking (simply because sinking takes less time). The study of Van Sebille is probably not very relevant for Mediterranean: with only six grid cells in the entire basin one can hardly expect that the circulation is realistically represented.

L112: Gulf of Lions

L125: stratified? I'm not entirely sure, so please explain, but I thought that BONGO are not depth stratified. I understand that all the observations mentioned here are integrated over the upper 200 m of the water column. Please explain precisely what was done; I also assume that the statement that the samples were taken at 200 m depth (L135) is not correct.

L154: what are unclassified specimens?

L166: remove location 1 from the list, abundances are clearly different there. Also refer to figures in this section.

L210: . . . dominance 'of a single species'. . .

L233: how was significance determined? P-value?

L243; 252: how were the locations grouped?

L249: add SNW after G. rubber

L293-295: all these species mentioned here are winter species of which the flux happens in a single short pulse (Bárcena et al., 2004; Rigual-Hernández et al., 2012).

Sampling at the end of spring could thus easily have missed them.

§5.2: please separate more clearly what are new results and what is existing knowledge.

L366: '. . . ranges as the . . .'

L367: '. . . and it was also found . . .'

L368: what characteristic of inflata shows this correlation? I don't follow this conclusion.

L377: 'In accordance with. . .' and '. . . Atlantic station. . .' (there is no causal link and only one Atlantic station).

L378: there isn't a station completely dominated by a single species, please reword.

L439: what is meant with the SNW is statistically significant? What was tested, with which confidence interval, using which test?

L445: again, see above. In addition, there seem to be two groups in O. universa (Fig. S3. 4). Have the authors looked at the spatial pattern of the SNW?

L478: the distributions are significant? Please reword.

L478-484: is there an inverse relationship between ruber abundance (absolute) and SNW? If so, it would be good if the authors could discuss why food availability has a different effect on abundance and SNW.

L511: reword '. . . heavier average weight-diameter relation. . . '.

L515: reproduction? Not calcification?

References

Bárcena, M.A., Flores, J.A., Sierro, F.J., Pérez-Folgado, M., Fabres, J., Calafat, A., Canals, M., 2004. Planktonic response to main oceanographic changes in the Alboran Sea (Western Mediterranean) as documented in sediment traps and surface sediments. Marine Micropaleontology 53, 423-445.

Bijma, J., Hemleben, C., 1994. Population dynamics of the planktic foraminifer Globigerinoides sacculifer (Brady) from the central Red Sea. Deep Sea Research Part I: Oceanographic Research Papers 41, 485-510.

Jonkers, L., Reynolds, C.E., Richey, J., Hall, I.R., 2015. Lunar periodicity in the shell flux of planktonic foraminifera in the Gulf of Mexico. Biogeosciences 12, 3061-3070.

Marshall, B.J., Thunell, R.C., Henehan, M.J., Astor, Y., Wejnert, K.E., 2013. Planktonic foraminiferal area density as a proxy for carbonate ion concentration: A calibration study using the Cariaco Basin ocean time series. Paleoceanography 28, 363-376.

Marshall, B.J., Thunell, R.C., Spero, H.J., Henehan, M.J., Lorenzoni, L., Astor, Y., 2015. Morphometric and stable isotopic differentiation of Orbulina universa morphotypes from the Cariaco Basin, Venezuela. Marine Micropaleontology 120, 46-64.

Rigual-Hernández, A.S., Sierro, F.J., Bárcena, M.A., Flores, J.A., Heussner, S., 2012. Seasonal and interannual changes of planktic foraminiferal fluxes in the Gulf of Lions (NW Mediterranean) and their implications for paleoceanographic studies: Two 12-year sediment trap records. Deep Sea Research Part I: Oceanographic Research Papers 66, 26-40.

---

## Referee Comment (RC2) · M. F. G. Weinkauf (Referee) · 29 Jul 2016

I have been reviewing the manuscript by Mallo et al. entitled 'Low planktic foraminiferal diversity and abundance observed in a 2013 West–East Mediterranean Sea transect', and submitted to the journal Biogeosciences.

This paper studies planktonic Foraminifera, sampled with plankton nets in the upper 200 m water column during spring/summer 2013 across the entire Mediterranean Sea. It reports abundance patterns of several species across a Mediterranean transect which is characterized by large differences in physical ocean properties (e.g. temperature, salinity). It further tries to infer the influence of those environmental parameters on the abundance and shell calcification intensity of selected (abundant) species. The

study finds that the species composition changes across the Mediterranean, with *Globigerina bulloides* and *Trilobatus sacculifer* dominating in the western part, *Globorotalia inflata* in the central part, and *Globigerinoides ruber* (white)/*Globigerinoides elongatus* in the east. The species investigated for their abundance and calcification intensity show distribution and calcification patterns that differ between regions in the Mediterranean Sea, and can partly be correlated with environmental factors.

I appreciate this study for its large potential in filling in gaps in our current knowledge about species distribution in the Mediterranean and their changes both seasonally and across longer timespans by comparison of their results with earlier studies. It can also be a significant contribution to the still relatively scarce set of literature about shell calcification in planktonic Foraminifera. The sections are logically ordered, and the abstract gives a sufficient and well structured overview over the manuscript, but some information is lacking throughout the manuscript (especially the Material and Methods section). Otherwise the manuscript has an appropriate length (although the discussion is rather long I do not think it is excessive). The presented figures and tables are suitable for the most part, but I think some additional figures are needed to present key results.

Having said that, I must unfortunately conclude that in its current state I do not see this manuscript fit for publication, because it shows a multitude of problems, which I will explain in detail in the sections below. Leaving aside that the manuscript definitely needs improvement in terms of style (especially the first sections), I have severe problems with the applied methodology, which is compromising the results reached by the authors. This starts with the application of possibly an inferior method of shell calcification intensity quantification (in fact I cannot say if that is the case, because the authors never explain which method they use or make even clear that they know that different methods exist). Much more severe even is the application of a completely unfit set of analytical techniques, which are fully inappropriate for the data at hand and the questions the authors want to answer. It is furthermore hard to follow the manuscript

in parts, because it contains thematic leaps, and it is often not clear about why the analysis described in a certain passage is needed or what the authors want to show with it. The authors are furthermore missing out of the chance to perform a proper assemblage analysis and broadly footed comparison with earlier studies.

I must therefore state that possibly part of the data acquisition and certainly virtually the entire data analysis must be repeated by the authors using adequate methods, leading to my suggestion of a major revision. After this has been done, and the manuscript has been streamlined (after which in my opinion it should be reviewed again), I would very much appreciate to see this study published in Biogeosciences.

**1 General comments**

In the section below, I give detailed comments (including line numbers) about very specific issues. However, in this section I already want to summarise some major points that are more relevant for the entire manuscript than at any specific place.

1. While reading the manuscript I noticed that the writing style needs some attention. The manuscript is understandable, but there are plenty of orthographical and grammatical errors or weird phrasing throughout. Those should be dealt with (and I noted some suggestions in the detailed comments), to make the manuscript more accessible for the reader.

2. The manuscript is partly missing important information, diverts from the topic, or promises undelivered results. Some examples: Parts of the manuscript, especially the section 'Oceanographic Setting', are lacking citations of information sources. Information on data sources and methods are largely missing. The temperature and salinity might come from the mentioned CTD casts, but the carbonate saturation values most certainly not: Have they been calculated on the

basis of water samples (on board or in the lab) or calculated on the basis of database oceanographic data? Which of the several existing methods to calculate size-normalized shell weight has been used? Which software has been used for statistical analyses? All this information belongs in the far too short Material and Methods section! The reason for several analyses (e.g. the correlation between shell size and shell weight) is not properly explained, thus leaving the reader guessing why the authors deem this necessary. A comparison of assemblage data with earlier studies to study long-term trends is promised but never really delivered (not on a reasonable analytical level at least).

3. The existing images are OK, for the most part (labels might be a bit small in several of them). However, several key findings of the study are not presented in any suitable graphical manner, instead referring to figures which cannot present these data in a suitable way. Most notably amongst these, while there are several claims made about the influence of environmental factors on abundance and SNW of the species, not a single such relationship is graphically shown in a cross plot.

4. The manuscript uses several wrong species names and species concepts. The most prominent one is the unfortunate use of the terms *Globigerinoides ruber* sensu stricto and *Globigerinoides ruber* sensu lato, which are pooled, together with *Globigerinoides ruber* (pink), within the same species. This is blatantly wrong. Aurahs et al. (2011) has established that *Globigerinoides ruber* (pink), *Globigerinoides ruber* (white) (your sensu stricto), and *Globigerinoides elongatus* (your sensu lato) are distinctly different species, both biologically and in terms of morphology; and has therefore rehauled their Linnean taxonomy. Could we please all agree that 5 years after this publication we could at last all start to call them by their proper names and abandon this unfortunate sensu stricto/sensu lato distinction. It would be one thing if it would only be about names (I would still request to use up-to-date terminology, but it would be a minor mistake). Rather,

*G. elongatus* is not even the adelphotaxon to *G. ruber* (white), but is more closely related to *Globigerinoides conglobatus*. Pooling them together under the same species name thus produces a polyphylum. If you want to pool them for some purposes (which can make sense) you can call them '*G. ruber*/*G. elongatus* plexus', or something along those lines. Second, the species *Globigerinella siphonifera* is reported from the samples. However, it is not clear whether this means that only *G. siphonifera* is present, or whether this is a collective term for the entire *Globigerinella siphonifera*/*Globigerinella calida*/*Globigerinella radians* plexus (Weiner et al., 2015), within which species have not been separated by the authors. Third, but less serious because this really is only a naming issue, the former *Globigerinoides sacculifer* should be referred to as *Trilobatus sacculifer* meanwhile (Spezzaferri et al., 2015). Furthermore, in that species your 'quadrocameratus' morphotype is correctly referred to as 'quadrilobatus' morphotype to my knowledge.

5. **The most important issue** is with the statistical analytical approach. According to lines 146–147 you are using a Pearson product moment correlation to test the relative abundances and shell calcification intensities of several species against environmental parameters. This is horribly wrong on a multitude of levels, as I will summarise hereafter. For further details you may have a look at Dytham (2011), Legendre and Legendre (2012), Faraway (2006), and McDonald (2009). **I**—You assume a causal relationship between environmental factors and SNW/species abundance. Correlation analyses are not appropriate here, regression analyses with SNW/abundance as the dependent variable against the independent environmental factors must be used. Occasionally this makes only a cosmetical difference (i.e. type I linear regression vs. Pearson product moment correlation), but even then it is of methodological and implicational importance (compare Legendre and Legendre, 2012, box 10.1). In this case, however, it is even more important because of the points below. **II**—Type I regression (as well as its correlation

equivalent for that matter) is only applicable under certain circumstances, one of which is that $x$-values are measured without errors (McDonald, 2009; Dytham, 2011; Legendre and Legendre, 2012). It is therefore nearly only usable for laboratory experiments. As long as you are testing for the influence of parameters that you actually measured on board (temperature, salinity, pH), you might this this still works with a lot of good will, but I would argue that even then you have an error on those values, because you only have a snapshot image, and not a mean (let alone constant) value covering the entire life-time of your specimens. Further, I assume (you never state that) that at least part of the data you needed to calculate the carbonate system comes from averaged database data anyway!? And at least then, and in my opinion under all circumstances, you have to use robust type II or type III regression methods. **III**—You cannot simply test the same dependent variable against several independent parameters in different tests. The simple reason is that each of those test has its own type I error chance, and those are summing up until (after a sufficient number of tests) you are guaranteed to get at least one type I error in your analyses (compare Dytham, 2011; Legendre and Legendre, 2012). It is imperative that under such conditions at the very least all multiple tests (i.e. all tests for the influence of individual environmental factors on SNW or abundance per species) are corrected for this problem. Either using a correction for the family-wise error rate (e.g. Bonferroni correction), or a correction for the false discovery rate (e.g. Benjamini and Hochberg, 1995). **IV**—Making several such analyses and correcting them per species is still not the ideal solution, mainly because (as usual in marine environments) all independent variables show a large degree of multicollinearity (just have a look at your own Fig. 1). This means that such simple parameter-wise tests may detect an influence of several parameters, but only because they are highly correlated, and it is unclear which factor influences the dependent variable the most (or at all, for that matter). For the case of SNW in particular it might be much better to use an approach that can test for all independent variables at once, while reducing the influence of the

multicollinearity between different environmental factors (Dormann et al., 2013). Such methods could for instance be generalized linear models (GLM) or generalized additive models (GAM), both of which have the added benefit over multiple linear regression that they are invariant to the order in which independent variables are added to the model (compare Faraway, 2006). For relative abundances you face the additional problem, that $y$-values are not independent of each other within a sample (e.g. if *G. ruber* already represents 50 % of the assemblage, then *G. bulloides* cannot be more abundant than 50 % anymore in that same sample). While there are ways around this (most notably, using absolute abundances with an appropriate link function in a GLM, or applying any of the methods described in van den Boogart and Tolosana-Delgado (2013)) you may also prefer to analyse the assemblage data using suitable ordination techniques (compare for instance Hammer and Harper, 2006). This would have the added benefit that such ordination techniques can also be adapted to properly compare your assemblage with that of earlier studies, in this way delivering on a promise made in the introduction and never fulfilled in the manuscript.

**2   Detailed comments**

**Line 33, 'calcareous zooplankton'**: I would be very careful talking about zooplankton here. While it is true that all planktonic Foraminifera can live heterotrophic, many are also able to harbour photosymbionts.

**Line 35, 'Hembelen et al., 1989'**: Should be 'Hemleben et al., 1989'.

**Line 36, 'due to'**: Should be 'and show'.

**Lines 36–37, 'The species are adapted [...] spines and test shape.'**: They are certainly adapted to different environments, because naturally there cannot be any two species which occupy exactly the same niche, but implying such a trivial form of adaptation is far too oversimplified.

**Line 37, 'test shape'**: Should be 'shape, which are partly related to those adaptations'.

**Lines 37–39, 'The distribution of foraminifera [...] which shifts during ontogeny.'**: A citation for this statement is needed.

**Lines 42–45, 'Ecological tolerance limits [...] departure from optimum conditions (Arnold and Parker, 1999).'**: Which is basically true for every organism, so what is the point here? Plus, this is hardly the best citation for this statement. What about Bé (1977) for example?

**Lines 48–50, 'The first modern study of planktic foraminifera [...] expedition of 1947–48.'**: Was this study published? Cite a source.

**Line 54, 'at 250 m depth'**: Should be 'of the upper 250 m water column'.

**Line 57, 'that'**: Should be 'that the'.

**Lines 57–61, 'Thunell (1978) studied samples [...] inside the Mediterranean.'**: Break up this sentence.

**Line 65, 'wide'**: Should be 'large'.

**Lines 65–66, 'They concluded [...] variable foraminifera assemblages,'**: This is not entirely correct. Pujol and Vergnaud Grazzini (1995) only state that the observed assemblage patterns 'cannot be entirely explained by the general temperature and salinity differences among the different Mediterranean Basins' and are also strongly correlated to more regional hydrogeographic patterns.

**Lines 70–72, 'The calcification of foraminifera [...] (Schiebel and Hemleben, 2005).'**: Those are neither the only factors influencing shell calcification intensity in planktonic Foraminifera, nor are all of the stated relationships universally true. Compare Marshall et al. (2013, tab. 1) and Weinkauf et al. (2016, tab. 7) for a summary of this matter.

**BGD**

**Lines 70–77**: I think the cited literature for calcification studies is by far not exhaustive. What about Broecker and Clark (2001b), Barker and Elderfield (2002), de Villiers (2004), Manno et al. (2012), and Marshall et al. (2013), to name but a few.

**Line 76, 'building'**: Should be 'formation'.

**Lines 82–83, 'In addition, few size-normalized weight (SNW) studies from water column foraminifera are available in the literature.'**: Then please provide such examples here in the form of citations.

**Line 91, 'more unbreakable tests'**: Should be 'tests with thicker walls'.

**Line 92, 'empty tests are passive particles that ocean currents may displace.'** Which is perfectly true for living Foraminifera as well; hence they are plankton, not nekton.

**Lines 97–98, '(2) characterize, at the species level their ecology through their seasonal and geographical distribution and abundance by comparison with previous studies,'**: This point is not really present in the paper, at least not above a relatively comparative level. The interpretation why abundances might be different now than they were 20 years ago, and any reliable analysis and graphical presentation that shows that in the first place, is largely missing.

**Line 103, 'with a strong thermohaline and wind-driven circulation,'**: Citation needed!

**Lines 105–106, 'These basins are composed of different sub-basins due to partial isolation caused by sills that influence the water circulation, and by different water properties.'**: Citation needed!

**Lines 106–107, 'World Ocean'**: Should be 'worlds oceans'.

**Lines 107–109, 'where the nutrient-rich Atlantic surface waters [...] (evaporation exceeding precipitation).'**: Citation needed!

**Line 111, 'until the'**: Should be 'and reach as far as the'.

**Lines 113–116, 'In the eastern basin, [...] and fresher toward the western basin.'**: Citation needed!

**Lines 117–118, 'Waters returning to the Atlantic through the Strait of Gibraltar at depth are cooler and saltier than the inbound waters, and compensate for the inflow from the Atlantic.'**: Citation needed!

**Line 135, 'at 200 m depth'**: Should be 'from 200 m depth to the surface'.

**Line 141, 'counted and separated by species and size'**: Should be 'split into fractions by size'.

**Line 142, 'to determine the absolute and relative abundances'**: Should be 'and planktonic Foraminifera were counted on the species level'. Furthermore, it is not mentioned which taxonomic system is used. It is most certainly not up-to-date (compare general comments).

**Lines 144–145, 'Individuals of the same station and species within a 50 $\mu$m diameter size constraint were weighed with a Mettler Toledo XS3DU microbalance ($\pm 1\,\mu$g of error).'**: So I assume they were weighed together (single shell measurements would require a more precise balance). But were the measurements afterwards actually corrected for mean shell size per sample (MBW approach, Barker and Elderfield (2002)), or was the simple SBW approach used (Lohmann, 1995; Broecker and Clark, 2001a). The main problem is that in the latter case, Beer et al. (2010a) has shown that the SBW method is not fully effective in eliminating the shell size effect. Additionally, results cannot be independently replicated and tested when the exact methodology is not sufficiently described. Also, 'error' is the wrong term in this context, and 'nominal precision' should be used instead.

**Lines 146–149**: It is not mentioned anywhere which software was used to carry out statistical analyses.

**Lines 147–149, 'Absolute abundances [...] observed within the environmental parameters.'**: This is no valid reason at all to skip this. It can be that you are not interested in this, then state why, or that you are concerned about the validity of the results, then state why. A large difference in values does not compromise such an analysis at all if the correct techniques are applied.

**Line 171, '*Globigerinoides ruber* sensu strict (s.s.)'**: As mentioned in the general comments, this species is correctly referred to as *Globigerinoides ruber* (white). Please change in the entire manuscript.

**Line 174, 'Globigerinella siphonifera'**: Your species list contains only *Globigerinella siphonifera*, but neither *G. calida* nor *G. radians* (compare Weiner et al., 2015, and the general comments above). This could mean that either you checked and the other two species are not present at all, or you lumped the entire plexus into one category. Please explain what is the case here.

**Lines 176–178, 'In addition, a higher percentage [...] and may not be generalized.'**: Given the fact that in plankton tows you have only little control over the growth stage of your individuals, one may wonder to what degree this size trend over time may represent a reproduction event.

**Line 180, 'sample'**: Should be 'assemblage'.

**Lines 183, 187, 191, 197, '(Fig. 3; Fig 4)'**: The referred information is illustrated by neither of these figures, because Fig. 3 does not give shell sizes and Fig. 4 does not distinguish between species. Unless Fig. 3 would only represent the fraction $> 350\,\mu$m, but then this is stated nowhere in the figure caption.

**Line 192, '*Globigerinoides sacculifer*'**: This should be *Trilobatus sacculifer* (compare Spezzaferri et al., 2015, and general comments).

**Line 198, 'quadrocameratus'**: Should be 'quadrilobatus' in the entire manuscript.

**Line 218, 'A Pearson test'**: This is the wrong method for the question that should be

answered (compare general comments). By the way, even if correlation per species was the correct approach, abundance data are by default not normally distributed but follow a Poisson distribution. This rules out any parametric test in the first place, and would leave Spearman rank-order correlation or Kendall rank-order correlation as the only reasonable alternative. Compare the general comments section, however, why neither of these is appropriate here.

**Lines 220–222**: All $p$-values reported here are invalid, because they have not been corrected for multiple testing on the species level. The general comments section gives more discussion about this. Additionally, why is nothing of that presented in a graphical form?

**Lines 222–223, 'Relative abundance was selected instead of absolute abundance to avoid bias due to the big differences between stations' results in absolute abundance.'**: This approach, however, introduces new problems because now the abundances per station are not independent; and the given reason for this decision is invalid anyway. Compositional regression (van den Boogart and Tolosana-Delgado, 2013) or other adequate approaches would be needed. Compare general comments section.

**Lines 223–225, 'The remaining species [...] abundance and environmental parameters.'**: This is no reasonable explanation. The mere lack of the species at some stations would not rule out such an analysis, if there are still enough stations with values $> 0$ left.

**Lines 229–230, 'The high two-dimensional (silhouette) area-to-diameter correlation is best fitted by a power regression (Fig. S2).'**: As would be expected. But why is this important in the context of that paper? Additionally, from a purely modelling-point-of-view I might argue that the regression should be fitted so that they are forced to have a zero intercept (everything else seems wrong).

**Lines 230–235, 'Comparing the average values [...] northwestern Mediterranean**

**(Fig. S2).'**: If the idea is to compare shell sizes between different basins, then this is hardly the best method of presentation. A boxplot or barplot would be much more appropriate here. Further, it is stated nowhere which statistical techniques were used to test the shell size differences between basins. I assume an ANOVA followed by post-hoc tests, but this is explained nowhere.

**Lines 237–239, 'The diameter-to-weight relation [...] ($r^2 = 0.516$; Fig. S3).'**: If you want to imply a dependency relationship (which can make sense, depending on your intention), then it would probably be more logically to assume that weight is dependent on size, so you should exchange the axis in your Fig. S3. Otherwise, here a correlation would be more appropriate. Furthermore, the question is again what is the sense of this analysis in the context of that paper. It should be made clear for the reader, why this analysis is performed.

**Lines 239–240, '*O. universa* was finally discarded for comparisons between SNWs at different locations due to a low area–weight correlation, while data from *G. ruber* s.s. correlate well (Fig. S4a).'**: I do not really see the reason for this. 1) The weight–size relationship is not that bad ($p$-values are not given, interestingly). 2) I do not understand why the authors would insist in such a relationship to be a necessity for the interpretation of SNW. Sure, if there is no good relationship it would be difficult to predict shell size from shell weight or vice versa. But especially if you imply a relationship between calcification intensity and the environment you would expect to see deviations from this relationship. Otherwise, shell weight would be a function purely of shell size, and size-normalized shell weight would not have any value in environmental interpretations. Now, a lower $R^2$ value in *O. universa* in my opinion only means, that its shell weight is to an even lower extant controlled by shell size than it is in other species. This could mean, that *O. universa* is more susceptible to environmental protrusions in regard to its ability to control calcification, which would by some standard make it an even better proxy species. I can think of no reason why a low correlation value itself would make SNW interpretations invalid, however.

**Lines 240–242, 'The eastern Mediterranean [...] *G. ruber s.s.* (Fig. S4d-e).'**: This is again not an appropriate way of presenting those results. Use a boxplot/barplot instead.

**Lines 243–244, 'The eastern Mediterranean individuals have the lowest median SNW'**: Is this just eyeballing or has it actually been tested somehow, which regions are different and which are not concerning SNW?

**Line 245, '$\mu$g $\cdot$ $\mu$m$^{-2}$'**: So from this unit I assume the authors yet used the MBW approach, instead of SBW!? It is imperative that this is made clear in the Methods section.

**Lines 248–251, 'A Pearson correlation test [...] correlation with fluorescence ($p = 0.01$).'**: Apart from the fact that this technique is again inappropriate for the data (compare general comments and discussion for the abundance data) it is interesting that this important result is not graphically presented in any form. If such relations really exist, you should show them in the form of a figure.

**Lines 252–253, 'The Atlantic has [...] opposite trend as in *G. ruber* s.s.'**: Again, eyeballing or tested?

**Lines 256–257, '*G. bulloides* is positively correlated with pH and $[\text{CO}_3^{2-}]$ ($p = 0.05$) in the Pearson test.'**: Which is again not shown in any graphical representation.

**Line 280, 'occurs in a'**: Should be 'come from'.

**Line 280, 'season of the year'**: Should be 'seasons'.

**Line 283**: Delete 'eastern'.

**Line 284**: Delete 'both'.

**Lines 285–286, 'no significant differences are observed between samples collected during day and night.'**: Is this a subjective impression or was it tested statistically, because only in the latter case you should use 'significantly'. Further, why is this

not presented graphically somewhere?

**Lines 287, 'accounting for a single species'**: Which is blatantly wrong for virtually every perceivable species concept.

**Lines 288–289, '*G. ruber*: sensu stricto, sensu lato (containing different cryptic species; Aurahs et al., 2009a),'**: This is no up-to-date information in this regard anymore. Furthermore, the references contain only one 'Aurahs et al. (2009)', not an 'a' and 'b' version; please correct this.

**Line 292, 'with Cifelli (1974)', 'with Pujol and Grazzini (1995)'**: 'with' should in both cases be 'by'.

**Line 294, 'reach'**: Should be 'reached'.

**Lines 294–295, '*Turborotalita quinqueloba*, *Neogloboquadrina pachyderma*, and *Globorotalia truncatulinoides*.'**: Another problem for some species (certainly not *G. truncatulinoides*, but probably *T. quinqueloba* and potentially *N. pachyderma*) is that you used a 150 $\mu$m mesh size. Most studies by default use 100 $\mu$m for plankton net hauls, and part of the discrepancy you see (also in terms of general abundances) might be that you missed a lot of the small specimens. From my experience (compare Weinkauf et al. (2016) vs. Storz (2006)/Storz et al. (2009)) you can miss the majority of specimens in some species by just switching from 125 $\mu$m to 150 $\mu$m. In this regard, Pujol and Vergnaud Grazzini (1995) used 120 $\mu$m, potentially explaining a lot of your observed differences.

**Lines 297–298, '*G. sacculifer* type quadrocameratus was not found in previous studies'**: A potential problem with this statement is whether in those previous studies *T. sacculifer* has been consequently subdivided. While most studies I am aware of distinguish between the sacculifer- and trilobus-morphotypes, it is often unclear whether the quadrilobatus- (or immaturus-) morphotypes would be counted separately if discovered and truly are absent in the samples, or if they are by default pooled in with the

trilobus-morphotype.

**Lines 300–302, 'The lower absolute abundance [...] recent years'**: Yes, it could. But again, given that Pujol and Vergnaud Grazzini (1995) definitely used a finer mesh size, this could simply be the result of you missing a lot of specimens. I would therefore be very cautious with this interpretation. Berger (1969) provides equations with which observed abundances could be calibrated for different hypothetical mesh sizes, and such a correction of your data might provide a much better comparability with earlier studies.

**Line 311, '(Fig. 4).'**: Again, as this figure does not distinguish between species it cannot illustrate the trends you describe here.

**Section '5.2. Factors controlling the abundance of the main species'**: The authors try to interpret each individual species' abundance in terms of seasonality and compare it with other studies. However, it is not fully clear what the purpose of this is supposed to be. Many of the described trends are not new, and while it is always good to replicate results, this should not be the main purpose of the manuscript. Rather, the comparison of abundances with studies from several years ago, and the interpretation of potential reasons for changes (as promised in the introduction) is largely missing.

**Line 314, 'results'**: Should be 'samples'.

**Line 324, 'Both varieties *G. ruber* sensu stricto (s.s.) and sensu lato (s.l.)'**: Those are not varieties but distinctly different species, *G. ruber* (white) and *G. elongatus* respectively. Moreover, they are not even sister-taxa, but *G. elongatus* is the adelpho-taxon to *G. conglobatus*. While they have comparable environmental preferences, and might thus be pooled for such an analysis as you intend to do, they should under no circumstances treated in a way that implies they are remotely the same species.

**Lines 324–325, 'share similar habitats'**: Yet they have different environmental preferences, with *G. elongatus* living deeper (Steinke et al., 2005; Numberger et al., 2009)

and showing different seasonality (Weinkauf et al., 2016).

**Lines 331–332, 'as demonstrated by positive significant correlations with temperature in the *G. ruber* s.s. variety ($p = 0.01$).':** Not that I would oppose this interpretation, but is is yet derived from inappropriate analytical techniques.

**Line 338, 'strong positive correlation with salinity ($p = 0.01$)':** Derived from invalid methods!

**Lines 340–341, 'The findings of Watkins et al. (1996) are supported by the negative correlations of standing stocks':** Are they? If Watkins was right, would you not expect no correlation at all between nutrient availability and abundance of *G. ruber*? Rather, it seems that *G. ruber* is faring less well in regions with more nutrients (if this trend is supported by proper statistical analyses, this is). This means that higher nutrient availabilities are negative for the species, maybe because it loses its competitive advantage against other species, or the higher nutrient concentration reduces light levels, thus hampering the photosymbiont activity.

**Lines 341–342, '*G. ruber* s.s. and fluorescence data of our study ($p = 0.05$).':** Derived from invalid methods!

**Lines 352–353, 'Hydrographic conditions and consequently food availability seem to be the factors limiting more its abundance once it has reached its habitable temperature range.':** Yes, but is this not what would be expected? Liebig's law of the minimum is equally valid for protists and animals as it is for plants.

**Line 359, 'shows':** Should be ', the species shows'.

**Line 368, 'positive correlation with fluorescence ($p = 0.05$),':** Derived from invalid methods!

**Line 372, 'Raden et al., 2012':** Should be 'van Raden et al., 2012'.

**Line 377, 'specie':** Should be 'species'.

**Line 383, 'opportunistic species'**: Opportunistic species are such species which can cope with highly unstable and/or unfavourable conditions better than other species can do. They thus massively dominate environments where few other species can live, resulting in very low diversities in those environments. This is often a transitional process until the environment becomes more stable/habitable, after which the opportunistic species are replaced by a more diverse community, because in developed environments they are at a competitive disadvantage to such species. I therefore do not believe that 'opportunistic' is the correct term to describe *G. bulloides*, which is cosmopolitan and often occurs in rather diverse assemblages.

**Line 384, 'It correlates with fluorescence peaks since it feeds on phytoplankton'**: Probably correct interpretation, but derived from invalid methods!

**Line 408, 'Its negative correlation with temperature ($p = 0.01$)'**: Derived from invalid methods!

**Line 417, 'only absent from'**: Should be 'being absent from only'.

**Line 428, 'even if this is not supported by our Pearson correlation.'**: Which is an inappropriate method anyways!

**Lines 438–439, 'The size-normalized weight (SNW) of tests of both *G. ruber* s.s. and *G. bulloides* are statistically significant'**: This statement is nonsensical, a value itself cannot be significant, it can only be significant in regard to a null hypothesis. I assume you refer to the fact reported in the Results section and Suppl. Fig. 4, that size and weight are not perfectly correlated in *O. universa* (otherwise I do not even know what you want to imply). However, as already mentioned above, this is in my opinion no prerequisite for the SNW to have a meaning. This is even leaving aside, that it is never established whether this relationship is really insignificant ($p > .05$) or if the $R^2$ value is simply to small for the authors taste.

**Line 439, 'follow a systematic change from the Atlantic towards the eastern**

**Mediterranean**': This might be, but it was never properly tested or depicted graphically.

**Lines 441–443, 'In contrast, [...] environmental effects.'**: Incorrect! The strict correlation between size and weight may not exist, but this only means that especially in this species there must be other factors influencing calcification intensity.

**Section '5.3.1 Unknown control of the SNW of *O. universa*'**: OK, now your regression between shell size and shell weight makes more sense, and it would have been good to explain this in the beginning already. I do appreciate that you discuss this possibility of cryptic diversity and gametogenic calcite meddling with your data. However, what André et al (2014) detected are subtypes, they do not even rank on the species level. On that level you have also several subtypes in *G. ruber* and *G. inflata*. To be honest, it could be that the lack of strong correlation between size and weight in *O. universa* results from such an effect that the subtypes react differently. But it can still as well be, that this species simply reacts more heavily towards environmental factors concerning its calcification. I would thus not go so far as to categorically rule out that species for a calcification analysis, because you simply do not know what is the case here. It is still interesting to see SNW values for that species as well, although they might suffer from higher uncertainty. Even more so since despite a large spread, the correlation between size and weight does not seem to show bimodality (indicative for the cryptic species problem), and possibly the SNW data would not do so either. Conversely, the values seem to show a wider spread for larger shells, which can mean that gametogenic calcite is more of a problem, or simply truly that this species is more variable in calcification intensity (then presumably influenced by environmental factors). After thoroughly discussing why this might be a less reliable signal, I would therefore still want to see how SNW in *O. universa* scales with environmental factors.

**Lines 448–449, 'Weight-area relation data do not show any statistically significant systematic distribution (Fig. S4c).'**: You probably mean 'correlation', not 'distribution'.

**Lines 453–454, 'their pore-size is also affected by environmental conditions including water temperature (e.g., Bé et al., 1973).'**: This statement is critical. Bé et al. (1973) did not know about different cryptic species. It might be that pore size is indeed influenced by environmental factors across all cryptic species, but also that cryptic species prefer different water temperatures and what Bé et al. (1973) interpreted as pore size changes within the species is simply the result of different species (with inherently different pore sizes) dominating different water masses.

**Line 476, 'nutrient concentration and food availability.'**: Which is basically the same thing in the context if this study, isn't it?

**Lines 476–478, 'However, in contrast to *O. universa*, the SNW data of *G. ruber* and *G. bulloides* follow systematic distributions, which are statistically significant.'**: It is again not clear what you mean with 'distributions'. All data have a distribution, and values themselves cannot be significant or insignificant. I assume you refer to a significant correlation between size and weight in those species.

**Lines 478–480, 'High SNW in the Atlantic [...] also noticeable in Fig. S2d-e and Fig. S4d-e).'**: Those graphs are all not appropriate to show that. Rather, an actual crossplot between SNW and the individual environmental factors must be shown. Interestingly, this trend is reversed to what has been reported from the Azores Front (Weinkauf et al., 2016).

**Lines 480–482, 'At the same sites, [...] interpretation of the data (Fig. 6).'**: Which could be shown effectively by calculating and presenting the coefficient of variation at those stations. Additionally, how could this trend then be interpreted?

**Lines 485–486, 'The relationship between food availability and SNW in *G. bulloides* is opposite to that in *G. ruber* s.s. (Fig. 6)'**: A better figure is needed to illustrate this.

**Lines 488–489, 'In both species *G. ruber* s.s. and *G. bulloides* larger IQRs are**

**found toward higher absolute SNW.'**: Which is perfectly normal stochastic behaviour. This is why it is important to normalize variation for expected value by reporting the coefficient of variation instead of raw variation under such circumstances.

**Lines 490–492, 'An opposite trend in SNW [...] growth conditions.'**: I assume this refers to Beer et al. (2010b). Please cite your sources properly.

**Line 494, 'Köhler-Rink and Kühl, 2005'**: This citation is missing in the list of references.

**Lines 496–497, 'additional calcite layers might be added to the proximal text surface before reproduction, similar to the process described for *O. universa* (see above).'**: Yet to my knowledge, those two species are not known for excessive amounts of gametogenic calcite (e.g. Deuser, 1987; Hamilton et al., 2008). Also, the alternative interpretation would be that more optimal conditions trigger faster growth and earlier reproduction, resulting in a trade-off for calcification intensity of each individual chamber already during growth (i.e. before gametogenic calcite is added). Additionaly, 'text' should be 'test' (Line 496)

**Lines 505–506, 'However, the comparison might be biased by the fact that *G. ruber* s.s. and s.l. morphotypes were analyzed together in the study of de Moel et al. (2009).'**: It most certainly is. Compare Weinkauf et al. (2016).

**Lines 514–516, 'All of these [...] in an increased SNW'**: They also support the interpretation, that a multitude of factors influences shell calcification in planktonic Foraminifera.

**Line 517, 'given that carbonate chemistry does not limit calcite formation in planktic foraminifera.'**: This is a blatant misrepresentation of basically the entirety of existing literature (compare Marshall et al. (2013, tab. 1) and Weinkauf et al. (2016, tab. 7)).

**Line 522, 'reflect high'**: Should be 'show large'.

**Line 526, 'ten morphospecies in total.'**: This is wrong since at least the individual species *G. ruber* (white), *G. ruber* (pink), and *G. elongatus* have been pooled together. Furthermore, it is unclear whether *G. calida* and *G. radians* also occur and have been pooled into *G. siphonifera*.

**Line 548, 'These observations highlight the need for more interdisciplinary studies on the causes of changing foraminiferal assemblages and decreasing shell production'**: If this is supposed to hint at the promised comparison with earlier studies then I must state again that 1) since you used a larger mesh size without correcting your data for that fact you cannot compare your abundances with those of earlier studies and 2) you never presented a thorough discussion whether species compositions have been significantly changing during the last 20 years and if so, why.

**Lines 588–589**: There is no Bijma et al., 1990a, so remove the 'b' after the year.

**Lines 625–626**: Ivanova et al. (2003) is not cited anywhere in the manuscript. Remove from list of references.

**Lines 650–651**: 'Grazzini' should be 'Vergnaud Grazzini'.

**Lines 682–683**: '*Orbulina universa*' should be set in italics.

**Caption Fig. 1, '(a) Temperature (°C), (b) salinity, (c) fluorescence ($\mu g \cdot l^{-1}$), (d) pH, and (e) $[CO_3^{2-}]$ ($\mu mol \cdot kg^{-1}$)**: Information where these data come from are missing completely. Additionally, the software used for plotting (I assume Ocean Data View, Schlitzer (2014)) has not been cited. Especially Section 2, and to a lesser extant Section 3 involves a huge amount of interpolation due to the large spatial distance between measurement profiles. This makes the reconstructions very unreliable.

**Caption Fig. 2, 'First leg: 1 to 13, second leg: 14 to 22.'**: It might be nice to distinguish the cruise-tracks of the two legs by colour.

**Caption Fig. 2, 'MODIS Aqua (L2),'**: What is this? This source has not been cited in any way (published article, url, ...) and was not mentioned in the Material and Methods

section.

**Caption Fig. 2, 'from the closest day as possible'**: Which means exactly what?
1 day, 10 days, 100 days,...? Also, I would have assumed the dates given in the map
are the dates for which chlorophyll *a* data have been plotted, or what else is displayed
there?

**References**

Aurahs, R., Treis, Y., Darling, K., and Kučera, M.: A Revised Taxonomic and Phylogenetic
Concept for the Planktonic Foraminifer Species *Globigerinoides ruber* Based on Molecular
and Morphometric Evidence, Marine Micropaleontology, 79, 1–14, doi:10.1016/j.marmicro.
2010.12.001, 2011.

Barker, S. and Elderfield, H.: Foraminiferal Calcification Response to Glacial–Interglacial
Changes in Atmospheric $CO_2$, Science, 297, 833–836, doi:10.1126/science.1072815, 2002.

Bé, A. W. H.: An Ecological, Zoogeographic and Taxonomic Review of Recent Planktonic
Foraminifera, in: Oceanic Micropalaeontology, edited by Ramsay, A. T. S., 1, chap. 1, pp.
1–100, Academic Press, London and New York and San Francisco, 1977.

Beer, Ch. J., Schiebel, R., and Wilson, P. A.: Technical Note: On Methodologies for Determining
the Size-Normalised Weight of Planktic Foraminifera, Biogeosciences, 7, 2193–2198, doi:
10.5194/bg-7-2193-2010, 2010a.

Beer, Ch. J., Schiebel, R., and Wilson, P. A.: Testing Planktic Foraminiferal Shell Weight as
a Surface Water $\left[\mathrm{CO_3^{2-}}\right]$ Proxy using Plankton Net Samples, Geology, 38, 103–106, doi:
10.1130/G30150.1, 2010b.

Benjamini, Y. and Hochberg, Y.: Controlling the False Discovery Rate: A Practical and Powerful
Approach to Multiple Testing, Journal of the Royal Statistical Society, Series B: Methodolog-
ical, 57, 289–300, doi:10.2307/2346101, 1995.

Berger, W. H.: Ecologic Patterns of Living Planktonic Foraminiferal, Deep-Sea Research and
Oceanographic Abstracts, 16, 1–24, doi:10.1016/0011-7471(69)90047-3, 1969.

Broecker, W. and Clark, E.: An Evaluation of Lohmann's Foraminifera Weight Dissolution Index,
Paleoceanography, 16, 531–534, doi:10.1029/2000PA000600, 2001a.

[Figure]

Broecker, W. S. and Clark, E.: Reevaluation of the $CaCO_3$ Size Index Paleocarbonate Ion Proxy, Paleoceanography, 16, 669–671, doi:10.1029/2001PA000660, 2001b.

de Villiers, S.: Optimum Growth Conditions as Opposed to Calcite Saturation as a Control on the Calcification Rate and Shell-Weight of Marine Foraminifera, Marine Biology, 144, 45–49, doi:10.1007/s00227-003-1183-8, 2004.

Deuser, W. G.: Seasonal Variations in Isotopic Composition and Deep-Water Fluxes of the Tests of Perennially Abundant Planktonic Foraminifera of the Sargasso Sea: Results from Sediment-Trap Collections and their Paleoceanographic Significance, Journal of Foraminiferal Research, 17, 14–27, doi:10.2113/gsjfr.17.1.14, 1987.

Dormann, C. F., Elith, J., Bacher, S., Buchmann, C., Carl, G., Carré, G., Marquéz, J. R. G., Gruber, B., Lafourcade, B., Leitão, P. J., Münkemüller, T., McClean, C., Osborne, P. E., Reineking, B., Schröder, B., Skidmore, A. K., Zurell, D., and Lautenbach, S.: Collinearity: A Review of Methods to Deal with it and a Simulation Study Evaluating their Performance, Ecography, 36, 27–46, doi:10.1111/j.1600-0587.2012.07348.x, 2013.

Dytham, C.: Choosing and Using Statistics: A Biologist's Guide, Wiley–Blackwell, Oxford and Chichester and Hoboken, 3 edn., 2011.

Faraway, J. J.: Extending the Linear Model with R: Generalized Linear, Mixed Effects and Nonparametric Regression Models, Texts in Statistical Science, Chapman & Hall and CRC Press, Taylor & Francis Group, Boca Raton, 2006.

Hamilton, Ch. P., Spero, H. J., Bijma, J., and Lea, D. W.: Geochemical Investigation of Gametogenic Calcite Addition in the Planktonic Foraminifera *Orbulina universa*, Marine Micropaleontology, 68, 256–267, doi:10.1016/j.marmicro.2008.04.003, 2008.

Hammer, Ø. and Harper, D.: Paleontological Data Analysis, Blackwell Publishing, Malden and Oxford and Carlton, 2006.

Legendre, P. and Legendre, L.: Numerical Ecology, no. 24 in Developments in Environmental Modelling, Elsevier, Amsterdam and Oxford, 3 edn., 2012.

Lohmann, G. P.: A Model for Variation in the Chemistry of Planktonic Foraminifera due to Secondary Calcification Selective Dissolution, Paleoceanography, 10, 445–457, doi:10.1029/95PA00059, 1995.

Manno, C., Morata, N., and Bellerby, R.: Effect of Ocean Acidification and Temperature Increase on the Planktonic Foraminifer *Neogloboquadrina pachyderma* (sinistral), Polar Biology, 35, 1311–1319, doi:10.1007/s00300-012-1174-7, 2012.

Marshall, B. J., Thunell, R. C., Henehan, M. H., Astor, Y., and Wejnert, K. R.: Planktonic

Foraminiferal Area Density as a Proxy for Carbonate Ion Concentration: A Calibration Study using the Cariaco Basin Ocean Time Series, Paleoceanography, 28, 1–14, doi:10.1002/palo.20034, 2013.

McDonald, J. H.: Handbook of Biological Statistics, Sparky House Publishing, Baltimore, 2 edn., http://www.biostathandbook.com/, 2009.

Numberger, L., Hemleben, Ch., Hoffmann, R., Mackensen, A., Schulz, H., Wunderlich, J.-M., and Kučera, M.: Habitats, Abundance Patterns and Isotopic Signals of Morphotypes of the Planktonic Foraminifer *Globigerinoides ruber* (d'Orbigny) in the Eastern Mediterranean Sea since the Marine Isotopic Stage 12, Marine Micropaleontology, 73, 90–104, doi:10.1016/j.marmicro.2009.07.004, 2009.

Pujol, C. and Vergnaud Grazzini, C.: Distribution Patterns of Live Planktic Foraminifers as Related to Regional Hydrography and Productive Systems of the Mediterranean Sea, Marine Micropaleontology, 25, 187–217, doi:10.1016/0377-8398(95)00002-I, 1995.

Schlitzer, R.: Ocean Data View, Alfred-Wegener-Institut, http://odv.awi.de, 2014.

Spezzaferri, S., Kučera, M., Pearson, P. N., Wade, B. S., Rappo, S., Poole, Ch. R., Morard, R., and Stalder, C.: Fossil and Genetic Evidence for the Polyphyletic Nature of the Planktonic Foraminifera *"Globigerinoides"*, and Description of the New Genus *Trilobatus*, PLOS ONE, 10, e0128 108, doi:10.1371/journal.pone.0128108, http://journals.plos.org/plosone/article?id=10.1371/journal.pone.0128108, 2015.

Steinke, S., Chiu, H.-Y., Yu, P.-S., Shen, C.-C., Löwemark, L., Mii, H.-S., and Chen, M.-T.: Mg/Ca Ratios of two *Globigerinoides ruber* (white) Morphotypes: Implications for Reconstructing Past Tropical/Subtropical Surface Water Conditions, Geochemistry, Geophysics, Geosystems, 6, Q11 005, doi:10.1029/2005GC000926, 2005.

Storz, D.: Die Saisonalität planktischer Foraminiferen im Bereich einer Sinkstoffallenstation in subtropischen östlichen Nordatlantik zwischen Februar 2002 bis April 2004 [The Seasonality of Planktonic Foraminifera in Vicinity of Sediment Trap in the Suptropical North Atlantic between February 2002 and April 2004], Diploma thesis, Eberhard–Karls Universität Tübingen, Tübingen, 2006.

Storz, D., Schulz, H., Waniek, J. J., Schulz-Bull, D. E., and Kučera, M.: Seasonal and Interannual Variability of the Planktic Foraminiferal Flux in the Vicinity of the Azores Current, Deep-Sea Research, Part I: Oceanographic Research Papers, 56, 107–124, doi:10.1016/j.dsr.2008.08.009, 2009.

van den Boogart, K. G. and Tolosana-Delgado, R.: Analyzing Compositional Data with R, Use

R!, Springer-Verlag, Berlin and Heidelberg, doi:10.1007/978-3-642-36809-7, 2013.

Weiner, A. K. M., Weinkauf, M. F. G., Kurasawa, A., Darling, K. F., and Kučera, M.: Genetic and Morphometric Evidence for Parallel Evolution of the *Globigerinella calida* Morphotype, Marine Micropaleontology, 114, 19–35, doi:10.1016/j.marmicro.2014.10.003, 2015.

Weinkauf, M. F. G., Kunze, J. G., Waniek, J. J., and Kučera, M.: Seasonal Variation in Shell Calcification of Planktonic Foraminifera in the NE Atlantic Reveals Species-Specific Response to Temperature, Productivity, and Optimum Growth Conditions, PLOS ONE, 11, e148 363, doi:10.1371/journal.pone.0148363, http://journals.plos.org/plosone/article?id=10.1371/journal.pone.0148363, 2016.
* * *

---

## Referee Comment (RC3) · Anonymous Referee #3 · 3 Aug 2016

Dear Biogeosciences Editorial Board

I hereby you receive my report on the MS " Low planktic foraminiferal diversity and abundance observed in a 2013 West-East Mediterranean Sea transect" by Mallo et al.

The authors provided new information on planktonic foraminiferal abundance from the upper part of the water column (200 m) in the Mediterranean Sea during May (spring) 2013 collected with BONGO nets (mesh size 150 micron and 40 cm of diameter). The authors documented a strong difference between western and eastern Mediterranean basins, and between different Mediterranean sub-basins, in terms of abundance and diversity in planktonic foraminiferal assemblage. They document 10 species and they proposed a study on the size-normalised weight (SNW) of two species (Globigerinoides

ruber s.s. and Globigerina bulloides) and their relation with change with food availability.

The manuscript is properly constructed and it is evident that the data support the interpretation proposed in the manuscript.

I think that the authors need to stress some issues: i) the statistical analysis (in my opinion the Principal Component Analysis is the appropriate approach) carried out of the planktonic foraminiferal data [maybe including data of other authors (ie., Pujol & Vergraud-Grazzini 1995; De Castro Coppa et al 1980) to produce a complete framework of the Mediterranean]; ii) the correlation with sediment trap data (Barcena et al. 2004, Alboran Sea; Rigual-Hernández et al 2012, Gulf of Lion); iii) the comparison with data from Gulf of Naples (De Castro Coppa et al 1980), iv) the Oceanographic setting chapter (in my opinion some planktonic foraminiferal difference between different Mediterranean sub-basins could be linked to different oceanographic settings) also adding more references; v) detailed comparison between data related to the spring season (this work) with past spring seasons documented by planktonic foraminifera in the Mediterranean (living and sediment traps data); vi) the authors need to improve the figures and maybe add new ones; vii) it could be interesting to propose contouring map of the planktonic foraminiferal species viii) add a small chapter (maybe in the material and methods) concerning the criteria used to classify the planktonic foraminifera ix) I would like to suggest to add in the title of the manuscript the word SPRING.

I think that it is very important to publish these data, because of the interpretation of marine fossil archives of the Mediterranean are basically based on data (interpretation) provided by Hemleben et al., (1989) and by Pujol & Vergraud-Grazzini (1995), and it results important to improve the information on living planktonic foraminifera to better reconstruct the past climate oscillation recorded in the fossil archives. Anyway, in my opinion, the present version of the manuscript needs still important modifications concerning the presentation of data (including comparison with literature data) and discussion.

[Figure]

Minor comments: Line 34: Hemleben et al. 1989 Line 36-38: please add Reference Line 47-49: please add the write reference for the Swedish Deep-Sea expedition 1947-1948 Line 61: the reference is Pujol & Vergraud-Grazzini (1995). Please modify in the entire manuscript Line 79: please modify the reference in De Castro Coppa et al., (1980) Line 87: it is necessary to compare the acquire data also with sediment trap data of Barcena et al. (2004) from Alboran Sea and of Rigual-Hernandez et al. (2012) from Gulf of Lion Line 98: SNW; please modified in Size-Normalized Weight (SNW) Line: please add Fig. 3 in the text Line 170: Globigerinoides ruber sensu strictu (ss) is correctly referable to G. ruber white variety. Please change the name in the manuscript. Anyway, I think that the authors due to the target of the manuscript have to add a small chapter where they report exactly the criteria followed to discriminate the different planktonic foraminiferal species as well as the species included in other. Line 175-176: the data clearly document higher percentages of individuals >500 micron between Sicily channel and Ionian Sea. It is important to be more precise about the geographic position of these abundances because of changes in abundance and size could be associated to change in oceanographic setting between the different parts of the Mediterranean. Line 180-181: the authors report that the G. ruber s.s abundance is low in the southern Mediterranean (station 16-18, 15 and 9). These data are strongly in contrast with the quantitative distribution of Thunell (1978) that reports for this area values >60%. Conversely in the Tyrrhenian Sea Thunell (1978) documents a decrease in abundance values of G. ruber respect to the Ionian Sea. This contradiction need to take in account in the discussion if you want to consider, for general comparison, the data proposed by Thunell (1978). Line 182: Fig. 3 is not necessary. Should maintain only Fig. 4 Line 186: see comment proposed in Line 182 Line 190: see comment proposed in Line 182 Line 191: if the authors want to use the classification proposed by Spezzaferri et al (2015), Globigerinoides sacculifer should be Trilobatus sacculifer. Once more, it is important to have short chapter concerning the criteria adopted for classification. Line 194: fraction are ≥350 micron, please add Fig. 4 at the end of the sentence. Line 195: but is usually less abundant, please add

[Figure]

Fig. 3 at the end of the sentence. Line 196: Fig. 3 is not necessary Line 197: Globigerinoides sacculifer of the quadricameratus-type, should change in Globigerinoides quadrilobatus in the manuscript Line 201: The authors report that they grouped some stations to achieve a minimum number of planktonic foraminifera. In my opinion is not correct and I think that also the low number of planktonic foraminifera need to take in account in the interpretation. The low number is related some specific environmental setting those characterised a specific part of the Mediterranean, and you cannot lose (or overlook) this datum in this manuscript. In addition, it is not necessary to plot the % abundance of the species, because of it is not useful for comparison with data from Pujol & Vergraud-Grazzini (1995) or from De Castro Coppa et al. (1980). If you want to use the % abundance you have to covert in % also data from literature. Probably it make sense for comparison. Line 217-224: In my opinion, I consider the PCA the correct statistical approach for these data, anyway, it is important to show the complete correlation matrix where the reader can see all the obtained values for each variables. In addition, please specify the software used of statistical analysis. Line 227-235: it is very hard to follow this discussion using the diagrams proposed in Fig. S2. If you want to compare the size of planktonic foraminifera between different parts of the Mediterranean, maybe the authors can chose other graphical representation. Line 261-275: I think that a table could be useful for a visual comparison between absolute abundance in the different areas Line 276-285: I think that a graphical representation is very useful to show this comparison. In addition, the authors need to take in account also the data reported in De Castro Coppa et al (1980) from Gulf of Naples that you could tentatively correlate with the station 19 in the Tyrrhenian Sea. Line 286-288: It is not correct to group these species. They are different Line 288-292: the authors compare G. sacculifer morphotype trilobus and quadricameratus (please modify in quadrilobatus) with literature data (Cifelli 1974; Pujol & Grazzini 1995 and Thunell 1978). Please be sure that in these papers are reported these species (i.e., in Thunell 1978, G. quadrilobatus is not reported). In addition, the authors have to consider also De Castro Coppa (1980). Once more, a graphic representation is useful. Line 292-294: A possible reason could

be the mesh size used in this work, even if in De Castro Coppa et al. (1980) where they used in the Gulf of Naples a mesh size of 145 micron, in May 1979, they found N. pachyderma, T. quinqueloba and G. truncatulinoides (no high number of individuals). However, I think that with this mesh size you lose small size planktonic foraminifera. Line 296-298: probably Globigerinoides sacculifer type quadrocameratus (quadriloba-tus) is not reported in the previous literature because of it was included in G. trilobus or G. sacculifer. I would suggest a graphical comparison between literature data concerning G. sacculifer and G. trilobus (Cifelli 1974, Thunell 1978, De Castro Coppa et al 1980 and Pujol & Grazzini 1995) and a group G. quadrilobatus of your data (where you can include sacculifer, sacculifer trilobus-type and quadrocameratus-type). Maybe it make sense. You can try. Line 298: I think that the authors can refer to a paper spanning a more recent time interval than the Eocene. In particular, it is necessary to select a paper where G. sacculifer type quadrocametarus (quadrilobatus) is present. Line 302: the reference is Cossarini et al. (2015). Please modify Line 323-324: they are two different species and not varieties and they have different environmental preferences. Line 344-346: please add a reference Line 350: the authors can report as reference also Rigual-Hernandez et al (2011) where in February from sediment trap G. ruber pink is not present. Line 365-367: I think that is necessary to report also the data from De Catro Coppa et al. (1980) where G. inflata is documented in May 1979. Line 370-371: these data are opposite to data reported in Barcena et al. (2004) for sediment trap in the Alboran Sea, where in spring season G. bulloides is more abundant than G. inflata. Can the authors try to explain this discrepancy? Line 371: is van Raden et al. (2011) Line 385-386: see comment reported in Line 370-371 Line 391-392: data from sediment trap (Gulf of Lion) of Rigual-Hernandez et al (2011) report a decrease in abundance of G. inflata respect to G. bulloides during May, while in April these species strongly reduce the difference in abundance. Line 394-395: add Fig. 3 at the end of the sentence. Line 416-417: the quantitative data of O. universa seem to suggest a strong decrease in abundance towards eastern Mediterranean and two possible decreasing trends, one versus the Gulf of Lion and the second one from Balearic versus Alboran

Sea. Can suggest these trends a possible explanation? Line 493: Kohler-Rink and Kuhl 2005 is missing in the references

Reference comments: Please add: Bárcena, M.A., Flores, J.A., Sierro, F.J., Pérez-Folgado, M., Fabres, J., Calafat, A., Canals, M., 2004. Planktonic response to main oceanographic changes in the Alboran Sea (Western Mediterranean) as documented in sediment traps and surface sediments. Marine Micropaleontology 53, 423-445.

Rigual-Hernández, A.S., Sierro, F.J., Bárcena, M.A., Flores, J.A., Heussner, S., 2012. Seasonal and interannual changes of planktic foraminiferal fluxes in the Gulf of Lions (NW Mediterranean) and their implications for paleoceanographic studies: Two 12-year sediment trap records. Deep Sea Research Part I: Oceanographic Research Papers 66, 26-40.

Modify Coppa et al. (1980) in De Castro Coppa et al. (1980).

Line 578-579: the reference is Bijma te al 1990. Please modify Line 615-616: this reference (Ivanov ate al. 203) in missing in the manuscript

Figure comments: Fig.1: the numbers are too small it is very hard to read. Please increase the size. If the station 8 was not sampled for planktonic foraminifera, please remove it from the Mediterranean location map. Fig. 2: In my opinion it is necessary to add close to the number of the station also the geographic location (i.e, 1-Atlantic or Gulf of Cadiz; 2 - Gibraltar; 3- Alboran Sea etc...). In addition, it is necessary to follow the same direction for the position of the columns (i.e., W versus E), so that for Fig. 3b the correct sequence is: 22, 20, 21, 19. The same modification you have to make for the other transect 17, 16, 16-18, 15, 14. Fig.4: see comments reported for Fig. 3

Appendix A: modify quadrocameratus-type in quadrilobatus, and G. ruber s.s. with G. ruber white or G. ruber alba

---

## Author Comment (AC1) · 14 Oct 2016

We appreciate the overall positive referee remarks and acknowledge the detailed and constructive comments that greatly helped to clarify a number of points and to improve the manuscript.

Below are our detailed responses to the referee's comments, including expected modifications of the manuscript:

Major Comments

REFEREE#1, COMMENT: Unclear methodology for SNW: what diameter was measured? Neither ruber or bulloides approximates a sphere, so this is important to men-
tion.

REPLY: In order to avoid any misunderstanding on the terminology "Size-Normalized Weight" (SNW), we agree to change SNW to "Density Area" (DA) in the revised manuscript. The latter denomination is less confusing and in agreement with previous work (Marshall et al., 2013). To obtain the foraminiferal DA, the 'diameter' measured was the longest straight line possible for each specimen. All unbroken individuals were analyzed for their maximum diameter and weight. DA determined from maximum diameter is assumed a solid measure to produce statistically significant data (Marshall et al., 2013; Marshall et al., 2015). Broken individuals were not analyzed for their SNW. To avoid confusion and potential misunderstanding, we will change in the manuscript the word "diameter" to the more precise terminology: "long axis".

REFEREE#1, COMMENT: It is unclear to me if the same specimens were used to measure diameter + area and weight, please explain. And I also understand – but I am not sure - that at each station only specimens were measured within a certain 50 m size range, is that right? If so, could that not bias the trends because samples weren't selected randomly and trends in shell weight may be affected by trends in shell size?

REPLY: The original plankton net used for the sampling had a mesh size of 150 $\mu$m, and the foraminifera $\leq$ 150 $\mu$m, including the ones with tests partially broken were discarded. All the weighed individuals were measured (area and long axis). The individuals for which the long axis and area were measured but were not able to be weighed were extrapolated from their diameter size (Blue stars in Fig. S4 of the original manuscript).

We will update the 'Material and Methods' section to clarify the methodology. The two paragraphs between lines 139-155 will be replaced by a more in-depth explanation of the methodology used. One part of that replacement clarifies that question:

"For the DA study, we selected 3 main species: G. ruber, G. bulloides and O. universa. All the specimens of these 3 species were photographed with a Canon EOS 650 D

camera device attached to a Leica Z16 AP0 microscope to measure their long axis and silhouette area using the software ImageJ (Schneider et al., 2012). For each station and each of the 3 selected species, the individuals were weighed together by triplicate with a Mettler Toledo XS3DU microbalance ($\pm 1$ $\mu$g of nominal precision) within 50 $\mu$m size fraction increments (150-200 $\mu$m, 200-250 $\mu$m, etc.). Cytoplasm-filled or empty dry-weighed foraminifera tests were weighed together since dry cytoplasm has no statistically significant effect on the weight of tests >150 $\mu$m (Schiebel et al., 2007). Specimens containing notable organic matter attached to the test were discarded. The maximum number of individuals weighed together was 5; in some stations individuals were measured individually as no more specimens were available. In all the cases the mean weight per specimen of the three weighings was applied. The silhouette area obtained was then used to obtain the DA measurements (as is also done in Marshall et al., 2013; Marshall et al. 2015)."

REFEREE#1, COMMENT: Analysis of species assemblages: I have a major problem with the fact that to analyse the species distribution the authors use relative abundances. This does not make sense if one looks at individual species (closed sum effect: the % of species A will because % B changes; see for instance station 10 and 13 where the absolute abundance of G. ruber ss is similar, yet the relative very different) and if one wants to investigate species assemblage variability other techniques (PCA, cluster analysis) that look at the entire assemblage are more appropriate. I don't see what bias large variability in absolute abundance could cause (L222), why would this be bias? The authors should decide what they want to do: investigate the assemblage and use a different technique or investigate individual species abundance and use absolute abundances. The discussion and conclusions will then need to be rewritten.

REPLY: Relative abundances are grouped to see which species dominate in each geographic region of the Mediterranean. There exists high variability in the sample size along the stations; we consider relative abundance a valuable data source to understand better the ecology and distribution of the different species. Also our relative abundance groupings were estimated to allow the comparison with previous studies in the Mediterranean using relative abundances in a sub-basin/regional location level of comparison (Cifelli, 1974; Thunell, 1978; Pujol & Grazzini, 1998 (in text, not in figures)). Absolute abundance data is also provided and used in the results and discussion sections.

We will change our Pearson test analysis for a PCA. For the analysis we compare the PCA factors with absolute abundance and DA, which will be treated in the results and discussion section, leaving the species assemblage only for comparison with previous literature. A new methodology chapter is included:

"3.3. Statistical methods

We performed a principal component analysis (PCA; Varimax rotation) using SPSS Statistic 23 software. The PCA was performed on the environmental parameters: temperature, salinity, oxygen, fluorescence, NO3, PO4, pH, pCO2, and [CO3-2], of every station. Two components, which together explain 77 % of the total variance, where obtained (REV Fig. 7), the first one (Factor 1) reflects the west-east Mediterranean gradient of temperature and salinity in opposition with the quantity of nutrients available. Factor 2 reflects the gradient in seawater carbonate chemistry. Then, absolute abundance for the main species and all the foraminifera overall plus the DA of the 3 selected species were plotted against the two factors (REV Fig. 7)."

REFEREE#1, COMMENT: SNW regressions: first of all, the rationale behind the regressions isn't clear to me. Why investigate the relation between area and diameter?

REPLY: The relation between area and long axis in the three selected main species allows detection of any anomaly or changes in their growth pattern. We will add the following text in the paragraph of lines 228-236 to clarify Fig. S2 of the original manuscript:

"...The high two-dimensional (silhouette) area-to-long axis correlation is best fitted by

a power regression (Fig. S2). The same growth pattern can be seen in G. ruber s.s., G. bulloides, and O. universa with that correlation, represented graphically in the shape of a power function (Fig. S2). They grow slightly faster when they are smaller (steepest in the lower left part of the regression line) and slightly slower when they are bigger (less steep in the upper right part of the regression line; Fig. S2). Comparing the average values from different locations sampled within the Mediterranean..."

REFEREE#1, COMMENT: Why is diameter of interest if one normalises to area in SNW?

REPLY: It is important to know both the foraminiferal size and the area in order to detect changes in their ratio and in their calcification pattern. See the previous answer. Also, the data on the long axis-weight makes possible the comparison with previous studies (see Bijma et al., 2002; Lombard et al., 2010; de Moel et al., 2009; Aldridge et al., 2012; Schiebel et al., 2007).

REFEREE#1, COMMENT: It is entirely unclear to me what we learn from this and hence how to these analyses can be used to exclude O. universa from further analysis. This really needs more explanation.

REPLY: High variability in long axis-area and long axis-weight correlation was detected for O. universa; this variability was also present within stations. Making a SNW study of O. universa leads us to no trend (REV Fig. 1). No specific cause for variable density is recognizable, as we have no clear DA differences between the different geographic locations.

REFEREE#1, COMMENT: Moreover, in this respect it may also be better to use the term area density (e.g. Marshall et al., 2013) rather than SNW to distinguish from sieve-based size measurements.

REPLY: We totally agree. We change all our "Size-Normalized Weight" terminology to Density Area (DA) in the revised manuscript, based on Marshall et al. (2013), and

Marshall et al. (2015). It will also be stated in the methodology section.

REFEREE#1, COMMENT: Then on to the actual regressions. What is the rationale/bio-physical reason that area and diameter should be linearly related in log-log space? First of all, this power regression implies that neither area nor diameter can actually be zero, which cannot be correct, the regression line should go through the origin. Secondly, why wouldn't a simple power regression model suffice (I'd expect the equation for O. universa to be close to pi*r$^2$). Something similar holds for the regression of area and weight and area and diameter (again why diameter?). Why, in the case for area, a different model for each species, are there any reasons for these differences and for the choice of any model in particular? The problem with the present equations is immediately clear when looking at Fig S4c: the predicted weight of shells with an area of 10$^5$ m$^2$ is 0 g, which is physically impossible. Since weight is linearly related to volume (through density) wouldn't one expect y to be related to x$^3$?

REPLY: Area and long axis are linearly related in log-log space as is the best fit found, and the shape of the regression highly coincides with the natural shape results (r2 = 0.975, 0.962 and 0.921 for O. universa, G. ruber s.s. and G. bulloides respectively). Size and mass of foraminifers relationship does not start at the origin. The proloculus of planktic foraminifera measures between 15-30 $\mu$m in average, and has a certain calcite mass, which has so far not been determined (see Hemleben et al., 1989). We will use the power fit in the three species treated in Fig. S4 of the original manuscript for consistency reasons.

REFEREE#1, COMMENT: Moreover, and this applies to both the analysis of the species abundance and the SNW, looking at Fig.1 it appears that with the exception of fluorescence, all parameters are strongly correlated; so how do the authors determine which of these parameters is really important for the prediction of the species assemblage or SNW? The actual correlations between water column characteristics and foraminifera abundance or SNW are not shown, yet this is the most important of the study. This should be amended and the predictive power of the proposed models

should be shown.

REPLY: We agree that proper statistical analysis should be conducted on our data set. This is why in the revised version we will include a principal component analysis (PCA) performed on the environmental parameters. Note that new environmental parameters will be added: the nutrients ($NO_3$ and $PO_4$), the oxygen concentrations and the $pCO_2$. The results of the PCA show that 2 factors explain about 77% of the total variance in the environmental parameters. The 1st factor exhibited positive loadings on the nutrients and the fluorescence and negative loadings on temperature and salinity (and to a lesser degree on carbonate ion concentrations). This factor explains 56.99% of the total variance and represents the strong west-east gradient characterizing the Mediterranean Sea as the water becomes warmer, saltier and more oligotrophic eastwards. The second factor explains about 20.02% of the total variance and is characterized by positive loadings on pH and oxygen concentrations (and to a lesser degree on carbonate ion concentrations) and a negative loading on the $pCO_2$. It is interpreted as the variations of the carbonate system properties in the Mediterranean Sea with more acidic conditions in the western basin compared to the eastern basin. The sample scores on the 2 first factors with overlay of absolute abundances of foraminifera species (G. ruber (white), G. bulloides, G. inflata, O. universa and T. sacculifer (without sac)) and density area (G. ruber (white), G. bulloides and O. universa) are presented and discussed in the revised manuscript.

REFEREE#1, COMMENT: Influence of wall thickness on shell weight: the authors mention this briefly in the discussion about O. universa. I'm surprised that the study by Marshall et al (2015) on exactly the same topic is not mentioned.

REPLY: We appreciate that reference. We modified text from the manuscript adding Marshall et al., 2015 reference: From line 452: "...Whereas the various types of O. universa differ in the size of pores (de Vargas et al., 1999; Morard et al., 2009; Marshall et al., 2015), their pore-size is also affected by environmental conditions including water temperature (e.g., Bé et al., 1973). Likewise, thickness of the test wall has been

described to vary between types (de Vargas et al., 1999; Morard et al., 2009; Marshall et al., 2015), and is as well affected by..." From line 468: "... (Lombard et al., 2010), may be caused either by differences in genotypes or environmental conditions, or both. Thinner walls overall in our specimens with respect to the mentioned studies might be a cause (Marshall et al., 2015). In our samples from the Mediterranean..." And a new paragraph addition after line 460 comparing our area-weight results with the ones in Marshall et al. (2015): "The O. universa weight-area results of our study are compared with Marshall et al. (2015) from Cariaco Basin sediment trap specimens, which differentiated O. universa in Type I (Mthick) and Type III (Mthin), suggesting thinner test walls in the latter. In the area range of $3 \cdot 105 - 4 \cdot 105 \; \mu m2$, our weight results coincide with the expected Mediterranean Type III variety (Fig. S4c; Marshall et al., 2015), but at $2 \cdot 105 - 2.5 \cdot 105$ we see a mix of both types until at $1.5 \cdot 105$ type I coincide more with our results (Fig. S4c; Marshall et al., 2015). We suggest that different sub-types of the Mediterranean O. universa variety coexist in the Mediterranean with differences in the wall thickness."

REFEREE#1, COMMENT: The conclusion that SNW and therefore calcite formation is not limited by carbonate chemistry in my opinion not supported by the data. Both pH and [CO32-] are high in the Med, so how can one exclude the possibility that both parameters limit calcification? If anything, but the authors need to firmly demonstrate this, it may be that at high pH and [CO32-] other parameters may be more important. But see also the comment above about the fact that in seawater everything appears correlated with everything making it very difficult to isolate the influence of a single parameter. Fig. 6 is not very revealing in this respect: it shows a spatial trend (based on unclear grouping of the data) that may or may not be statistically significant and that may or may not be related to carbonate chemistry or food availability. The authors have a unique dataset including ancillary data and could do better to explain the variability in SNW.

REPLY: We will clarify this point raised by the reviewer. In fact the overall conclusion of the paper is not that seawater carbonate chemistry cannot be a key driver for foraminifera calcification. The results of this study are related to the modern Mediterranean conditions where pH and [CO32-] are relatively high, well above the carbonate saturation, compared to the critical values tested in ocean acidification experiments and other oceanographic settings. The pH in the upper 200 meters is ranging from 8.047 (St.1) to 8.126 (St.20) and the [CO32-] 178.88 $\mu$mol Kg-1(St.1) to 243.560 $\mu$mol Kg-1 (St.11). The Mediterranean Sea is an oligotrophic to ultra-oligotrophic environment having a strong physical and biogeochemical gradient from the Atlantic to the Eastern Mediterranean (Fig. 1 of the original manuscript; Fig. 2 of the original manuscript; MEDAR: http://modb.oce.ulg.ac.be/backup/medar/medar_med_phph_spring.html; Touratier et al., 2012: http://images.slideplayer.com/31/9579232/slides/slide_2.jpg). A main point of the paper is to show that since the seawater carbonate saturation at the studied sites is negligible compared to other oceanic regions, the effect of parameters other than carbonate saturation could be detected as observed in other studies (e.g. Weinkauf et al., 2016). We conclude that planktic foraminifera calcification in the modern Mediterranean Sea is likely more affected by factors other than carbonate saturation. In oligotrophic regions, food availability can be critical for the fitness and growth conditions since there is the hypothesis that food availability can free more energy for calcification (Beer et al., 2010; de Villiers et al., 2004; Horigome et al., 2012).

G. ruber (white) is dominant in the eastern basin, whereas G. bulloides show its dominance in the western basin, accentuating more the differences in food availability for both species. Our conclusions also might work in similar highly oligotrophic areas,. Our conclusions do not exclude that in a future with the ongoing accelerating emission of anthropogenic carbon and its uptake by the Mediterranean sea surface, carbonate chemistry will have a major effect on the SNW of planktic foraminifera, even if this is of relatively low influence today.

Figure 6 (of the original manuscript) grouping was set by location proximity in which

foraminiferal assemblages were similar, also, the grouping was done in order to achieve a minimum number of foraminifera (in Fig. 6 (of the original manuscript): $\geq$ 8 tests). We also notice that each grouping also has similar water mass characteristics. It is important to note that we work with small quantities of individuals (9 groupings of 13 in Fig. 6 (of the original manuscript) does not exceed 20 individuals) that come from a single collection in May. Further conclusions could be taken from further data availability (e.g. at different seasons).

REFEREE#1, COMMENT: Lunar and seasonal abundance variations, diel migration: the possible effect of a lunar reproductive cycle on the abundance should be mentioned in the introduction and discussed (Bijma and Hemleben, 1994; Jonkers et al., 2015). And even though the authors discuss the effect of seasonality in the discussion, it would be good to mention it also in the introduction. There at least two long-term sediment trap studies from the Western Med that could be used to place these new observations in perspective (Bárcena et al., 2004; Rigual-Hernández et al., 2012). Moreover, the authors also allude to diel migration, yet don't really make anything out of this (I wonder if it's possible with nets that integrate over the upper 200 m).

REPLY: We are aware that lunar cycle can influence the distribution of foraminifera. However, in our study the lunar day influence on the total absolute abundances (REV Fig. 2) was negligible. In fact no significant correlation was detected with our results and we decided that for this study this topic was not presented in the introduction.

The abundance distribution affected by seasonal variations will be mentioned in the introduction in the following way (ending of L33-45 paragraph):

"The abundance distribution of foraminifera is also affected by a predictable and distinct seasonal cycle for each species driven by the food source content in the watermass (Hemleben, 1989; Bé and Tolderlund, 1971; for Mediterranean examples see e.g.: Pujol and Vergraud-Grazzini, 1995; de Castro Coppa et al., 1980; Bárcena et al., 2004; Rigual-Hernández et al., 2012)."

The suggested references were added and discussed in the revised version, as well as Castro Coppa et al. (1980) and Hernández-Almeida et al. (2011). For example we changed some parts of the discussion as follows: "Despite no new plankton tow study covering the Mediterranean, three regional studies based on sediment traps were realized in the Alboran Sea (Bárcena et al., 2004; Hernández-Almeida et al., 2011) and the Gulf of Lions (Rigual-Hernández et al., 2012). The one year time series of the Alboran Sea sediment traps (July 1997 – May 1998) showed big differences in the main species relative abundances and daily fluxes through the different seasons, driven by food availability (related with water mixing/stratification periods) and temperature (Bárcena et al., 2004; Hernández-Almeida et al., 2011). The 12-year sediment trap records at Gulf of Lions (October 1993 – January 2006) showed a big seasonal pattern of the species, being more than 80% of the data from winter and spring in correlation with the nutrient supply and mixed water column (Rigual-Hernández et al., 2012)." "Comparisons are made with older similar studies from Pujol and Vergraud-Grazzini (1995), Cifelli (1974), de Castro Coppa et al. (1980); Bárcena et al. (2004), Hernández-Almeida et al. (2011), Rigual-Hernández et al. (2012), and Thunell (1978)." "The presence of G. inflata is related with cool waters and high food availability (Pujol and Vergraud-Grazzini, 1995; Rigual-Hernández et al., 2012), following high phosphate concentrations (Ottens, 1992)." "In winter, with cooler temperatures, the opposite process happens, and G. inflata becomes the dominant species in the Alboran Sea (Bárcena et al., 2004) and the southwestern basin, with high frequencies in the Strait of Sicily and just east of it." No allusion to diel migration was stated inside the manuscript.

REFEREE#1, COMMENT: Suggested trend in abundance and diversity (L28-30; 84-85; 300-305): while such a trend would be very interesting I really don't think that this is anything else than speculation. Two cruises almost 20 years apart are not enough to constrain intra and interannual variability in foram abundance and diversity, so there is simply not enough data to support this statement. There are also important sampling differences between the present study and the one by Pujol and Grazzini: 1) the maximum depth of observations (350 m vs 200 m) and the spatial distribution, which

both affect the observed abundance and diversity and a simple comparison of mean abundance or total diversity compromised. I suggest that the authors remove this speculative remark from the paper.

REPLY: The results of our study may be the first indication of a possible long term-change in diversity, which might be caused by interannual variability, and affected by climate change. Possible causes of such changes may include variability in the North Atlantic Oscillation and other regional or larger scale changes of the climate system.

We are aware of the limitations to announce that the trend in abundance and diversity is happening (few similar studies in the Mediterranean, just one with absolute abundance data, a long time span between them, sampling at different times of the year, and with different methodologies) ; we cannot prove that, we just found first insights of a possible indication. During our study the DCM is situated close to 200 m depth; between 200 and 350 m depth we do not expect to find higher numbers of foraminifera (Fig. 1c of the original manuscript), so comparison with Pujol and Vergraud-Grazzini (1995) can be made effectively.

Factors supporting that might be a future trend in reduced abundance and diversity exists too: Environmental parameters are changing in the Mediterranean (e.g. see Yáñez et al., 2010; Hassoun et al., 2015a; Hassoun et al., 2015b, Cossarini, 2015), and our absolute abundance numbers, sampled in a period of the year in which the productivity is supposed to be at its highest in the Mediterraneann (e.g. see Rigual-Hernández et al., 2012; Barcena et al., 2014), are the lowest found in the literature, even lower than recent studies in other oligotrophic areas, suggesting the Mediterranean is a critical location where possible future problems of planktonic foraminifera scarcity might occur.

For the reasons above, we think our statements are appropriate. Notice in the manuscript we are not stating that a reduced trend in abundance and diversity is a fact: see wording "could be" (L 28-30), "might have" (L 84-85), and "may indicate";

"could also imply" (L 300-305).

Minor/technical comments

COMMENT: Frequent use of locations that are not indicated on a map. Please make sure that each locality is indicated. REPLY: Text added in Fig. 1 legend (of the original manuscript): "Fig. 1. (a) Temperature (°C), (b) salinity, (c) fluorescence ($\mu$g·l-1), (d) pH, and (e) [CO3]-2 ($\mu$mol·kg-1) values of the water column of the transect. Values follow a color scale (under every graph), also values shown in the isometric lines. X axis: water depth. Y axis: longitude (degrees). Measurement locations indicated with white dots, with the coinciding stations numbered at top. The station number and the map section correlates with the map at right of this description. For station code names see Table 1. Note reversed color scale at (d) and (e). Software used: Ocean Data View (Schlitzer, 2016)" Text added in Fig. 2 legend (of the original manuscript): Fig. 2. Sampled stations with BONGO nets (dots). The numbers in the picture represent the station codes: First leg: 1 to 13, second leg: 14 to 22. For station code names see Table 1. Colour scale at right represents the values of surface chlorophyll concentration (in $\mu$g/l), retrieved from MODIS Aqua (L2), from the closest day as possible of the first leg transect.

COMMENT: Fig.1: use same x axis scale for each panel. REPLY: We agree that the 3 panels should be on the same scale. However, it appears to be barely possible as a result would be that section 2 and section 3 won't be readable (or at least the stations presented on each of these panels would be so close together so the reader can't distinguish them).

COMMENT: Fig. 3: use same y axis scale (perhaps log based?). A better representation of the data (Figs 3-6) may be to plot them on a map, or add a small map inset to the figures. REPLY: Fig. 3 of the original manuscript was modified (REV Fig. 3) and now has the same Y axis scale (see figure below). We do not consider it necessary to plot a map on Figs. 3-6, as Fig. 2 fills that purpose.

COMMENT: Fig. 4: what is 'n' below the graphs (16-18 > 16?)? REPLY: 'n' = sample size = number of individuals. It was missing at the legend, and is now included in the revised manuscript. 16: station code: Otranto Strait. 16-18: station code: Northern Ionian Sea. See Table 1.

COMMENT: L38: perhaps replace radiation with sunlight. REPLY: Changed in the revised manuscript.

COMMENT: L39: not only depth habitat, also seasonality. REPLY: We agree. Changed in the revised manuscript: "...these factors provoke an overall water depth preference, which shifts during ontogeny, and seasonal priority for each species."

COMMENT: L50: provide a reference for the expedition. REPLY: Reference added: Pettersson (1953).

COMMENT: L52: large not high abundance variations. REPLY: Changed in the revised manuscript.

COMMENT: L57-69: From this seems that the controls on the sedimentary assemblage are different from those on the water column assemblage. The main difference of course is that fact that water column observations are mere snap shots in time, whereas the sediment integrates centuries to millennia. Could the controls really be different, could the sedimentary signal integrate enough to obscure intra- and interannual variability in food availability? I'd find it interesting if the authors could spend a bit of time on this. REPLY: See the answer to your question in L88-95.

COMMENT: L66: correlation with what? The authors appear use correlation and statistically significant quite often without referring to what was tested, how and with what confidence interval. REPLY: Correlation between foraminiferal assemblage variability, and temperature and salinity gradients regarding Pujol and Vergraud-Grazzini (1995). The sentence is one of the main conclusions from the Pujol and Vergraud-Grazzini (1995) article, not from our study.

COMMENT: L70: Consider changing 'Its weight..' by 'Their shell mass...' to be more consistent. REPLY: Changed in the revised manuscript.

COMMENT: L73: What's the conclusion, implication of mentioning of the De Beer study? REPLY: Beer et al. (2010a) supports our results for G. ruber s.s. SNW being negatively correlated with [CO3-2]. Citing this article provides a point of view that shows that seawater chemistry might be independent of the shell mass of foraminifera in the Mediterranean Sea presently. We corrected the citation to Beer et al. 2010a to avoid confusion with Beer et al. 2010b (see References).

COMMENT: L78: Studies of the water column in the Mediterranean (or similar); not Mediterranean studies. REPLY: We agree: "Studies of the water column foraminifera in the Mediterranean and accurate knowledge..."

COMMENT: L88-95: while all of this holds true of course, the most important difference between the living (water) assemblages and the dead (sedimentary) is the time integrated in the sample (see also above) and (post)depositional changes to the assemblage. This needs to be mentioned. Living specimens are of course also advected; in fact, advection during life is probably more important than during sinking (simply because sinking takes less time). The study of Van Sebille is probably not very relevant for Mediterranean: with only six grid cells in the entire basin one can hardly expect that the circulation is realistically represented. REPLY: We are aware of the differences between sedimentary and water column samples, that paragraph is focused on proving that Thunell (1978) results are consistent for a comparison with water samples. We assume the reader knows the main differences between both sampling methods. Thunell (1978) states that its samples represent well the present foraminifera distribution, as they are from the very upper sediment (0-2 cm) and are recovered by trigger cores with little mixing. Note that the Mediterranean Sea is very CaCO3 saturated, with a good preservation of the samples. We propose a slight comment on the manuscript (see below).

We agree, at the horizontal scales we are working at, advection can be neglected. Moreover, live foraminifera are advected in their "own" water mass (e.g. plankton) and are indicative of their ambient seawater. Sebille et al. (2015) reference and text will be removed from the manuscript. We provided a better reference to show that the quick vertical settling provoke minimal horizontal advection of foraminifera. Modified manuscript text (L 88-95): "The study by Thunell (1978) is based on surface sediments, which can provide information, but might be biased towards faster-sinking and more hydrodynamic tests due to shorter exposition to dissolution processes (Caromel et. al., 2014; Schiebel et al., 2007), and towards tests with thicker walls that are better preserved (Thunell, 1978). The top (0-2 cm) sediment samples recovered by little disturbed and mixed trigger cores are suitable to represent modern times data according to Thunell (1978), although this sedimentary data can have a time span of some centuries and our sampling is a snap shot in time (Mortyn and Charles, 2003). In additional, empty tests are passive particles that ocean currents may displace horizontally, but that displacement is negligible due to their quick settling velocities (Caromel et al., 2014). Correlated results between plankton tows (Pujol and Vergraud-Grazzini, 1995) and surface sediments (Vergraud-Grazzini et al., 1986) at coincident places inside the Mediterranean confirm the data of Thunell (1978)." COMMENT: L112: Gulf of Lions REPLY: Changed in the revised manuscript.

COMMENT: L125: stratified? I'm not entirely sure, so please explain, but I thought that BONGO are not depth stratified. I understand that all the observations mentioned here are integrated over the upper 200 m of the water column. Please explain precisely what was done; I also assume that the statement that the samples were taken at 200 m depth (L135) is not correct. REPLY: We agree, the word "stratified" is eliminated from the manuscript. BONGO nets collect specimens from 200 m depth, and also the ones that are caught while the net is descending and ascending (above 200 m depth).

COMMENT: L154: what are unclassified specimens? REPLY: Unknowns, impossible to recognize at species level with the technology we have (most of them were juveniles not well shaped yet). We replaced "unclassified" by "unknowns" in the revised manuscript.

COMMENT: L166: remove location 1 from the list, abundances are clearly different there. Also refer to figures in this section. REPLY: True. Station 1 average removed and figure references added.

COMMENT: L210: : : : dominance 'of a single species': : : REPLY: Changed in the revised manuscript.

COMMENT: L233: how was significance determined? P-value? REPLY: We agree that proper statistical analysis should be conducted on our data set. This is why in the revised version we will include a principal component analysis (PCA) performed on the environmental parameters. See a more extended explanation of our PCA in the answer above in the major comments section.

COMMENT: L243; 252: how were the locations grouped? REPLY: By geographic proximity in which water mass properties were similar. See L 202 – 204.

COMMENT: L249: add SNW after G. rubber REPLY: Changed in the revised manuscript: adding DA instead of SNW.

COMMENT: L293-295: all these species mentioned here are winter species of which the flux happens in a single short pulse (Bárcena et al., 2004; Rigual-Hernández et al., 2012). Sampling at the end of spring could thus easily have missed them. REPLY: Reasons for those species missing will be incorporated in the manuscript as follows: "Some of the species not found reached high frequencies in the aforementioned studies: e.g., the winter species Turborotalita quinqueloba, Neogloboquadrina pachyderma, and Globorotalia truncatulinoides. The fact that these species were not sampled in the present study may be due to absence or presence at extremely low abundances of adult specimens at the sampled stations in May, as they use to have low abundances at that time according to a 12-year sediment trap record in the Gulf of Lions (Rigual-

Hernández et al., 2012). Another possibility is their presence in sizes smaller than 150 $\mu$m, escaping from our BONGO nets mesh size, a possibility potentially supported by previous Mediterranean studies with thinner mesh sizes that found these species (see Pujol and Vergraud-Grazzini, 1998, 120 $\mu$m mesh size; Rigual-Hernández et al., 2012, 63-150 $\mu$m mesh size)."

COMMENT: §5.2: please separate more clearly what are new results and what is existing knowledge. REPLY: We consider that exists an appropriate separation between our study results and existing knowledge. Latter points are always indicated by their references or named inside the text; also, many times the season of the cruise is named before the reference (e.g. "The G. ruber results confirm the findings of the June 1969 cruise of Cifelli (1974), where...", "in winter", "in late summer"). For our results words like "our data set", "our study", "in May" and references to our figures, avoid confusion. To avoid any confusion, we will add on the manuscript "this study" when we discuss our results on that section on sentences that might provoke doubt to the reader: i.e.: L366-367: "Despite having similar temperature ranges as the southwestern Mediterranean, G. inflata is absent in the Tyrrhenian Sea and the northwestern Mediterranean in this study."

COMMENT: L366: ': : : ranges as the : : :' REPLY: Changed in the revised manuscript.

COMMENT: L367: ': : : and it was also found : : :' REPLY: Changed in the revised manuscript.

COMMENT: L368: what characteristic of inflata shows this correlation? I don't follow this conclusion. REPLY: Abundance. Re-written sentence to avoid confusion and adapted to the new PCA statistical analysis performed in the revised manuscript.

COMMENT: L377: 'In accordance with: : :' and ': : : Atlantic station: : :' (there is no causal link and only one Atlantic station). REPLY: Changed in the revised manuscript.

COMMENT: L378: there isn't a station completely dominated by a single species,

please reword. REPLY: We agree. Sentence changed to: "whereas in our study it shares dominance with other species".

COMMENT: L439: what is meant with the SNW is statistically significant? What was tested, with which confidence interval, using which test? REPLY: A proper statistical analysis should be conducted on our data set. This is why in the revised version we will include a principal component analysis (PCA) performed on the environmental parameters. See a more extended explanation of our PCA in the answer above in the major comments section.

COMMENT: L445: again, see above. In addition, there seem to be two groups in O. universa (Fig.S3. 4). Have the authors looked at the spatial pattern of the SNW? REPLY: See figure of O. universa density area by location groupings attached on the 3rd question in "Major Comments": "SNW regressions". Also see answer of the 5th question in "Major Comments": "Influence of wall thickness on shell weight", where we compare O. universa weight-area relation with Marshall et al. (2015) results. It really seems two different O. universa types, further genetic research should be useful for that species inside the Mediterranean.

COMMENT: L478: the distributions are significant? Please reword. REPLY: We agree. The final part of the sentence ("…which are statistically significant.") is removed from the manuscript.

COMMENT: L478-484: is there an inverse relationship between ruber abundance (absolute) and SNW? If so, it would be good if the authors could discuss why food availability has a different effect on abundance and SNW. REPLY: Abundance of G. ruber is related to sunlight (as it is symbiont-bearing species) and food availability. Whereas shell mass is related to $[CO_3^{2-}]$ (Schiebel et al., 2004, Arabian Sea; Beer et al., 2010a). Theoretically we do not expect an inverse relationship.

COMMENT: L511: reword ': : : heavier average weight-diameter relation: : : '. REPLY: We guess that Referee #1 meant to quit the "a" before "heavier average…" here. Done.

Now the sentence remains like that: "Schiebel et al. (2007) found heavier average weight-long axis relation..."

COMMENT: L515: reproduction? Not calcification? REPLY: Changed by the word "calcification".

References

Aldridge, D., Beer, C. J., and Purdie, D. A.: Calcification in the planktonic foraminifera Globigerina bulloides linked to phosphate concentrations in surface waters of the North Atlantic Ocean, Biogeosciences, 9, 1725-1739, 2012.

André, A., Quillévéré, F., Morard, R., Ujiié, Y., Escarguel, G., de Vargas, C., de Garidel-Thoron, T., and Douady, C.J.: SSU rDNA divergence in planktonic Foraminifera: Molecular taxonomy and biogeographic implications, PLoS One, 9 (8), doi: 10.1371/journal.pone.0104641, 2014.

Bárcena, M.A., Flores, J.A., Sierro, F.J., Pérez-Folgado, M., Fabres, J., Calafat, A., and Canals, M.: Planktonic response to main oceanographic changes in the Alboran Sea (Western Mediterranean) as documented in sediment traps and surface sediments, Mar. Micropaleontol., 53, 423-445, 2004.

Bé, A.W.H., Harrison, S.M., and Lott, L.: Orbulina universa (d'Orbigny) in the Indian Ocean, Micropaleontology, 19 (2), 150–192, 1973.

Beer, C. J., Schiebel, R., and Wilson, P.A.: Technical note: On methodologies for determining the size-normalised weight of planktic foraminifera, Biogeosciences, 7, 2193-2198, 2010b.

Beer, J., Schiebel, R., and Wilson, P. A.: Testing planktic foraminiferal shell weight as a surface water [CO32-] proxy using plankton net samples, Geol. Soc. Am., 38, 103-106, 2010a.

Bijma, J., and Hemleben, C.: Population dynamics of the planktic foraminifer Globigerinoides sacculifer (Brady) from the central Red Sea, Deep Sea Res. I, 41, 485-510, 1994.

Bijma, J., Hönisch, B., and Zeebe, R. E.: Impact of the ocean carbonate chemistry on living foraminiferal shell weight: comment on "carbonate ion concentration in glacial-age deep waters of the Caribbean Sea" by W. S. Broecker and E. Clark, Geochem. Geophy. Geosy., 3 (11), 1064, doi: 10.1029/2002GC000388, 2002.

Caromel, A. G. M., Schmidt, D. N., Phillips, J. C., Rayfield, E. J.: Hydrodynamic constraints on the evolution and ecology of planktic foraminifera, Mar. Micropaleontol., 106, 69-78, 2014. Cossarini, G., Lazzari, P., and Solidoro, C.: Spatiotemporal variability of alkalinity in the Mediterranean Sea, Biogeosciences, 12, 1647-1658, 2015.

Darling, K.F. and Wade, C.M.: The genetic diversity of planktic Foraminifera and the global distribution of ribosomal RNA genotypes, Mar. Micropaleontol., 67 (3), 216–238, 2008.

de Moel, H., Ganssen, G. M., Peeters, F. J. C., Jung, S. J. A., Kroon, D., Brummer, G. J. A., and Zeebe, R. E.: Planktic foraminiferal shell thinning in the Arabian Sea due to anthropogenic ocean acidification?, Biogeosciences, 6, 1917-1925, 2009.

de Vargas, C., Norris, R., Zaninetti, L., Gibb, S. W., and Pawlowski, J.: Molecular evidence of cryptic speciation in planktonic foraminifers and their relation to oceanic provinces, Proc. Natl. Acad. Sci. USA, 96, 2864-2868, 1999.

de Villiers S. Optimum growth conditions as opposed to calcite saturation as a control on the calcification rate and shell-weight of marine Foraminifera. Mar Biol. 2004; 144: 45–49.

Hassoun, A. E. R., Gemayel, E., Krasakopoulou, E., Goyet, C., Saab, M. A., Ziveri, P., Touratier, F., Guglielmi, V., and Falco, C.: Modeling of the total alkalinity and the total inorganic carbon in the Mediterranean Sea, J. Water Res. Ocean Sci., 4 (1), 24-32, 2015a.

Hassoun, A. E. R., Guglielmi, V., Gemayel, E., Goyet, C., Saab, M. A., Giani, M., Ziveri, P., Ingrosso, G., and Touratier, M.: Is the Mediterranean Sea circulation in a steady state, J. Water Res. Ocean Sci., 4 (1), 6-17, 2015b.

Hembelen, Ch., Spindler, M., and Anderson, O.R.: Modern Planktonic Foraminifera, Springer-Verlag, New York, Berlin, Heidelberg, 363 pp., 1989.

Horigome, MT, Ziveri, P, Grelaud, M, Baumann, KH, Marino, G, Mortyn, PG, Environmental controls on the Emiliania huxleyi calcite mass, Biogeosciences, 10, 9285-9313, 2014.

Jonkers, L., Reynolds, C.E., Richey, J., and Hall, I.R.: Lunar periodicity in the shell flux of planktonic foraminifera in the Gulf of Mexico, Biogeosciences, 12, 3061-3070, 2015.

Lombard, F., Rocha, R. E., Bijma, J., and Gattuso, J. P.: Effect of carbonate ion concentration and irradiance on calcification in planktonic foraminifera, Biogeosciences, 7, 247-255, 2010.

Marshall, B. J., Thunell, R. C., Spero, H. J., Henehan, M. J., Lorenzoni, L., Astor, Y.: Morphometric and stable isotopic differentiation of Orbulina universa morphotypes from the Cariaco Basin, Venezuela, Mar. Micropaleontol., 120, 46-64, 2015.

Marshall, B.J., Thunell, R.C., Henehan, M.J., Astor, Y., and Wejnert, K.E.: Planktonic foraminiferal area density as a proxy for carbonate ion concentration: A calibration study using the Cariaco Basin ocean time series, Paleoceanography, 28, 363-376, 2013.

Morard, R., Quillévéré, F., Escarguel, G., Ujiie, Y., Garidel-Thoron, T., Norris, R. D., and de Vargas, C.: Morphological recognition of cryptic species in the planktonic foraminifer Orbulina universa, Mar. Micropaleontol., 71, 148-165, 2009.

Mortyn, P. G. and Charles, C. D.: Planktonic foraminiferal depth habitat and $\delta18O$ calibrations: plankton tow results from the Atlantic sector of the Southern Ocean, Paleoceanography, 18 (2), 1037, doi: 10.1029/2001PA000637, 2003.

Pettersson, H.: The Swedish Deep-Sea Expedition, 1947-48, Deep-Sea Res., 1, 17-24, 1953.

Pujol, C. and Vergraud-Grazzini, C.: Distribution patterns of live planktic foraminifers as related to regional hydrography and productive systems of the Mediterranean Sea, Mar. Micropaleontol., 25, 187-217, 1995.

Rigual-Hernández, A., Sierro, F. J., Bárcena, M. A., Flores, J. A., and Heussner, S.: Seasonal and interannual changes of planktic foraminiferal fluxes in the Gulf of Lions (NW Mediterranean) and their implications for paleoceanographic studies: Two 12-year sediment trap records, Deep-Sea Res I, 66, 26-40, 2012.

Schiebel, R. and Hemleben, C.: Modern planktic foraminifera, Palaeont. Z., 79 (1), 135-148, 2005.

Schiebel, R., Barker, S., Lendt, R., Thomas, H., and Bollmann, J.: Planktic foraminiferal dissolution in the twilight zone, Deep-Sea Res. II, 54, 676-686, 2007.

Schiebel, R., Zeltner, A., Treppke, U. F., Waniek, J. J., Bollmann, J., Rixen, T., Hemleben, C.: Distribution of diatoms, coccolithophores and planktic foraminifers along a trophic gradient during SW monsoon in the Arabian Sea, Mar. Micropaleontol., 51, 345-371, 2004.

Schlitzer, R.: Ocean Data View, http://odv.awi.de,2016.

Sebille, E., Scussolini, P., Durgadoo, J. V., Peeters, F. J. C., Biastoch, A., Weijer, W., Turney, C., Paris, C. B., and Zahn, R.: Ocean currents generate large footprints in marine palaeoclimate proxies, Nature Communications, 6, 6521, doi: 10.1038/ncomms7521, 2015.

Spero, H.J., Eggins, S.M., Russell, A.D., Vetter, L., Kilburn, M.R., and Hönisch, B.: Timing and mechanism for intratest Mg/Ca variability in a living planktic foraminifer, Earth Planet Sci. Lett., 409, 32–42, doi: 10.1016/j.epsl.2014.10.030, 2015.

Spero, H.J.: Ultrastructural examination of chamber morphogenesis and biomineralization in the planktonic foraminifer Orbulina universa, Mar. Biol., 99 (1), 9–20, 1988.

Thunell, R. C.: Distribution of recent planktonic foraminifera in surface sediments of the Mediterranean Sea, Mar. Micropaleontol., 3, 147-173, 1978.

Touratier, F., Guglielmi, V., Goyet, C., Prieur, L., Pujo-Pay, M., Conan, P., and Falco, C.: Distributions of the carbonate system properties, anthropogenic CO2, and acidification during the 2008 BOUM cruise (Mediterranean Sea), Biogeosciences, 9, 2709-2753, 2012.

Weinkauf, M. F. G., Kunze, J. G., Waniek, J. J., and Kucera, M.: Seasonal variation in shell calcification of planktonic foraminifera in the NE Atlantic reveals species-specific response to temperature, productivity, and optimum growth conditions, PLOS One, 11 (2), doi: 10.1371/journal.pone.0148363, 2016.

Vergraud Grazzini, C., Glaçon, C., Pierre, C., Pujol, C., and Urrutiaguer, M. J.: Foraminifères planctoniques de Méditerranée en fin d'été. Relations avec les structures hydrologiques, Mem. Soc. Geol. Ital., 36, 175-188, 1986.

Yáñez, M. V., Martínez, M. C. G., and Ruiz, F. M.: Cambio climático en el Mediterráneo español, edited by: Instituto Español de Oceanografía, Ministerio de Educación y Ciencia, Madrid, España, 2010.
* * *
[Figure]

REV Figure 7: Sample scores on the two PCA factors with (a) the loadings of the enviromental parameters on each factor, (b) with overlay of the absolute abundance values (individuals·10 m$^{-3}$) of every station of all the foraminifera sample, (c) *G. inflata*, (d) *T. sacculifer* (without sac), (e) *G. ruber* (white), (f) *G. bulloides*, and (g) *O. universa*. With overlay of the $\rho_A$ values (µg·µm$^{-2}$) of (h) *G. ruber* (white), (i) *G. bulloides*, and (j) *O. universa*. In blue color western Mediterranean stations (incl. Atlantic and Strait of Gibraltar), in red colour the eastern Mediterranean stations.

**Fig. 1.**

---

## Author Comment (AC3) · 14 Oct 2016

We appreciate the overall positive referee remarks and acknowledge the detailed and constructive comments that greatly helped to clarify a number of points and to improve the manuscript.

Below are our detailed responses to the referee's comments, including expected modifications of the manuscript.

REFEREE #3: The authors provided new information on planktonic foraminiferal abundance from the upper part of the water column (200 m) in the Mediterranean Sea during May (spring) 2013 collected with BONGO nets (mesh size 150 micron and 40 cm of

diameter). The authors documented a strong difference between western and eastern Mediterranean basins, and between different Mediterranean sub-basins, in terms of abundance and diversity in planktonic foraminiferal assemblage. They document 10 species and they proposed a study on the size-normalised weight (SNW) of two species (Globigerinoides ruber s.s. and Globigerina bulloides) and their relation with change with food availability. The manuscript is properly constructed and it is evident that the data support the interpretation proposed in the manuscript. I think that the authors need to stress some issues:

REFEREE #3 COMMENT: i) the statistical analysis (in my opinion the Principal Component Analysis is the appropriate approach) carried out of the planktonic foraminiferal data [maybe including data of other authors (ie., Pujol & Vergraud-Grazzini 1995; De Castro Coppa et al 1980) to produce a complete framework of the Mediterranean]; REPLY: We agree that proper statistical analysis should be conducted on our data set. This is why in the revised version we will include a principal component analysis performed on the environmental parameters. Note that new environmental parameters will be added: the nutrients (NO3 and PO4), the oxygen concentrations and the pCO2. The results of the PCA show that 2 factors explain about 77% of the total variance in the environmental parameters. The 1st factor exhibited positive loadings on the nutrients and the fluorescence and negative loadings on temperature and salinity (and to a lesser degree on carbonate ion concentrations). This factor explains 56.99% of the total variance and represents the strong west-east gradient characterizing the Mediterranean Sea as the water becomes warmer, saltier and more oligotrophic eastward. The second factor explains about 20.02% of the total variance and is characterized by positive loadings on pH and oxygen concentrations (and to a lesser degree on carbonate ion concentrations) and a negative loading on the pCO2. It is interpreted as the variations of the carbonate system properties in the Mediterranean Sea with more acidic conditions in the western basin compared to the eastern basin. The sample scores on the 2 first factors with overlay of absolute abundances of foraminifera species (G. ruber (white), G. bulloides, G. inflate, O. universa and T. sacculifer (without sac)) and density

area (G. ruber (white), G. bulloides and O. universa) are presented and discussed in the revised manuscript (REV Fig. 1).

REFEREE #3 COMMENT: ii) the correlation with sediment trap data (Barcena et al. 2004, Alboran Sea; Rigual-Hernández et al 2012, Gulf of Lion); REPLY: The references are added to the manuscript and compared with our data. Also, we added Hernández-Almeida et al. (2011). See minor comment about Line 87 for more details.

REFEREE #3 COMMENT: iii) the comparison with data from Gulf of Naples (De Castro Coppa et al 1980), REPLY: The data presented by de Castro Coppa et al. (1980) are compared to our results. See minor comment about Line 87 and line 180-181 for more details.

REFEREE #3 COMMENT: iv) the Oceanographic setting chapter (in my opinion some planktonic foraminiferal difference between different Mediterranean sub-basins could be linked to different oceanographic settings) also adding more references; REPLY: The oceanographic settings section has been changed in the revised manuscript as follows: "The Mediterranean Sea, with a strong thermohaline and wind-driven circulation, and a surface of approximately 2,500,000 km2, is divided into two main basins near the Strait of Sicily: the western and eastern basins (Rohling et al., 2015; Rohling et al., 2009). These basins are composed of different sub-basins due to partial isolation caused by sills that influence the water circulation, and by different water properties (Rohling et al., 2015; Rohling et al., 2009). Natural connection with the ocean is through the narrow Strait of Gibraltar, where the nutrient-rich Atlantic surface waters enter the Mediterranean and experience an eastward increase of temperature and salinity (Fig. 1) driven by insolation and evaporation, having a negative hydrological balance (evaporation exceeding precipitation; Rohling et al., 2015; Rohling et al., 2009). The Mediterranean also becomes increasingly oligotrophic towards the east (Fig. 1; Fig. 2). In addition, the incoming Atlantic waters enter the Algero–Provençal Basin and reach as far as the Tyrrhenian Sea, and contribute to deep water formation in the Gulf of Lions in cold winters (Rohling et al., 2015; Rohling et al., 2009). In the eastern

basin, two main sources of deep water formation are active mainly during winter in the Adriatic and the Aegean Seas. Cold dry winds cause evaporation and cooling forming denser and more saline water masses that sink to depth (Rohling et al., 2015; Rohling et al., 2009; Hassoun et al., 2015b). The same process is active in the Levantine basin, forming an intermediate water mass, which becomes progressively cooler and fresher toward the western basin. Some waters reach the Tyrrhenian Sea (Rohling et al., 2015; Rohling et al., 2009). Waters returning to the Atlantic through the Strait of Gibraltar at depth are cooler and saltier than the inbound waters, and compensate for the inflow from the Atlantic (Rohling et al., 2015; Rohling et al., 2009). The Mediterranean Sea has a large physicochemical gradient for such a small marginal sea (Rohling et al., 2015; Rohling et al., 2009; Fig. 1)."

REFEREE #3 COMMENT: v) detailed comparison between data related to the spring season (this work) with past spring seasons documented by planktonic foraminifera in the Mediterranean (living and sediment traps data); REPLY: New references were added to the revised manuscript and compared to our data, such as the work by Hernández-Almeida et al. (2011). In the discussion of the revised manuscript we compare as well our data with samples from late spring (Cifelli (1974), Pujol and Vergraud-Grazzini (1995)). We consider this enough detailed description for purposes of the manuscript.

REFEREE #3 COMMENT: vi) the authors need to improve the figures and maybe add new ones; REPLY: See comments in the "minor comments: figures" section.

REFEREE #3 COMMENT: vii) it could be interesting to propose contouring map of the planktonic foraminiferal species REPLY: Unfortunately, for each station and for a given species we only have one data point. Then a contour map would create excessive interpretation.

REFEREE #3 COMMENT: viii) add a small chapter (maybe in the material and methods) concerning the criteria used to classify the planktonic foraminifera REPLY: The following paragraph will be added in the Methodology section of the revised manuscript: "We classified the different foraminifera species with visual identification with the optical microscopy with the option of picking and turning the specimens to see their different sides. We followed the morphometric guidelines and genetic nomenclature proposed by Aurahs et al. (2011) for Globigerinoides ruber (white), Globigerinoides ruber (pink) and Globigerinoides elongatus. For Trilobatus sacculifer (with sac) and T. sacculifer (without sac) we used Spezzaferri et al. (2015). Hemleben et al. (1989) was used as a guide to classify Globigerinoides bulloides, Orbulina universa, Globorotalia inflata, Globorotalia menardii, and Hastigerina pelágica. Globigerinoides quadrilobatus was inferred from Papp and Schmid (1985). G.bulloides could not be differentiated from Globigerina falconensis in our samples and are treated together; the G. bulloides/G. falconensis plexus is referred as G. bulloides in our study. Globigerinella siphonifera/G. calida/ G. radians plexus (see Weiner et al., 2015) is treated as G. siphonifera in our study."

REFEREE #3 COMMENT: ix) I would like to suggest to add in the title of the manuscript the word SPRING. REPLY: We agree with the comment of the referee. The title will be changed as follow: "Low planktic foraminiferal diversity and abundance observed in a spring 2013 West-East Mediterranean Sea transect"

REFEREE #3 COMMENT: I think that it is very important to publish these data, because of the interpretation of marine fossil archives of the Mediterranean are basically based on data (interpretation) provided by Hemleben et al., (1989) and by Pujol & Vergraud-Grazzini (1995), and it results important to improve the information on living planktonic foraminifera to better reconstruct the past climate oscillation recorded in the fossil archives. Anyway, in my opinion, the present version of the manuscript needs still important modifications concerning the presentation of data (including comparison with literature data) and discussion. REPLY: We appreciate the comment and agree on the importance of publishing these kinds of observations, still relatively rare in the world of planktic foraminifera and their interpretive use for examining past environments. Some

of those landmark studies mentioned do lay the ground work for detailed ecologic descriptions of key species and their preferred environments, however they are dated by decades now and more modern observations are critical to publish in order to illustrate perhaps rapidly changing marine plankton responses to ocean climate conditions.

Minor comments: REFEREE #3 COMMENT: Line 34: Hemleben et al. 1989 REPLY: Changed in the revised manuscript.

REFEREE #3 COMMENT: Line 36-38: please add Reference REPLY: Schiebel et al. (2005) and Hembelen et al. (1989) added to the revised manuscript.

REFEREE #3 COMMENT: Line 47-49: please add the write reference for the Swedish Deep-Sea expedition 1947-1948 REPLY: Pettersson (1953) added to the revised manuscript.

REFEREE #3 COMMENT: Line 61: the reference is Pujol & Vergraud-Grazzini (1995). Please modify in the entire manuscript REPLY: Changed in the revised version.

REFEREE #3 COMMENT: Line 79: please modify the reference in De Castro Coppa et al., (1980) REPLY: Changed in the revised version.

REFEREE #3 COMMENT: Line 87: it is necessary to compare the acquire data also with sediment trap data of Barcena et al. (2004) from Alboran Sea and of Rigual-Hernandez et al. (2012) from Gulf of Lion REPLY: The suggested references were added and discussed in the revised version as well as Castro Coppa et al. (1980) and Hernández-Almeida et al. (2011). For example we changed some parts of the discussion as follows: "Despite no new plankton tow study was published covering the Mediterranean, three regional studies based on sediment traps were realized in the Alboran Sea (Bárcena et al., 2004; Hernández-Almeida et al., 2011) and the Gulf of Lions (Rigual-Hernández et al., 2012). The one year time series of the Alboran Sea sediment traps (July 1997 – May 1998) showed big differences in the main species relative abundances and daily fluxes through the different seasons, driven by food avail-

ability (related with water mixing/stratification periods) and temperature (Bárcena et al., 2004; Hernández-Almeida et al., 2011). The 12-year sediment trap records at Gulf of Lions (October 1993 – January 2006) showed a big seasonal pattern of the species, being more than 80% of the data from winter and spring in correlation with the nutrient supply and mixed water column (Rigual-Hernández et al., 2012)." "Comparisons are made with older similar studies from Pujol and Vergraud-Grazzini (1995), Cifelli (1974), de Castro Coppa et al. (1980); Bárcena et al. (2004), Hernández-Almeida et al. (2011), Rigual-Hernández et al. (2012), and Thunell (1978)." "The presence of G. inflata is related with cool waters and high food availability (Pujol and Vergraud-Grazzini, 1995; Rigual-Hernández et al., 2012), following high phosphate concentrations (Ottens, 1992)." "In winter, with cooler temperatures, the opposite process happens, and G. inflata becomes the dominant species in the Alboran Sea (Bárcena et al., 2004) and the southwestern basin, with high frequencies in the Strait of Sicily and just east of it."

REFEREE #3 COMMENT: Line 98: SNW; please modified in Size-Normalized Weight (SNW) REPLY: We decided to change "Size-Normalized Weight" to "Density Area" in the revised manuscript. The latter denomination is less confusing and in agreement with previous study (Marshall et al., 2013).

REFEREE #3 COMMENT: Line: please add Fig. 3 in the text REPLY: Referee did not indicate which line.

REFEREE #3 COMMENT: Line 170: Globigerinoides ruber sensu strictu (ss) is correctly referable to G. ruber white variety. Please change the name in the manuscript. Anyway, I think that the authors due to the target of the manuscript have to add a small chapter where they report exactly the criteria followed to discriminate the different planktonic foraminiferal species as well as the species included in other. REPLY: As mentioned above, a new paragraph was added to methodology section. Moreover, we changed the names in the revised manuscript in agreement with Aurahs et al. (2011) as follows: Globigerinoides ruber sensu stricto changed to Globigerinoides ruber (white) Globigerinoides ruber sensu lato changed to Globigerinoides elongatus

REFEREE #3 COMMENT: Line 175-176: the data clearly document higher percentages of individuals >500 micron between Sicily channel and Ionian Sea. It is important to be more precise about the geographic position of these abundances because of changes in abundance and size could be associated to change in oceanographic setting between the different parts of the Mediterranean. REPLY: We agree and the text in the revised version was changed as follows: "Overall, higher percentages of individuals >500 $\mu$m are found in the western part of the Mediterranean compared to the eastern part (Fig. 4). The highest percentages are found at the Strait of Sicily and the Northern Ionian Sea (St. 7a, 16-18; Fig. 4; Fig. S1; Appendix A)."

REFEREE #3 COMMENT: Line 180-181: the authors report that the G. ruber s.s abundance is low in the southern Mediterranean (station 16-18, 15 and 9). These data are strongly in contrast with the quantitative distribution of Thunell (1978) that reports for this area values >60%. Conversely in the Tyrrhenian Sea Thunell (1978) documents a decrease in abundance values of G. ruber respect to the Ionian Sea. This contradiction need to take in account in the discussion if you want to consider, for general comparison, the data proposed by Thunell (1978). REPLY: We agree and the text was modified according to referee's suggestion as follows: "G. ruber (white) remains scarce or absent in May in the Ionian Sea stations, increasing its abundance towards the Tyrrhenian Sea, on the other hand, in the Ionian Sea it shows values of >60% of relative abundance in Thunell (1978) surface sediments, and decreases towards the Tyrrhenian Sea. That situation could be due to more food availability in the Tyrrhenian Sea relative to the Ionian Sea during May 2013 (Fig. 1c) plus a small difference in temperature between both seas (Fig. 1a). This fact could not be the typical spring situation, as due to surface sediment evidence, the Ionian Sea is more abundant in G. ruber tests (Thunell, 1978) and May is the most productive season in foraminiferal tests (Rigual-Hernández, 2012; Bárcena et al., 2004; Hernández-Almeida et al., 2011). Also, we note that in May 1979, scarce presence of G. ruber was reported in the Bay of Naples (de Castro Coppa et al., 1980), meanwhile our study shows a 46.8 % presence in the Tyrrhenian Sea, being the main species, something only previously achieved in

August, September and December (de Castro Coppa et al., 1980), accentuating more the atypical situation of May 2013."

REFEREE #3 COMMENT: Line 182: Fig. 3 is not necessary. Should maintain only Fig. 4. REPLY: Changed in the revised manuscript.

REFEREE #3 COMMENT: Line 186: see comment proposed in Line 182. REPLY: Changed in the revised manuscript.

REFEREE #3 COMMENT: Line 190: see comment proposed in Line 182 REPLY: Changed in the revised manuscript.

REFEREE #3 COMMENT: Line 191: if the authors want to use the classification proposed by Spezzaferri et al (2015), Globigerinoides sacculifer should be Trilobatus sacculifer. Once more, it is important to have short chapter concerning the criteria adopted for classification. REPLY: We agree and use the classification proposed by Spezzaferri et al. (2015). We changed the names in the revised manuscript as follows: Globigerinoides sacculifer sacculifer type changed to Trilobatus sacculifer (with sac) Globigerinoides sacculifer trilobus type changed to Trilobatus sacculifer (without sac) Globigerinoides sacculifer quadrocameratus type changed to Globigerinoides quadrilobatus

REFEREE #3 COMMENT: Line 194: fraction are ≥350 micron, please add Fig. 4 at the end of the sentence. REPLY: Changed in the revised manuscript.

REFEREE #3 COMMENT: Line 195: but is usually less abundant, please add Fig. 3 at the end of the sentence. REPLY: Changed in the revised manuscript.

REFEREE #3 COMMENT: Line 196: Fig. 3 is not necessary REPLY: Changed in the revised manuscript.

REFEREE #3 COMMENT: Line 197: Globigerinoides sacculifer of the quadricameratus-type, should change in Globigerinoides quadrilobatus in the manuscript REPLY: We agree and changed it in the revised manuscript.

REFEREE #3 COMMENT: Line 201: The authors report that they grouped some stations to achieve a minimum number of planktonic foraminifera. In my opinion is not correct and I think that also the low number of planktonic foraminifera need to take in account in the interpretation. The low number is related some specific environmental setting those characterised a specific part of the Mediterranean, and you cannot lose (or overlook) this datum in this manuscript. In addition, it is not necessary to plot the % abundance of the species, because of it is not useful for comparison with data from Pujol & Vergraud-Grazzini (1995) or from De Castro Coppa et al. (1980). If you want to use the % abundance you have to covert in % also data from literature. Probably it make sense for comparison. REPLY: We set the minimum number of tests to 95 because our samples come from a single picking in each station (a snap shot in time), the remaining sample from the BONGO collectors come from aliquots of $\frac{1}{2}$, $\frac{1}{4}$, 1/6, and 1/8 (information not added before, now actualized in the Methodology section). That makes a difference of one individual much more significant, meaning that a different picking could change substantially the small sample results, especially the relative abundances, and having the risk of showing no realistic data. We decided to not discuss the groupings with <95 tests but include the data in Appendix A for giving further information to the readers and to help promote future studies. Moreover, relative abundance (%) data is useful for comparison with Cifelli (1974), Thunell, (1978), the regional studies of Bárcena et al. (2004), Hernández-Almeida et al. (2011) and some text information of Pujol and Vergraud-Grazzini (1995) presented in percentages. We consider useful the relative abundance data for comparison with the mentioned studies and for future researcher utility; we consider important to have the absolute values as well.

REFEREE #3 COMMENT: Line 217-224: In my opinion, I consider the PCA the correct statistical approach for these data, anyway, it is important to show the complete correlation matrix where the reader can see all the obtained values for each variables. In addition, please specify the software used of statistical analysis. REPLY: We agree and as mentioned above a PCA analysis was performed and added to the revised

manuscript.

REFEREE #3 COMMENT: Line 227-235: it is very hard to follow this discussion using the diagrams proposed in Fig. S2. If you want to compare the size of planktonic foraminifera between different parts of the Mediterranean, maybe the authors can chose other graphical representation. REPLY: We consider Fig. S2 appropriate. We decided to investigate the relation between area and long axis in the three selected main species to see their growth pattern. We clarify it in the text of the revised manuscript: "...The high two-dimensional (silhouette) area-to-long axis correlation is best fitted by a power regression (Fig. S2). The same growth pattern can be seen in G. ruber s.s., G. bulloides, and O. universa with that correlation, represented graphically in the shape of a power function (Fig. S2). They grow slightly faster when they are smaller (steepest in the lower left part of the regression line) and slightly slower when they are bigger (less steep in the upper right part of the regression line; Fig. S2). Comparing the average values from different locations sampled within the Mediterranean, G. ruber s.s. individuals from the Atlantic have the largest size followed by individuals from the Tyrrhenian Sea, and tests from east of the Strait of Sicily...."

REFEREE #3 COMMENT: Line 261-275: I think that a table could be useful for a visual comparison between absolute abundance in the different areas REPLY: We already tabulated the data by station, such that future readers and researchers can group similarly to our area grouping, or they can do it differently. If it is already grouped by area, then we short-circuit the opportunity for other grouping schemes in the future.

REFEREE #3 COMMENT: Line 276-285: I think that a graphical representation is very useful to show this comparison. In addition, the authors need to take in account also the data reported in De Castro Coppa et al (1980) from Gulf of Naples that you could tentatively correlate with the station 19 in the Tyrrhenian Sea. REPLY: We consider that a graphical representation is not indispensable for that purpose. Now we use de Castro Coppa et al. (1980) for our discussion comparison (see answers to your questions at line 87, line 180-181). In that paragraph (276-285), we compare absolute

abundance values from past studies with respect to our study. The study of de Castro Coppa et al. (1980) gives their absolute values in individuals obtained (they just give a general and approximate value of how many m3 they filter in each towing), making the results incomparable (i.e. individuals·m-3). We obtained the valuable information of the relative distribution of the foraminifera assemblage through the seasons, used in the discussion section 5.2.

REFEREE #3 COMMENT: Line 286-288: It is not correct to group these species. They are different REPLY: We agree and the text was changed as follows in the revised manuscript: "Comparing with previous studies that covered the Mediterranean, we notice that Thunell (1978) and Pujol and Vergraud-Grazzini (1995) did not find G. menardii, despite it being found in this study and Cifelli (1974), both in very low quantities. The lack of data from surface sediments and their tropical water preference suggest that is a new species in the Mediterranean (Cifelli, 1974), possibly caused by warmer conditions than in past times. The rest of the species found in our study are found in the past studies covering the Mediterranean Sea (Cifelli, 1974; Thunell, 1978; Pujol and Vergraud-Grazzini, 1995), but it remains in doubt if whether Pujol and Vergraud-Grazzini found G. falconensis and classified it as G. bulloides; or if Thunell (1978) found G. elongatus and T. sacculifer (without sac) and classified them as G. ruber and G. sacculifer. The former problem is also found in Pujol and Vergraud-Grazzini (1995). Also, it is not certain if Cifelli (1974) found G. calida and classified it as G. aequilateralis (old equivalent of G. siphonifera). For the figures in Cifelli (1974) we deduce that G. elongatus was classified as G. ruber in the study. In the same way, we do not find any evidence of finding T. sacculifer (with sac) from the Cifelli (1974) figures, but we cannot discard the possibility of it being classified as G. trilobus (T. sacculifer without sac). Finally, we do not have the evidence if Cifelli (1974) found G. ruber (pink) and classified it together with the white variety into G. ruber.

G. quadrilobatus was not found in previous studies working with plankton tows in the Mediterranean, despite its abundance in sedimentary cores (i.e. Cramp et al., 1988;

Rio et al., 1990); there exists the possibility to classify it as G. sacculifer or G. trilobus in previous studies as was suggested by Hemleben et al. (1989). Some of the species not found reached high frequencies in the aforementioned studies: e.g., the winter species Turborotalita quinqueloba, Neogloboquadrina pachyderma, and Globorotalia truncatulinoides. The fact that these species were not sampled in the present study may be caused by their absence or presence at extremely low abundances of adult specimens at the sampled stations in May, as they use to have low abundances at that time according to a 12-year sediment trap record in the Gulf of Lions (Rigual-Hernández et al., 2012). Another possibility is their presence in a size smaller than 150 $\mu$m, escaping from our BONGO nets mesh size. That possibility could be supported by the fact that previous Mediterranean studies with thinner mesh sizes found that species (see Pujol and Vergraud-Grazzini, 1998: 120 $\mu$m mesh size; Rigual-Hernández et al., 2012: 63-150 $\mu$m mesh size).

To be able do a quantitative comparison of the number of species found with previous Mediterranean studies , first, we make the following simplification: G. bulloides and G. falconensis count as one species for that comparison; the same is applied for G. siphonifera and G. calida, and G. ruber (white) and G. ruber (pink). Secondly, we made the assumption that all the doubtful species found in previous studies (see two paragraphs above) were found (e.g.: we assume that Thunell (1978) found G. elongatus and he classified it as G. ruber). After applying these conditions we arrive at a smaller number of species able to be compared. Our number of apparent species becomes 11, clearly inferior to Cifelli (1974) with 19 apparent species, and Thunell (1978) and Pujol and Vergraud-Grazzini (1995) with 17 apparent species. In station 3 of this study (Alboran Sea), we found 8 species; meanwhile the number ascends to 12 in Rigual-Hernández et al. (2012) species flux in the same month."

REFEREE #3 COMMENT: Line 288-292: the authors compare G. sacculifer morpho-type trilobus and quadricameratus (please modify in quadrilobatus) with literature data (Cifelli 1974; Pujol & Grazzini 1995 and Thunell 1978). Please be sure that in these

papers are reported these species (i.e., in Thunell 1978, G. quadrilobatus is not reported). In addition, the authors have to consider also De Castro Coppa (1980). Once more, a graphic representation is useful. REPLY: See the answer the previous comment (Line 286-288). We consider de Castro Coppa et al. (1980) for the discussion section 5.2., here the purpose of the paragraph was to compare our species number with the studies that covered the Mediterranean. Regional studies are not included in that comparison as they miss the rest of the areas of the Mediterranean. Comparison of regional studies with individual stations in our study was discarded, as results can be more biased (as it only depends on 1 station of our study instead of several ones that can reduce the bias of no collected specimens in a single plankton tow). But a comparison of our station 19 with de Castro de Coppa et al. (1980) at 200 m depth can be mentioned here: Our study yields 7 different apparent species (read the answer to your question about line 286-288) in the Tyrrhenian Sea station, meanwhile de Castro Coppa (1980) found 12 apparent species at the Bay of Naples. We consider that a graphical representation is not indispensable for that purpose.

REFEREE #3 COMMENT: Line 292-294: A possible reason could be the mesh size used in this work, even if in De Castro Coppa et al. (1980) where they used in the Gulf of Naples a mesh size of 145 micron, in May 1979, they found N.pachyderma, T. quinqueloba and G. truncatulinoides (no high number of individuals). However, I think that with this mesh size you lose small size planktonic foraminifera. REPLY: We agree. Pujol and Vergraud-Grazzini (1995) used a mesh size of 120 $\mu$m and found the three species. Rigual-Hernández et al. (2012) used a 63-150 $\mu$m size fraction and collected the three species too. We note, as well, that Cifelli (1974) with 158 $\mu$m mesh size collected specimens of G. truncatulinoides and T. quinqueloba. We modify the manuscript as follows: "Some of the species not found reached high frequencies in the aforementioned studies: e.g., the winter species Turborotalita quinqueloba, Neoglobo-quadrina pachyderma, and Globorotalia truncatulinoides. The fact that these species were not sampled in the present study may be caused by their absence or presence at extremely low abundances of adult specimens at the sampled stations in May, as

they use to have low abundances at that time according to a 12-year sediment trap record in the Gulf of Lions (Rigual-Hernández et al., 2012). Another possibility is their presence in sizes smaller than 150 $\mu$m, escaping from our BONGO nets mesh size, a possibility that could be supported by previous Mediterranean studies with thinner mesh sizes finding that species (see Pujol and Vergraud-Grazzini, 1998, 120 $\mu$m mesh size; Rigual-Hernández et al., 2012, 63-150 $\mu$m mesh size)."

REFEREE #3 COMMENT: Line 296-298: probably Globigerinoides sacculifer type quadrocameratus (quadrilobatus) is not reported in the previous literature because of it was included in G. trilobus or G. sacculifer. I would suggest a graphical comparison between literature data concerning G. sacculifer and G. trilobus (Cifelli 1974, Thunell 1978, De Castro Coppa et al 1980 and Pujol & Grazzini 1995) and a group G. quadrilobatus of your data (where you can include sacculifer, sacculifer trilobus-type and quadrocameratus-type). Maybe it make sense. You can try. REPLY: See the answer to the comment about Line 286-288. We consider that a graphical comparison is not indispensable for that purpose.

REFEREE #3 COMMENT: Line 298: I think that the authors can refer to a paper spanning a more recent time interval than the Eocene. In particular, it is necessary to select a paper where G. sacculifer type quadrocametarus (quadrilobatus) is present. REPLY: We agree and references: Cramp et al. (1988) and Rio et al. (1990) were added to the revised manuscript.

REFEREE #3 COMMENT: Line 302: the reference is Cossarini et al. (2015). Please modify REPLY: Changed in the revised manuscript.

REFEREE #3 COMMENT: Line 323-324: they are two different species and not varieties and they have different environmental preferences. Reply: We agree. This paragraph will be deleted.

REFEREE #3 COMMENT: Line 344-346: please add a reference REPLY: Pujol and Vergraud-Grazzini (1995) added to the revised manuscript.

REFEREE #3 COMMENT: Line 350: the authors can report as reference also Rigual-Hernandez et al (2011) where in February from sediment trap G. ruber pink is not present. REPLY: We agree the reference was added to the revised manuscript.

REFEREE #3 COMMENT: Line 365-367: I think that is necessary to report also the data from De Catro Coppa et al. (1980) where G. inflata is documented in May 1979. REPLY: We agree. See our answer to your question about line 87.

REFEREE #3 COMMENT: Line 370-371: these data are opposite to data reported in Barcena et al. (2004) for sediment trap in the Alboran Sea, where in spring season G. bulloides is more abundant than G. inflata. Can the authors try to explain this discrepancy? REPLY: We modify the text in the manuscript as follows: "Alboran Sea spring distribution of G. inflata, with G. bulloides as a clear secondary species, matches with other studies (Pujol and Vergraud-Grazzini, 1995; van Raden et al., 2011). Although, in May 1998, G. bulloides clearly exceeds G. inflata in abundance, but seems an exceptional year in which G. inflata productivity is unfavored by high temperature anomalies that might be influenced by the El Niño-Southern Oscillation event (Bárcena et al., 2004; Hernández-Almeida et al., 2011)."

REFEREE #3 COMMENT: Line 371: is van Raden et al. (2011) REPLY: Changed in the revised manuscript.

REFEREE #3 COMMENT: Line 385-386: see comment reported in Line 370-371 REPLY: Changed in the revised manuscript: "In April (Pujol and Vergraud-Grazzini, 1995; van Raden et al., 2011) and May, it is found to be the second most abundant species, surpassed by G. inflata, in the westernmost Alboran Sea. High temperature anomalies provoke an inverse situation, thanks to G. bulloides faster reproduction plus G. inflata being further for its optimum temperature (Bárcena et al., 2004)."

REFEREE #3 COMMENT: Line 391-392: data from sediment trap (Gulf of Lion) of Rigual-Hernandez et al (2011) reports a decrease in abundance of G. inflata respect to G. bulloides during May, while in April these species strongly reduce the difference

in abundance.

REPLY: The text here was stating that G. bulloides abundances were higher in winter than in late summer overall. The Rigual-Hernández et al. (2012) reference is now added, as it demonstrated what was stated by Pujol and Vergraud-Grazzini (1995) as well.

REFEREE #3 COMMENT: Line 394-395: add Fig. 3 at the end of the sentence. REPLY: We consider no need for placing Fig. 3 here as the last sentence speaks about Pujol & Vergraud-Grazzini (1995) and does not mention or compare with our results in May.

REFEREE #3 COMMENT: Line 416-417: the quantitative data of O. universa seem to suggest a strong decrease in abundance towards eastern Mediterranean and two possible decreasing trends, one versus the Gulf of Lion and the second one from Balearic versus Alboran Sea. Can suggest these trends a possible explanation? REPLY: We do not observe any strong decrease in absolute abundance towards the eastern Mediterranean (see Appendix A: absolute abundance, total numbers, O. universa). We think that those differences are not large enough to be certainly caused by environmental factors or ecological competition for food. Those differences can merely be coincidental since we did one plankton tow in each station, meaning a "snapshot" in time. Thus, small differences can be misinterpreted.

REFEREE #3 COMMENT: Line 493: Kohler-Rink and Kuhl 2005 is missing in the references REPLY: Changed in the revised manuscript.

Reference comments: REFEREE #3 COMMENT: Please add: Bárcena, M.A., Flores, J.A., Sierro, F.J., Pérez-Folgado, M., Fabres, J., Calafat, A., Canals, M., 2004. Planktonic response to main oceanographic changes in the Alboran Sea (Western Mediterranean) as documented in sediment traps and surface sediments. Marine Micropaleontology 53, 423-445. REPLY: Added to the revised manuscript. REFEREE #3 COMMENT: Rigual-Hernández, A.S., Sierro, F.J., Bárcena, M.A., Flores, J.A., Heussner, S.,

2012. Seasonal and interannual changes of planktic foraminiferal ïnĔĞC′ uxes in the Gulf of Lions (NW Mediterranean) and their implications for paleoceanographic studies: Two 12-year sediment trap records. Deep Sea Research Part I: Oceanographic Research Papers 66, 26-40. REPLY: Added to the revised manuscript.

REFEREE #3 COMMENT: Modify Coppa et al. (1980) in De Castro Coppa et al. (1980). REPLY: Changed to de Castro Coppa et al., (1980) in the revised manuscript.

REFEREE #3 COMMENT: Line 578-579: the reference is Bijma te al 1990. REPLY: Changed to the revised manuscript.

REFEREE #3 COMMENT: Please modify Line 615-616: this reference (Ivanov ate al. 203) in missing in the manuscript REPLY: Changed to the revised manuscript.

Figure comments:

REFEREE #3 COMMENT: Fig.1: the numbers are too small it is very hard to read. Please increase the size. If the station 8 was not sampled for planktonic foraminifera, please remove it from the Mediterranean location map. REPLY: Numbers size changed in the revised version. The station 8 is mentioned here as the values were used for the interpolation of the environmental parameters.

REFEREE #3 COMMENT: Fig. 2: In my opinion it is necessary to add close to the number of the station also the geographic location (i.e, 1-Atlantic or Gulf of Cadiz; 2 - Gibraltar; 3- Alboran Sea etc: : :). REPLY: We consider this unnecessary as the geographic location of the station codes is presented in Table 1, and the naming of all stations in the figure can be a problem for good visibility of the transect and the names themselves. We explain where to check the names of the station codes in the new legend for Figure 2:

"Fig. 2. Sampled stations with BONGO nets (dots). The numbers in the picture represent the station codes: First leg: 1 to 13, second leg: 14 to 22. For station code names see Table 1. Colour scale at right represents the values of surface chlorophyll concentration (in $\mu$g/l), retrieved from MODIS Aqua (L2), from the closest day as possible, specified in the upper part, of the first leg transect."

REFEREE #3 COMMENT: In addition, it is necessary to follow the same direction for the position of the columns (i.e., W versus E), so that for Fig. 3b the correct sequence is: 22, 20, 21, 19. The same modification you have to make for the other transect 17, 16, 16-18, 15, 14. Fig.4: see comments reported for Fig. 3 REPLY: Changed in the revised figures 3 and 4 (REV Fig.3 and REV Fig. 4).

Appendix A: REFEREE #3 COMMENT: modify quadrocameratus-type in quadrilobatus, and G. ruber s.s. with G. ruber white or G. ruber alba REPLY: Changed in the revised manuscript.

References

Aurahs, R., Treis, Y., Darling, K., Kucera, M.: A revised taxonomic and phylogenetic concept for the planktonic foraminifer species Globigerinoides ruber based on molecular and morphometric evidence, Mar. Micropaleontol., 79, 1-14, 2011. Bárcena, M.A., Flores, J.A., Sierro, F.J., Pérez-Folgado, M., Fabres, J., Calafat, A., and Canals, M.: Planktonic response to main oceanographic changes in the Alboran Sea (Western Mediterranean) as documented in sediment traps and surface sediments, Mar. Micropaleontol., 53, 423-445, 2004. Bé, A. W. H. and Tolderlund, D. S.: Distribution and ecology of living planktonic foraminifera in surface waters of the Atlantic and Indian Oceans, in: The micropaleontology of oceans, edited by: Funnel, B. M. and Riedel, W. R., Cambridge University Press, London, U.K., 105-149, 1971. Cifelli, R.: Planktonic foraminifera from the Mediterranean and adjacent Atlantic waters (Cruise 49 of the Atlantis II, 1969), J. Foramin. Res., 4, 171-183, 1974. Cramp, A., Collins, M., and West, R.: Late Pleistocene-Holocene sedimentation in the NW Aegean Sea: A palaeoclimatic palaeoceanographic reconstruction, Palaeogeogr. Palaeoclimatol. Palaeocecol., 68, 61-77, 1988. de Castro Coppa, M. G., Zei, M. M., Placella, B., Sgarella, F., and Ruggiero, E. T.: Distribuzione stagionale e verticale dei foraminiferi planctonici del Golfo di

Napoli, Boll. Soc. Natur. Napoli, 89, 1-25, 1980. Glaçon, G., Grazzini, C. V., and Sigal, J.: Premiers resultants d'une série d'observations saisonnières des foraminifères du plankton méditerranéen, in: Procceedings of the 2nd Plankton Conference, Rome, 1970, 555-581, 1971. Hassoun, A. E. R., Guglielmi, V., Gemayel, E., Goyet, C., Saab, M. A., Giani, M., Ziveri, P., Ingrosso, G., and Touratier, M.: Is the Mediterranean Sea circulation in a steady state, J. Water Res. Ocean Sci., 4 (1), 6-17, 2015b. Hembelen, Ch., Spindler, M., and Anderson, O.R.: Modern Planktonic Foraminifera, Springer-Verlag, New York, Berlin, Heidelberg, 363 pp., 1989. Hernández-Almeida, I., Bárcena, M. A., Flores, J. A., Sierro, F. J., Sanchez-Vidal, A., and Calafat, A.: Microplankton response to environmental conditions in the Alboran Sea (Western Mediterranean): One year sediment trap record, Mar. Micropaleontol., 78, 14-24, 2011. Mortyn, P. G. and Charles, C. D.: Planktonic foraminiferal depth habitat and $\delta$18O calibrations: plankton tow results from the Atlantic sector of the Southern Ocean, Paleoceanography, 18 (2), 1037, doi: 10.1029/2001PA000637, 2003. Ottens, J. J.: April and August Northeast Atlantic surface water masses reflected in planktic foraminifera, Neth. J. Sea Res., 28 (4), 261-283, 1992. Papp A., Schmid M. E.: Die fossilien Foraminiferen des tertiären Beckens von Wien, Revision der Monographic von Alcide d'Orbigny (1846). Wien: Abhandlungen der Geologischen Bundesanstalt, 1985. Pettersson, H.: The Swedish Deep-Sea Expedition, 1947-48, Deep-Sea Res., 1, 17-24, 1953. Pujol, C. and Vergraud-Grazzini, C.: Distribution patterns of live planktic foraminifers as related to regional hydrography and productive systems of the Mediterranean Sea, Mar. Micropaleontol., 25, 187-217, 1995. Rigual-Hernández, A., Sierro, F. J., Bárcena, M. A., Flores, J. A., and Heussner, S.: Seasonal and interannual changes of planktic foraminiferal fluxes in the Gulf of Lions (NW Mediterranean) and their implications for paleoceanographic studies: Two 12-year sediment trap records, Deep-Sea Res I, 66, 26-40, 2012. Rio, D., Sprovieri, R., Thunell, R., Vergnaud-Grazzini, C., and Glaçon, G.: Pliocene-Pleistocene paleoenvironmental history of the western Mediterranean: A synthesis of ODP site 653 results, Proc. Ocean Drill Prog. Sci. Results, 107, 695-704, 1990. Rohling E. J., Marino, G., and Grant, K. M.: Mediterranean climate and

oceanography, and the periodic development of anoxic events (sapropels), Earth Sci., 143, 62-97, 2015. Rohling, E., Ramadan, A., Casford, J., Hayes, A., and Hoogakker, B.: The marine environment: present and past, in: The physical geography of the Mediterranean, edited by: Woodward, J., Oxford University Press, New York, United States, 33-67, 2009. Schiebel, R. and Hemleben, C.: Modern planktic foraminifera, Palaeont. Z., 79 (1), 135-148, 2005. Spezzaferri, S., Kucera, M., Pearson, P. N., Wade, B. S., Rappo, S., Poole, C. R., Morard, R., and Stalder, C.: Fossil and genetic evidence for the polyphyletic nature of the planktonic foraminifera "Globigerinoides", and description of the new genus Trilobatus, PLOS One 10 (5), doi: 10.1371/journal.pone.0128108, 2015. Thunell, R. C.: Distribution of recent planktonic foraminifera in surface sediments of the Mediterranean Sea, Mar. Micropaleontol., 3, 147-173, 1978. van Raden, U. J., Groeneveld, J., Raitzsch, M, and Kučera, M.: Mg/Ca in the planktonic foraminifera Globorotalia inflata and Globigerinoides bulloides from Western Mediterranean plankton tow and core top samples, Mar. Micropaleontol., 78, 101-112, 2011. Weiner A. K. M., Weinkauf, M. F. G., Kurasawa, A., Darling, K. F., and Kucera, M.: Genetic and morphometric evidence for parallel evolution of the Globigerinella calida morphotype, Mar. Micropaleontol., 114, 19-35, 2015.

[Figure]

a)

b)

c) *G. inflata*

d) *T. sacculifer* (without sac)

e) *G. ruber* (white)

f) *G. bulloides*

g) *O. universa*

h) *G. ruber* (white)

i) *G. bulloides*

j) *O. universa*

REV Figure 7: Sample scores on the two PCA factors with (a) the loadings of the enviromental parameters on each factor, (b) with overlay of the absolute abundance values (individuals•10 m$^{-3}$) of every station of all the foraminifera sample, (c) *G. inflata*, (d) *T. sacculifer* (without sac), (e) *G. ruber* (white), (f) *G. bulloides*, and (g) *O. universa*. With overlay of the pA values (µg•µm$^{-2}$) of (h) *G. ruber* (white), (i) *G. bulloides*, and (j) *O. universa*. In blue color western Mediterranean stations (incl. Atlantic and Strait of Gibraltar), in red colour the eastern Mediterranean stations.

**Fig. 1.**

[Figure]

REV Figure 3: Absolute abundance of planktic foraminifera from BONGO nets during leg 1 (stations 1 to 13) and leg 2 (stations 22 to 14). Category 'Others' is comprised by *G. siphonifera*/ *G. calida*/ *G. radians*, *G. quadrilobatus*, *H. pelagica*, *G. ruber* (pink), *G. menardii* and *T. sacculifer* (with sac).

**Fig. 2.**

[Figure]

REV Figure 4: Percentage of each planktic foraminifera size fraction in leg 1 (stations 1 to 13) and leg 2 (stations 22 to 14).

**Fig. 3.**

---

## Author Comment (AC5) · 14 Oct 2016

We appreciate the constructive referee remarks and acknowledge the detailed comments that greatly helped to clarify a number of points and to improve the manuscript. Below are our detailed responses to the referee's comments, including expected modifications of the manuscript.

1 General comments

REFEREE #3, COMMENT: 1. While reading the manuscript I noticed that the writing style needs some attention. The manuscript is understandable, but there are plenty of orthographical and grammatical errors or weird phrasing throughout. Those should

be dealt with (and I noted some suggestions in the detailed comments), to make the manuscript more accessible for the reader.

REPLY: Writing style and grammatical errors are now improved. We appreciated your suggestions.

REFEREE #3, COMMENT: 2. The manuscript is partly missing important information, diverts from the topic, or promises undelivered results. Some examples: Parts of the manuscript, especially the section 'Oceanographic Setting', are lacking citations of information sources. Information on data sources and methods are largely missing. The temperature and salinity might come from the mentioned CTD casts, but the carbonate saturation values most certainly not: Have they been calculated on the basis of water samples (on board or in the lab) or calculated on the basis of database oceanographic data? Which of the several existing methods to calculate size-normalized shell weight has been used? Which software has been used for statistical analyses? All this information belongs in the far too short Material and Methods section!

REPLY: The Oceanographic Setting was written in a way that the references cited at the end of the paragraph were the ones used to reconstruct the paragraph. Now, in the revised manuscript, we change the way of referencing and we apply the references needed after each statement.

A new, more complete, methodology was written, explaining the data sources, the software analysis citation, and the methodology for the SNW. We decided to change "Size-Normalized Weight" to "Density Area" (A) in the revised manuscript. The latter denomination is less confusing and in agreement with previous work (Marshall et al., 2013). Here we present the fragments of the Methodology section that cover that information:

"...The sampling device was equipped with a flow-meter to have data of the volume filtered in each tow. From the upper 200 m of the conductivity-temperature-depth (CTD) stations, located near the sampling sites, was obtained water column data of temperature, salinity, oxygen, fluorescence (for the complete dataset see Ziveri and Grelaud, 2015). Seawater carbonate data (Total alkalinity (AT), and dissolved inorganic carbon (DIC)) was retrieved from Goyet et al. (2015), which was used to calculate pH, pCO2, and [CO3-2] using the software CO2Sys (Lewis and Wallace, 1998) with the equilibrium constants of Mehrbach (1973) refitted by Dickson and Millero (1987). The Italaian National Institute of Oceanography and Experimental Geophysics obtained [PO4] and [NO3] onboard, filtering in glass fiber filters (Whatman GF/F; 0.7 $\mu$m) the water samples, which were keep it at -20°C. After in the laboratory, samples were analyzed with a Bran+Luebbe3 AutoAnalyzer, as did Grasshoff et al. (1999). Surface chlorophyll a concentration was obtained from MODIS Aqua L2 satellite (NASA Goddard Space Flight Center: http://oceandata.sci.gsfc.nasa.gov/)." "For the density area (A) study, we selected 3 main species: G. ruber, G. bulloides and O. universa. All the specimens of these 3 species were photographed with a Canon EOS 650 D camera device attached to a Leica Z16 AP0 microscope to measure their long axis and silhouette area using the software ImageJ (Schneider et al., 2012). For each station and each of the 3 selected species, the individuals were weighed together by triplicate with a Mettler Toledo XS3DU microbalance ($\pm$1 $\mu$g of nominal precision) within 50 $\mu$m size fraction increments (150-200 $\mu$m, 200-250 $\mu$m, etc.). Cytoplasm-filled or empty dry-weighed foraminifera tests were weighted together since dry cytoplasm has no statistically significant effect on the weight of tests >150 $\mu$m (Schiebel et al., 2007). Specimens containing notable organic matter attached to the test were discarded. The maximum number of individuals weighed together was 5, in some stations individuals were measured individually as no more specimens were available. In all the cases the mean weigh per specimen of the three weightings was applied. The silhouette area obtained was then used to obtain the A measurements (as is also done in Marshall et al., 2013; Marshall et al. 2015)." On the revised manuscript we will include a principal component analysis (PCA; Varimax rotation) using SPSS Statistic 23 software.

REFEREE #3, COMMENT: (continuation) The reason for several analyses (e.g. the correlation between shell size and shell weight) is not properly explained, thus leaving

the reader guessing why the authors deem this necessary. A comparison of assemblage data with earlier studies to study long-term trends is promised but never really delivered (not on a reasonable analytical level at least). REPLY: The relation between area and long axis in the three selected main species did not allow detection of any anomaly or changes in their growth pattern. The data on the long axis-weight make possible the comparison with previous studies (see Bijma et al., 2002; Lombard et al., 2010; de Moel et al., 2009; Aldridge et al., 2012; Schiebel et al., 2007), also for the area-weight analysis (compared with Marshall et al. (2015) on the revised manuscript). Especially in the latter, we obtain useful information (in our case, specially for G. ruber (white) and G. bulloides) of their calcification intensity in different locations of the Mediterranean.

Our study does make detailed comparisons against prior studies (Thunell, 1978; Cifelli, 1974; Pujol and Vergnaud-Grazzini, 1998, in the revised manuscript we will include: Rigual-Hernández et al., 2012; Bárcena et al., 2004; Hernández-Almeida et al., 2011; de Castro Coppa et al., 1980). Water column plankton tow data from the Mediterranean is extremely limited, and consequently we are forced to make our detailed assemblage comparisons against sediment trap and surface sediments studies. Therefore we do as sensibly as we can, given the very real limits of existing data.

REFEREE #3, COMMENT: 3. The existing images are OK, for the most part (labels might be a bit small in several of them). However, several key findings of the study are not presented in any suitable graphical manner, instead referring to figures which cannot present these data in a suitable way. Most notably amongst these, while there are several claims made about the influence of environmental factors on abundance and SNW of the species, not a single such relationship is graphically shown in a cross plot.

REPLY: Labels of the figures that need it will be increased in size on the revised manuscript. Our Figures 3 and 4 were modified for the revised manuscript (see REV Fig. 3 and REV Fig. 4). We agree that proper statistical analysis should be conducted

on our data set. This is why in the revised version we will include a principal component analysis performed on the environmental parameters. Such analysis will include a graphical representation in which the absolute abundance and density area values are overlain (REV Fig. 7).

REFEREE #3, COMMENT: 4. The manuscript uses several wrong species names and species concepts. The most prominent one is the unfortunate use of the terms Globigerinoides ruber sensu stricto and Globigerinoides ruber sensu lato, which are pooled, together with Globigerinoides ruber (pink), within the same species. This is blatantly wrong. Aurahs et al. (2011) has established that Globigerinoides ruber (pink), Globigerinoides ruber (white) (your sensu stricto), and Globigerinoides elongatus (your sensu lato) are distinctly different species, both biologically and in terms of morphology; and has therefore rehauled their Linnean taxonomy. Could we please all agree that 5 years after this publication we could at last all start to call them by their proper names and abandon this unfortunate sensu stricto/sensu lato distinction. It would be one thing if it would only be about names (I would still request to use up-to-date terminology, but it would be a minor mistake). Rather, G. elongatus is not even the adelphotaxon to G. ruber (white), but is more closely related to Globigerinoides conglobatus. Pooling them together under the same species name thus produces a polyphylum. If you want to pool them for some purposes (which can make sense) you can call them 'G. ruber/G. elongatus plexus', or something along those lines. Second, the species Globigerinella siphonifera is reported from the samples. However, it is not clear whether this means that only G. siphonifera is present, or whether this is a collective term for the entire Globigerinella siphonifera/Globigerinella calida/Globigerinella radians plexus (Weiner et al., 2015), within which species have not been separated by the authors. Third, but less serious because this really is only a naming issue, the former Globigerinoides sacculifer should be referred to as Trilobatus sacculifer meanwhile (Spezzaferri et al., 2015). Furthermore, in that species your 'quadrocameratus' morphotype is correctly referred to as 'quadrilobatus' morphotype to my knowledge.

REPLY: We changed the names in the revised manuscript in agreement with Spezzaferri et al. (2015) and Aurahs et al. (2011) as follows: Globigerinoides ruber sensu stricto changed to Globigerinoides ruber (white) Globigerinoides ruber sensu lato changed to Globigerinoides elongatus Globigerinoides sacculifer sacculifer type changed to Trilobatus sacculifer (with sac) Globigerinoides sacculifer trilobus type changed to Trilobatus sacculifer (without sac) Globigerinoides sacculifer quadrocameratus type changed to Globigerinoides quadrilobatus Globigerinella siphonifera changed to Globigerinella siphonifera/ G. calida/ G. radians plexus

REFEREE #3, COMMENT: 5. The most important issue is with the statistical analytical approach. According to lines 146–147 you are using a Pearson product moment correlation to test the relative abundances and shell calcification intensities of several species against environmental parameters. This is horribly wrong on a multitude of levels, as I will summarise hereafter. For further details you may have a look at Dytham (2011), Legendre and Legendre (2012), Faraway (2006), and McDonald (2009).

I—You assume a causal relationship between environmental factors and SNW/species abundance. Correlation analyses are not appropriate here, regression analyses with SNW/abundance as the dependent variable against the independent environmental factors must be used. Occasionally this makes only a cosmetical difference (i.e. type I linear regression vs. Pearson product moment correlation), but even then it is of methodological and implicational importance (compare Legendre and Legendre, 2012, box 10.1). In this case, however, it is even more important because of the points below.

II—Type I regression (as well as its correlation equivalent for that matter) is only applicable under certain circumstances, one of which is that x-values are measured without errors (McDonald, 2009; Dytham, 2011; Legendre and Legendre, 2012). It is therefore nearly only usable for laboratory experiments. As long as you are testing for the influence of parameters that you actually measured on board (temperature, salinity, pH), you might this this still works with a lot of good will, but I would argue that even

then you have an error on those values, because you only have a snapshot image, and not a mean (let alone constant) value covering the entire life-time of your specimens. Further, I assume (you never state that) that at least part of the data you needed to calculate the carbonate system comes from averaged database data anyway!? And at least then, and in my opinion under all circumstances, you have to use robust type II or type III regression methods.

III—You cannot simply test the same dependent variable against several independent parameters in different tests. The simple reason is that each of those test has its own type I error chance, and those are summing up until (after a sufficient number of tests) you are guaranteed to get at least one type I error in your analyses (compare Dytham, 2011; Legendre and Legendre, 2012). It is imperative that under such conditions at the very least all multiple tests (i.e. all tests for the influence of individual environmental factors on SNW or abundance per species) are corrected for this problem. Either using a correction for the family-wise error rate (e.g. Bonferroni correction), or a correction for the false discovery rate (e.g. Benjamini and Hochberg, 1995).

IV—Making several such analyses and correcting them per species is still not the ideal solution, mainly because (as usual in marine environments) all independent variables show a large degree of multicollinearity (just have a look at your own Fig. 1). This means that such simple parameter-wise tests may detect an influence of several parameters, but only because they are highly correlated, and it is unclear which factor influences the dependent variable the most (or at all, for that matter). For the case of SNW in particular it might be much better to use an approach that can test for all independent variables at once, while reducing the influence of the multicollinearity between different environmental factors (Dormann et al., 2013). Such methods could for instance be generalized linear models (GLM) or generalized additive models (GAM), both of which have the added benefit over multiple linear regression that they are invariant to the order in which independent variables are added to the model (compare Faraway, 2006). For relative abundances you face the additional problem, that y-values

are not independent of each other within a sample (e.g. if G. ruber already represents 50% of the assemblage, then G. bulloides cannot be more abundant than 50% anymore in that same sample). While there are ways around this (most notably, using absolute abundances with an appropriate link function in a GLM, or applying any of the methods described in van den Boogart and Tolosana-Delgado (2013)) you may also prefer to analyse the assemblage data using suitable ordination techniques (compare for instance Hammer and Harper, 2006). This would have the added benefit that such ordination techniques can also be adapted to properly compare your assemblage with that of earlier studies, in this way delivering on a promise made in the introduction and never fulfilled in the manuscript.

REPLY: We agree that proper statistical analysis should be conducted on our data set. This is why in the revised version we will include a principal component analysis performed on the environmental parameters. Note that new environmental parameters will be added: the nutrients (NO3 and PO4), the oxygen concentrations and the pCO2. The results of the PCA show that 2 factors explain about 77% of the total variance in the environmental parameters. The 1st factor exhibited positive loadings on the nutrients and the fluorescence and negative loadings on temperature and salinity (and to a lesser degree on carbonate ion concentrations). This factor explains 56.99% of the total variance and represents the strong west-east gradient characterizing the Mediterranean Sea as the water become warmer, saltier and more oligotrophic eastward. The second factor explains about 20.02% of the total variance and is characterized by positive loadings on pH and oxygen concentrations (and to a lesser degree on carbonate ion concentrations) and a negative loading on the pCO2. It is interpreted as the variations of the carbonate system properties in the Mediterranean Sea with more acidic conditions in the western basin compared to the eastern basin. The sample scores on the 2 first factors with overlay of absolute abundances of foraminifera species (G. ruber (white), G. bulloides, G. inflate, O. universa and T. sacculifer (without sac)) and density area (G. ruber (white), G. bulloides and O. universa) are presented and discussed in the revised manuscript.

**2 Detailed comments**

COMMENT: Line 33, 'calcareous zooplankton': I would be very careful talking about zooplankton here. While it is true that all planktonic Foraminifera can live heterotrophic, many are also able to harbour photosymbionts. REPLY: We decided to change it to "calcareous plankton" to avoid possible confusion.

COMMENT: Line 35, 'Hembelen et al., 1989': Should be 'Hemleben et al., 1989'. REPLY: Changed in the revised manuscript.

COMMENT: Line 36, 'due to': Should be 'and show'. REPLY: Changed in the revised manuscript.

COMMENT: Lines 36–37, 'The species are adapted [...] spines and test shape.': They are certainly adapted to different environments, because naturally there cannot be any two species which occupy exactly the same niche, but implying such a trivial form of adaptation is far too oversimplified. _ Line 37, 'test shape': Should be 'shape, which are partly related to those adaptations'. REPLY: Changed as it follows in the revised manuscript. "The species are adapted to different environments and show differences in wall structure, pores, spines and test shape, which are partly related to those adaptations."

COMMENT: Lines 37–39, 'The distribution of foraminifera [...] which shifts during ontogeny.': A citation for this statement is needed. REPLY: We added the following references for that statement in the revised manuscript : Schiebel and Hemleben (2005); Hemleben et al. (1989).

COMMENT: Lines 42–45, 'Ecological tolerance limits [...] departure from optimum conditions (Arnold and Parker, 1999).': Which is basically true for every organism, so what is the point here? Plus, this is hardly the best citation for this statement. What about Bé (1977) for example? REPLY: We consider that sentence can help some readers to understand better the article, despite others not having any new information

from reading it. In that sentence we include the citations that prove the cause of having more or less abundance of foraminifera in a location. We include Bé (1977) in the citations. Also, that sentence provides the information that presently these boundaries are not completely defined, and work for it is still needed.

COMMENT: Lines 48–50, 'The first modern study of planktic foraminifera [...] expedition of 1947–48.': Was this study published? Cite a source. REPLY: We added the following reference in the revised manuscript: Petterson (1953).

COMMENT: Line 54, 'at 250 m depth': Should be 'of the upper 250m water column'. REPLY: Changed in the revised manuscript.

QUESTION: Line 57, 'that': Should be 'that the'. REPLY: Changed in the revised manuscript.

COMMENT: Lines 57–61, 'Thunell (1978) studied samples [...] inside the Mediterranean.': Break up this sentence. REPLY: We change it as it follows in the revised manuscript: "Thunell (1978) studied samples from the upper 2 cm of cores covering the Mediterranean, concluding that the distribution of planktic foraminifera is closely linked with the distribution of the different surface water masses. There are specific temperature and salinity ranges for each water mass, as Bé and Tolderlund (1971) stated for the Atlantic, and a partial isolation effect in the different basins and sub-basins inside the Mediterranean. Those phenomena result in different species assemblages in each region."

COMMENT: Line 65, 'wide': Should be 'large'. REPLY: Changed in the revised manuscript.

COMMENT: Lines 65–66, 'They concluded [...] variable foraminifera assemblages,': This is not entirely correct. Pujol and Vergnaud Grazzini (1995) only state that the observed assemblage patterns 'cannot be entirely explained by the general temperature and salinity differences among the different Mediterranean Basins' and are also

strongly correlated to more regional hydrogeographic patterns. REPLY: It is true that in the sentence of the Abstract of Pujol and Vergnaud-Grazzini (1995) that they do not discard the temperature and salinity to explain their results, but also they state that the hydrogeographic patterns that regulate the nutrient dynamics have stronger weight on them. In the conclusion section they state it more clearly than in the abstract. From from Pujol and Vergnaud-Grazzini (1995): "Although the distribution patterns of many species display strong differences between the two sampling periods, there is no direct correlation with sea surface temperature or salinity gradient changes. In fact, the rather large west to east gradients in temperature and salinity are not reflected in the relative or absolute abundances of the different species. The strong seasonal and regional variability of other hydrochemical parameters such as nutrients and of physical structures such as eddies or fronts may explain part of the observed differences in the distribution patterns."

As in our sentence, we are not discarding the possibility of a temperature-salinity effect on them, despite these two parameters alone not varying enough to justify the extremely variable foraminifera assemblages, we think that there is no need to modify it.

COMMENT: Lines 70–72, 'The calcification of foraminifera [...] (Schiebel and Hemleben, 2005).': Those are neither the only factors influencing shell calcification intensity in planktonic Foraminifera, nor are all of the stated relationships universally true. Compare Marshall et al. (2013, tab. 1) and Weinkauf et al. (2016, tab. 7) for a summary of this matter. REPLY: We appreciate your references here. We propose the next modification: "The calcification of foraminifera is affected by the chemical state of their surrounding waters. Theoretically their shell mass is positively related to temperature, pH, [Ca+2], and alkalinity from its ambient water and negatively related with [CO2] (Schiebel and Hemleben, 2005). In the different practical studies with water column plankton their shell mass was tested as positively related with [CO2] (Aldridge et al., 2012; Beer et al., 2010a; Marshall et al., 2013; Moy et al., 2009) but also negatively

(Beer et al. 2010a). Also, other studies relate positively foraminifera shell mass with temperature (Mohan et al. 2015; Aldridge et al., 2012; Marshall et al., 2013)."

COMMENT: Lines 70–77: I think the cited literature for calcification studies is by far not exhaustive. What about Broecker and Clark (2001b), Barker and Elderfield (2002), de Villiers (2004), Manno et al. (2012), and Marshall et al. (2013), to name but a few. REPLY: We focus on living plankton from tows and how the environmental parameters affect their calcification. The literature cited, despite not being exhaustive, represents the living foraminifera calcification studies. We propose to include the following sentence at line 75: "For further studies relating foraminiferal calcification influenced by environmental parameters see Weinkauf et al. (2016); Table 7. Since the industrial era. . ."

COMMENT: Line 76, 'building': Should be 'formation'. REPLY: Changed in the revised manuscript.

COMMENT: Lines 82–83, 'In addition, few size-normalized weight (SNW) studies from water column foraminifera are available in the literature.': Then please provide such examples here in the form of citations. REPLY: We provide the following citations in the revised manuscript: Schiebel et al., 2007; Beer et al., 2010a; Aldridge et al., 2012; Marshall et al., 2013; Mohan et al., 2015; Marshall et al., 2015; Weinkauf et al., 2016).

COMMENT: Line 91, 'more unbreakable tests': Should be 'tests with thicker walls'. REPLY: Changed in the revised manuscript.

COMMENT: Line 92, 'empty tests are passive particles that ocean currents may displace.' Which is perfectly true for living Foraminifera as well; hence they are plankton, not nekton. REPLY: We are in agreement that this characteristic is accomplished for plankton and not nekton. We considered no modifications in that sentence.

COMMENT: Lines 97–98, '(2) characterize, at the species level their ecology through their seasonal and geographical distribution and abundance by comparison with previous studies,': This point is not really present in the paper, at least not above a relatively comparative level. The interpretation why abundances might be different now than they were 20 years ago, and any reliable analysis and graphical presentation that shows that in the first place, is largely missing. REPLY: The numbers of available studies generating data of this kind (water column planktonic foraminifera abundances, etc.) are extremely rare overall, and especially in the Mediterranean Sea. Therefore we are forced to compare with sediment trap and surface sediment (core-top) results from prior decades in this marginal sea (Rigual-Hernández et al., 2012; Bárcena et al., 2004; Hernández-Almeida et al., 2011; Thunell, 1978). Based on these sample format differences, time differences (e.g. late 20th century vs. early 21st century), and likely other differences as well, the basis for such comparisons is of course very far from perfect and ideal. However, given the rarity and recency of this new water column data set, we naturally use it to speculate on the comparisons and what they might reveal about changes going on the surface ocean environment in this region. This is what we can do and what anyone can do with the data in hand. To compare with comparable data from prior studies is a natural discussion aim based on it, even if the basis for the comparison is far from ideal.

COMMENT: Line 103, 'with a strong thermohaline and wind-driven circulation,': Citation needed! _ Lines 105–106, 'These basins are composed of different sub-basins due to partial isolation caused by sills that influence the water circulation, and by different water properties.': Citation needed! _ Lines 107–109, 'where the nutrient-rich Atlantic surface waters [...] (evaporation exceeding precipitation).': Citation needed! _ Lines 113–116, 'In the eastern basin, [...] and fresher toward the western basin.': Citation needed! _ Lines 117–118, 'Waters returning to the Atlantic through the Strait of Gibraltar at depth are cooler and saltier than the inbound waters, and compensate for the inflow from the Atlantic.': Citation needed! REPLY: See the answer of these questions in the major comment (2.).

COMMENT: Lines 106–107, 'World Ocean': Should be 'worlds oceans'. REPLY: We

propose to change it to "ocean" instead of "worlds oceans". Now the sentence would be like the following: "Natural connection with the ocean is through the narrow Strait of Gibraltar,"

COMMENT: Line 111, 'until the': Should be 'and reach as far as the'. REPLY: Changed in the revised manuscript.

COMMENT: Line 135, 'at 200m depth': Should be 'from 200m depth to the surface'. REPLY: The towing is realized mainly at 200 m depth, but meanwhile it goes down and it returns up to the vessel, also the tows can catch samples. To clarify that we change "at 200m depth" for the following: "primarily 200 m depth, but also including tow time integrating the upper water column from 200m to the surface".

COMMENT: Line 141, 'counted and separated by species and size': Should be 'split into fractions by size'. _ Line 142, 'to determine the absolute and relative abundances': Should be 'and planktonic Foraminifera were counted on the species level'. Furthermore, it is not mentioned which taxonomic system is used. It is most certainly not up-to-date (compare general comments). REPLY: We do not agree here. First, the separation was made by species, then by size. If we correct like that it would seem that the process was done in the opposite way. We change, in order to clarify, that sentence to the following in the revised manuscript: "From each sampling station, the foraminifera were isolated and identified at species level. [. . .] For each sample, each species was counted and isolated according to 3 size fractions (150–350 $\mu$m, $\geq$350–500 $\mu$m, and >500 $\mu$m) to determine the absolute and relative abundances." We include, in the revised manuscript, the references used for the taxonomic nomenclature of our found species, being part of the Methodology section: "We classified the different foraminifera species with visual identification with optical microscopy with the option of picking and turning the specimens to see their different sides. We followed the morphometric guidelines and taxonomic nomenclature proposed by Aurahs et al. (2011) for Globigerinoides ruber (white), Globigerinoides ruber (pink) and Globigerinoides elongatus. For Trilobatus sacculifer (with sac) and T. sacculifer (without sac)

we used Spezzaferri et al. (2015). Hemleben et al. (1989) was used as a guide to classify Globigerinoides bulloides, Orbulina universa, Globorotalia inflata, Globorotalia menardii, and Hastigerina pelágica. Globigerinoides quadrilobatus was inferred from Papp and Schmid (1985). G.bulloides could not be differentiated from Globigerina falconensis in our samples and are treated together; the G. bulloides/G. falconensis plexus is referred as G. bulloides in our study. Globigerinella siphonifera/G. calida/ G. radians plexus (see Weiner et al., 2015) is treated as G. siphonifera in our study."

COMMENT: Lines 144–145, 'Individuals of the same station and species within a 50 _m diameter size constraint were weighed with a Mettler Toledo XS3DU microbalance (_1 _g of error).': So I assume they were weighed together (single shell measurements would require a more precise balance). But were the measurements afterwards actually corrected for mean shell size per sample (MBW approach, Barker and Elderfield (2002)), or was the simple SBW approach used (Lohmann, 1995; Broecker and Clark, 2001a). The main problem is that in the latter case, Beer et al. (2010a) has shown that the SBW method is not fully effective in eliminating the shell size effect. Additionally, results cannot be independently replicated and tested when the exact methodology is not sufficiently described. Also, 'error' is the wrong term in this context, and 'nominal precision' should be used instead. REPLY: We weighed together a maximum of 5 individuals, always within the same 50 $\mu$m size constraint. We decided to change "Size-Normalized Weight" to "Density Area" (A) in the revised manuscript. The latter denomination is less confusing and in agreement with previous work (Marshall et al., 2013). For further details see the methodology text addition to the new manuscript, found on the major comments section, comment (2.). Changed the word "error" for "nominal precision" on the revised manuscript.

COMMENT: Lines 146–149: It is not mentioned anywhere which software was used to carry out statistical analyses. REPLY: On the revised manuscript we will include a principal component analysis (PCA; Varimax rotation) using SPSS Statistic 23 software.

COMMENT: Lines 147–149, 'Absolute abundances [...] observed within the environmental parameters.': This is no valid reason at all to skip this. It can be that you are not interested in this, then state why, or that you are concerned about the validity of the results, then state why. A large difference in values does not compromise such an analysis at all if the correct techniques are applied. REPLY: The relative abundances were used, as the samples have less variability and results correlate more, giving more importance to the species assemblages than the highly variable quantity of foraminifera in each station. Furthermore, now we carried a different statistical analysis (PCA) in which absolute abundances are considered.

COMMENT: Line 171, 'Globigerinoides ruber sensu strict (s.s.)': As mentioned in the general comments, this species is correctly referred to as Globigerinoides ruber (white). Please change in the entire manuscript. REPLY: In agreement with Aurahs et al. (2011) we change the nomenclature in the revised manuscript.

COMMENT: Line 174, 'Globigerinella siphonifera': Your species list contains only Globigerinella siphonifera, but neither G. calida nor G. radians (compare Weiner et al., 2015, and the general comments above). This could mean that either you checked and the other two species are not present at all, or you lumped the entire plexus into one category. Please explain what is the case here. REPLY: In agreement with Weiner et al. (2015) we change the nomenclature in the revised manuscript. Globigerinella siphonifera will be changed to Globigerinella siphonifera/ G. calida/ G. radians plexus. In the methodology section will be noted the use of the name G. siphonifera to represent the whole plexus further on the article.

COMMENT: Lines 176–178, 'In addition, a higher percentage [...] and may not be generalized.': Given the fact that in plankton tows you have only little control over the growth stage of your individuals, one may wonder to what degree this size trend over time may represent a reproduction event. REPLY: That is a reason why we add the last sentence of the paragraph referred in that question.

COMMENT: Line 180, 'sample': Should be 'assemblage'. REPLY: Changed in the

revised manuscript.

COMMENT: Lines 183, 187, 191, 197, '(Fig. 3; Fig 4)': The referred information is illustrated by neither of these figures, because Fig. 3 does not give shell sizes and Fig. 4 does not distinguish between species. Unless Fig. 3 would only represent the fraction >350_m, but then this is stated nowhere in the figure caption. REPLY: Now references to Fig. 3 and Fig. 4 are separated and located after the exact sentence each one. We also include the citation of Appendix A (where absolute abundance data for each size fraction in each species is provided) in the revised manuscript.

COMMENT: Line 192, 'Globigerinoides sacculifer': This should be Trilobatus sacculifer (compare Spezzaferri et al., 2015, and general comments). REPLY: In agreement with Spezzaferri et al. (2015) we change the nomenclature in the revised manuscript.

COMMENT: Line 198, 'quadrocameratus': Should be 'quadrilobatus' in the entire manuscript. REPLY: Changed in the revised manuscript.

COMMENT: Line 218, 'A Pearson test': This is the wrong method for the question that should be answered (compare general comments). By the way, even if correlation per species was the correct approach, abundance data are by default not normally distributed but follow a Poisson distribution. This rules out any parametric test in the first place, and would leave Spearman rank-order correlation or Kendall rank-order correlation as the only reasonable alternative. Compare the general comments section, however, why neither of these is appropriate here. REPLY: See the answer of this question in the "major comment (5.)".

COMMENT: Lines 222–223, 'Relative abundance was selected instead of absolute abundance to avoid bias due to the big differences between stations' results in absolute abundance.': This approach, however, introduces new problems because now the abundances per station are not independent; and the given reason for this decision is invalid anyway. Compositional regression (van den Boogart and Tolosana-Delgado, 2013) or other adequate approaches would be needed. Compare general comments

section. REPLY: Relative abundances are grouped to see which species dominate in each geographic region of the Mediterranean. There exists high variability in the sample size along the stations; we consider relative abundance a valuable data source to understand better the ecology and distribution of the different species. Also our relative abundance groupings were estimated to allow the comparison with previous studies in the Mediterranean using relative abundances in a sub-basin/regional location level of comparison (Cifelli, 1974; Thunell, 1978; Pujol & Grazzini, 1998 (in text, not in figures)). Absolute abundance data is also provided and used in the results and discussion sections. For the analysis we compare the PCA factors with absolute abundance and SNW, which will be treated in the results and discussion section, leaving the species assemblage only for comparison with previous literature.

COMMENT: Lines 220–222: All p-values reported here are invalid, because they have not been corrected for multiple testing on the species level. The general comments section gives more discussion about this. Additionally, why is nothing of that presented in a graphical form? REPLY: We performed a PCA analysis on the revised manuscript. See the answer at "major comment (5.)".

COMMENT: Lines 223–225, 'The remaining species [...] abundance and environmental parameters.': This is no reasonable explanation. The mere lack of the species at some stations would not rule out such an analysis, if there are still enough stations with values >0 left. REPLY: See the answer to that question in "major comments (5.)".

COMMENT: Lines 229–230, 'The high two-dimensional (silhouette) area-to-diameter correlation is best fitted by a power regression (Fig. S2).': As would be expected. But why is this important in the context of that paper? Additionally, from a purely modeling-point-of-view I might argue that the regression should be fitted so that they are forced to have a zero intercept (everything else seems wrong). REPLY: The relation between area and long axis in the three selected main species does not allow detection of any anomaly or changes in their growth pattern. We will add the following text in the paragraph of lines 228-236 to clarify Fig. S2: "...The high two-dimensional

(silhouette) area-to-long axis correlation is best fitted by a power regression (Fig. S2). The same growth pattern can be seen in G. ruber s.s., G. bulloides, and O. universa with that correlation, represented graphically in the shape of a power function (Fig. S2). They grow slightly faster when they are smaller (steepest in the lower left part of the regression line) and slightly slower when they are bigger (less steep in the upper right part of the regression line; Fig. S2). Comparing the average values from different locations sampled within the Mediterranean…"

Size and mass of foraminifers relationship does not start at the origin (zero). The proloculus of planktic foraminifera measures between 15-30 $\mu$m in average, and has a certain calcite mass, which has so far not been determined (see Hemleben et al., 1989).

COMMENT: Lines 230–235, 'Comparing the average values [...] northwestern Mediterranean (Fig. S2).': If the idea is to compare shell sizes between different basins, then this is hardly the best method of presentation. A boxplot or barplot would be much more appropriate here. Further, it is stated nowhere which statistical techniques were used to test the shell size differences between basins. I assume an ANOVA followed by post-hoc tests, but this is explained nowhere. REPLY: We consider our graphical representation appropriate for the function it has. We will change the word "Comparing" to "Presenting", to avoid confusion in the interpretation of the sentence.

COMMENT: Lines 237–239, 'The diameter-to-weight relation [...] (r2 = 0:516; Fig. S3).': If you want to imply a dependency relationship (which can make sense, depending on your intention), then it would probably be more logically to assume that weight is dependent on size, so you should exchange the axis in your Fig. S3. Otherwise, here a correlation would be more appropriate. Furthermore, the question is again what is the sense of this analysis in the context of that paper. It should be made clear for the reader, why this analysis is performed. In agreement with that question we will exchange the axis in our figure in the revised manuscript. We find useful our "Weight vs. Long axis" study as its comparable with other studies in the literature (see Bijma et al.,

2002; Lombard et al., 2010; de Moel et al., 2009; Aldridge et al., 2012; Schiebel et al., 2007) and make our own conclusions after its comparison (in our discussion section).

COMMENT: Lines 239–240, 'O. universa was finally discarded for comparisons between SNWs at different locations due to a low area–weight correlation, while data from G. ruber s.s. correlate well (Fig. S4a).': I do not really see the reason for this.

1) The weight–size relationship is not that bad (p-values are not given, interestingly).

2) I do not understand why the authors would insist in such a relationship to be a necessity for the interpretation of SNW. Sure, if there is no good relationship it would be difficult to predict shell size from shell weight or vice versa. But especially if you imply a relationship between calcification intensity and the environment you would expect to see deviations from this relationship. Otherwise, shell weight would be a function purely of shell size, and size-normalized shell weight would not have any value in environmental interpretations. Now, a lower R2 value in O. universa in my opinion only means, that its shell weight is to an even lower extant controlled by shell size than it is in other species. This could mean, that O. universa is more susceptible to environmental protrusions in regard to its ability to control calcification, which would by some standard make it an even better proxy species. I can think of no reason why a low correlation value itself would make SNW interpretations invalid, however. REPLY: We decided not to show O. universa density area in Fig. 6 as no pattern was seen in its data but the data are presented in REV Fig. 1.

In the PCA realized for the revised manuscript we overlay the results of O. universa density area on the two factors obtained, which reflect environmental parameters of our sampled stations.

COMMENT: Lines 240–242, 'The eastern Mediterranean [...] G. ruber s.s. (Fig. S4d-e).': This is again not an appropriate way of presenting those results. Use a box-plot/barplot instead. REPLY: We consider our graphical representation appropriate for the function it has.

COMMENT: Lines 243–244, 'The eastern Mediterranean individuals have the lowest median SNW': Is this just eyeballing or has it actually been tested somehow, which regions are different and which are not concerning SNW? REPLY: The median values where obtained from the density area approach; we observed that its values were lower in the eastern Mediterranean. We do not need a statistical test to know which is the smallest value. No statistical test was done regarding Fig. 6; on the other hand, statistically robust results regarding density area are presented in the revised manuscript with a PCA (see the answer to the question "major comments (5.)" for further details).

COMMENT: Line 245, '_g _ _môĂĂĂ2': So from this unit I assume the authors yet used the MBW approach, instead of SBW!? It is imperative that this is made clear in the Methods section. REPLY: Yes, see answer to your comment at Lines 144–145 and the new methodology section written in the major comment (2.).

COMMENT: Lines 248–251, 'A Pearson correlation test [...] correlation with fluorescence (p = 0:01).': Apart from the fact that this technique is again inappropriate for the data (compare general comments and discussion for the abundance data) it is interesting that this important result is not graphically presented in any form. If such relations really exist, you should show them in the form of a figure. REPLY: See the answer to that question in "major comment (5.)". A graphical representation of the PCA overlaid with the absolute abundance and density area results will be included in the revised manuscript.

COMMENT: Lines 252–253, 'The Atlantic has [...] opposite trend as in G. ruber s.s.': Again,eyeballing or tested? REPLY: The median and IQR values were obtained from the density area approach and whisker-box plot conversion. We do not need a statistical test to know which is the smallest value. No statistical test was done regarding Fig. 6; on the other hand, statistically robust results regarding density area are presented in the revised manuscript with a PCA (see the answer to the question "major comments (5.)" for further details).

COMMENT:Lines 256–257, 'G. bulloides is positively correlated with pH and [$CO_2$ôĂĂĂ3 ] (p = 0:05) in the Pearson test.': Which is again not shown in any graphical representation. REPLY: See the answer to that question in "major comment (5.)". A graphical representation of the PCA overlaid with the absolute abundance and density area results will be included in the revised manuscript.

COMMENT: Line 280, 'occurs in a': Should be 'come from'. REPLY: Changed in the revised manuscript.

COMMENT: Line 280, 'season of the year': Should be 'seasons'. REPLY: Changed in the revised manuscript.

COMMENT: Line 283: Delete 'eastern'. REPLY: Changed in the revised manuscript as follows: "...western Mediterranean abundances are higher than the eastern ones overall, due to more oligotrophic conditions and higher temperature and salinity values in the east that limit foraminiferal production during winter and late summer."

COMMENT: Line 284: Delete 'both'. REPLY: Changed in the revised manuscript.

COMMENT: Lines 285–286, 'no significant differences are observed between samples collected during day and night.': Is this a subjective impression or was it tested statistically, because only in the latter case you should use 'significantly'. Further, why is this not presented graphically somewhere? REPLY: We delete "significant" in the revised manuscript, as no statistically test was performed on that matter.

COMMENTS: Lines 287, 'accounting for a single species': Which is blatantly wrong for virtually every perceivable species concept. _ Lines 288–289, 'G. ruber: sensu stricto, sensu lato (containing different cryptic species; Aurahs et al., 2009a),': This is no up-to-date information in this regard anymore. Furthermore, the references contain only one 'Aurahs et al. (2009)', not an 'a' and 'b' version; please correct this. REPLY: In agreement with Aurahs et al. (2011) we delete that paragraph and we change: Globigerinoides ruber sensu stricto changed to Globigerinoides ruber (white) Globigerinoides ruber sensu lato changed to Globigerinoides elongatus. Aurahs et al. (2009) has been removed from our references.

The deleted paragraph will be substituted by the following: "Comparing with previous studies that covered the Mediterranean, we notice that Thunell (1978) and Pujol and Vergraud-Grazzini (1995) did not find G. menardii, despite it being found in this study and Cifelli (1974), both in very low quantities. The lack of data from surface sediments and their tropical water preference suggest that is a new species in the Mediterranean (Cifelli, 1974), possibly caused by warmer conditions than in past times. The rest of the species found in our study are found in the past studies covering the Mediterranean Sea (Cifelli, 1974; Thunell, 1978; Pujol and Vergraud-Grazzini, 1995), but it remains in doubt if whether Pujol and Vergraud-Grazzini found G. falconensis and classified it as G. bulloides; or if Thunell (1978) found G. elongatus and T. sacculifer (without sac) and classified them as G. ruber and G. sacculifer. The former problem is also found in Pujol and Vergraud-Grazzini (1995). Also, it is not certain if Cifelli (1974) found G. calida and classified as G. aequilateralis (old equivalent of G. siphonifera). For the figures in Cifelli (1974) we deduce that G. elongatus was classified as G. ruber in the study. In the same way, we do not find any evidence of finding T. sacculifer (with sac) from the Cifelli (1974) figures, but we cannot discard the possibility of it being classified as G. trilobus (T. sacculifer without sac). Finally, we do not have the evidence if Cifelli (1974) found G. ruber (pink) and classified it together with the white variety into G. ruber."

"To be able do a quantitative comparison of the number of species found with previous Mediterranean studies , first, we make the following simplification: G. bulloides and G. falconensis count as one species for that comparison; the same is applied for G. siphonifera and G. calida, and G. ruber (white) and G. ruber (pink). Secondly, we made the assumption that all the doubtful species found in previous studies (see two paragraphs above) were found (e.g.: we assume that Thunell (1978) found G. elongatus and he classified it as G. ruber). After applying these conditions we arrive at an "apparent number of species" able to be compared. Our apparent species becomes

11, clearly inferior to Cifelli (1974) with 19 apparent species, and Thunell (1978) and Pujol and Vergraud-Grazzini (1995) with 17 apparent species. In station 3 of this study (Alboran Sea), we found 8 apparent species; meanwhile the number ascends to 12 in Rigual-Hernández et al. (2012) apparent species flux in the same month."

COMMENT: Line 292, 'with Cifelli (1974)', 'with Pujol and Grazzini (1995)': 'with' should in both cases be 'by'. REPLY: Changed in the revised manuscript.

COMMENT: Line 294, 'reach': Should be 'reached'. REPLY: Changed in the revised manuscript.

COMMENT: Lines 294–295, 'Turborotalita quinqueloba, Neogloboquadrina pachyderma, and Globorotalia truncatulinoides.': Another problem for some species (certainly not G. truncatulinoides, but probably T. quinqueloba and potentially N. pachyderma) is that you used a 150 _m mesh size. Most studies by default use 100 _m for plankton net hauls, and part of the discrepancy you see (also in terms of general abundances) might be that you missed a lot of the small specimens. From my experience (compare Weinkauf et al. (2016) vs. Storz (2006)/Storz et al. (2009)) you can miss the majority of specimens in some species by just switching from 125 _m to 150 _m. In this regard, Pujol and Vergnaud Grazzini (1995) used 120 _m, potentially explaining a lot of your observed differences. REPLY: We treat that problem in the revised manuscript discussion section "5.1. Abundance and diversity patterns". The problem in this question is addressed there as follows: "Some of the species not found reached high frequencies in the aforementioned studies: e.g., the winter species Turborotalita quinqueloba, Neogloboquadrina pachyderma, and Globorotalia truncatulinoides. The fact that these species were not sampled in the present study may be caused by their absence or presence at extremely low abundances of adult specimens at the sampled stations in May, as they use to have low abundances at that time according to a 12-year sediment trap record in the Gulf of Lions (Rigual-Hernández et al., 2012). Another possibility is their presence in sizes smaller than 150 $\mu$m, escaping from our BONGO nets mesh size, a possibility that could be supported by previous Mediterranean studies with

thinner mesh sizes finding that species (see Pujol and Vergraud-Grazzini, 1998, 120 $\mu$m mesh size; Rigual-Hernández et al., 2012, 63-150 $\mu$m mesh size)."

COMMENT: Lines 297–298, 'G. sacculifer type quadrocameratus was not found in previous studies': A potential problem with this statement is whether in those previous studies T. sacculifer has been consequently subdivided. While most studies I am aware of distinguish between the sacculifer- and trilobus-morphotypes, it is often unclear whether the quadrilobatus- (or immaturus-) morphotypes would be counted separately if discovered and truly are absent in the samples, or if they are by default pooled in with the trilobus-morphotype. REPLY: We treat that problem in the revised manuscript discussion section "5.1. Abundance and diversity patterns". The problem in this question is addressed there as follows: "G. quadrilobatus was not found in previous studies working with plankton tows in the Mediterranean, despite its abundance in sedimentary cores (i.e. Cramp et al., 1988; Rio et al., 1990); there exists the possibility to classify it as G. sacculifer or G. trilobus in previous studies as was suggested by Hemleben et al. (1989)."

COMMENT: Lines 300–302, 'The lower absolute abundance [...] recent years': Yes, it could. But again, given that Pujol and Vergnaud Grazzini (1995) definitely used a finer mesh size, this could simply be the result of you missing a lot of specimens. I would therefore be very cautious with this interpretation. Berger (1969) provides equations with which observed abundances could be calibrated for different hypothetical mesh sizes, and such a correction of your data might provide a much better comparability with earlier studies. REPLY: We treat that problem in the revised manuscript discussion section "5.1. Abundance and diversity patterns". The problem in this question is addressed there as follows: "Note that our mesh size is bigger than Pujol and Vergraud-Grazzini (1995) and Rigual-Hernández et al. (2012), but is similar to Cifelli (1974): mesh size of 158 $\mu$m. The wider mesh size could be a cause of our lower numbers in absolute abundance and reduced diversity, but the bigger results in species diversity of Cifelli (1974) in June, a theoretical lower foraminiferal presence month than

May (Rigual-Hernández et al. 2012) supports our statement."

COMMENT: Line 311, '(Fig. 4).': Again, as this figure does not distinguish between species it cannot illustrate the trends you describe here. REPLY: We add Appendix A reference here, where abundance per each size fraction is found.

COMMENT: Section '5.2. Factors controlling the abundance of the main species': The authors try to interpret each individual species' abundance in terms of seasonality and compare it with other studies. However, it is not fully clear what the purpose of this is supposed to be. Many of the described trends are not new, and while it is always good to replicate results, this should not be the main purpose of the manuscript. Rather, the comparison of abundances with studies from several years ago, and the interpretation of potential reasons for changes (as promised in the introduction) is largely missing. REPLY: We disagree with the notion that we do not deliver on the promise of detailed comparison against other studies. As described earlier, the nature of this data set is rare and we make comparisons as well as we can to other works. We also highlight that the basis for such comparisons is far from perfect given a number of factors such as sampling format, time of study (late 20th century vs. early 21st century), and more. Given this rare opportunity with our new data however, we profit from it as much as possible and exploit it as much as possible, and compare against prior works as well as we can. This naturally becomes a major discussion point of the paper.

COMMENT: Line 314, 'results': Should be 'samples'. REPLY: We propose "sample" better than "samples".

COMMENT: Line 324, 'Both varieties G. ruber sensu stricto (s.s.) and sensu lato (s.l.)': Those are not varieties but distinctly different species, G. ruber (white) and G. elongatus respectively. Moreover, they are not even sister-taxa, but G. elongatus is the adelphotaxon to G. conglobatus. While they have comparable environmental preferences, and might thus be pooled for such an analysis as you intend to do, they should under no circumstances treated in a way that implies they are remotely the

same species. REPLY: We agree (see you major comment (4.)). The discussion is focused now on G. ruber (white), also some information about its difference with G. ruber (pink) is provided. G. elongatus is discarded for discussion and no pool with G. ruber is done anymore. Paragraph of lines 324-327 is deleted.

COMMENT: Lines 324–325, 'share similar habitats': Yet they have different environmental preferences, with G. elongatus living deeper (Steinke et al., 2005; Numberger et al., 2009) and showing different seasonality (Weinkauf et al., 2016). REPLY:We appreciate those references. That paragraph is now deleted from the article.

COMMENTS: Lines 331–332, 'as demonstrated by positive significant correlations with temperature in the G. ruber s.s. variety (p = 0:01).': Not that I would oppose this interpretation, but is is yet derived from inappropriate analytical techniques. _ Line 338, 'strong positive correlation with salinity (p = 0:01)': Derived from invalid methods! REPLY: See the answer to those questions in "major comment (5.)".

COMMENT: Lines 340–341, 'The findings of Watkins et al. (1996) are supported by the negative correlations of standing stocks': Are they? If Watkins was right, would you not expect no correlation at all between nutrient availability and abundance of G. ruber? Rather, it seems that G. ruber is faring less well in regions with more nutrients (if this trend is supported by proper statistical analyses, this is). This means that higher nutrient availabilities are negative for the species, maybe because it loses its competitive advantage against other species, or the higher nutrient concentration reduces light levels, thus hampering the photosymbiont activity. REPLY: We rephrase the sentence according to the PCA results conducted in the revised manuscript. The sentence will change as follows: "The findings of Watkins et al. (1996) are supported by our PCA results, where higher abundances are affine with low nutrient and fluorescence concentrations, with the exception of station 19 (REV Fig. 7e)."

COMMENT:Lines 341–342, 'G. ruber s.s. and fluorescence data of our study (p = 0:05).': Derived from invalid methods! REPLY: See the answer to those questions in

"major comment (5.)".

COMMENT: Lines 352–353, 'Hydrographic conditions and consequently food availability seem to be the factors limiting more its abundance once it has reached its habitable temperature range.': Yes, but is this not what would be expected? Liebig's law of the minimum is equally valid for protists and animals as it is for plants. REPLY: We are announcing what features limit its abundance (food availability determined by the lowered or enhanced stratification of the water column inside its temperature range). That sentence gives information of the environmental preferences of G. bulloides. We consider that no modification has to be done here.

COMMENT: Line 359, 'shows': Should be ', the species shows'. REPLY: Changed in the revised manuscript.

COMMENT:Line 368, 'positive correlation with fluorescence (p = 0:05),': Derived from invalid methods! REPLY: See the answer to those questions in "major comment (5.)".

COMMENT: Line 372, 'Raden et al., 2012': Should be 'van Raden et al., 2012'. REPLY: Changed in the revised manuscript.

COMMENT: Line 377, 'specie': Should be 'species'. REPLY: Changed in the revised manuscript. COMMENT: Line 383, 'opportunistic species': Opportunistic species are such species which can cope with highly unstable and/or unfavourable conditions better than other species can do. They thus massively dominate environments where few other species can live, resulting in very low diversities in those environments. This is often a transitional process until the environment becomes more stable/habitable, after which the opportunistic species are replaced by a more diverse community, because in developed environments they are at a competitive disadvantage to such species. I therefore do not believe that 'opportunistic' is the correct term to describe G. bulloides, which is cosmopolitan and often occurs in rather diverse assemblages. REPLY: We decide to maintain the term "opportunistic" as is used also in Rigual-Hernández et al. (2012), Schiebel and Hemleben (2005), Ottens (1992), among others.

COMMENTS: Line 384, 'It correlates with fluorescence peaks since it feeds on phytoplankton': Probably correct interpretation, but derived from invalid methods! _ Line 408, 'Its negative correlation with temperature (p = 0:01)': Derived from invalid methods! REPLY: See the answer to those questions in "major comment (5.)".

COMMENT: Line 417, 'only absent from': Should be 'being absent from only'. REPLY: Changed in the revised manuscript.

COMMENT:Line 428, 'even if this is not supported by our Pearson correlation.': Which is an inappropriate method anyways! REPLY: See the answer to those questions in "major comment (5.)".

COMMENT: Lines 438–439, 'The size-normalized weight (SNW) of tests of both G. ruber s.s. and G. bulloides are statistically significant': This statement is nonsensical, a value itself cannot be significant, it can only be significant in regard to a null hypothesis. I assume you refer to the fact reported in the Results section and Suppl. Fig. 4, that size and weight are not perfectly correlated in O. universa (otherwise I do not even know what you want to imply). However, as already mentioned above, this is in my opinion no prerequisite for the SNW to have a meaning. This is even leaving aside, that it is never established whether this relationship is really insignificant (p > :05) or if the R2 value is simply to small for the authors taste. REPLY: We appreciate the question of Referee #2 for noticing our mistake. We change the following sentence as follows: "The size-normalized weight (SNW) of tests of both G. ruber (white) and G. bulloides follow a systematic change from the Atlantic towards the eastern Mediterranean (Fig. 6)."

COMMENT:Line 439, 'follow a systematic change from the Atlantic towards the Eastern Mediterranean': This might be, but it was never properly tested or depicted graphically. REPLY: We consider Fig. 6 a proper way to show that pattern. With our PCA graphical representation that will be included on the revised manuscript also is noticeable that trend (REV Fig. 7).

COMMENT: Section '5.3.1 Unknown control of the SNW of O. universa': OK, now your

regression between shell size and shell weight makes more sense, and it would have been good to explain this in the beginning already. I do appreciate that you discuss this possibility of cryptic diversity and gametogenic calcite meddling with your data. However, what André et al (2014) detected are subtypes, they do not even rank on the species level. On that level you have also several subtypes in G. ruber and G. inflata. To be honest, it could be that the lack of strong correlation between size and weight in O. universa results from such an effect that the subtypes react differently. But it can still as well be, that this species simply reacts more heavily towards environmental factors concerning its calcification. I would thus not go so far as to categorically rule out that species for a calcification analysis, because you simply do not know what is the case here. It is still interesting to see SNW values for that species as well, although they might suffer from higher uncertainty. Even more so since despite a large spread, the correlation between size and weight does not seem to show bimodality (indicative for the cryptic species problem), and possibly the SNW data would not do so either. Conversely, the values seem to show a wider spread for larger shells, which can mean that gametogenic calcite is more of a problem, or simply truly that this species is more variable in calcification intensity (then presumably influenced by environmental factors). After thoroughly discussing why this might be a less reliable signal, I would therefore still want to see how SNW in O. universa scales with environmental factors. REPLY: See the density area of O. universa in Fig. 6 format (REV Fig. 1) in the answer of the question about lines 239-240. Also, O. universa density area results are considered in the new PCA. Here is the graphical representation of it (to see the environmental parameters loading the two factors see REV Fig. 7a .

COMMENT: Lines 441–443, 'In contrast, [...] environmental effects.': Incorrect! The strict correlation between size and weight may not exist, but this only means that especially in this species there must be other factors influencing calcification intensity. REPLY: We change the following sentence as follows: "In contrast, changes of the SNW of O. universa do not create any trend within locations (Figs. S2c, S3c, and S4c; Fig. 7j), and cannot be used to identify and quantify particular environmental effects."

COMMENT: Lines 448–449, 'Weight-area relation data do not show any statistically significant systematic distribution (Fig. S4c).': You probably mean 'correlation', not 'distribution'. REPLY: Changed in the revised manuscript.

COMMENT: Lines 453–454, 'their pore-size is also affected by environmental conditions including water temperature (e.g., Bé et al., 1973).': This statement is critical. Bé et al. (1973) did not know about different cryptic species. It might be that pore size is indeed influenced by environmental factors across all cryptic species, but also that cryptic species prefer different water temperatures and what Bé et al. (1973) interpreted as pore size changes within the species is simply the result of different species (with inherently different pore sizes) dominating different water masses. REPLY: It is true that we do not know if in cryptic species it will be like this without genetic studies; the citation of Bé et al. (1973) just contributes to the possibility of such a relation in cryptic species as well.

COMMENT: Line 476, 'nutrient concentration and food availability.': Which is basically the same thing in the context if this study, isn't it? REPLY: We agree, "nutrient concentration" eliminated from that sentence in the revised manuscript.

COMMENT: Lines 476–478, 'However, in contrast to O. universa, the SNW data of G. ruber and G. bulloides follow systematic distributions, which are statistically significant.': It is again not clear what you mean with 'distributions'. All data have a distribution, and values themselves cannot be significant or insignificant. I assume you refer to a significant correlation between size and weight in those species. REPLY: We agree. "distributions" removed by "correlations" in the revised manuscript.

COMMENT: Lines 478–480, 'High SNW in the Atlantic [...] also noticeable in Fig. S2d-e and Fig. S4d-e).': Those graphs are all not appropriate to show that. Rather, an actual crossplot between SNW and the individual environmental factors must be shown. Interestingly, this trend is reversed to what has been reported from the Azores Front (Weinkauf et al., 2016). REPLY: We consider our graphical representation appropriate

for the function it has.

COMMENT: Lines 480–482, 'At the same sites, [...] interpretation of the data (Fig. 6).': Which could be shown effectively by calculating and presenting the coefficient of variation at those stations. Additionally, how could this trend then be interpreted? REPLY: Figure 6 is simply a box plot comparison that yields information on how population statistics differ across regions. There is no correlation among properties here and therefore no "coefficient of variation" to present on a station-specific basis, as far as we understand the reviewer comment. We currently fail to understand what calculation method the reviewer is suggesting here, and wonder also if it is beyond the scope of this study.

COMMENT: Lines 485–486, 'The relationship between food availability and SNW in G. bulloides is opposite to that in G. ruber s.s. (Fig. 6)': A better figure is needed to illustrate this. REPLY: We consider our graphical representation appropriate for the function it has. In the revised manuscript, we add the figure of the overlaid density area results of both species: shown in the answer to the question about line 439.

COMMENT: Lines 488–489, 'In both species G. ruber s.s. and G. bulloides larger IQRs are found toward higher absolute SNW.': Which is perfectly normal stochastic behaviour. This is why it is important to normalize variation for expected value by reporting the coefficient of variation instead of raw variation under such circumstances. REPLY: As also described above, in our comment to the reviewer comment about lines 480-482, we are unsure about what statistical method and / or calculation the reviewer is referring to here. Is there a distinct suggestion of some kind, with a reference? We are not sure how to calculate a "coefficient of variation" with regard to box plots and their statistics.

COMMENT: Lines 490–492, 'An opposite trend in SNW [...] growth conditions.': I assume this refers to Beer et al. (2010b). Please cite your sources properly. REPLY: We add the following citation to the revised manuscript: Beer et al. (2010a): Beer, J.,

Schiebel, R., and Wilson, P. A.: Testing planktic foraminiferal shell weight as a surface water [CO32-] proxy using plankton net samples, Geol. Soc. Am., 38, 103-106, 2010a. COMMENT: Line 494, 'Köhler-Rink and Kühl, 2005': This citation is missing in the list of references. REPLY: We agree. Done.

COMMENT: Lines 496–497, 'additional calcite layers might be added to the proximal text surface before reproduction, similar to the process described for O. universa (see above).': Yet to my knowledge, those two species are not known for excessive amounts of gametogenic calcite (e.g. Deuser, 1987; Hamilton et al., 2008). Also, the alternative interpretation would be that more optimal conditions trigger faster growth and earlier reproduction, resulting in a trade-off for calcification intensity of each individual chamber already during growth (i.e. before gametogenic calcite is added). Additionaly, 'text' should be 'test' (Line 496) REPLY: It has not necessarily to be an excessive calcite addition, just enough to be detected. We appreciate your interpretation here. "Text" substituted by "test" in the revised manuscript.

COMMENT: Lines 505–506, 'However, the comparison might be biased by the fact that G. ruber s.s. and s.l. morphotypes were analyzed together in the study of de Moel et al. (2009).': It most certainly is. Compare Weinkauf et al. (2016). REPLY: Weinkauf et al. (2016) will be taken in to account for the useful density area results that it presents.

COMMENT: Lines 514–516, 'All of these [...] in an increased SNW': They also support the interpretation, that a multitude of factors influences shell calcification in planktonic Foraminifera. REPLY: We agree with that statement. In oligotrophic regions, like the Mediterranean Sea, planktic foraminifera calcification is affected by a combination of factors like carbonate saturation and food availability (Beer et al., 2010a; de Villiers et al., 2004).

COMMENT: Line 517, 'given that carbonate chemistry does not limit calcite formation in planktic foraminifera.': This is a blatant misrepresentation of basically the entirety of existing literature (compare Marshall et al. (2013, tab. 1) and Weinkauf et

al. (2016, tab. 7)). REPLY: We will clarify this point raised by the reviewer. In fact the overall conclusion of the paper is not that seawater carbonate chemistry cannot be a key driver for foraminifera calcification. The results of this study are related to the modern Mediterranean conditions where pH and [CO32-] are relatively high, well above the carbonate saturation, compared to the critical values tested in ocean acidification experiments and other oceanographic settings. The pH in the upper 200 meters is ranging from 8.047 (St.1) to 8.126 (St.20) and the [CO32-] 178.88 $\mu$mol Kg-1(St.1) to 243.560 $\mu$mol Kg-1 (St.11). The Mediterranean Sea is an oligotrophic to ultra-oligotrophic environment having a strong physical and biogeochemical gradient from the Atlantic to the Eastern Mediterranean (Fig. 1; Fig. 2; MEDAR: http://modb.oce.ulg.ac.be/backup/medar/medar_med_phph_spring.html; Touratier et al., 2012: http://images.slideplayer.com/31/9579232/slides/slide_2.jpg). A main point of the paper is to show that since the seawater carbonate saturation at the studied sites is negligible compared to other oceanic regions, the effect of parameters other than carbonate saturation could be detected as observed in other studies (e.g. Weinkauf et al., 2016). We conclude that planktic foraminifera calcification in the modern Mediterranean Sea is likely more affected by factors other than carbonate saturation. In oligotrophic regions, food availability can be critical for the fitness and growth conditions since there is the hypothesis that food availability can free more energy for calcification (Beer et al., 2010a; de Villiers et al., 2004; Horigome et al., 2012).

G. ruber (white) is dominant in the eastern basin, whereas G. bulloides show its dominance in the western basin, accentuating more the differences in food availability for both species. Our conclusions also might work in similar highly oligotrophic areas. Our conclusions do not exclude that in a future with the ongoing accelerating emission of anthropogenic carbon and its uptake by the Mediterranean surface sea carbonate chemistry will have a major effect on the SNW of planktic foraminifera, even if this is of relatively low influence today.

COMMENT: Line 522, 'reflect high': Should be 'show large'. REPLY: Changed in the

revised manuscript.

COMMENT: Line 526, 'ten morphospecies in total.': This is wrong since at least the individual species G. ruber (white), G. ruber (pink), and G. elongatus have been pooled together. Furthermore, it is unclear whether G. calida and G. radians also occur and have been pooled into G. siphonifera. REPLY: See the answer to the question about line 287, from the Discussion section. We will change it as well in the Conclusions section of the revised manuscript.

COMMENT: Line 548, 'These observations highlight the need for more interdisciplinary studies on the causes of changing foraminiferal assemblages and decreasing shell production': If this is supposed to hint at the promised comparison with earlier studies then I must state again that 1) since you used a larger mesh size without correcting your data for that fact you cannot compare your abundances with those of earlier studies and 2) you never presented a thorough discussion whether species compositions have been significantly changing during the last 20 years and if so, why. REPLY: See the answer to your question about lines 300–302 for point 1). See the answer to your question about lines 97–98 for point 2).

COMMENT: Lines 588–589: There is no Bijma et al., 1990a, so remove the 'b' after the year. REPLY: Changed in the revised manuscript.

COMMENT: Lines 625–626: Ivanova et al. (2003) is not cited anywhere in the manuscript. Remove from list of references. REPLY: Changed in the revised manuscript.

COMMENT: Lines 650–651: 'Grazzini' should be 'Vergnaud Grazzini'. REPLY: Changed in the revised manuscript.

COMMENT: Lines 682–683: 'Orbulina universa' should be set in italics. REPLY: Changed in the revised manuscript.

COMMENT: Caption Fig. 1, '(a) Temperature (°C), (b) salinity, (c) fluorescence (_g _

lôĂĂĂ1), (d) pH, and (e) [CO2ôĂĂĂ3 ] (_mol _ kgôĂĂĂ1): Information where these data come from are missing completely. Additionally, the software used for plotting (I assume Ocean Data View, Schlitzer (2014)) has not been cited. Especially Section 2, and to a lesser extant Section 3 involves a huge amount of interpolation due to the large spatial distance between measurement profiles. This makes the reconstructions very unreliable. REPLY: The source of our data is solved on the methodology section of the revised manuscript; see the answer to "major comment (2.)", where it is specified. Fig. 1 software source will be cited on the figure legend with Schlitzer (2016): Schlitzer, R.: Ocean Data View, http://odv.awi.de,2016. We consider that the aim of Fig. 1 to show the environmental parameters of the Mediterranean Sea is suitable, even if local hydrographic features are not presented here.

COMMENT: Caption Fig. 2, 'First leg: 1 to 13, second leg: 14 to 22.': It might be nice to distinguish the cruise-tracks of the two legs by colour. REPLY: The two legs represents two different lines in Fig. 2, which is why we consider it unnecessary to distinguish by colour.

COMMENT: Caption Fig. 2, 'MODIS Aqua (L2),': What is this? This source has not been cited in any way (published article, url, ...) and was not mentioned in the Material and Methods section. REPLY: MODIS (Moderate Resolution Imaging Spectroradiometer) Aqua L2 is a NASA Satellite that view the Earth's water surface to acquire data to understand global processes and dynamics. In the new methodology we mention it; see the answer to your question in major comments (2.). The reference is: NASA Goddard Space Flight Center, Ocean Ecology Laboratory, Ocean Biology Processing Group; (2013): MODIS-Aqua L2 Data; NASA Goddard Space Flight Center, Ocean Ecology Laboratory, Ocean Biology Processing Group. http://oceandata.sci.gsfc.nasa.gov/. Accessed on 06/06/2013.

COMMENT: Caption Fig. 2, 'from the closest day as possible': Which means exactly what? 1 day, 10 days, 100 days,...? Also, I would have assumed the dates given in the map are the dates for which chlorophyll a data have been plotted, or what else is

displayed there? REPLY: Chlorophyll a data was measured the day that the satellite image was available, noted in the upper part of the figure. The reader can see the days that separate the towing and the chlorophyll a values looking Table 1: i.e. Stations 1, 2, 3 were sampled days 3rd to 5th (Table 1) and the satellite image and its corresponding Chlorophyll a values are from day 5th (Fig. 2). We clarify Fig. 2 legend to avoid confusion as follows: "
[revised manuscript text omitted]

a)

b)

c) *G. inflata*

d) *T. sacculifer* (without sac)

e) *G. ruber* (white)

f) *G. bulloides*

g) *O. universa*

h) *G. ruber* (white)

i) *G. bulloides*

j) *O. universa*

REV Figure 7: Sample scores on the two PCA factors with (a) the loadings of the enviromental parameters on each factor, (b) with overlay of the absolute abundance values (individuals•10 m-3) of every station of all the foraminifera sample, (c) *G. inflata*, (d) *T. sacculifer* (without sac), (e) *G. ruber* (white), (f) *G. bulloides*, and (g) *O. universa*. With overlay of the ρA values (µg•µm-2) of (h) *G. ruber* (white), (i) *G. bulloides*, and (j) *O. universa*. In blue color western Mediterranean stations (incl. Atlantic and Strait of Gibraltar), in red colour the eastern Mediterranean stations.

**Fig. 1.**

[Figure]

[Figure]

REV Figure 3: Absolute abundance of planktic foraminifera from BONGO nets during leg 1 (stations 1 to 13) and leg 2 (stations 22 to 14). Category 'Others' is comprised by *G. siphonifera*/*G. calida*/ *G. radians*, *G. quadrilobatus*, *H. pelagica*, *G. ruber* (pink), *G. menardii* and *T. sacculifer* (with sac).

**Fig. 2.**

[Figure]

REV Figure 4: Percentage of each planktic foraminifera size fraction in leg 1 (stations 1 to 13) and leg 2 (stations 22 to 14).

**Fig. 3.**

[Figure]

REP Figure 1: Average density area for *O. universa* according to the different sub-regions of the Mediterranean Sea.

**Fig. 4.**

---

## Author Response (AR1)

**Low planktic foraminiferal diversity and abundance observed in a spring 2013 West-East Mediterranean Sea plankton tow transect**

3 Miguel Mallo1, Patrizia Ziveri1,2, P. Graham Mortyn1,3, Ralf Schiebel4 and Michael Grelaud1

4 5 1. Institute of Environmental Science and Technology (ICTA), Autonomous University of Barcelona (UAB), Bellaterra 08193, Spain

6

 Catalan Institution for Research and Advanced Studies (ICREA), Pg. Lluís Companys 23, 08010 Barcelona, Spain

8 3. Geography Department, UAB, Bellaterra 08193, Spain

9 10  LPG-BIAF, University of Angers, 2 bd Lavoisier, 49045 Angers, France, now at Climate Geochemistry, Max Planck Institute for Chemistry, Hahn-Meitner-Weg 1, 55128 Mainz, Germany

12

11

**13 Abstract**

14 Planktic foraminifera were collected with 150 µm BONGO nets from the upper 200 m water depth at 20 15 stations across the Mediterranean Sea between 02 May and 02 June, 2013. The main aim was to characterize the species distribution and the size-normalized shell weight area density ( $\rho_A SNW$ ). Average 16 for a bundances and diversity are  $1.42 \pm 1.43$  ind. 10 m-3 (ranging from 0.11 to 5.20 ind. 10 m-3). 17 with ten-a total of twelve<del>overall</del> morphospecies found, respectively. Large differences in species 18 19 assemblages and absolute abundances values are observed between the different Mediterranean sub-20 basins, with an overall dominance of spinose, symbiont-bearing species indicating oligotrophic 21 conditions. The highest values in absolute abundance are-were found in the Strait of Gibraltar and the 22 Alboran Sea. The western basin is dominated by Globorotalia inflata and Globigerina bulloides at 23 slightly lower standing stocks than in the eastern basin. In contrast, the planktic foraminiferal assemblage 24 in the warmer, saltier and more nutrient-limited eastern basin is dominated by Globigerinoides ruber 25 sensu stricto (s.s.)(white). These new collective results in combination with comparison to previous 26 findings, suggest that temperature-induced stratification of the surface water column, nutrient 27 concentration and hence food availability, and temperature seem to be the main factors controlling foraminiferal abundances and distribution. In the highly alkaline and supersaturated with respect to calcite 28 and aragonite Mediterranean surface water, sStanding stocks and size normalized weight ( $\rho_A$ SNW) of G. 29 30 ruber s.s.(white) and G. bulloides seem moreare related to affected by food availability than-and only 31 secondarily by seawater carbonate chemistry. Increasing temperature, salinity, surface ocean stratification 32 and trophic conditions could be the causes of reduced abundance, diversity and species-specific changes 33 in calcification in planktic foraminifera.

**35 **1. Introduction**

36 The single-celled foraminifera comprise the most diverse group of calcareous zooplankton of the modern 37 ocean. The majority of foraminifer species are benthic. About 50 morphospecies are planktic, which have a calcareous exoskeleton organized in chambers (ie.eg., d'Orbigny, 1826; Hemlbeblen et al., 1989; 38 39 Goldstein, 1999). The-species are adapted to from different environments can be characterized by due to 40 differences in wall structure, pores size and spatial density, spines and test shape, which are partly related 41 to adaptation. The distribution of foraminifera is thought to be influenced by food availability, 42 temperature, salinity, turbidity, radiationsunlight, and predatory presence; these factors provoke an overall 43 water depth preference, which shifts during ontogeny, and seasonal preference for each species, which 44 shifts during ontogeny (Schiebel and Hemleben, 2005; Hemleben et al., 1989). Some of them are found 45 only in the photic zone because they are symbiont-bearing and depend on light for photosynthesis. After 46 reproduction, the empty shells sink to the seafloor, where their fossils are useful for paleoceanographic 47 studies (e.g., Shackleton, 1968; Rohling et al., 2004; Mojtahid et al., 2015). Ecological tolerance limits of 48 modern foraminifera are not completely defined, but progressive reduction in abundance (caused by 49 worsening of their organic functions like nutrient-food uptake, growth and reproduction, until death) is related with their departure from optimum conditions (Bé, 1977; Arnold and Parker, 1999). The absolute 50 51 abundance of foraminifera is also affected by a predictable and distinct seasonal cycle for each species 52 driven by the food source content of the watermass (Hemleben, 1989; Bé and Tolderlund, 1971; for 53 Mediterranean examples see: Pujol and Vergraud-Grazzini, 1995; Bárcena et al., 2004; Hernández-54 Almeida et al., 2011; Rigual-Hernández et al., 2012; de Castro Coppa et al., 1980).

55 The A vast majority of studies on planktic foraminifera globally are based on samples from bottom sediments and sediment cores, mainly for paleoceanographic purposes, with few studies considering the 56 57 modern population in the water column-foraminifera, including the Mediterranean Sea. The first modern 58 study of planktic foraminifera in this specific areathe Mediterranean was based on surface sediment 59 samples collected by the Swedish Deep-Sea expedition of 1947-48 (Pettersson, 1953). A subsequent 60 study found different species assemblages between the western basin, the eastern basin, and the Aegean 61 Sea (Parker, 1955). The pioneering study of foraminifera population variability in the water column 62 for a minifera in of the Mediterranean was achieved conducted by Glacon et al. (1971) in the Ligurian Sea, 63 showing high-large seasonal variations of the relative abundances variations of the different species 64 throughout the seasons. Such variations of planktic foraminiferal assemblages in the water column were 65 also reported for the Bay of Naples (de Castro Coppa et al., 1980). Cifelli (1974) was the first to cover the 66 broader Mediterranean, with plankton tows at-of the upper 250-m depth-of the water column from west 67 Madeira to the Isle of Rhodes in June 1969; they identified prominent differenced relative abundances of 68 subtropical and subpolar species in different parts of the Mediterranean.

69 Thunell (1978) studied samples from the upper 2 cm of sediment cores covering retrieved in different

70 sites of the Mediterranean Sea and concluded, concluding that the distribution of planktic foraminifera is

71 was closely linked related to with the distribution of the different surface water masses, with There are

72 specific temperature and salinity ranges for each of themwater mass, as Bé and Tolderlund (1971) stated 73 for the Atlantic, helped by theand a partial isolation effect of in the different basins and sub-basins inside 74 of the Mediterranean. Those phenomena hydrographis differences result in different species assemblages in each region. This contradicts somewhat with Pujol and Vergraud-Grazzini (1995), who gained 75 76 quantitative data with flow-metered plankton tows in the upper 350 m of the water column, through a 77 NW-SE Mediterranean transect from September-October 1986 and February 1988, and the Alboran Sea 78 in April 1990. They concluded that despite the W-E temperature and salinity gradients observed, those 79 were not wide large enough and no close correlation was found to justify the extremely variable 80 foraminifera assemblages, with high seasonal and geographical variations in absolute and relative 81 abundances. They suggested that food availability is the main factor controlling their seasonal and 82 geographical distribution and abundance; and when nutrients are sufficient, hydrographic structures like 83 eddies and fronts are the ones that play the main role.

84 Despite no new plankton tow study being carried out in the entire Mediterranean Sea, three regional 85 studies based on sediment traps were realized in the Alboran Sea (Bárcena et al., 2004; Hernández-86 Almeida et al., 2011) and the Gulf of Lion (Rigual-Hernández et al., 2012). The one-year time series of 87 the Alboran Sea sediment traps (July 1997 - May 1998) showed big differences in the main species 88 distribution and daily fluxes, driven by food availability (related with water mixing/stratification periods) 89 and temperature (Bárcena et al., 2004; Hernández-Almeida et al., 2011). The 12-year sediment trap 90 records in the Gulf of Lion (October 1993 - January 2006) showed a strong seasonal pattern of the species, with more than 80% of the abundances from winter and spring in correlation with the nutrient 91 92 supply and mixed water column conditions (Rigual-Hernández et al., 2012).

93 The calcification of foraminifera is affected by the chemical state of their surrounding waters. Theoretically, their shell massIts weight is positively related to temperature, pH,  $[Ca^{+2\pm}]$ , and alkalinity 94 and  $[CO_3^{2-}]$  from its ambient water and negatively related with to the  $[CO_2]$  of the surrounding waters 95 (Schiebel and Hemleben, 2005). Different studies conducted on water column foraminifera show 96 97 differential results, as their shell mass can either be positively (Aldridge et al., 2012; Beer et al., 2010a; 98 Marshall et al., 2013; Moy et al., 2009) but also negatively related to [CO2] (Beer et al. 2010a). Also, 99 other studies report a positive effect of the temperature on foraminifera shell mass (Mohan et al. 2015; Aldridge et al., 2012; Marshall et al., 2013; Weinkauf et al., 2016). Beer et al. (2010a) discussed the 100 101 <del>positive</del>suggest a species-specific relation between weight shell mass and  $[CO_3^{-2}]$ , depending on the presence or absence of symbionts. suggesting that it is not a significant parameter for calcification. It 102 103 seems-Some authors suggest that biotic-other factors like ecological stress do not affect the calcification 104 intensity (Weinkauf et al., 2013). For further studies that relate foraminiferal calcification with 105 environmental parameters see Weinkauf et al. (2016); Table 7. Since From the onset of the industrial era, 106 anthropogenic emissions of CO2 have led to ocean acidification, decreasing its seawater pH and [CO3-2-], 107 which provokes reduced stability of CaCO3 that may obstruct reduce the building formation of 108 foraminiferal tests (Zeebe, 2012; de Moel et al., 2009; Moy et al., 2009).

Mediterranean studies of Studies of the water columnecology of foraminifera in the Mediterranean waters
 and accurate knowledge of its different species ecology remain scarce. Few studies exist covering the

111 entire Mediterranean Sea; most are focused at-on specific regions, ie.eg., the Tyrrhenian BasinGulf of 112 Naples (de Castro Coppa et al., 1980), the Alboran Sea plus the southwestern Mediterranean (van Raden 113 et al., 2011), among others. Data on live living planktic foraminiferal abundances were provided by Cifelli (1974; spring only) and more recently by Pujol and Vergraud-Grazzini (1995). In addition, few 114 115 size-normalized weight (SNW) and area density  $(\rho_{\Lambda})$  studies from water column foraminifera are 116 available in the literature (see Schiebel et al., 2007; Beer et al., 2010a; Aldridge et al., 2012; Marshall et 117 al., 2013; Mohan et al., 2015; Marshall et al., 2015; Weinkauf et al., 2016).-New data are needed, since 118 environmental conditions of the water column and associated foraminiferal assemblages might have 119 changed over the past 20 years.

120 In this paperstudy, new quantitative and qualitative data are presented on living planktic foraminifera, 121 across the Mediterranean Sea during springMay 2013. Comparisons are made with older similar previous 122 studies from Pujol and Vergraud-Grazzini (1995), Cifelli (1974), de Castro Coppa et al. (1980), Bárcena 123 et al. (2004), Hernández-Almeida et al. (2011), Rigual-Hernández et al. (2012) and Thunell (1978). The 124 study by Thunell (1978) is based on surface sediments, which can provide information, which but might 125 be biased towards faster-sinking and more hydrodynamic tests due to shorter exposureition to dissolution 126 processes (Caromel et. al., 2014; Schiebel et al., 2007), and towards more unbreakable tests with thicker 127 walls that are better preserved (Thunell, 1978). Although core top samples (0-2 cm) are suitable to infer 128 modern variability (Thunell; 1978), they can cover the last few decades to few centuries, depending on 129 the sedimentation rate, while our plankton tow sampling represents a relative "snap shot" (Mortyn and 130 Charles, 2003). In additional, empty tests are passive particles that ocean currents may displace 131 horizontally, but that displacement is negligible due to their quick settling velocities (Caromel et al., 132 2014). On the other hand, average drift distances of foraminiferal test are estimated to be less than 10 km 133 in the Mediterranean (Sebille et al., 2015), and Ceorrelated results between plankton tows (Pujol and 134 Vergraud-Grazzini, 1995) and surface sediments (Vergraud-Grazzini et al., 1986) at coincident places 135 inside the Mediterranean confirm the data-results obtained byof Thunell (1978).

136 The objectives here are to (1) delineate new absolute abundances data of spring planktic foraminifera 137 within the different regions of the Mediterranean Sea\_during spring, (2) characterize, at the species level 138 their ecology through their seasonal and geographical distribution and abundance by comparison with 139 previous studies, and (3) contribute provide new  $\rho_A$ SNW data for comparisons between basins and with 140 other studies from the literature in the context of ocean warming and acidification over the past 20 to 40 141 years.

142

**143 **2. Oceanographic Setting**

[revised manuscript text omitted]

229Selected foraminifera for the SNW study were photographed with a Canon EOS 650 D camera device230attached to a Leica Z16 AP0 microscope to measure their diameter and silhouette area. Individuals of the231same station and species within a 50  $\mu$ m diameter size constraint were weighed with a Mettler Toledo232XS3DU microbalance (±1  $\mu$ g of error). A Pearson correlation test was applied to study the relation of233foraminiferal SNW and relative abundance with temperature, salinity, pH, [CO3-2] and fluorescence.234Absolute abundances were discarded for Pearson test as the magnitude of the variability observed235between each station was much higher than the variability observed within the environmental parameters.

Foraminiferal samples were collected either at daytime or nighttime. Individuals were not necessarily
 alive when collected and no distinction was made between eytoplasm-bearing tests: alive or dead but still
 containing cytoplasm (see also Boltovskoy and Lena, 1970) and empty tests (dead). Cytoplasm-filled or
 empty dry-weighed foraminifera tests were weighted together since dry cytoplasm has no statistically
 significant effect on the weight of tests >150 µm (Schiebel et al., 2007). Unclassified specimens are not
 included in the test-size analyses presented in the following.

**242 3.3. Statistical methods**

[revised manuscript text omitted]

338 eastern Mediterranean stations were dominanted by species (G. ruber s.s.(white), at the Tyrrhenian Sea 339 and the eastern Mediterranean; the Alboran Sea by G. inflata-at the Alboran Sea), whereas The 340 dominance of a single species in the southwestern Mediterranean the dominance is less clear, which 341 might be due to a low numbers of individuals (G. inflata being the main species followed by G. bulloides as at in the Alboran Sea station). GT. sacculifer type trilobus(without sac) has a high relative abundance 342 343 in the Atlantic Ocean and in the Strait of Gibraltar, -(being the main and the second most abundant 344 species, respectively. At all other stations analyzed,); elsewhere T. sacculifer (without sac) it is less 345 abundant. G. bulloides is most frequent in the entire western Basin and the Atlantic Ocean, being the 346 main species at in the Strait of Gibraltar. It is less frequent in the Tyrrhenian Sea, and in the eastern Basin 347 and its sub-basins. G. bulloides contrasts with G. ruber s.s.(white), which always represents a small 348 percentage in the western Mediterranean but dominates the Tyrrhenian Sea and the eastern Basin (Fig. 5; 349 Appendix A).

A Pearson test was applied to the main species to see their relative abundance correlation with 350 temperature, salinity, and fluorescence. The correlations found are: G. ruber s.s. is positively correlated 351 with temperature and salinity (p = 0.01), and negatively with fluorescence (p = 0.05). G. inflata is 352 353 positively correlated with fluorescence (p = 0.05) and G. bulloides has a negative correlation with temperature (p = 0.01). Relative abundance was selected instead of absolute abundance to avoid bias due 354 355 to the big differences between stations' results in absolute abundance. The remaining species did not pass 356 through a Pearson test as they are not present in all the stations, which makes it difficult to assess a 357 relation between abundance and environmental parameters.

358 359

**4. 2. Size-normalized weightArea density (ρASNW)**

360 Due to their abundance, G. ruber s.s. (white), G. bulloides, and O. universa where analyzed for their size-361 normalized weight (SNW) area density ( $\rho_A$ ; Fig. 6; Fig. 7g-i). The high two-dimensional (silhouette) area-362 to-diameterlong axis- correlation is best fitted by a power regression (Fig. S2). The same growth pattern 363 can be seen in G. ruber (white), G. bulloides, and O. universa with that correlation, represented graphically in the shape of a power function (Fig. S2). They grow slightly faster when they are younger 364 365 and smaller (steepest in the lower left part of the regression line) and slightly slower when they grow older and bigger (less steep in the upper right part of the regression line; Fig. S2). Comparing the average 366 367 values from different locations sampled within the Mediterranean, The specimens of G. ruber s.s. (white) 368 individuals from the Atlantic have the largest size followed by individuals from the Tyrrhenian Sea, and 369 tests those from the ecastern-of the Ionian SeaStrait of Sicily. For the other two species G. bulloides and 370 O. universa, the results are statistically not significant, but a similar trend is observed regarding the two 371 basins, with the eastern Mediterranean having the smallest individuals, while the largest individuals 372 occurred in the Atlantic and the northwestern Mediterranean (Fig. S2). The different locations were 373 grouped using the same criteria as in Fig. 5.

374 The diameterlong axis-to-weight relation of *G. ruber* s.s.(white) specimens yielded an  $r^2 = 0.8412$  (linear 375 regression throughout this paragraph; Fig. S3), followed by *O. universa* ( $r^2 = 0.630$ ), and *G. bulloides* ( $r^2$

- 376 = 0.516; Fig. S3). *O. universa* was finally discarded for comparisons between  $p_A$ SNWs at different 377 locations due to a low area-weight correlation and no remarkable trend observable between locations (Fig. 378 S4c; Fig. 7i);, while data from *G. ruber* s.s.(white) correlate well (Fig. S4a). The eastern Mediterranean 379 specimens are the lightest for both species (*G. ruber* s.s.(white), *G. bulloides*), with more extreme W-E 380 differences for *G. ruber* s.s.(white) (Fig. S4d-e).
- The  $\rho_A$  of G. ruber s.s. (white) specimens from six locations were compared in a SNW study (Fig. 6). The 381 eastern Mediterranean individuals have the lowest median  $\rho_A SNW$  (approximately between 7.5·10-5 and 382  $9 \cdot 10^{-5} \,\mu\text{g}\cdot\mu\text{m}^{-2}$ ), with lower values eastward, and a small interquartile range (IQR = Q3 – Q1). The Atlantic 383 individuals of G. ruber s.s. (white) show the highest median value  $(1.55 \cdot 10^{-4} \,\mu g \cdot \mu m^{-2})$  and IQR. The 384  $\rho_{A}$ SNW of Tyrrhenian individuals ranges between those from the eastern Mediterranean and Atlantic 385 Ocean (1.2·10-4 µg·µm-2). A Pearson correlation test was done to assess the correlations between the SNW 386 values and the environmental parameters of the section above plus pH and [CO3-2]. In the Pearson 387 correlation test, G. ruber s.s. shows a negative correlation with salinity,  $[CO_3^{-2}]$  (p = 0.01), pH (p = 0.05) 388 and a positive correlation with fluorescence (p = 0.01). The  $\rho_A$  of G. ruber (white) for each station was 389 compared with the two PCA factors; higher pA are related to slightly lower pH and higher food 390 391 availability in the western Mediterranean and Atlantic stations (Fig. 7g).
- 392 For G. bulloides specimens, seven locations were compared (Fig. 6). The Atlantic has the lowest median  $\rho_{A}$ SNW (8.75·10-5 µg·µm-2) and the smallest IQR, showing an opposite trend as in G. ruber s.s. (white). 393 Also contrary to G. ruber s.s. (white) individuals, the G. bulloides from the eastern Mediterranean 394 individuals yield the highest median tend to have a higher median  $\rho_A \frac{\text{SNW}}{\text{SNW}}$  (9.75·10-5 µg·µm-2) and a larger 395 IOR. The differences in  $\rho_{A}$ SNW between the eastern and western Mediterranean are smaller in G. 396 *bulloides* than in *G. ruber* s.s.(white). *G. bulloides* is positively correlated with pH and  $[CO_3^{-2}]$  (p = 0.05) 397 in the Pearson test. The  $\rho_A$  of G. bulloides at each station was compared with the two PCA factors. 398 Results show a less clear overall trend for G. bulloides than for G. ruber (white), with the higher  $\rho_A$ 399 associated with slightly higher pH in the eastern Mediterranean sea-water (Fig. 7h). 400
- 401

**403 5. Discussion**

404

**405 5. 1. Abundance and diversity patterns**

406 The aAbsolute abundance values of up to 4.2 individuals per 10 m-3 ( $\geq$ 150 µm) on average are low in 407 comparison with other water column foraminifera studies found in the literature, even for oligotrophic 408 regions. For example, in the oligotrophic northern Red Sea, less than 100 ind.·10 m-3 (>125 µm) were\_-not 409 reported from surface waters, and standing stocks were much higher than 100 ind.·10 m-3 at most of the 410 sites sampled in 1984 and 1985 (Auras-Schudnagies et al., 1989). In the oligotrophic to mesotrophic

Caribbean and Sargasso Seas, standing stocks were up to 786 ind. 10 m-3 (>100 µm) and 907 ind. 10 m-3 411 (>202 µm), respectively (Schmuker and Schiebel, 2002, and references therein). In the more proximal 412 413 Atlantic, south of the Azores Islands, Schiebel et al. (2002) counted an average of 66.15 ind. 10 m-3 for the upper 100 m in August 1997, and 422.97 ind. 10 m-3 in January 1999 (>100 µm). Other similar studies 414 continue to show higher abundances of results with one or two orders of magnitude higher abundance 415 416 (ie.eg. Sousa et al., 2014; Boltovskoy et al., 2000; Kuroyanagi and Kawahata, 2004; Rao et al., 1991; 417 Ottens, 1992; Schiebel et al., 1995). At higher latitudes, in the Fram Strait (Arctic SeaOcean), Pados and Spielhagen (2014) obtained approximate values of  $117 \pm 74$ -ind.  $10 \text{ m}^{-3}$  from the upper 500 m in late June-418 early July of 2011. Mortyn and Charles (2003), in February-March 1996, at 200 m depth range in the 419 Atlantic sector of the Southern Ocean, found as a minimum value 0.1 ind. 10 m-3, with an approximate 420 421 mean of 73  $\pm 160$  ind.  $\cdot 10 \text{ m}^{-3}$ .

422 Within the Mediterranean, a previous study with results comparable to ours, results sampled the upper 350 423 m (Pujol and GrazziniPujol and Vergraud-Grazzini, 1995). For the Alboran Sea, samples were obtained 424 duringat a similar period time of the year (April 1990) with values around 16, 6 and 9 ind. 10 m-3, greater than in theour Station 3 with (4.14 ind. 10 m-3). The rest of their sSamples occurs in a from different 425 426 seasons of the year and also have notably higher abundances, with highest values in with larger ones in 427 February (Pujol and Vergraud-Grazzini, 1995), and a high annual average of than during September-428 October. Their sampling mean is also higher and approximates to 9.3  $\pm$ 8.9 ind. 10 m-3. Regarding Pujol 429 and GrazziniPujol and Vergraud-Grazzini (1995), western Mediterranean abundances are higher than the 430 eastern ones-overall, due to more eastern oligotrophic conditions and higher temperature and salinity 431 salinities values in the east that limit for a production both, during winter and late summer. In concordance with Pujol and GrazziniPujol and Vergraud-Grazzini (1995), no significant-differences are 432 433 observed between samples collected during day and night.

Ten different species are recognized in our study, accounting for a single species (to have comparable 434 results with previous studies) the three varieties of G. ruber: sensu stricto, sensu lato (containing different 435 ervptic species; Aurahs et al., 2009a), and the pink variety. To facilitate comparison, the different G. 436 437 sacculifer morphotypes trilobus and quadrocameratus are here treated separately, despite belonging to the same genotype (André et al., 2013). Our findings contrast with previous studies covering the 438 Mediterranean, where more species were found: 18 species with Cifelli (1974), and 17 species with Pujol 439 and Grazzini (1995) and with the surface sediments of Thunell (1978). Some of the species not found 440 441 reach high frequencies in the aforementioned studies: e.g., Turborotalita quinqueloba, Neogloboquadrina 442 pachyderma, and Globorotalia truncatulinoides. The fact that these species were not sampled in the present study may be caused by their absence or presence at extremely low abundances of adult 443 specimens at the sampled stations in May 2013. G. sacculifer type quadrocameratus was not found in 444 445 previous studies working with plankton tows in the Mediterranean, despite its abundance in sedimentary 446 cores (i.e. Živkovic and Glumac, 2007).

447 Comparing with previous studies that covered the Mediterranean, we notice that Thunell (1978) and Pujol
448 and Vergraud-Grazzini (1995) did not find *G. menardii*, while it was reported by Cifelli (1974) in very

449 low abundances. The fact that G. menardii, which has a preference for tropical waters, is not found in the 450 surface sediments suggests that it is a new species in the Mediterranean Sea (Cifelli, 1974). Its recent 451 presence in the Mediterranean Sea could be related to the warming of the waters. All other species found in our study were also found in the past studies covering the Mediterranean Sea (Cifelli, 1974; Thunell, 452 453 1978; Pujol and Vergraud-Grazzini, 1995). It remains unclear whether Pujol and Vergraud-Grazzini 454 (1998) found G. falconensis and classified it with G. bulloides, or if Thunell (1978) found G. elongatus 455 and T. sacculifer (without sac) and classified them as G. ruber and G. sacculifer, respectively. Also, it is 456 not certain if Cifelli (1974) found G. calida and classified it with G. aequilateralis (older synonym of G. 457 siphonifera). From the figures in Cifelli (1974), we suspect that G. elongatus was classified as G. ruber. 458 In the same way, we do not find any evidence of T. sacculifer (with sac) from the figures presented by 459 Cifelli (1974), but we cannot discard the possibility that this species was classified as Globigerinoides 460 trilobus (T. sacculifer without sac).

461 462

463

464

465

466

467 468

469

470

471

472

473

474

Globigerinoides quadrilobatus was not found in any previous plankton tow studies in the Mediterranean, but is abundant in sedimentary cores (i.e. Cramp et al., 1988; Rio et al., 1990); there exists the possibility to classify it with *G. sacculifer* or *G. trilobus* in previous studies as suggested by Hemleben et al. (1989). Some species, which are absent from our samples, reached high frequencies in the aforementioned studies, i.e., *Turborotalita quinqueloba, Neogloboquadrina pachyderma*, and *Globorotalia truncatulinoides*. The fact that these species were not sampled in the present study may be due to their absence or presence at extremely low abundances of adult specimens at the sampled stations in May, as they present generally low abundances in spring according to a 12-year sediment trap record in the Gulf of Lion (Rigual-Hernández et al., 2012). Another possibility is their presence in test sizes smaller than 150 µm, which is smaller than the mesh size of our BONGO nets, a possibility potentially supported by previous Mediterranean studies using smallesr mesh sizes (see Pujol and Vergraud-Grazzini, 1998, 120 µm mesh size; Rigual-Hernández et al., 2012, 63-150 µm mesh size).

475 To propose a quantitative comparison of the number of species found in previous studies in the 476 Mediterranean, we used the morphospecies identified in them by the authors of each study. We identified 477 12 morphospecies, clearly less than Cifelli (1974), Thunell (1978) and Pujol and Vergraud-Grazzini 478 (1995), with 18 morphospecies in total. At Station 3 of this study (Alboran Sea), we found 8 479 morphospecies; whereas Rigual-Hernández et al. (2012) found 12 morphospecies during the same season. The lower absolute abundance of individuals in our study compared with to Pujol and GrazziniPujol and 480 481 Vergraud-Grazzini (1995), together with low species diversity in the Mediterranean, may indicate a trend 482 of changing conditions in recent yearsover the last decades, as it has been reported for temperature and 483 salinity (Yáñez et al., 2010), alkalinity (Cossariniet al., 2015; Hassoun et al., 2015a), and water mass 484 mixings (Hassoun et al., 2015b). These changing conditions could also imply changes in the ecology and 485 distribution of planktic foraminifera, as discussed below. Note that our mesh size is larger than that of 486 Pujol and Vergraud-Grazzini (1995) and Rigual-Hernández et al. (2012), but is similar to that of Cifelli 487 (1974): mesh size of 158 µm. A larger mesh size would explain the lower numbers in absolute abundance

490

and reduced diversity, but the higher diversity observed by Cifelli (1974) in June supports our idea of changing ecological conditions.

491The western part of the first leg-transect (from the Atlantic to the Strait of Sicily) has a higher percentage492of larger size fractions than the eastern part.; The mainthe main cause of the increase in test sizeat trend is493the a change in species composition. The results are conditioned by the presence of For example, large494sized G. inflata (especially in the 350-500  $\mu$ m fraction) are present with higher abundances in the west495than in the east. The same is true for the presence of large O. universa (especially in the >500  $\mu$ m), plus496the contribution of G. siphonifera, which grow-is largerst inat stations in which they are most where it is497more frequent (Appendix A; Fig. 4).

498

**499 5. 2. Factors controlling the abundance of the main species**

500 This discussion is focuses on the five main species of our results samples. The spinose and symbiontbearing species: G. ruber (white), O. universa, and TG. sacculifer (always referring to the trilobus 501 502 typewithout sac), which mainly inhabit tropical and subtropical waters. G. ruber (white) is found as the 503 main species of in the Atlantic. O. universa has a quite cosmopolitan standing stockis rather ubiquitous, 504 also being present in warm transitional Atlantic waters (Bé and Tolderlund, 1971). The spinose and 505 nonsymbiotic species G. bulloides, is typical of subpolar and transitional regions as well as upwelling 506 areas, but-and is also found in subtropical and tropical waters at a much lower abundances, highlighting 507 characterized by its wide temperature range (Thunell, 1978; Bé and Tolderlund, 1971). The non-spinose 508 species G. inflata is considered indigenous from the transitional typical of the temperate region in the 509 Atlantic Ocean (Bé and Tolderlund, 1971).

510 5. 2. 1. Globigerinoides ruber (white)

Both varieties *G. ruber* sensu stricto (s.s.) and sensu lato (s.l.) are warm water shallow dwellers and share
similar habitats. Regarding some studies, *G. ruber* s.s. is found slightly shallower than *G. ruber* s.l.
(Kuroyanagi and Kawahata, 2004; Wang, 2000); a reason could be that *G. ruber* s.l. may be less
dependent on symbiont activity than *G. ruber* s.s. (Kuroyanagi and Kawahata, 2004).

515 In our study, G. ruber s.s. (white) and s.l. varieties are is found in the Atlantic with slightly larger higher 516 absolute abundances and higher relative abundances than in the western Mediterranean Basin, where it is 517 found in low abundances. Temperature-related factors may be the main cause, with warmer Atlantic 518 waters (16.1 °C) with respect to the western Mediterranean (14.3 °C in the SW, 14.0 °C in the NW; Fig. 519 1), as demonstrated by positive significant correlations with temperature in the G. ruber s.s. variety (p =520 0.01). These-G. ruber results are in agreement witheonfirm the observations made by Cifelli (1974) in 521 findings of the June 1969-eruise of Cifelli (1974), where it-G. ruber (white) was by far more abundant in 522 the eastern than the western Mediterranean Basin, being the most abundant, clearly being the main 523 species found-in the Levantine Basin and the south Ionian Sea; for these two locations it seems that G. 524 ruber (white) is present independent of the during the different seasons, winter included, which is also 525 true for the pink variety of G. ruber (see also Thunell, 1978; Pujol and Grazzini Pujol and Vergraud526 Grazzini, 1995). The increasing dominance of G. ruber s.s. (white) in from the western to the eastern 527 Mediterranean Basin coincides with the eastward increasing salinity relative to the western Basin causes a strong positive correlation with salinity (p = 0.01 Fig. 7d) in our data set. Its higher relative abundance in 528 529 the eastern basin may result from symbiont activity in G. ruber, supporting survival in oligotrophic regions, and some independence from chlorophyll a and macronutrient concentrations (Watkins et al., 530 1996). The findings of Watkins et al. (1996) are supported by the negative correlations of standing stocks 531 532 of G. ruber s.s. and fluorescence data of our study (p = 0.05). Its higher relative abundance in the eastern 533 basin results from the ability of G. ruber to thrive in food-depleted conditions (Hemleben et al., 1989).

534 G. ruber (white) remains scarce or absent in May in the Ionian Sea stations (Fig. 3), increasing its 535 abundance towards the Tyrrhenian Sea. On the other hand, in the Ionian Sea it exhibits relative abundance 536 below 60% in the surface sediments (Thunell, 1978), and decreases towards the Tyrrhenian Sea. This 537 situation could be due to higher food availability in the Tyrrhenian Sea in comparison to the Ionian Sea 538 during May 2013 (Fig. 1c; Fig. 7d) plus a small difference in temperature between both seas (Fig. 1a; Fig. 539 7d). This may not be the typical spring situation, as due to surface sediment evidence, the Ionian Sea 540 sediments are enriched in G. ruber tests (Thunell, 1978) and May is the most productive season in terms 541 of foraminiferal tests (Rigual-Hernández, 2012; Bárcena et al., 2004; Hernández-Almeida et al., 2011). 542 Also, we note that in May 1979, a scarce presence of G. ruber was reported in the Bay of Naples (de 543 Castro Coppa et al., 1980), whereas in our study G. ruber is present at 47 % in the Tyrrhenian Sea, being 544 the main species.

545 The dominance of G. ruber (white) and abundance peaks in May in the eastern Mediterranean (this 546 study), coincides with the positive temperature gradient between Station 9 and Station 13 (16.2–17.3 °C; 547 Fig. 1), being more evident for the G. ruber s.s. than for the G. ruber s.l. morphotype. 
[revised manuscript text omitted]
 642 643 area is clearoccurs in spring in (our data) and in the report that of Cifelli (1974), but abundances are slightly higher abundances in the western basin compared to to than the east-are modest. That small 644 645 difference can be caused by more nutrient-rich upwelling areas (Sousa et al., 2014; Morard et al., 2013) in 646 the western basin or by high salinities in the eastern basin.

**647 5. 2. 5. *Globigerinoides sacculifer* type *trilobus*Trilobatus sacculifer (without sac)**

648 In June, the distribution of GT. sacculifer (without sac) is guite ubiquitous and has-represents 5 % 649 presence of the assemblage inat the Strait of Gibraltar (Cifelli, 1974). At our stations, T. sacculifer constituted up to; meanwhile our results show a 25 % presence one month before of the assemblages in 650 May, and was absentee at from seven stations (St. 5, 7a, 14, 15, 16-18, 20, 22). Also, Llower percentages 651 652 are were found in April at in the Alboran Sea (Pujol and GrazziniPujol and Vergraud-Grazzini, 1995). In 653 September-October it-T. sacculifer shows high abundances and is one of the main species from north of 654 Minorca to the southwestern Mediterranean until the Strait of Sicily, where it is rare. ly found, 655 presumably due to warmer waters than in May, even if this is not supported by our Pearson correlation. In 656 late summer it decreases considerably and progressively eastwards, where the highly dominant G. ruber 657 is maintained as the most abundantimportant species (Pujol and GrazziniPujol and Vergraud-Grazzini, 658 1995), probably due to slightly higher temperature and salinity tolerance (see also Bijma et al., 1990). On the other hand, in February GT. sacculifer (without sac) disappears from the north Levantine Basin and its 659 660 abundances lowers considerably, being a residual species in terms of relative abundance in all the Mediterranean (Pujol and GrazziniPujol and Vergraud-Grazzini, 1995), suggesting temperatures too cold 661 662 for--.<del>it.</del>

664

663

**665 5. 3. Factors controlling planktic foraminiferal test weight**

666The size normalized weight area density ( $\rho_{\Lambda}$ SNW) of tests of tests of both *G. ruber* s.s.(white) and *G.*667*bulloides* are statistically significant, and follow a systematic change from the Atlantic towards the668eastern Mediterranean (Fig. 6). Therefore, the  $\rho_{\Lambda}$ SNW of these two species is interpreted and discussed669for possible environmental effects and biological prerequisites in the following. In contrast, changes of670the  $\rho_{\Lambda}$ SNW of *O. universa* are statistically insignificantdoes not show any change between the western671and eastern basins (Figs. S2e, S3e, and S4eFig. 7i), and cannot be used to identify and quantify particular672environmental effects.

**673** 5.3.1 Unknown control of the  $\rho_{A}$ SNW of *O. universa*

674 The lack of statistical significanceNo systematic change between the western and eastern basins in the 675  $\rho_A$ SNW data of *O. universa*\_in our data set\_could possibly\_be explainedeaused by an insufficient 676 understanding of the ecology of the different morphotypes and genotypes of O. universa. Despite the 677 finding that only-Only one out of three genotypes (i.e. Type III, after Darling and Wade, 2008) occurs-is 678 recorded\_in the Mediterranean Sea (Mediterranean species, after de Vargas et al., 1999), Weight-area relation data do not show any statistically significant systematic distribution (Fig. S4c). The 679 680 Mediterranean Type III has been found to include two sub-types, Type IIIa and Type IIIb (André et al., 681 2014). The different genotypes and morphotypes of O. universa tolerate wide ranges of salinity and 682 temperature in surface waters (ie.eg., de Vargas et al., 1999). Whereas the various types of O. universa 683 differ in the size of porespore-size (de Vargas et al., 1999; Morard et al., 2009; Marshall et al., 2015), 684 their pore-size is also affected by environmental conditions including water temperature (ie.eg., Bé et al., 1973). Likewise, thickness of the test wall has been described to vary between types (de Vargas et al., 685 686 1999; Morard et al., 2009; Marshall et al., 2015), and is as well affected by environmental conditions and 687 ontogenetic stage of specimens. Adult O. universa have been shown to continuously add calcite layers to 688 the proximal surface of the same sphere (Spero, 1988; Spero et al., 2015). Since environmental and 689 biological factors may affect individuals of the different genotypes of O. universa to varying degrees, we could not detect any systematic change in  $\underline{p}_A \underline{SNW}$  in the data presented here. 690

691The O. universa weight-area data of our study are compared with those of Marshall et al. (2015) from692Cariaco Basin sediment trap specimens, including O. universa Type I ( $M_{thick}$ ) and Type III ( $M_{thin}$ )693specimens, suggesting thinner test walls in the latter. In the area range of  $3 \cdot 10^5 - 4 \cdot 10^5 \, \mu m^2$ , our weight694data coincide with the expected Mediterranean Type III variety (Fig. S4c; Marshall et al., 2015), but at695 $2 \cdot 10^5 - 2.5 \cdot 10^5 \, \mu m^2$  we see a mix of both types until at  $1.5 \cdot 10^5 \, \mu m^2$  Type I coincides more with our results696(Fig. S4c; Marshall et al., 2015). We suggest that different groups of the Mediterranean O. universa697variety coexist in the Mediterranean with differences in the wall thickness.

698 The various interfering effects, which control the  $\rho_A SNW$  of O. universa in the Mediterranean Sea, may 699 also explain differences in the weight-diameterlong axis relation data reported from other regions of the 700 world ocean: Bijma et al. (2002) weighed O. universa of in the 500-600 µm size fraction in the Caribbean 701 Sea and reported a weight ranging at from 28 to  $-60 \mu g$ . Lombard et al. (2010) giveneasured a weight of 702  $20-70 \ \mu g$  for specimens sampled off Catalina Island, California, in the same size fraction of the 500-600 703  $\mu$ m. Our weight-diameterlong axis relation data range at from 24 to -45  $\mu$ g (Fig. S3c) for the same size 704 fraction of the 500-600 µm, ranging at the lower limit of the weight-diameterlong axis relations measured 705 in the Caribbean (Bijma et al., 2002) and off California (Lombard et al., 2010), which may be caused 706 either by differences in genotypes or environmental conditions, or both. Thinner walls overall in our specimens with respect to the mentioned studies could be a possible explanation for the differences in  $\rho_A$ 707 708 (Marshall et al., 2015). In our samples from the Mediterranean, individuals exceeding 60  $\mu$ g have 709 diameterlong axis larger than 650  $\mu$ m. The reason why the  $\rho_A$  SNW-of O. universa is particularly low 710 and highly variable in the Mediterranean despite  $\frac{\partial f}{\partial h}$  high carbonate ion concentration ([CO32-]) and pH 711 (Fig. 1) might be sought in factors other than, and in addition to, chemical and physical conditions, 712 namely the changing availability of food along the transect from the Atlantic Ocean to the Levantine 713 Basin.

716

717

718

719

720

721

722

723

724

In the same way as in *O. universa*, the  $\rho_A$ SNW of *G. ruber* (white)s.s. seems-is only partlynot to be controlled by carbonate chemistry, and being to beinstead affected by other factors like nutrient concentration and food availability.\_-However, in contrast to *O. universa*, the  $\rho_A$ SNW data of *G. ruber* and *G. bulloides* follow systematic distributionscorrelations, which are statistically significant.\_ High  $\rho_A$ SNW in the Atlantic and Tyrrhenian Sea correlates with enhanced primary production: (enhanced fluorescence, (Fig. 1d; Fig. 7g) and presumably enhanced food availability (Fig. 6; Fig. 7g; Fig. 2, also noticeable in Fig. S2d-e and Fig. S4d-e). At the same sites, larger IQR indicates more variability in test calcite production of *G. ruber* (white)s.s. specimens, although a limited number of samples together with the low and uneven sampling size impede any further interpretation of the data (Fig. 6). Under more oligotrophic conditions, low  $\rho_A$  SNW-of *G. ruber* (white)s.s. might be caused by limited food availability.

An opposite trend occurs in *G. ruber* (white) sediment trap samples from the Madeira Basin, in which,
apart from showing a negative significant correlation between calcification intensity and productivity, ρA
shows a positive correlation with temperature (Weinkauf et al., 2016).

The relationship between food availability and  $\rho_{A}$ SNW in *G. bulloides* is opposite to that in *G. ruber* (white)s.s. (Fig. 6; Fig. 7g-h). The  $\rho_{A}$ SNW of *G. bulloides* tests increases from the Atlantic toward the eastern Mediterranean. At the same time, variability in  $\rho_{A}$ SNW data increases with increasing absolute  $\rho_{A}$ SNW, which resembles the distribution of data in *G. ruber* (white)s.s. (Fig. 6): In both species *G. ruber* s.s. and *G. bulloides*-larger IQRs are found toward higher absolute  $\rho_{A}$ SNW.

733 An opposite trend in  $p_A SNW$  of the two species G. ruber (white)s.s. and G. bulloides had earlier been described from the Arabian Sea, and could neither be assigned to changes in [CO3-2-] of ambient seawater 734 nor growth conditions (Beer et al., 2010a). Due to its symbionts, G. ruber would rather have an advantage 735 736 over symbiont-barren G. bulloides in oligotrophic waters, and support formation of test calcite through  $CO_2$  consumption and increasing  $[CO_3^{-2_2}]$  and pH (see also Köhler-Rink and Kühl, 2005). Those findings 737 738 may still point toward differences in growth conditions: Reproduction of both G. ruber and G. bulloides 739 might be retarded under less optimal conditions, and additional calcite layers might be added to the 740 proximal tesxt surface before reproduction, similar to the process described for O. universa (see above). 741 Therefore, tests may grow heavier under less optimal than optimal alimentation food availability, given 742 that carbonate chemistry of ambient seawater does not seems to limit the formation of test calcite in our 743 samples.

744 Comparing weight-diameterlong axis relations, G. ruber (255–350 µm size fraction) from plankton tows 745 of the western Arabian Sea have an average weight of  $11.5 \pm 0.69 \ \mu g$  (de Moel et al., 2009), which is 746 heavier than the individuals from our study (5.9  $\pm$ 0.31 µg; Fig. S3a; Appendix A). The difference in 747 weight-diameterlong axis relation may indicate that G. ruber was produced under more ideal suited 748 conditions for shell calcite formation in the Arabian Sea especially during non-upwelling periods and still 749 higher overall primary productivity and food availability. However, the comparison might be biased by 750 the fact that G. ruber (white)s.s. and s.l.G. elongatus morphotypes were analyzed together in the study of 751 de Moel et al. (2009).

Data for supra-regional comparison of weight-diameterlong axis relation of G. bulloides from the water 752 753 column are found for the 200-250 µm size fraction: in the north Atlantic (56-63 °N) in June 2009 (Aldridge et al., 2012) with a range of 1.75–2.92  $\mu$ g (r2 = 0.52). For that size fraction our results (36 °N) 754 show heavier tests in the Alboran Sea ( $3.46 \pm 0.15 \mu g$ ), and similar weights at the Strait of Gibraltar (2.57 755 756  $\pm 0.00 \ \mu$ g; Fig. S3b). For the same water depth as in our samples, Schiebel et al. (2007) found a heavier 757 average weight-diameterlong axis relation in fall (5.19  $\pm 0.25 \ \mu$ g) than during spring (4.21  $\pm 0.2 \ \mu$ g) in the 758 eastern north Atlantic (47 °N), and 5.51 ±0.31 µg during the SW monsoon in the Arabian Sea (16 °N). In general, higher  $\rho_A SNW$  occurs at lower latitudes and lower  $\rho_A SNW$  at higher latitudes (see also Schmidt 759 760 et al., 2004). All of these findings support our idea of an effect of limited alimentation on reproductioncalcification. Increased longevity and ongoing production of additional calcite layers at the 761 proximal side of shells may result in an increased  $\rho_{A}$  SNW, given that seawater carbonate chemistry does 762 763 only partially affecting thenot limit calcite formation in planktic foraminifera in our samples.

764

765

**766 **6.** Conclusions**

[revised manuscript text omitted]

**804 (Appendix A, cont.).**

| Location                    | Atlantic | Gibraltar | Alboran | South-
Central
Western | Strait of | Strait of | South of | Off
Southern
Crete | Eastern | Off Nile | Off     | Antikythera | a Eastern | Adriatic | Otranto | Northern
Ionian Sea | Tyrrhenian | North-
Central
Western
Med | Central
Western
Med | Catalano-Balear |
|-----------------------------|----------|-----------|---------|------------------------------|-----------|-----------|----------|--------------------------|---------|----------|---------|-------------|-----------|----------|---------|------------------------|------------|-------------------------------------|---------------------------|-----------------|
| Station                     | 1        | 2         | 3       | 5                            | 6         | 7a        | 9        | 10                       | 11      | 12       | 13      | 14          | 15        | 17       | 16      | 16-18                  | 19         | 20                                  | 21                        | 22              |
| >500 um size fraction       |          |           |         | 5                            | -         |           |          |                          | -       |          | -       | -           |           | .,       | -       |                        | .,         | 20                                  |                           |                 |
| G milar size machon         | 0.010    | 0         | 0       | 0                            | 0         | 0         | 0        | 0                        | 0       | 0        | 0       | 0           | 0         | 0        | 0       | 0                      | 0          | 0                                   | 0                         | 0               |
| T. sacculifer (without sac) | 0.001    | 0.010     | 0       | 0                            | 0         | 0         | 0        | 0                        | 0       | 0        | 0       | 0           | 0         | 0        | 0       | 0                      | 0          | 0                                   | 0                         | 0               |
| G inflata                   | 0.001    | 0.019     | 0 135   | 0.022                        | 0.047     | 0.022     | 0        | 0.031                    | 0       | 0        | 0       | 0           | 0         | 0        | 0       | 0                      | 0          | 0                                   | 0                         | 0               |
| O universe                  | 0.079    | 0.015     | 0       | 0.022                        | 0.017     | 0.224     | 0        | 0.051                    | 0.027   | 0.028    | 0.050   | 0           | 0.077     | 0        | 0 130   | 0 117                  | 0 102      | 0                                   | 0 102                     | 0.059           |
| G sinhonifera               | 0.010    | 0.019     | 0.007   | 0.022                        | 0         | 0.089     | 0        | 0.031                    | 0.027   | 0.020    | 0.020   | 0           | 0.077     | 0        | 0       | 0                      | 0.102      | 0                                   | 0.102                     | 0               |
| G avadrilobatus             | 0.010    | 0.015     | 0.007   | 0                            | 0         | 0.022     | 0        | 0.051                    | 0       | 0        | 0       | 0           | 0         | 0        | 0       | 0                      | 0          | 0                                   | 0                         | 0               |
| Tatal                       | 0 108    | 0.056     | 0 1/2   | 0.044                        | 0.047     | 0.358     | 0        | 0.063                    | 0.027   | 0.027    | 0.050   | 0           | 0.077     | 0        | 0 120   | 0.117                  | 0 102      | 0                                   | 0 102                     | 0.059           |
| Relative abundance (%)      | 0.100    | 0.050     | 0.145   | 0.044                        | 0.047     | 0.550     | 0        | 0.005                    | 0.027   | 0.027    | 0.050   | 0           | 0.077     | 0        | 0.150   | 0.117                  | 0.102      | 0                                   | 0.102                     | 0.057           |
| G mbar (white)              | 8.00     | 0.72      | 0.17    | 1.49                         | 0         | 0         | 21.02    | 12 75                    | 53 57   | 56.25    | 74.62   | 42.22       | 22.22     | 0        | 22.81   | 0                      | 46.91      | 0                                   | 0                         | 0               |
| G. alongatus                | 12.00    | 0.72      | 0.17    | 0                            | 2 22      | 0         | 0        | 43.75                    | 7.14    | 6.25     | 11.04   | 45.55       | 0         | 0        | 12.01   | 27.27                  | 14.80      | 0                                   | 4.00                      | 0               |
| T sacculifer (without sac)  | 24.00    | 0.50      | 0.17    | 0                            | 3.33      | 0         | ( 00     | 9.56                     | 2.57    | 10.25    | 2.00    | 50.00       | 0         | 20.00    | 12.28   | 21.21                  | 7.00       | 0                                   | 4.00                      | 0               |
|                             | 24.00    | 25.45     | 0.69    | 24.22                        | 0.07      | 0         | 0.90     | 7.29                     | 3.37    | 18./5    | 2.99    | 0           | 0         | 20.00    | 15.79   | 0.00                   | 7.09       | 52.05                               | 4.00                      | 0               |
| G. bulloides                | 15.00    | 44.44     | 01.02   | 34.33                        | 20.00     | 0         | 24.14    | 3.13                     | 7.14    | 0        | 4.48    | 0           | 33.33     | 0        | 3.51    | 9.09                   | 8.51       | 53.85                               | 16.00                     | 21./4           |
| G. inflata                  | 12.00    | 9.68      | 84.85   | 37.31                        | 65.55     | 35.56     | 10.34    | 4.17                     | 3.57    | 0        | 0       | 0           | 0         | 20.00    | 0       | 0                      | 0          | 0                                   | 0                         | 0               |
| O. universa                 | 13.00    | 1.79      | 0.34    | 14.93                        | 0         | 28.89     | 0        | 7.29                     | /.14    | 6.25     | 2.99    | 0           | 25.00     | 20.00    | 31.58   | 54.55                  | 7.80       | 7.69                                | 28.00                     | 26.09           |
| G. siphonifera              | 3.00     | 1.08      | 1.03    | 1.49                         | 0         | 31.11     | 0        | 2.08                     | 0       | 0        | 1.49    | 0           | 0         | 0        | 0       | 0                      | 0          | 0.00                                | 16.00                     | 0               |
| G. quaarnooauus             | 1.00     | 6.45      | 0.17    | 5.97                         | 0         | 4.44      | 17.24    | 2.08                     | 3.57    | 0        | 0       | 0           | 0         | 20.00    | 0       | 0                      | 6.38       | 30.77                               | 32.00                     | 34.78           |
| H. pelagica                 | 0        | 0         | 0       | 0                            | 0         | 0         | 0        | 4.17                     | 0       | 6.25     | 0       | 0           | 0         | 0        | 0       | 0                      | 0          | 0                                   | 0                         | 0               |
| 1. saccutifer (with sac)    | 0        | 0         | 0       | 0                            | 0         | 0         | 0        | 0                        | 0       | 0        | 0       | 0           | 0         | 0        | 0       | 0                      | 0.71       | 0                                   | 0                         | 0               |
| G. ruber (pink)             | 0        | 1.43      | 0       | 0                            | 3.33      | 0         | 3.45     | 4.17                     | 0       | 6.25     | 0       | 13.33       | 0         | 0        | 0       | 0                      | 0          | 0                                   | 0                         | 0               |
| G. menardii                 | 0        | 0         | 0       | 0                            | 0         | 0         | 0        | 0                        | 0       | 0        | 0       | 0           | 0         | 0        | 0       | 0                      | 0          | 0                                   | 0                         | 4.35            |
| Unknowns                    | 12.00    | 8.60      | 1.55    | 4.48                         | 5.55      | 0         | 6.90     | 12.50                    | 14.29   | 0        | 1.49    | 13.33       | 8.33      | 20.00    | 14.04   | 9.09                   | 7.80       | 7.69                                | 0                         | 13.04           |
| Weight and size             |          |           |         |                              |           |           |          |                          |         |          |         |             |           |          |         |                        |            |                                     |                           |                 |
| G. ruber (white)            | 250 200  |           |         |                              |           |           |          | 200.250                  | 200.250 |          | 200.250 | 250 200     |           |          | 250 200 |                        | 200.250    |                                     |                           |                 |
| size traction (µm)          | 250-300  |           |         |                              |           |           |          | 200-250                  | 200-250 |          | 200-250 | 250-300     |           |          | 250-300 |                        | 200-250    |                                     |                           |                 |
| n° of individuals           | 1        |           |         |                              |           |           |          | 4                        | 4       |          | 4       | 2           |           |          | 4       |                        | 4          |                                     |                           |                 |
| average size (µm)           | 285      |           |         |                              |           |           |          | 221                      | 215.25  |          | 221.5   | 281         |           |          | 268     |                        | 218.5      |                                     |                           |                 |
| average weight (µg)         | 4.667    |           |         |                              |           |           |          | 1.583                    | 2.417   |          | 2       | 3.167       |           |          | 5.5     |                        | 2.083      |                                     |                           |                 |
| SD (µg)                     | 0.577    |           |         |                              |           |           |          | 0.144                    | 0.289   |          | 0       | 0.577       |           |          | 0       |                        | 0.144      |                                     |                           |                 |
| size freation (um)          | 250 400  |           |         |                              |           |           |          | 250 250                  | 250 200 |          | 250 200 | 200.250     |           |          |         |                        | 250 200    |                                     |                           |                 |
| size fraction (µm)          | 350-400  |           |         |                              |           |           |          | 250-350                  | 250-300 |          | 250-300 | 300-350     |           |          |         |                        | 250-300    |                                     |                           |                 |
| II OI Individuals           | 4        |           |         |                              |           |           |          | 267                      | 261     |          | 264     | 217         |           |          |         |                        | 280.6      |                                     |                           |                 |
| average size (µiii)         | 14 2 2 2 |           |         |                              |           |           |          | 207                      | 201     |          | 5 111   | 6 667       |           |          |         |                        | 200.0      |                                     |                           |                 |
| average weight (µg)         | 0.280    |           |         |                              |           |           |          | 0.115                    | 0.577   |          | 0.102   | 0.577       |           |          |         |                        | 4.0        |                                     |                           |                 |
| 3D (µg)                     | 0.289    |           |         |                              |           |           |          | 0.115                    | 0.577   |          | 0.192   | 0.577       |           |          |         |                        | 0.2        |                                     |                           |                 |
| size fraction (um)          | 400 450  |           |         |                              |           |           |          | 200 250                  | 250 400 |          | 200 250 |             |           |          |         |                        | 200.250    |                                     |                           |                 |
| nº of individuals           | 1        |           |         |                              |           |           |          | 3                        | 1       |          | 2       |             |           |          |         |                        | 5          |                                     |                           |                 |
| average size (um)           | 412      |           |         |                              |           |           |          | 212 222                  | 256     |          | 222.5   |             |           |          |         |                        | 242.4      |                                     |                           |                 |
| average weight (µg)         | 14 667   |           |         |                              |           |           |          | 7 444                    | 5 667   |          | 11      |             |           |          |         |                        | 9 867      |                                     |                           |                 |
| arenage weight (µg)         | 1 155    |           |         |                              |           |           |          | 0.385                    | 1 155   |          | 0       |             |           |          |         |                        | 0.231      |                                     |                           |                 |
| 5D (µg)                     | 1.155    |           |         |                              |           |           |          | 0.565                    | 1.155   |          | 0       |             |           |          |         |                        | 0.251      |                                     |                           |                 |
| size fraction (um)          |          |           |         |                              |           |           |          | 350-400                  |         |          |         |             |           |          |         |                        | 350-400    |                                     |                           |                 |
| nº of individuals           |          |           |         |                              |           |           |          | 2                        |         |          |         |             |           |          |         |                        | 4          |                                     |                           |                 |
| average size (um)           |          |           |         |                              |           |           |          | 374                      |         |          |         |             |           |          |         |                        | 366        |                                     |                           |                 |
| average weight (ug)         |          |           |         |                              |           |           |          | 8.833                    |         |          |         |             |           |          |         |                        | 9.083      |                                     |                           |                 |
| SD (ug)                     |          |           |         |                              |           |           |          | 0 764                    |         |          |         |             |           |          |         |                        | 0 144      |                                     |                           |                 |
| 5.2 (µg)                    |          |           |         |                              |           |           |          | 0.704                    |         |          |         |             |           |          |         |                        | 0.144      |                                     |                           |                 |

**806 (Appendix A, cont.).**

|                                                                                                |          |           |         | South-
Central |           |           |            | Off      |         |          |         |             |            |          |         |            |                                        | North-
Central | Central  |           |
|------------------------------------------------------------------------------------------------|----------|-----------|---------|-------------------|-----------|-----------|------------|----------|---------|----------|---------|-------------|------------|----------|---------|------------|----------------------------------------|-------------------|----------|-----------|
|                                                                                                |          |           | Alboran | Western           | Strait of | Strait of | South of   | Southern | Eastern | Off Nile | Off     | Antikythera | Eastern    | Adriatic | Otranto | Northern   | Tyrrhenian                             | Western           | Western  | Catalano- |
| Location                                                                                       | Atlantic | Gibraltar | Sea     | Med.              | Sardinia  | Sicily    | Ionian Sea | Crete    | Basin   | Delta    | Lebanon | Strait      | Ionian Sea | Sea      | Strait  | Ionian Sea | Sea                                    | Med.              | Med.     | Balear    |
| Station                                                                                        | 1        | 2         | 3       | 5                 | 6         | 7a        | 9          | 10       | 11      | 12       | 13      | 14          | 15         | 17       | 16      | 16-18      | 19                                     | 20                | 21       | 22        |
| size fraction (μm)
n° of individuals
average size (μm)
average weight (μg)
SD (μg) |          |           |         |                   |           |           |            |          |         |          |         |             |            |          |         |            | 400-450
2
413
16.167
1.258 |                   |          |           |
| G. bulloides                                                                                   |          |           |         |                   |           |           |            |          |         |          |         |             |            |          |         |            |                                        |                   |          |           |
| size fraction (µm)                                                                             | 300-350  | 200-250   | 200-250 | 350-400           | 300-350   |           |            |          |         |          |         |             |            |          |         |            |                                        |                   | 400-450  | 300-350   |
| nº of individuals                                                                              | 2        | 7         | 8       | 1                 | 1         |           |            |          |         |          |         |             |            |          |         |            |                                        |                   | 1        | 3         |
| average size (µm)                                                                              | 326.5    | 228.143   | 227.875 | 364               | 337       |           |            |          |         |          |         |             |            |          |         |            |                                        |                   | 414      | 318.333   |
| average weight ( $\mu g$ )                                                                     | 4.5      | 2.571     | 3.458   | 4.667             | 4         |           |            |          |         |          |         |             |            |          |         |            |                                        |                   | 11.667   | 8.222     |
| SD (µg)                                                                                        | 0.5      | 0         | 0.144   | 0.577             | 1         |           |            |          |         |          |         |             |            |          |         |            |                                        |                   | 0.577    | 0.385     |
| cize fraction (um)                                                                             |          | 250 200   | 250 300 |                   |           |           |            |          |         |          |         |             |            |          |         |            |                                        |                   |          | 400 450   |
| size fraction (µm)                                                                             |          | 12        | 250-500 |                   |           |           |            |          |         |          |         |             |            |          |         |            |                                        |                   |          | 400-450   |
| ni or individuais                                                                              |          | 262.75    | 270     |                   |           |           |            |          |         |          |         |             |            |          |         |            |                                        |                   |          | 1         |
| average size (µiii)                                                                            |          | 2 833     | 2 833   |                   |           |           |            |          |         |          |         |             |            |          |         |            |                                        |                   |          | 20 333    |
| SD (ug)                                                                                        |          | 2.855     | 0.280   |                   |           |           |            |          |         |          |         |             |            |          |         |            |                                        |                   |          | 1 155     |
| 3D (µg)                                                                                        |          | 0         | 0.289   |                   |           |           |            |          |         |          |         |             |            |          |         |            |                                        |                   |          | 1.155     |
| size fraction (µm)                                                                             |          | 300-350   | 350-400 |                   |           |           |            |          |         |          |         |             |            |          |         |            |                                        |                   |          |           |
| nº of individuals                                                                              |          | 2         | 4       |                   |           |           |            |          |         |          |         |             |            |          |         |            |                                        |                   |          |           |
| average size (µm)                                                                              |          | 310.5     | 386.5   |                   |           |           |            |          |         |          |         |             |            |          |         |            |                                        |                   |          |           |
| average weight ( $\mu g$ )                                                                     |          | 4.5       | 9.667   |                   |           |           |            |          |         |          |         |             |            |          |         |            |                                        |                   |          |           |
| SD (µg)                                                                                        |          | 0.5       | 0.144   |                   |           |           |            |          |         |          |         |             |            |          |         |            |                                        |                   |          |           |
| size fraction (um)                                                                             |          | 350-400   | 400-450 |                   |           |           |            |          |         |          |         |             |            |          |         |            |                                        |                   |          |           |
| nº of individuals                                                                              |          | 2         | 2       |                   |           |           |            |          |         |          |         |             |            |          |         |            |                                        |                   |          |           |
| average size (um)                                                                              |          | 375.5     | 429     |                   |           |           |            |          |         |          |         |             |            |          |         |            |                                        |                   |          |           |
| average weight (µg)                                                                            |          | 5.833     | 11      |                   |           |           |            |          |         |          |         |             |            |          |         |            |                                        |                   |          |           |
| SD (μg)                                                                                        |          | 0.289     | 0       |                   |           |           |            |          |         |          |         |             |            |          |         |            |                                        |                   |          |           |
|                                                                                                |          |           |         |                   |           |           |            |          |         |          |         |             |            |          |         |            |                                        |                   |          |           |
| size fraction (µm)                                                                             |          | 400-450   | 450-500 |                   |           |           |            |          |         |          |         |             |            |          |         |            |                                        |                   |          |           |
| nº of individuals                                                                              |          | 1         | 1       |                   |           |           |            |          |         |          |         |             |            |          |         |            |                                        |                   |          |           |
| average size (µm)                                                                              |          | 447       | 477     |                   |           |           |            |          |         |          |         |             |            |          |         |            |                                        |                   |          |           |
| average weight (µg)                                                                            |          | 9.333     | 7.333   |                   |           |           |            |          |         |          |         |             |            |          |         |            |                                        |                   |          |           |
| SD (µg)                                                                                        |          | 0.577     | 0.577   |                   |           |           |            |          |         |          |         |             |            |          |         |            |                                        |                   |          |           |
| O. universa                                                                                    |          |           |         |                   |           |           |            |          |         |          |         |             |            |          |         |            |                                        |                   |          |           |
| size fraction (µm)                                                                             | 350-400  | 250-300   | 500-550 | 400-450           |           | 450-500   |            | 300-350  | 350-400 | 700-750  | 650-700 |             | 700-750    | 450-500  | 300-350 | 400-450    | 400-450                                | 400-450           | 450-500  | 350-400   |
| nº of individuals                                                                              | 3        | 1         | 1       | 2                 |           | 1         |            | 1        | 1       | 1        | 1       |             | 2          | 1        | 1       | 1          | 1                                      | 1                 | 2        | 1         |
| average size (µm)                                                                              | 390      | 286       | 501     | 445               |           | 479       |            | 342      | 398     | 719      | 687     |             | 722.5      | 452      | 347     | 444        | 441                                    | 441               | 479.5    | 377       |
| average weight ( $\mu g$ )                                                                     | 17.667   | 7         | 20.667  | 11.667            |           | 31        |            | 3        | 6.333   | 47       | 43      |             | 24.167     | 14.333   | 5.333   | 18.667     | 24.333                                 | 22.667            | 31       | 20        |
| SD (µg)                                                                                        | 0.333    | 0         | 0.577   | 0.289             |           | 1         |            | 0        | 0.577   | 1        | 0       |             | 0.289      | 0.577    | 0.577   | 0.577      | 0.577                                  | 0.577             | 0.5      | 1         |
| cize fraction (um)                                                                             | 400 450  |           |         | 450 500           |           | 500 550   |            | 350 400  | 500 550 |          | 750 800 |             | 750 800    |          | 350 400 | 550 600    | 450 500                                |                   | 5 50 600 | 400 450   |
| nº of individuals                                                                              | 1        |           |         | 450-500           |           | 2         |            | 330=400  | 1       |          | 1       |             | 1          |          | 1       | 1          | 450-500                                |                   | 1        | 400-450   |
| average size (um)                                                                              | 444      |           |         | 479               |           | 530.5     |            | 373 667  | 530     |          | 781     |             | 785        |          | 369     | 559        | 455                                    |                   | 571      | 425.5     |
| average weight (ug)                                                                            | 28 667   |           |         | 22.889            |           | 33 833    |            | 6 556    | 25 667  |          | 54 667  |             | 53 667     |          | 6 667   | 34 333     | 23 667                                 |                   | 45       | 24 167    |
| SD (ug)                                                                                        | 1 155    |           |         | 0 192             |           | 0 289     |            | 0.385    | 0.577   |          | 0 577   |             | 0.577      |          | 0.577   | 0.577      | 0.577                                  |                   | 1        | 0 577     |
| 4.67                                                                                           |          |           |         |                   |           |           |            |          |         |          |         |             |            |          |         |            |                                        |                   |          |           |
| size fraction (µm)                                                                             | 500-550  |           |         | 650-700           |           | 600-650   |            | 400-450  |         |          |         |             |            |          | 400-450 | 600-650    | 500-550                                |                   | 650-700  | 450-500   |
| nº of individuals                                                                              | 1        |           |         | 1                 |           | 1         |            | 1        |         |          |         |             |            |          | 1       | 2          | 6                                      |                   | 2        | 1         |
| average size (µm)                                                                              | 527      |           |         | 656               |           | 603       |            | 439      |         |          |         |             |            |          | 412     | 640        | 534.5                                  |                   | 676      | 482       |
| average weight (µg)                                                                            | 36.667   |           |         | 25.667            |           | 50.667    |            | 13.667   |         |          |         |             |            |          | 13      | 54.833     | 30.278                                 |                   | 84.333   | 35        |
| SD (µg)                                                                                        | 0.577    |           |         | 1.155             |           | 0.577     |            | 1.155    |         |          |         |             |            |          | 0       | 0.289      | 0.096                                  |                   | 0.289    | 1         |
| size fraction (µm)                                                                             | 550-600  |           |         |                   |           | 650-700   |            | 450-500  |         |          |         |             |            |          | 450-500 | 650-700    |                                        |                   | 750-800  | 500-550   |
| nº of individuals                                                                              | 6        |           |         |                   |           | 6         |            | 1        |         |          |         |             |            |          | 1       | 2          |                                        |                   | 1        | 1         |
| average size (µm)                                                                              | 578.667  |           |         |                   |           | 674.333   |            | 460      |         |          |         |             |            |          | 476     | 656.5      |                                        |                   | 762      | 509       |
| average weight (µg)                                                                            | 45.389   |           |         |                   |           | 47.889    |            | 17.333   |         |          |         |             |            |          | 24      | 63.333     |                                        |                   | 136      | 42        |
| SD (µg)                                                                                        | 0.096    |           |         |                   |           | 0.096     |            | 1.155    |         |          |         |             |            |          | 1       | 0.289      |                                        |                   | 0        | 0         |
|                                                                                                | (00      |           |         |                   |           |           |            |          |         |          |         |             |            |          |         |            |                                        |                   |          |           |
| size fraction (µm)                                                                             | 600-650  |           |         |                   |           | /00-750   |            |          |         |          |         |             |            |          | 500-550 |            |                                        |                   |          |           |
| n° of individuals                                                                              | 1        |           |         |                   |           | 2         |            |          |         |          |         |             |            |          | 3       |            |                                        |                   |          |           |
| average size (µm)                                                                              | 18407    |           |         |                   |           | 720       |            |          |         |          |         |             |            |          | 21 779  |            |                                        |                   |          |           |
| average weight (µg)                                                                            | +0.00/   |           |         |                   |           | 54        |            |          |         |          |         |             |            |          | 21.//8  |            |                                        |                   |          |           |
| SD (µg)                                                                                        | 0.577    |           |         |                   |           | 0         |            |          |         |          |         |             |            |          | 0.192   |            |                                        |                   |          |           |

**808 (Appendix A, cont.).**

|                     |          |           |         | South-
Central |           |           |            | Off      |         |          |         |             |            |          |         |            |            | North-
Central | Central |           |
|---------------------|----------|-----------|---------|-------------------|-----------|-----------|------------|----------|---------|----------|---------|-------------|------------|----------|---------|------------|------------|-------------------|---------|-----------|
|                     |          |           | Alboran | Western           | Strait of | Strait of | South of   | Southern | Eastern | Off Nile | Off     | Antikythera | Eastern    | Adriatic | Otranto | Northern 7 | Tyrrhenian | Western           | Western | Catalano- |
| Location            | Atlantic | Gibraltar | Sea     | Med.              | Sardinia  | Sicily    | Ionian Sea | Crete    | Basin   | Delta    | Lebanon | Strait      | Ionian Sea | Sea      | Strait  | Ionian Sea | Sea        | Med.              | Med.    | Balear    |
| Station             | 1        | 2         | 3       | 5                 | 6         | 7a        | 9          | 10       | 11      | 12       | 13      | 14          | 15         | 17       | 16      | 16-18      | 19         | 20                | 21      | 22        |
| size fraction (µm)  | 650-700  |           |         |                   |           | 750-800   |            |          |         |          |         |             |            |          | 550-600 |            |            |                   |         |           |
| nº of individuals   | 1        |           |         |                   |           | 1         |            |          |         |          |         |             |            |          | 1       |            |            |                   |         |           |
| average size (µm)   | 651      |           |         |                   |           | 772       |            |          |         |          |         |             |            |          | 570     |            |            |                   |         |           |
| average weight (µg) | 50.667   |           |         |                   |           | 48        |            |          |         |          |         |             |            |          | 17.333  |            |            |                   |         |           |
| SD (µg)             | 0.577    |           |         |                   |           | 1         |            |          |         |          |         |             |            |          | 1.528   |            |            |                   |         |           |
| size fraction (µm)  |          |           |         |                   |           |           |            |          |         |          |         |             |            |          | 600-650 |            |            |                   |         |           |
| nº of individuals   |          |           |         |                   |           |           |            |          |         |          |         |             |            |          | 1       |            |            |                   |         |           |
| average size (µm)   |          |           |         |                   |           |           |            |          |         |          |         |             |            |          | 625     |            |            |                   |         |           |
| average weight (µg) |          |           |         |                   |           |           |            |          |         |          |         |             |            |          | 23      |            |            |                   |         |           |
| SD (µg)             |          |           |         |                   |           |           |            |          |         |          |         |             |            |          | 0       |            |            |                   |         |           |
|                     |          |           |         |                   |           |           |            |          |         |          |         |             |            |          |         |            |            |                   |         |           |
| size fraction (µm)  |          |           |         |                   |           |           |            |          |         |          |         |             |            |          | 650-700 |            |            |                   |         |           |
| nº of individuals   |          |           |         |                   |           |           |            |          |         |          |         |             |            |          | 2       |            |            |                   |         |           |
| average size (µm)   |          |           |         |                   |           |           |            |          |         |          |         |             |            |          | 654.5   |            |            |                   |         |           |
| average weight (µg) |          |           |         |                   |           |           |            |          |         |          |         |             |            |          | 31.167  |            |            |                   |         |           |
| SD (µg)             |          |           |         |                   |           |           |            |          |         |          |         |             |            |          | 0.289   |            |            |                   |         |           |
|                     |          |           |         |                   |           |           |            |          |         |          |         |             |            |          |         |            |            |                   |         |           |

**810 Acknowledgments**

- 811 We thank the captain and crew of the Spanish research vessel R/V Ángeles Alvariño. B. d'Amario is thanked for her software
- 812 guidance and overall advice as well. The work was funded by the EC FP7 'Mediterranean Sea Acidification in a changing climate'
- 813project(MedSeA;grantagreement265103).814

| 887
888        | Grasshoff, K., Ehrhardt, M., Kremling, K.: Methods of seawater analysis, third ed., Verlag Chemie, Weinheim, Deerfield Beach, Florida, Basel, 1983.                                                                                                                                                                                    |
|-------------------|----------------------------------------------------------------------------------------------------------------------------------------------------------------------------------------------------------------------------------------------------------------------------------------------------------------------------------------|
| 889
890        | Vergraud-Grazzini, CV., Glaçon, C., Pierre, C., Pujol, C., and Urrutiaguer, M. J.: Foraminifères planctoniques de Méditerranée en fin d'été. Relations avec les structures hydrologiques, Mem. Soc. Geol. Ital., 36, 175-188, 1986.                                                                                                    |
| 891
892
893 | Hassoun, A. E. R., Gemayel, E., Krasakopoulou, E., Goyet, C., Saab, M. A., Ziveri, P., Touratier, F., Guglielmi, V., and Falco, C.:
Modeling of the total alkalinity and the total inorganic carbon in the Mediterranean Sea, J. Water Res. Ocean Sci., 4 (1), 24-32,
2015a.                                                     |
| 894
895        | Hassoun, A. E. R., Guglielmi, V., Gemayel, E., Goyet, C., Saab, M. A., Giani, M., Ziveri, P., Ingrosso, G., and Touratier, M.: Is the
Mediterranean Sea circulation in a steady state, J. Water Res. Ocean Sci., 4 (1), 6-17, 2015b.                                                                                                |
| 896
897        | Hembelen, Ch., Spindler, M., and Anderson, O.R.: Modern Planktonic Foraminifera, Springer-Verlag, New York, Berlin, Heidelberg, 363 pp., 1989.                                                                                                                                                                                         |
| 898
899
900 | Hernández-Almeida, I., Bárcena, M. A., Flores, J. A., Sierro, F. J., Sanchez-Vidal, A., and Calafat, A.: Microplankton response to
environmental conditions in the Alboran Sea (Western Mediterranean): One year sediment trap record, Mar. Micropaleontol., 78,
14-24, 2011.                                                    |
| 901
902        | Ivanova, E., Schiebel, R., Singh, A.D., Schmiedl, G., Niebler, H.S., and Hemleben, C.: Primary production in the Arabian Sea
during the last 135 000 years. Palaeogeogr. Palaeoclimatol. Palaeoecol., 197, 61 82, 2003.                                                                                                             |
| 903
904        | Köhler-Rink, S. and Kühl, M.: The chemical microenvironment of the symbiotic planktonic foraminifer Orbulina universa , Mar. Biol. Res., 1, 68-78, 2005.                                                                                                                                                                        |
| 905
906        | Kuroyanagi, A. and Kawahata, H.: Vertical distribution of living planktonic foraminifera in the seas around Japan, Mar. Micropaleontol., 53, 173-196, 2004.                                                                                                                                                                            |
| 907
908
909 | Lewis, E., Wallace, D., and Allison, L. J.: Program developed for CO2 system calculations, Tennessee: Carbon Dioxide Information
Analysis Center, managed by Lockheed Martin Energy Research Corporation for the US Department of Energy, 1998.                                                                                     |
| 910
911        | Lombard, F., Rocha, R. E., Bijma, J., and Gattuso, J. P.: Effect of carbonate ion concentration and irradiance on calcification in planktonic foraminifera, Biogeosciences, 7, 247-255, 2010.                                                                                                                                          |
| 912
913        | Marshall, B. J., Thunell, R. C., Henehan, M. J., Astor, Y., and Wejnert, K. E.: Planktonic foraminiferal area density as a proxy for carbonate ion concentration: A calibration study using the Cariaco Basin ocean time series, Paleoceanography, 28, 363-376, 2013.                                                                  |
| 914
915        | Marshall, B. J., Thunell, R. C., Spero, H. J., Henehan, M. J., Lorenzoni, L., Astor, Y.: Morphometric and stable isotopic differentiation of Orbulina universa morphotypes from the Cariaco Basin, Venezuela, Mar. Micropaleontol., 120, 46-64, 2015.                                                                           |
| 916
917        | Mehrbach, C.: Measurement of the apparent dissociation constants of carbonic acid in seawater at atmospheric pressure, M.S., Oregon State University, Oregon, United States, 1973.                                                                                                                                                     |
| 918
919        | Mohan, R., Shetye, S. S., Tiwari, M., and Anilkumar, N.: Secondary calcification of planktic foraminifera from the Indian sector of Southern Ocean, Acta Geol. SinEngl., 89 (1), 27-37, 2015.                                                                                                                                          |
| 920
921
922 | Mojtahid, M., Manceau, R., Schiebel, R., Hennekam, R., and de Lange, G.J.: Thirteen thousand years of southeastern Mediterranean climate variability inferred from an integrative planktic foraminiferal-based approach: Holocene climate in the SE Mediterranean, Paleoceanography, 30 (4), 402–422, doi: 10.1002/2014PA002705, 2015. |
| 923               | Morard, R., Quillévéré, F., Escarguel, G., Ujiie, Y., Garidel-Thoron, T., Norris, R. D., and de Vargas, C.: Morphological recognition                                                                                                                                                                                                  |

[revised manuscript text omitted]

---

## Referee Report (RR1)

**Review by Manuel F. G. Weinkauf Manuel.Weinkauf@unige.ch**

I have been reviewing the manuscript by Mallo et al. entitled 'Low planktic for aminiferal diversity and abundance observed in a 2013 West–East Mediterranean Sea transect', and submitted to the journal Biogeosciences, in its first revised version.

This paper studies planktonic Foraminifera, sampled with plankton nets in the upper 200 m water column during spring/summer 2013 across the entire Mediterranean Sea. It reports abundance patterns of several species across a Mediterranean transect which is characterized by large differences in physical ocean properties (e.g. temperature, salinity). It further tries to infer the influence of those environmental parameters on the abundance and shell calcification intensity of selected (abundant) species. The study finds that the species composition changes across the Mediterranean, with *Globigerina bulloides* and *Trilobatus sacculifer* dominating in the western part, *Globorotalia inflata* in the central part, and *Globigerinoides ruber* (white)/*Globigerinoides elongatus* in the east. The species investigated for their abundance and calcification intensity show distribution and calcification patterns that differ between regions in the Mediterranean Sea, and can partly be correlated with environmental factors.

I appreciate this study for its large potential in filling in gaps in our current knowledge about species distribution in the Mediterranean and their changes both seasonally and across longer timespans by comparison of their results with earlier studies. It can also be a significant contribution to the still relatively scarce set of literature about shell calcification in planktonic Foraminifera. The sections are logically ordered, and the abstract gives a sufficient and well structured overview over the manuscript, but some information is lacking throughout the manuscript (especially the Material and Methods section). Otherwise the manuscript has an appropriate length (although the discussion is rather long I do not think it is excessive). The figures and tables are suitable.

The manuscript have been significantly improved since the original version in most regards. There are still some problems however, that in my opinion make it unpublishable in its current state: (1) The studies compared used a variety of different sampling techniques which necessarily lead to different results. Why the authors inisist on interpreting them as they are instead of correcting their data to make them comparable is beyond me. (2) The PCA applied by the authors is a downgrade from the faulty but more eloquent approach in the first manuscript version. The statistics is still not appropriate for the data, and many trends are extracted by eyeballing instead of proper hypothesis testing (see General comments for details). I think, many of the conclusions reached by the authors are too bold in light of the very basic data analysis. The manuscript must be either toned down in terms of interpretation, or the quantitative analysis must be improved considerably. After this has been done, I would very much appreciate to see this study published in Biogeosciences.

**1 General comments**

In the section below, I give detailed comments (including line numbers) about very specific issues. However, in this section I already want to summarise some major points that are more relevant for the entire manuscript than at any specific place.

- 1. The work does still not normalize its data for the consistent differences in sampling employed by the other studies, with which comparisons of assemblages are anticipated. Cifelli (1974) sampled the upper 250 m water depth, while Pujol and Vergnaud-Grazzini sampled the upper 350 m. This study uses mainly the association at 200 m and partly an integrated column of the upper 200 m. Furthermore, mesh sizes have been different between most studies. In addition, the authors now state that their net had a diameter of only  $40 \,\mathrm{cm} \,(0.12 \,\mathrm{m}^2 \,\mathrm{opening})$ , in contrast to the  $0.5 \,\mathrm{m}^2$ common with most plankton nets (e.g. Pujol and Vergnaud Grazzini 1995). While absolute abundances are certainly normalized for filtered water volume, this much smaller net opening means that the authors have much larger errors in their assemblage data than the compared studies, because of the much lower volume of filtered sea water. All this has already been criticised in my first review, but the authors did not change anything, although I for instance suggested already there to use equations provided by Berger (1969) to normalize all studies concerning mesh sizes. The authors try to argue that Cifelli (1974), who actually used a comparable mesh size, argue in favour of their interpretation of changing abundances due to changing environments. However, they totally ignore that Cifelli (1974) used another depth range in their studies, so certainly they found other abundances. In my opinion, the authors cannot succesfully show, that the assemblage differences they observe between studies with employing such different sampling techniques are not an artefact of the data, but a real trend.
- 2. The systematics are still not consistent. Why is quadrilobatus designated as belonging to the genus Globigerinoides? From André et al. (2013), which the authors cite themselves, it is very clear that the species genetically belongs to the trilobus-sacculifer plexus (at least as long as recent specimens are concerned). It makes absolutely no sense to not only treat it as a separate species from Trilobatus sacculifer, but even put it into another genus. It should instead be correctly categorized as another morphotype of T. sacculifer.
- 3. The statistical analyses is still a huge problem. The authors state they applied a principal components analysis (PCA), which by the way is data visualization and no proper statistics (because it lacks any possibility to infer significance), and thus a step back from the faulty approach the authors applied in the first iteration of this paper. However, PCA does not include explanatory variables such as environmental parameters. So it is

first not clear to me what have been done, i.e. what are Factors 1 and 2 in Fig. 7? Have samples (as it seems) been ordinated by environment, and then somehow overlain by assemblages? Or is it indeed a redundancy analysis that have been applied, and if so, constrained for which environmental parameters? Furthermore, since PCA is using euclidean distances for ordination, it is very unsuitable for abundance data, and other methods like principal coordinates analysis are much more suitable for comparing assemblages (Hammer and Harper, 2006; Legendre and Legendre, 2012). The authors also still do not use proper techniques to interpret their findings in relation to the hefty multicollinearity in their data. I suggested some techniques in my first review (e.g. GLM, GAM). The authors may also use any of the techniques applied by the Thunell-work group, who also do an excellent job in that (e.g. Marshall et al., 2013; Osborne et al., 2016). As it is now, however, the authors only visually interpret trends in the PCA by eye, which is no proper and robust method when reliable interpretations should be reached.

**2 Detailed comments**

Line 50, 'Pujol and Vergraud-Grazzini, 1995': This work is consistently misspelled. It should be Pujol and Vergnaud Grazzini, 1995!

Line 52, 'bottom sediments': Should be 'surface sediments'.

Line 63, 'prominent differenced': Should be 'prominently different'.

Line 65, 'retrieved in different sites': Should be 'retrieved from different sites'.

Line 69, 'hydrographis': Should be 'hydrographic'.

Line 79, 'study being carried out': Should be 'study have been carried out'. Lines 97f, 'For further studies that relate foraminiferal calcification with environmental parameters see Weinkauf et al. (2016); Table 7.': You should also cite Marshall et al. (2013) in this regard.

Lines 106f, 'In addition, few size-normalized weight (SNW) and area density ( $\rho_A$ ) studies from water column foraminifera are available in the literature': Area density is a form of size-normalized weight.

Line 112, 'spring2013': Should be 'spring 2013'.

Lines 120–122, 'In addition, empty tests are passive particles that ocean currents may displace horizontally, but that displacement is negligible due to their quick settling velocities (Caromel et al., 2014).': This is not always correct, and it might be good to show that drift distances in the Mediterranean are actually very low (van Sebille et al., 2015).

Line 146, 'become': Should be 'becomes'.

Line 166. 'primarily 200 m depth': Should be 'primarily from 200 m depth'. Line 179, 'MODIS Aqua L2 satellite': Should be 'MODIS Aqua L2 satellite data'.

Lines 186f, 'Samples were studied from the collecting bottles and the bottom collector, the latter representing 52.33% of the total sample

were treated in aliquots of 1/2, 1/4, 1/6, until 1/8.': I do not understand this sentence.

Line 188, ' $\geq$ 350–500  $\mu$ m': Should be '350–500  $\mu$ m'.

Line 199, 'Globigerinella siphonifera/G. calida/G. radians plexus': Should be 'The Globigerinella siphonifera/G. calida/G. radians plexus'.

Line 204f, 'the individuals were weighed together by triplicate with a Mettler Toledo XS3DU microbalance': Which means the authors were actually applying the mean area density approach as described in Weinkauf et al. (2013) instead of the more advanced area density approach as described by Marshall et al. (2013).

Lines 216, 'The PCA was performed on the environmental parameters:': So how to understand this? The samples were ordinated by environmental parameters? What then are the scores of the black axes, passively projected assemblage scores? Or is this indeed a redundancy analysis instead of PCA? Compare also general comments why PCA is unsuitable anyways.

Line 218, '(Fig. 7)': What happened to Figs 3–6, which should be cited in the text before Fig. 7?

Lines 218–228, 'The first factor exhibited positive loadings... are shown in Figure 7).': This entire passage belongs into the Results section.

Line 244, 'The exceptions are at Station 3...': And what about stations 1 and 6?

Lines 246f, 'The 350–500- $\mu$ m size fraction dominates in the western Mediterranean and is progressively reduced eastwards (Fig. 4)': I do not see this trend. This could be due to the bad layout of figs 3 and 4 (see below).

Line 272, 'G. quadrilobatus': Incorrect genus (see General Comments).

Lines 274–276, 'The PCA performed on the environmental parameters and the sample scores on the two first components clearly shows a separation, regarding Factor 1, between the western and eastern Mediterranean stations (Fig. 7).': I do not understand how this 'PCA' was performed. Did it ordinate the samples on environmental data (as seems the case), then what are the black factors in fig. 7? Or is it indeed an RDA, then constrained for which environmental factors?

Line 278, 'station 10 is an exception': But stations 1, 6, 20, 21, and 22 (all Western Mediterranean) all have low a abundances as well.

Line 279, 'Factor 2': Should be 'principal component 2' or 'PC 2'.

Lines 283–285, 'Overall, the highest absolute abundance of all foraminifera seems related to food availability and only secondarily to the carbonate system (Fig. 7a).': While it makes the impression to be true, as it is this is eyeballing, because PCA cannot yield any significance but is only ordinating datapoints. Since many of your environmental factors show multicollinearity (as I already pointed out in my first revision) you need much more advanced, real statistical methods to say exactly whith which factors correlation is greatest. At the very least, you should use a more appropriate ordination method for abundances (probably constrained ordination, which at least delivers a significance for the overall correlation of data with environmental factors) than PCA, which uses euclidean distances.

Lines 286–292, 'With the exception ... path of Atlantic waters (Fig. 7b)': Where do you see this? *Globigerinoides ruber (white)* shows a peak (the richest sample) on the cold side of the ordination space, and *G bulloides* seems to be more correlated with pH. To convince me that those trends are true, you would have to show me something more robust than just a PCA impression (i.e. a compositional multiple regression as described by van den Boogart and Tolosana-Delgado (2013), as I also already suggested last time).

Lines 298f, 'The Atlantic and the Ionian–Adriatic–Aegean grouping have similar proportions of species.': Except that from Atlantic to Ionian–Adriatic–Aegean grouping dominances are completely shifted: *G. ruber* becomes much more dominant, *G. bulloides* and *T. sacculifer* are strongly reduced in abundance, *O. universa* is much more prevalent, and *G. inflata* is hardly there anymore.

Lines 313f, 'The high two-dimensional (silhouette) area-to-long axis correlation is best fitted by a power regression (Fig. S2).': Which, as I already argued in the first review, should be forced to have zero offset. The authors argued concerning this 'Size and mass of foraminifers relationship does not start at the origin (zero). The proloculus of planktic foraminifera measures between  $15-30 \,\mu$ m in average, and has a certain calcite mass, which has so far not been determined (see Hemleben et al., 1989).'. This, however, only means that the model should stop short of zero. Especially when the authors argue that a zero-intercept model would not make sense because it would imply the existence of individuals with zero mass and size, is it not logical to them that non-zero-intercept model which allows a foraminifer to have mass at size zero or have a certain size without mass is even more problematic!

Lines 314f, 'The same growth pattern can be seen in *G. ruber* (white), *G. bulloides*, and *O. universa*': But this assumption is wrong at least in *O. universa*. There, size increase cannot be growth, because the spherical form is the terminal form and cannot grow considerably anymore.

Lines 318f, 'The specimens of G. ruber (white) from the Atlantic have the largest size followed by individuals from the Tyrrhenian Sea, and those from the eastern Ionian Sea.': If this statement is made, I already requested a statistical proof in the last review, to which the authors responded 'We do not need a statistical test to know which is the smallest value'. Since this shows a complete lack of understanding for the nature of any quantitative analysis, here is a short Statistics 101 (I again refer the authors to basic introductory literature such as Hammer and Harper (2006) or Dytham (2011): When dealing with natural values, one value will **always** be larger than the other when measured accurately enough. The question you want to answer is not, is one value larger, to which you know the answer beforehand, but is one value **significantly** larger. This means, is the difference you observe between the values in two random samples large enough that, taking into account uncertainty from the fact that you only sampled a couple of randomly selected specimens from the population, you can be reasonably sure that the populations the samples were drawn from differ in this value. An easy example:

I measure a difference of 0.3 cm between two samples. Do the populations from which those samples have been drawn differ in size? Well, when I use the variation in the samples to estimate the uncertainty in the estimate of the mean, I can tell with a certain probability. When the standard deviation in both samples (of, say, 100 specimens each) is 0.2 cm, then the 95% confidence interval is  $\pm 0.02$  cm, so the two populations do differ in size with a probability of more than 95%. If the standard deviation is 5 cm, in contrast, the 95% confidence interval is  $\pm 0.5$  cm, so the two populations do not show a significant difference in size. This is, what statistics is for, and in this sense, yes, you do need statistics to know which value is smaller!

Line 337f, 'higher  $\rho_A$  are related to slightly lower pH and higher food availability in the western Mediterranean and Atlantic stations': This must be proven, and from the PCA I doubt the pH relationship.

Line 340, 'opposite trend as in *G. ruber* (white)': Should be 'opposite trend than *G. ruber* (white)'.

Line 367f, 'Within the Mediterranean, a previous study with results comparable to ours, sampled the upper 350 m (Pujol and Vergraud-Grazzini, 1995).': They also sampled with another mesh size, for which still no corrections have been applied.

Line 401, 'smallesr': Should be 'smaller'.

Lines 409–411, 'The lower absolute abundance of individuals in our study compared to Pujol and Vergraud-Grazzini (1995), together with low species diversity in the Mediterranean, may indicate a trend of changing conditions over the last decades, ...': I still believe that this has to do more with the different mesh-sizes. The size fraction between  $120 \,\mu\text{m}$  and  $150 \,\mu\text{m}$  in my experience contains a lot of the standing stock of foraminifers. Section Factors controlling the abundance of the main species: All trends described here are purely derived from the PCA by eye, without any appropriate test. While their explanation can be valuable, their interpretation should be toned down considerably.

Lines 445f, 'The increasing dominance of G. ruber (white) from the western to the eastern Mediterranean Basin coincides with the east-ward increasing salinity (Fig. 7d).': Or Temperature, or  $CO_2$ . It is hard to say without proper analytical techniques under this degree of multicollinearity. Line 537: Remove second 'its'.

Line 548, 'but abundances are slightly higher in the western basin to than the east.': I highly doubt that from the PCA alone. You could prove it though.

Line 569f, 'In contrast, the  $\rho_A$  of *O. universa* does not show any change between the western and eastern basins (Fig. 7i), and cannot be used to identify and quantify particular environmental effects.:' I also doubt that there is a difference between basins in *G. bulloides*, and since the authors still refuse to use proper quantitative techniques to prove it ...

Line 615, 'larger IQR indicates ...': This is only true, when the variation in the sample is normalized for expected value (i.e. mean). This means, calculating the coefficient of variation, which I already requested in the first review. The authors replied 'As also described above, in our comment to the reviewer comment about lines 480–482, we are unsure about what statistical method and/or calculation the reviewer is referring to here. Is there a distinct suggestion of some kind, with a reference? We are not sure how to calculate a "coefficient of variation" with regard to box plots and their statistics.'. No, I do not have a reference for it, because the coefficient of variation is such a basic and old method that its origins are lost in the mist of time, and you would not cite a reference as you would not cite a reference when calculating a mean value. Rather, the coefficient of variation is explained (and listed in the index) in every basic statistics book I suggested the authors to consult in my first review. It is also very easily found using Google and the search term 'coefficient of variation'. Again, in short, variation is always correlated to mean value, so variations of samples which mean value differs must be corrected for this stochastic effect. An example: Let's say you measured the length of twenty mice and found it to be  $3\pm0.5$  cm. You also measured the length of 20 elephants and found it to be  $4\pm0.5$  m. Which species has the higher variation? The absolute value is much larger for elephants (0.5 m) than for mice (0.5 cm), but when calculating the coefficient of variation you actually find mice to be more variable in size (0.166) than elephants (0.125). Since none of the IQRs in the manuscript are corrected (and I would recommend to use the standard deviation instead of the IQR anyways) all conclusions drawn by the authors concerning variation in their samples are invalid.

Line 624, 'variability in  $\rho_A$  data increases with increasing absolute  $\rho_A$ ': Exactly as stochastically predicted. Calculate the coefficient of variation and compare again.

Line 633, 'retarded': Should be 'hampered'.

Line 636, 'seems': Should be 'seem'.

Line 640, 'suited conditions': Should be 'suitable conditions'.

Line 648, ''heavier average': Should be 'steeper average', maybe.

Line 651f, 'All of these findings support our idea of an effect of limited alimentation on calcification.': I do not understand this sentence.

Caption Fig 4 'Sample size is indicated by n below each station code.': This information is not present in the figure.

Figs 3 and 4: A lot of the interpretation by the authors in concerned with east-west trends. Then why are the graphs not ordered west-east, instead of by station number?

**References**

André, A., Weiner, A., Quillévéré, F., Aurahs, R., Morard, R., Douady, Ch. J., de Garidel-Thoron, Th., Escarguel, G., de Vargas, C., and Kučera, M.: The Cryptic and the Apparent Reversed: Lack of Genetic Differentiation within the Morphologically Diverse Plexus of the Planktonic Foraminifer *Globigerinoides sacculifer*, Paleobiology, 39, 21–39, doi:10.1666/0094-8373-39.1.21, 2013.

- Berger, W. H.: Ecologic Patterns of Living Planktonic Foraminiferal, Deep-Sea Research and Oceanographic Abstracts, 16, 1–24, doi:10.1016/0011-7471(69)90047-3, 1969.
- Dytham, C.: Choosing and Using Statistics: A Biologist's Guide, Wiley–Blackwell, Oxford and Chichester and Hoboken, 3 edn., 2011.
- Hammer, Ø. and Harper, D.: Paleontological Data Analysis, Blackwell Publishing, Malden and Oxford and Carlton, 2006.
- Legendre, P. and Legendre, L.: Numerical Ecology, no. 24 in Developments in Environmental Modelling, Elsevier, Amsterdam and Oxford, 3 edn., 2012.
- Marshall, B. J., Thunell, R. C., Henehan, M. J., Astor, Y., and Wejnert, K. R.: Planktonic Foraminiferal Area Density as a Proxy for Carbonate Ion Concentration: A Calibration Study using the Cariaco Basin Ocean Time Series, Paleoceanography, 28, 1–14, doi:10.1002/palo.20034, 2013.
- Osborne, E. B., Thunell, R. C., Marshall, B. J., Holm, J. A., Tappa, E. J., Benitez-Nelson, C., Cai, W.-J., and Chen, B.: Calcification of the Planktonic Foraminifera *Globigerina bulloides* and Carbonate Ion Concentration: Results from the Santa Barbara Basin, Paleoceanography, 31, 1083–1102, doi: 10.1002/2016PA002933, 2016.
- van den Boogart, K. G. and Tolosana-Delgado, R.: Analyzing Compositional Data with R, Use R!, Springer-Verlag, Berlin and Heidelberg, doi: 10.1007/978-3-642-36809-7, 2013.
- van Sebille, E., Scussolini, P., Durgadoo, J. V., Peeters, F. J. C., Biastoch, A., Weijer, W., Turney, Ch., Paris, C. B., and Zahn, R.: Ocean Currents Generate Large Footprints in Marine Palaeoclimate Proxies, Nature Communications, 6, Article 6521, doi:10.1038/ncomms7521, 2015.
- Weinkauf, M. F. G., Moller, Т., Koch, M. С., and Kučera, Planktonic M.: Calcification Intensity inForaminifera Reflects Ambient Conditions Irrespective of Environmental Stress. Bio-6639-6655, doi:10.5194/bg-10-6639-2013, URL geosciences, 10,http://www.biogeosciences.net/10/6639/2013/bg-10-6639-2013.html, 2013.

---

## Referee Report (RR2)

I have been reviewing the manuscript by Mallo et al. entitled 'Low planktic foraminiferal diversity and abundance observed in a spring 2013 West–East Mediterranean Sea plankton tow transect', and submitted to the journal Biogeosciences, in its second revised version.

This paper studies planktonic Foraminifera, sampled with plankton nets in 200 m water depth during spring/summer 2013 across the entire Mediterranean Sea. It reports abundance patterns of several species across a Mediterranean transect which is characterized by large differences in physical ocean properties (e.g. temperature, salinity). It further tries to infer the influence of those environmental parameters on the abundance and shell calcification intensity of selected (abundant) species. The study finds that the species composition changes across the Mediterranean, with *Globigerina bulloides* and *Trilobatus sacculifer* dominating in the western part, *Globorotalia inflata* in the central part, and *Globigerinoides ruber* (white)/*Globigerinoides elongatus* in the east. The species investigated for their abundance and calcification intensity show distribution and calcification patterns that differ between regions in the Mediterranean Sea, and can partly be correlated with environmental factors.

I appreciate this study for its large potential in filling in gaps in our current knowledge about species distribution in the Mediterranean and their changes both seasonally and across longer timespans by comparison of their results with earlier studies. It can also be a significant contribution to the still relatively scarce set of literature about shell calcification in planktonic Foraminifera. The sections are logically ordered, and the abstract gives a sufficient and well structured overview over the manuscript. Otherwise the manuscript has an appropriate length (although the discussion is rather long I do not think it is excessive). The figures and tables are suitable.

The manuscript has again been improved since the first revised version. Doing so, however, revealed a new problem of which I was not aware. The major problem is that the data are even less comparable to earlier studies than I believed so far, because they do not represent the depth range that has so far been indicated by the authors (compare General comments). As a result, part of the comparison with older works must be further toned down. Additionally, there are still some small issues with the analyses, but those can now be dealt with reasonably quickly.

I must admit that the analytical quality in the manuscript is not to the highest standards (I still believe that principal component analysis (PCA) is a subpar analysis for what the authors want to do). But I do not see how this can be helped, and although I would have wished for a more thorough and robust analysis, PCA as applied by the authors (i.e. for ordinating on the environmental parameters) is not wrong. I therefore believe that once

the authors dealt with the remaining problems, we finally reached the state where the manuscript can be published in Biogeosciences.

**General comments**

I have only one major problem left with the new iteration of the manuscript, that is a change in the description of the data. So far I, was led to believe, that the assemblages investigated by the authors have a bias towards coming from 200 m depth, but that they are representative of the entire water column from 0–200 m. This was indicated by the phrasing used by the authors in the first revised manuscript version, lines 165–167 'Twenty samples were collected with BONGO nets (mesh size 150 μm and 40 cm diameter, for further details see Posgay, 1980) primarily 200 m depth, but also including tow time integrating the upper water column from 200 m to the surface (Table 1).'. This statement has now been revised to 'Those nets sampled primarily 200 m depth, but also caught foraminifera during the net descent and ascent to the ocean surface, which both involve **negligible** towing and capturing time compared to the sampling at 200 m depth (Table 1).' (lines 167–169, emphasis by me).

This statement is in stark contrast to the statement in the first revised version of the manuscript. It now means that the data available to the authors are even less suitable for any comparison with earlier studies, which always used the data from the surface to a particular depth. The authors now not only have a snap-shot in time, but also in space, which is very critical given the distinct depth distribution of planktonic foraminiferal species (Rebotim et al., in press) when one wants to compare abundance data with other studies. It in fact means, that the authors should have missed representative populations of all species identified as shallow and most species from the intermediate depth range *sensu* Rebotim et al. (in press). As it stands, I do not see how this can be corrected in any way, and actually I doubt that the assemblages mainly represent the standing stock at 200 m depth, because of the large abundances of shallow species like *G. ruber* (white) and *T. sacculifer* (both occurring mainly above 70 m, compare Rebotim et al. (in press)) in some samples. Alas, the authors seem not to know what exactly they were catching themselves either, which is why I have to conclude that any attempt to compare the assemblages presented here with any of the older studies on a more than very basic quantitative level are futile and should be taken out of the manuscript. It also raises the question of how well and with what attention the experimental design was thought through, because if the goal was a comparison with earlier studies, it would have been wise to stick as close as possible to sample schemes of such earlier studies, instead of doing something basically incomparable.

My request is therefore to further tone down the comparison with earlier studies, specifically removing the text passages between lines 397 and 416.

**Detailed comments**

**Line 40, 'predatory presence':** I have never heard that. In fact, we do not have the faintest idea if there are selective predators targeting Foraminifera.

**Lines 41f, '(i.e., Schiebel and Hemleben, 2005; Hemleben et al., 1989':** Replace 'i.e.' with 'e.g.', consider citing Rebotim et al. (in press).

**Lines 50f, 'Pujol and Vergnaud-Grazzini, 1995':** I still believe it should be 'Pujol and Vergnaud Grazzini, 1995' (without hyphen) in the entire manuscript, since this is how the author wants to be referred to according to the original publication.

**Lines 105f, 'Gulf of Naples (de Castro Coppa et al., 1980), the Alboran Sea':** Should be 'Gulf of Naples (de Castro Coppa et al., 1980) or the Alboran Sea'.

**Lines 167–169, 'Those nets sampled primarily 200 m depth, but also caught foraminifera during the net descent and ascent to the ocean surface, which both involve negligible towing and capturing time compared to the sampling at 200 m depth (Table 1).':** This new information makes the data collected by the authors now fully unsuitable for any in-depth quantitative comparison with earlier studies. Compare General comments.

**Lines 176f, ' These three parameters of the carbonate system were then integrated for the upper 200 m water depth.':** Which does not make any sense when your specimens are basically exclusively from 200 m water depth.

**Line 188, 'When necessary, samples were split into aliquots of 1/4 and 1/6':** How do you split to $\frac{1}{6}$? All splitters I am aware of split a sample in two. So you get $\frac{1}{2}$ from the first split, $\frac{1}{4}$ by splitting one of the $\frac{1}{2}$ samples, and then $\frac{1}{8}$ when splitting one of the $\frac{1}{4}$ samples. There is no reasonable way to end up with $\frac{1}{6}$ using this technique.

**Line 202, 'G. ruber':** Should probably be 'G. ruber (white)'.

**Line 207, '($\pm 1\,\mu g$ of nominal precision)':** As I already requested in my first review, the measurement error must be analysed in more detail. A precision of $1\,\mu g$ is rather low for such a study (normally it should be at $0.2\,\mu g$ or below). Given that many measurement values in the appendix are only 2–4 times larger than the precision of the balance, this leads to theoretical errors of 25–50 %. The authors must show, that the measurement in triplicate is enough to yield meaningful results across all weights.

**Line 217, 'Varimax rotation':** Varimax is an orthogonal rotation method, which is not suitable if factors are correlated, which is likely the case here. The authors should either use an oblique rotation method (e.g. oblimin or promax), or at the very least show that

no correlation problem exists according to Tabachnik and Fidell (1996) (i.e. correlations between all relevant factors (PCs 1 and 2) after oblique rotation must be $< 0.32$).

**Line 221, 'salinity (and to a lesser degree on $\left[CO_3^{2-}\right]$;':** I would say salinity and carbonate saturation leaks into both axes.

**Lines 249–250, 'no differences are observed between samples collected during day and night.':** Consider showing this graphically.

**Lines 280f, 'stations influenced by the incoming waters from the Atlantic and lowest $\left[CO_3^{-2}\right]$ values score highest.':** But this makes no sense with the PCA. Highest scores on the second PC are correlated with high $\left[CO_3^{2-}\right]$ values, which also makes more sense with the description of the authors that stations close to the Atlantic indeed plot at the lower end of the second axis. Additionally, $\left[CO_3^{-2}\right]$ should be $\left[CO_3^{2-}\right]$ in the entire manuscript.

**Line 288, 'and at stations where pH is higher':** Where? The highest abundances of *G. bulloides* are all on the negative side of the second PC (stations 3, 5, 2), correlating with low pH values.

**Lines 311-313, 'Similar growth patterns can be seen in *G. ruber* (white), *G. bulloides*, and *O. universa* with that correlation, graphically represented by the shape of a power function (Fig. S2).':** This is a misconception by the authors, that I already noted again in their response to both Reviewer #1 and myself. You cannot call those curves growth curves, as they do not represent ontogenetic growth! You simply have a lot of individuals at different sizes, which can be at very different or very similar ontogenetic stages depending on their individual ontogeny. The size of adult Foraminifera of the same species can be very different, and all shells above $150\,\mu\text{m}$ diameter can reasonably be considered adult and in a reproductive state (Peeters et al., 1999). What you have there is a diameter–cross-section scaling, and you have no idea how much of it is ontogenetic growth and how much is intra-specific variation.

**Lines 313–315, 'Planktic foraminifera grow faster when they are younger and smaller (steepest in the lower left part of the regression line) and slower when they are older and bigger (less steep in the upper right part of the regression line; Fig. S2).':** This is nonsense (see comment to lines 311–313), and could only be studied if you followed single individuals during their ontogeny by repeated measurements designs, or at least made a chamber-by-chamber analysis as in Caromel et al. (2015). What you see from the curves is rather the very trivial and predictable fact that a 1D and a 2D size measurement are linked via some kind of power function.

**Lines 315–317:** Those *p*-values must again be corrected for multiple testing, using for example corrections for the family-wise error rate (e.g. Bonferroni correction) or the false discovery rate (e.g. Benjamini and Hochberg, 1995). Since those analyses are basically the

post-hoc tests for an ANOVA, you may alternatively want to use the classical correction in that case, which would be Tukeys honest significance difference, as implemented in most packages. Certainly, all those pairwise comparisons should be performed for all possible pairs of groups defined in fig. 6, not just subjectively selected ones, as this artificially inflates the significance of the results. Compare my review on the original version of the manuscript for details.

**Lines 317f, ' In the other two species *G. bulloides* and *O. universa*, a similar trend is observed regarding the two basins,':** Then where are *p*-values to prove that?

**Line 322, 'The long axis-to-weight relation':** Those size–weight curves (most notably those shown in fig S4) are still not forced to go through origin, and the authors completely misrepresent the comments of both Reviewer #1 and myself in this regard in their response. Reviewer #1 wrote '...but disagree with assertion that the curves ...should not go through origin. That is physically impossible.' I wrote 'Especially when the authors argue that a zero-intercept model would not make sense because it would imply the existence of individuals with zero mass and size, is it not logical to them that non-zero-intercept model which allows a foraminifer to have mass at size zero or have a certain size without mass is even more problematic!'. In both cases, the authors effectively seem to believe (at least they argue as such in their response) that we would both agree that curves should **not** go through the origin. **This is demonstrably false!** Reviewer #1 seems adamant, that they have to go through origin. I would personally relax that necessity, depending on the purpose, as I already argued in my first review. If those curves are only supposed to be local approximations of a relationship, that should not be extrapolated, I am fine with them using simply the best fitting curve. However, it has been the authors who were constantly talking about those things as a growth curve, and it has been the authors who were arguing that a foraminifer of size and weight zero would not make sense, because the proloculus already has a certain size and weight. This implies that the authors believe the curves to be a biological model, that can be extrapolated across the reasonable size range of Foraminifera. As soon as this is the case, **the curves must be forced to go through the origin**, because otherwise they would allow a foraminifer with zero size and positive or negative mass, or zero mass and positive or negative size to exist. The authors must therefore either stop trying to sell these curves as some biological growth model, or at last apply proper regression to ensure a zero intercept of the fitted curves. Under no circumstances is it acceptable that they try to blatantly misinterpret what either of the reviewers criticised, just to fit their needs.

**Lines 324f, '*O. universa* was finally discarded for comparisons between $\rho_A$ at different locations due to a low area–weight correlation and no remarkable trend observable between locations (Fig. S4c; Fig. 3i);':** I absolutely do not understand this argumentation. The correlation (i.e. $R^2$ value) is only marginally lower for *O. universa* (0.64) than for *G. bulloides* (0.69). The correlation between cross-sectional area and weight is probably significant for both species, if it is not then *G. bulloides*

is the problematic species, not *O. universa*. And the PCA of area densities shows no clearer signal in *G. bulloides* than in *O. universa*. Yet *G. bulloides* is used in the ensuing analyses, while *O. universa* is not. This makes absolutely no sense!

**Lines 326–328, 'The eastern Mediterranean specimens are the lightest in both species (*G. ruber* (white), *G. bulloides*), with more extreme W–E differences in *G. ruber* (white) than in *G. bulloides* (Fig. S4d–e).':** Any prove of this statement, while probably true judged by the graphic, is still missing though.

**Lines 329f, 'The data of all the locations show a similar CV value.':** Well, there is certainly some variation. Calculating the 95 % confidence interval for the coefficient of variation (e.g. using the method by Vangel (1996) or a bootstrap approach) could help to interpret those values more meaningful.

**Lines 332f, 'highest median value ($1.55 \cdot 10^{-4}$ µg µm$^{-2}$) and IQR.':** I believed we now finally established, that variation is always correlated with mean value due to stochastic reasons. This is why you should, and finally did, calculate the coefficient of variation. So why do you still compare IQRs here?

**Line 337, 'seven locations were compared (Fig. 7).':** Why are the locations not the same for *G. ruber* and *G. bulloides*?

**Lines 338f, 'Specimens from the Atlantic have the lowest median $\rho_A$ ($8.75 \cdot 10^{-5}$ µg µm$^{-2}$) and the smallest IQR':** I believed we now finally established, that variation is always correlated with mean value due to stochastic reasons. This is why you should, and finally did, calculate the coefficient of variation. So why do you still compare IQRs here?

**Lines 343f, 'Results show a less clear overall trend for *G. bulloides* than for *G. ruber* (white), with higher $\rho_A$ associated with slightly higher pH in the eastern Mediterranean (Fig. 3h).':** I actually fail to see any trend in *G. bulloides* at all.

**Lines 397–416:** Given the new evidence of your true sampling depth (i.e. only 200 m, instead of the upper 200 m integrated with only a bias towards the 200 m level) I consider all attempts of in-depth quantitative comparisons to be futile. However, you can very well keep the qualitative comparisons in the paragraphs before here and in section 5.2.

**Line 434, 'Basinit':** Should be 'Basin it'.

**Line 449, 'abundanceat':** Should be 'abundance at'.

**Line 453, 'tempreatures':** Should be 'temperatures'.

**Line 487, 'Atlanticclose':** Should be 'Atlantic close'.

**Line 506, 'itdecreases':** Should be 'it decreases'.

**Line 550, '*G. bulloides*':** Neither from fig. 3 nor fig. 7, I see any such trend for *G. bulloides*, and I am certain that a quantitative comparison of area density between the basins shows no significant differences either.

**Line 569, 'The reason why the $\rho_\mathbf{A}$ of *O. universa* is particularly low and highly variable':** Where do you take this information from. Fig. 3i shows that the area density of *O. universa* is the highest of all investigated species, and its spatial variation does not seem to be exceedingly high.

**Line 574, 'The $\rho_\mathbf{A}$ of *G. ruber* (white) is only partly controlled by carbonate chemistry':** Just by eye from fig. 3g, I would say that the highest correlation of *G. ruber* area density is with $\left[\mathrm{CO_3}^{2-}\right]$ (i.e. diagonal trend from bottom right to top left).

**Line 575, 'similar to *O. universa*':** So now all of a sudden there is a trend in *O. universa*!?

**Lines 575f, 'In contrast to *O. universa*, the $\rho_\mathbf{A}$ data of *G. ruber* and *G. bulloides* follow systematic correlations.':** You have indeed more significant correlations in *O. universa* than in any of the other species, according to your table 2, so I do not know what you are talking about. Needless to say, that all correlations presented in table 2 must be corrected for multiple testing anyways, and I do not know how many will remain significant after this is done. For a further explanation of this I refer you to my review of the first iteration of this manuscript, where I already explained to you in great detail, why this is necessary and how it is done.

**Lines 584f, 'The $\rho_\mathbf{A}$ of *G. bulloides* tests increases from the Atlantic toward the eastern Mediterranean.':** I do see such a trend in neither fig. 3h nor fig. 7. This is also apparent from fig. S4, where the relationship between size and weight indicates that the area density is nearly constant: $3\,\mu\mathrm{g}/30\,000\,\mu\mathrm{m}^2 = 6\,\mu\mathrm{g}/60\,000\,\mu\mathrm{m}^2$.

**Line 585:** Delete 'In both species larger IQRs are found toward higher absolute $\rho_\mathbf{A}$ (Fig. 7).'

**Line 596, 'weight-to-long axis relations':** In the following paragraph you are comparing raw weights in a particular size fraction, not weight to size relations.

**Line 603, 'comparison of the weight-to-long axis relation':** In the following paragraph you are comparing raw weights in a particular size fraction, not weight to size relations.

**Line 621, 'upper 200 m of the water column':** According to your own words, you do not have data from the upper 200 m but only from more or less exactly 200 m depth.

**Caption table 2, 'r-values in bold are significant at p < 0.05, *p < 0.1.':** This is, as the analyses in the first version of the manuscript, again a case of hefty multiple testing. As such, all $p$-values of correlations per species would have to be corrected for this fact, either using family-wise error rate (e.g. Bonferroni) or false discovery rate (e.g. Benjamini and Hochberg, 1995) corrections! Compare my review to the original manuscript for an in-depth explanation, why this is necessary, and where to find further information. A graphical depiction of all significant correlations could further be very helpful. Please also note again, that $r$, $p$ and all such variables need to be typeset in italics.

**Table 2:** 'Density area' should be 'Area density'.

**Figure 5:** For station 12, a sample size of 1 is given. This can hardly be right. How can one individual belong to different size classes (as depicted in the barplot)?

**Figure 7:** Why are the regions different for *G. ruber* and *G. bulloides*?

**References**

Benjamini, Y. and Hochberg, Y.: Controlling the false discovery rate: A practical and powerful approach to multiple testing, J. R. Stat. Soc. B Met., 57, 289–300, doi:10.2307/2346101, 1995.

Caromel, A. G. M., Schmidt, D. N., Fletcher, I., and Rayfield, E. J.: Morphological change during the ontogeny of the planktic Foraminifera, J. Micropalaeontol., 35, 2–19, doi:10.1144/jmpaleo2014-017, 2015.

Peeters, F., Ivanova, E., Conan, S., Brummer, G.-J., Ganssen, G., Troelstra, S., and van Hinte, J.: A size analysis of planktic Foraminifera from the Arabian Sea, Mar. Micropaleontol., 36, 31–63, doi:10.1016/S0377-8398(98)00026-7, 1999.

Pujol, C. and Vergnaud Grazzini, C.: Distribution patterns of live planktic foraminifers as related to regional hydrography and productive systems of the Mediterranean Sea, Mar. Micropaleontol., 25, 187–217, doi:10.1016/0377-8398(95)00002-I, 1995.

Rebotim, A., Voelker, A. H. L., Jonkers, L., Waniek, J. J., Meggers, H., Schiebel, R., Fraile, I., Schulz, M., and Kučera, M.: Factors controlling the depth habitat of planktonic Foraminifera in the subtropical eastern North Atlantic, Biogeosciences, doi:10.5194/bg-2016-348, in press.

Tabachnik, B. G. and Fidell, L. S.: Using Multivariate Statistics, Harper Collins, New York, 3rd edn., 1996.

Vangel, M. G.: Confidence intervals for a normal coefficient of variation, Am. Stat., 50, 21–26, URL `http://www.jstor.org/stable/2685039`, 1996.

---

## Author Response (AR2)

**REFEREE #1**

REF.1 COMMENT: **Statistical analyses:** While reading the ms I get repeatedly the impression that the authors relate foraminifera abundance or area density to a single environmental parameter (e.g. L276-283, L286-287, L613, these are just some examples). How did they distinguish between the influence of covarying environmental parameters? (and how is that possible by doing a PCA on the environmental parameters?). They have used the PCA to describe the variability in the environmental parameters and plotted the foraminifera data in the PCA space. These plots go some way towards a meaningful statistical analysis, but it is still hard to see (and impossible to quantify) the correlations to which the authors repeatedly refer. It would make more sense to then use the PCA scores and see how these correlate (in a scatter plot) with the abundance or area density. Better still would be to use CCA to take both the environmental and the foraminiferal data into account.

ANSWER: Plankton tow results are usually not related to a *unique* environmental parameter but to co-varying parameters; as mentioned in the above comment there are likely multiple factors affecting their physiology and distribution. We decided to apply the PCA for the statistical analysis of the environmental parameters as we consider it as appropriate and sufficient for the purpose of this study (e.g., Schiebel et al., 2001; Horigome et al., 2014). In order to discuss the foraminiferal results we characterize and distinguish different sea surface water masses and the PCA is a tool for achieving this. The PCA doesn't strictly allow the distinction between the influence of covarying environmental parameters. However, it can produce valuable results to better understand in which water masses/environment the foraminifera were retrieved. We will put emphasis in the revised manuscript on the outcomes of the PCA and as suggested by the referee we provide the scatter plots combining both the PCA scores (for each factor) and the abundances or the density area (see fig R1), although this new figure presents the exact same characteristics presented in figure 3 of the revised manuscript. The two first factors produced by the PCA performed on the environmental parameters account for more than 77% of the total variance of all the parameters taken together. As such, we attributed the 1st factor to the temperature and the food availability (inferred here from the nutrients concentrations and the fluorescence (Fig. 3 of the revised manuscript)). This 1st factor explaining more than 55% of the variance depicts well the general trend observed in the Mediterranean Sea with in general colder and more productive waters in the western basin and warmer and less productive waters in the eastern one (see Fig. 1c of the revised manuscript for the fluorescence). The 2nd factor accounts for about 22 % of the total variance and is attributed to the carbonate system. Once again this reflects the general trend observed within the Mediterranean Sea with in general lower $[CO_3^{2-}]$ waters in the western basin compared to the eastern basin (see Fig. 1d-e of the revised manuscript for the distribution of pH and $[CO_3^{2-}]$). We added a new table in the revised manuscript presenting the loadings of the environmental parameters in the PCA and additional Pearson correlation coefficients (r) for relationships between the environmental parameters, the PCA factors, the abundances of the selected species and the density area of the selected species (Table 2).

From Fig R1 and Table 2, we can see that in general the total abundances are higher when the factor 1 is >0 and factor 2 <0 (Fig R1 a and g) in other words when the temperature is lower, the food availability id higher and the pH lower such as in the western basin, with the exception of st. 15 and 16-18 (factor 1, fig R1a) and st. 10 and 12 (factor 2, fig R1 g). The same pattern is observed for *T. sacculifer* (without sac) (fig R1 c and i), *G. bulloides* (fig R1 e and k) and to a lesser degree for *G. inflata* (fig R1 b and h) as no significant correlations are found between the abundances of this species and the 2 factors (fig R1 b and h). The opposite trend is observed for *G. ruber* (white), with in general higher abundances observed when the temperature is higher, the food availability is lower and the pH higher such as in the eastern basin (fig R1 d and g) although no significant correlations are observed (Table 2). Finally no significant correlations were found between the abundances of *O. universa* and the 2 factors (fig R1 f and l; Table 2).

When we compare the density area to factor 1 and factor 2 (Fig R2), it shows for *G. ruber* (white) that $\rho_A$ is higher when the temperature are lower, the food availability higher and the pH lower (Fig R2 a and d). The opposite occurs for *G. bulloides* and *O. universa,* for which the $\rho_A$ is higher when the temperature is higher, the food availability lower and the pH higher (fig R2, b and e).

For both *G. bulloides* and *G. ruber* (white) these observations taken together show that the two species have a higher $\rho_A$ when they are less abundant.

[Figure]

Fig R1: scatter plots between the PCA scores (factor 1 on the left and factor 2 on the right) and the total abundances (a and g), *G. inflata* (b and h), *T sacculifer* (without sac) (c and i), *G. ruber* (white) (d and j), *G. bulloides* (e and k) and *O. universa* (f and l). The red dotted lines show the zero of each factor.

[Figure]

Fig R2: scatter plots between the PCA scores (factor 1 on the left and factor 2 on the right) and the density area of *G. ruber* (white) (a and d), *G. bulloides* (b and e) and *O. universa* (c and f). The red dotted lines show the zero of each factor.

REF.1 COMMENT: **Lunar cycles:**
In response to my previous comment, the authors write:
We are aware that lunar cycle can influence the distribution of foraminifera. However, in our study the lunar day influence on the total absolute abundances (REV Fig. 2) was negligible.
First of all, it is unclear to me how that is evident from Fig. 2 (map of sample locations with chlorophyll-a concentration as background), so please explain. Also, I think this should be part of the manuscript, even a negative result is important and show that lunar paced abundance variability was at least considered as a possible mechanism. Moreover, was the influence of a lunar cycle on area density checked? It seems to me that this parameter would be extremely sensitive to the ontogenetic stage of the organism.

ANSWER: We clarify this point by providing the following plots. We cannot exclude that there is a possible influence of the lunar cycle on the foraminiferal distribution, however our results showed that other factors are probably more important and drive the observed changes. The figure related to this in the previous revised manuscript is REP Fig. 2 and not Figure 2 of the manuscript. We show below the figure, adding the density area plots in it:

[Figure]

REF.1 COMMENT: **Area to long axis relationship:** In response to my previous comment, the authors write: "Size and mass of foraminifers relationship does not start at the origin. The proloculus of planktic foraminifera measures between 15-30 m in average, and has a certain calcite mass, which has so far not been determined (see Hemleben et al., 1989). We will use the power fit in the three species treated in Fig. S4 of the original manuscript for consistency reasons."

I'm happy to see that the species are treated consistently, but disagree with assertion that the curves – which are apparently interpreted as growth curves - should not go through origin. That is physically impossible. As to the interpretation that these relationships reflect growth patterns, it seems that the relationships reflect regional differences in size, rather than ontogeny (Fig. S2). Do the authors think that they sampled shells at different ontogenetic stages at different locations (synchronized reproduction?)? It would be good if they commented on this. Also, were juvenile O. universa (without the spherical chamber) recognized/found?

ANSWER: We agree about the fact that a growth curves should not go through origin since it is physically impossible. This indeed is not the case in the figure presented (Fig. S2). Above and below the regression line, the relation between the area and long axis would be certainly different, with a different slope of the line. In any case, the data points for Fig. S2 (also Fig. S4) are the result of image analysis by incident light microscope of foraminifera, and the graph just reflects the results obtained and the pattern that follow the data (a power regression in this case).

The data points together with the regression line show the general increasing size of the studied foraminifera. This suggests different foraminiferal ontogenetic stages such that smaller/younger ones and older/bigger ones are the end members. As we probably picked individuals in a wide range of ontogenetic stages, we can see a curve of growth. This is what we mean by "growth pattern" in the manuscript.

We did not found any juvenile *O. universa* by recognizing by incident light microscope. We were aware of the shape that juvenile *O. universa* has, previous to its terminal chamber formation, when we identified the different species distribution (i.e. Vilks and Walker, 1974).

**Minor comments:**

REF.1 COMMENT: L16: it would be good to mention here the reason why are density was investigated.
ANSWER: Changed in the revised manuscript as follows: "The main aim was to characterize the species distribution and test the hypothesis of covariance between foraminiferal area density ($\rho_A$) and seawater carbonate chemistry in a biogeochemical gradient including ultraoligotrophic conditions."

REF. 1 COMMENT: L49: perhaps delete 'source' and mention the influence of temperature on seasonal abundance variability (Jonkers and Kučera, 2015; Zaric et al., 2005).
ANSWER: Changed in the revised manuscript as follows: "The absolute abundance of foraminifera is also affected by a predictable and distinct seasonal cycle for each species driven by the food content and temperature of the water mass (Hemleben, 1989; Bé and Tolderlund, 1971; Jonkers and Kučera, 2015; Žarić et al., 2005; for Mediterranean examples see: Pujol…".

REF. 1 COMMENT: L67-69: I don't quite understand this sentence about the different temperature and salinity ranges for water masses (and doubt that Be and Tolderlund is the right citation for that); please clarify.
ANSWER: To clarify better we substitute the sentence: "There are specific temperature and salinity ranges for each water mass," by "Each water mass has a characteristic range of temperature and salinity (Brown et al., 2001)".
We deleted the sentence: "as Bé and Tolderlund (1971) stated for the Atlantic,"

Biogeochemical parameters are used for differentiate Mediterranean water such as Mediterranean Intermediate Water (MIW), Modified Atlantic Water (MAW),…; (see Rohling et al., 2015).

REF. 1 COMMENT: L81: is it Gulf of Lion or of Lions?
ANSWER: Both terminologies are accepted, but the original term is "Lion". Note that in French, Spanish and Italian is written in singular (Lion, León, Leone). The name comes originally from the animal name. This part of the Mediterranean is historically known by the sailors to be a dangerous area; in consequence they named it as the mammal.

REF. 1 COMMENT: L83: the terms distribution and daily fluxes are confusing to me, sediment traps do not provide daily fluxes, but integrate the flux over a certain time interval. Perhaps leaving distribution and daily out is better.
ANSWER: We change 'fluxes' with 'export production'. Authors of the cited papers express daily flux as organisms·m$^{-2}$·day$^{-1}$. The daily export production was estimated considering that the sediment trap sampling period per cup was of 10-11 days (See their methodology sections plus Fig. 5 of Bárcena et al., 2004, and Fig. 4 of Hernández-Almeida et al., 2011). "Distribution" refers to the species relative abundance, which varies considerably between seasons.

REF. 1 COMMENT: L85-87: this sentence is also unclear to me. Sediment traps provide information about the export flux of shells, not an abundance that is directly comparable to measurements from plankton nets. And if there is a correlation (or relationship?) between flux and nutrient supply (why nutrients? Forams don't rely on nutrients) and water column conditions, then how does this work (which direction)?
ANSWER: Rigual-Hernández et al. (2012) analyzed the foraminiferal assemblage in sediment trap samples with sampling period of 14 days to one month (see their methodology section for further details). They calculated the relative abundance of foraminifera species in the total foraminiferal assemblages (see Table 1 in Rigual-Hernández et al. (2012)).
We changed that sentence in the revised manuscript to avoid any confusion as follows: "The 12-year sediment trap foraminiferal export production record in the Gulf of Lion (October 1993 – January 2006) shows a strong seasonal pattern, with more than 80% of the annual export production recorded from winter to spring related to higher food supply and mixing state of the upper water column (Rigual-Hernández et al., 2012)."

REF. 1 COMMENT: L106-108: it would be good if the authors explicitly stated what SNW or area density can be used for, i.e. say that it may tell something about calcification intensity.
ANSWER: Changed in the revised manuscript as follows: "In addition, very few size-normalized weight (SNW) and area density ($\rho_A$) studies to infer the calcification intensity of water column foraminifera are available in the literature…"

REF. 1 COMMENT: L112: add a space between spring and 2013. There are more cases where the space is missing between words (I guess because of the many changes in the document), please double check.
ANSWER: Changed in the revised manuscript. Thanks, true, sometimes the text justification hide this problem to our eyes too.

REF. 1 COMMENT: L138: is there really a longitudinal increase in insolation?
ANSWER: We don't understand this comment on the longitudinal increase in insolation. We did not mention any longitudinal increase in insolation; the main text is: "Natural connection with the ocean is through the narrow Strait of Gibraltar, where nutrient-rich Atlantic surface waters enter the Mediterranean and experience an eastward increase of temperature and salinity (Fig. 1) driven by insolation and evaporation, having a negative hydrological balance (evaporation exceeding precipitation)."

REF. 1 COMMENT: L146: add 's' after become.
ANSWER: Changed in the revised manuscript.

REF. 1 COMMENT: L165-167: please reword this sentence.
ANSWER: Changed in the revised manuscript as follows: "Twenty samples were collected with BONGO nets (mesh size 150 μm and 40 cm diameter, for further details see Posgay, 1980). Those nets sampled primarily 200 m depth, but also caught foraminifera during the net descent and ascent to the surface, which both involve negligible towing and capturing time compared to the sampling at 200 m depth (Table 1)."

REF. 1 COMMENT: L175: replace PO4 and NO3 with PO43- and NO3- throughout the manuscript.
ANSWER: We consider that is not needed to change as it is just an acronym to name it.

REF. 1 COMMENT: L186-187: Can you please reword this sentence, it is unclear to me what has been done.
ANSWER: We appreciate the referee comment here, and changed the sentence in the revised manuscript as follows: "When necessary, samples were split into aliquots of 1/4 and 1/6."

REF. 1 COMMENT: L190: I guess the last statement about the exclusion of tests with attached organic matter only applies to the size/weight analyses, or not? If not, then please explain the reason why.
ANSWER: Yes. Changed in the revised manuscript as follows:

L189-190: "Foraminifera smaller than 150 μm and/or with tests partially broken, making them unrecognizable or unmeasurable, were discarded."

L 202-203: "For the area density ($\rho_A$) study, we selected three main species: *G. ruber*, *G. bulloides* and *O. universa*. All specimens without partially broken tests and/or with organic matter attached of these three species were photographed…"

REF. 1 COMMENT: L191: consider changing 'under optical microscopy.' to 'using optical microscopy.'
ANSWER: Changed in the revised manuscript by: "using incident light microscopy".

REF. 1 COMMENT: L218: I think the figure order needs to be updated, Fig. 7 should not appear before Fig. 3.
ANSWER: Figure order changed by order of appearance, in the text and the figure section of the manuscript: Fig. 7 turned to Fig. 3, Fig. 3 to Fig. 4, Fig. 4 to Fig. 5, Fig. 5 to Fig. 6, and Fig. 6 to Fig. 7.

REF. 1 COMMENT: L248: space after 'Overall,'.
ANSWER: Changed in the revised manuscript.

REF. 1 COMMENT: L277: is food availability estimated/inferred from the fluorescence, make this explicit because food availability is not directly in the PCA? Perhaps this is unneeded, but it is important to realise that foraminifera rely on food, not on nutrients. But again, why do a PCA and then discuss individual environmental parameters and not use the scores?
ANSWER: Yes, food availability is inferred from the CTD fluorescence values (Fig. 1d; Fig. 3) and the nutrient concentration (nitrate and phosphate; Fig. 3). For the PCA scores please to the first comment of this review.

REF. 1 COMMENT: L286-287: it would be helpful if the authors indicate the station number(s) in the text. Also, please explain how this correlation is evident from this plot.
ANSWER: Sentence modified in the revised manuscript as follows: "With the exception of the Tyrrhenian Sea (St. 19), *G. ruber* (white) abundance is related with warmer and saltier waters, and lower pH (St. 9, 10, 11, 12, 13, 14, 16; Fig. 3d)."

REF. 1 COMMENT: L334: to what degree is the large IQR due to the low n?
ANSWER: We plotted the Coefficient of Variation (CV) in Fig. 7 of the revised manuscript. We found no relation between the CV and higher or lower sample size, or with IQR length. We can say that the dispersion of the data is not an artefact of the sample size.

[Figure]

REF. 1 COMMENT: L374-375: this statement should be part of the results.
ANSWER: We agree with the referee. The sentence now is relocated in L250, between the sentence: "The highest percentages are found at the Strait of Sicily and the Northern Ionian Sea (St. 7a, 16-18; Fig. 5; Fig. S1; Appendix A)." and the sentence: "However, due to the extremely low standing stocks the above observations are mere snapshots, and may not be generalized."

REF. 1 COMMENT: L398: reword. Sediment traps provide shell flux.

ANSWER: The data of Rigual-Hernández et al. (2012) also provide relative abundances. See the answer to referee's REF. 1 COMMENT about L85-87 for further details.

REF. 1 COMMENT: L401: not Mediterranean studies (even though all the authors were from countries bordering the Med). Rigual-Hernandez analysed the >150 micron fraction, not the 63-150.
ANSWER: True. Changed in the revised manuscript as follows: "…a possibility potentially supported by Pujol and Vergraud-Grazzini (1995): 120 μm mesh size." Deleted the Rigual-Hernández et al. (2012) reference, also deleted in L415: "and Rigual-Hernández et al. (2012)".

REF. 1 COMMENT: L408: the comparison with R-H is appropriate. This study is not from the Alboran Sea as is suggested here. Better compare to the other sediment trap studies from the Alboran Sea.
ANSWER: Sentence in L408-409 deleted in the revised manuscript.

REF. 1 COMMENT: L410-411: I still think one cannot and should not extrapolate a few observations spread over several decades to suggest a trend. Moreover, 'in the Mediterranean' (L410) should be replaced with 'this study' and the 'trend of changing conditions' is a very vague description of what is happening to temperature and salinity and water mass mixing. If the authors insist on leaving this speculative statement in the ms they should at least analyse this properly (i.e. take the counted number of shell into account, calculate a rarefaction curve etc) and explain what and how the Med is changing and suggest a mechanism how this can affect species diversity.
ANSWER: We replace it "in the Mediterranean" replaced by "in this study".
We do not extrapolate a few observations spread over several decades to suggest a trend. However, it is well known that the Mediterranean is one of the most impacted seas in the world and climate change interacts synergistically with many other disturbances. We consider that our results highlight the need of further work addressing the impact of climate change on plankton diversity in areas particularly vulnerable to rapid environmental change. The Mediterranean Sea is changing rapidly under anthropogenic climate change forcing (e.g. see Giorgi 2006; IPCC 2007, 2013); Yáñez et al., 2010; Hassoun et al., 2015a; Hassoun et al., 2015b, Cossarini, 2015) being among the ocean regions warming fastest. Warming, increased stratification and acidification footprints on the biota can be detected (e.g. Marbá et al., 2015, Meier et al., 2014). Mediterranean biodiversity is undergoing rapid alteration under the combined pressure of climate change and human impact, but detailed studies and biodiversity monitoring are still scarce (Bianchi and Morri, 2000).

We would like to highlight that although our absolute abundance results are obtained by sampling in a relatively high productivity annual period (i.e. see Rigual-Hernández et al., 2012; Barcena et al., 2014) they are the lowest ever recorded in the literature, even lower than recent studies in other oligotrophic areas (i.e.: Auras-Schudnagies et al., 1989; Schmuker and Schiebel, 2002). This surely deserve attention and future studies to clarify the impacts of climate change and human activities on the Mediterranean marine plankton biodiversity.

REF. 1 COMMENT: L413: I somehow missed where it is discussed how 'the ecology and distribution of planktic foraminifera' could change due to these changing conditions. Please explain and provide evidence that the ecology is really changing (and what is meant with that).
ANSWER: In our study, we just provide a suggestion (note the word "could" in that sentence) since our data are based on a single oceanographic expedition. We added a new reference to provide evidence of recent measurable changes in planktonic foraminiferal distributions in another oceanographic region (off southern California: see Field et al. (2006)).

REF. 1 COMMENT: L430: add a reference.
ANSWER: The second and third sentences of that paragraph share reference with the fourth sentence: Bé and Tolderlund, 1971. We considered it enough to provide it once and not repeat it twice in such a short text space.

REF. 1 COMMENT: L433-434: reword 'characterized by its wide temperature range'.
ANSWER: Modified in the revised manuscript as follows: "The spinose and symbiont-barren species *G. bulloides* tolerates a wide temperature range and is typical of subpolar and transitional regions as well as upwelling areas, it is also found in subtropical and tropical waters at lower abundances (Thunell, 1978; Bé and Tolderlund, 1971)."

REF. 1 COMMENT: L440-442: the comparison with Cifelli doesn't make sense here: L437-440 compare the Atlantic with the western Med, Cifelli compares western and eastern Med. The sentence is also very long and complicated, please reword.
ANSWER: We appreciate the referee comment here, and we are clarifying this point and change the sentence: "In our study and the one by Cifelli (1974), *G. ruber* (white) occurs with higher abundances in the eastern compared to the western Mediterranean Basin, being the most abundant species in the Levantine Basin and the South Ionian Sea. Also like Cifelli (1974), in our study, *G. ruber* (white) from the Atlantic station is found with slightly higher relative abundances than in the western Mediterranean Basin. Temperature-related factors may be the main cause, i.e.: warmer Atlantic waters (16.1 ºC) compared to the western Mediterranean (14.3 ºC in the SW, 14.0 ºC in the NW; Fig. 1a)."

REF. 1 COMMENT: L449: please add station numbers.
ANSWER: Changed in the revised manuscript as follows: "*G. ruber* (white) remains scarce (St. 9, 14, 15) or absent (St. 16-18) in the Ionian Sea stations (Fig. 4), increasing its abundance towards the Tyrrhenian Sea. On the other hand, in the Ionian Sea it exhibits relative abundance around 40 to more than 60% in the surface sediments (Thunell, 1978), and decreases towards the Tyrrhenian Sea."

REF. 1 COMMENT: L454: reword 'as due to....'
ANSWER: That sentence is now deleted in the revised manuscript.

REF. 1 COMMENT: L455-456: May is not the month when G. ruber fluxes are highest in the Gulf of Lions or the Alboran Sea (it's perhaps the month when total shell fluxes are highest).

ANSWER: Sentence deleted in the revised manuscript: "This may not be the typical spring situation, as due to surface sediment evidence, the Ionian Sea sediments are enriched in G. ruber tests (Thunell, 1978) and May is the most productive season in terms of foraminiferal tests (Rigual-Hernández, 2012; Bárcena et al., 2004; Hernández-Almeida et al., 2011)."

REF. 1 COMMENT: L483-485: add reference and what is the rational behind the link between seasonal and spatial distribution?

ANSWER: Last sentence of that paragraph is a summary/conclusion of the whole paragraph of *G. inflata*. We consider it unnecessary for a reference here as the reader would have the references after each sentence of that paragraph and our study results.
"Seasonal distribution" is the change of foraminiferal assemblage or a concrete species during periods of time (seasons). "Spatial distribution" is the change in geographical locations of foraminiferal assemblages or a concrete species. The seasonal distribution is linked with the spatial distribution when every season or concrete period of time the foraminiferal assemblage move from one place to another, or it expands or diminishes its presence from determinate locations.

REF. 1 COMMENT: L495-496: I don't understand this sentence. How does it match and why is bulloides mentioned?
ANSWER: In spring, our study, together with Pujol and Vergraud-Grazzini (1995) and van Raden et al., 2011) identified G. inflata as the main species of the assemblage (the highest relative abundance, %) and G. bulloides the second main species (the second highest relative abundance), with a clear difference (in %) from the third main species and the others. The mention of *G. bulloides* here is useful to understand the behavior of these two species regarding temperature in a food-abundant scenario. This is clarified later on L511-L516: "In April (Pujol and Vergnaud-Grazzini, 1995; van Raden et al., 2011) and May (this study), *G. bulloides* is found to be the second most abundant species, surpassed by *G. inflata*, in the westernmost Alboran Sea. High temperature anomalies could provoke an inverse situation, thanks to more suitable environmental conditions for *G. bulloides*, which profits from successful reproduction than *G. inflata*, which instead stays further from its optimum temperature (Bárcena et al., 2004). One month later it is found to be the dominant species replacing *G. inflata*, which is still dominant in the eastern Alboran Sea (Cifelli, 1974)."

REF. 1 COMMENT: L501-502: reword, it cannot be 'In accordance with Cifelli' and 'whereas in our study'.
ANSWER: Changed in the revised manuscript as follows: "Following Cifelli (1974), *G. bulloides …*"

REF. 1 COMMENT: L506: replace 'all the transect' with 'the whole transect'.
ANSWER: Changed in the revised manuscript.

REF. 1 COMMENT: L509: what characteristic of bulloides correlates with fluorescence peaks (its abundance, relative abundance) and if it really correlates, then show the scatter plot.

ANSWER: Its absolute abundance (see Fig. 3e). To avoid any confusion we rephrased the sentence as follows: "Consequently, higher standing stocks of *G. bulloides* are related with higher fluorescence values (i.e., Morthyn and Charles, 2003; Fig. 1; Fig. 3e)." See also the answer to comment 1 about statistical analysis.

REF. 1 COMMENT: L513: faster reproduction, how would that work? I know that Barcena et al say this, but struggle to grasp it. Where is the evidence? Reproductive success may be higher, population growth too, but faster reproduction?

ANSWER: We appreciate the point of view raised by the referee here. According to Schiebel and Hemleben (2005) both species (*G. bulloides* and *G. inflata*) rely on a synodic lunar cycle, but *G. bulloides* could have a better fertilization success and higher growth and survival of the offspring compared with *G. inflata*. Despite no study proving this concretely, we infer this by looking at the behavior (in abundance terms) of the adult specimens: concretely, *G. inflata* would have a better fertilization and offspring legacy in the westernmost Alboran Sea with colder temperatures compared with *G. bulloides*, as we found *G. inflata* less abundant in the warmer June than *G. bulloides*, when before (colder April and May) it was the opposite. We retain that only as a possibility as we do not have enough evidence from our study, or those of Pujol and Vergraud-Grazzini (1995) and van Raden et al. (2011) which consist of "snapshots in time" samples.

We modify the sentence as follows: "High temperature anomalies could provoke an inverse situation, thanks to more suitable environmental conditions for *G. bulloides*, which profits from successful reproduction than *G. inflata*, which instead stays further from its optimum temperature (Bárcena et al., 2004)."

REF. 1 COMMENT: L517: replace 'higher' with 'larger'.
ANSWER: Changed in the revised manuscript.

REF. 1 COMMENT: L524: water 'column' stratification.
ANSWER: Added in the revised manuscript.

REF. 1 COMMENT: L525: 'being more present': there seems to be a word missing.
ANSWER: Changed in the revised manuscript by "is more abundant".

REF. 1 COMMENT: L537: foraminiferal prey? Or fluorescence/chl a?
ANSWER: Fluorescence acts as a proxy for Chlorophyll-*a* concentration. Fluorescence is related with phytoplankton presence, which is a food source, and Chlorophyll-*a* high values very often relate with high presence of foraminifera, and viceversa (i.e. Fairbanks et al., 1982; Mortyn and Charles, 2003). Other zooplankton and other foraminiferal prey concentrations are generally linked with the phytoplankton concentration. Because of that, we consider an area of low Chlorophyll-*a* and fluorescence values, an area with less foraminiferal prey.

REF. 1 COMMENT: L541: planktonic foraminifera rely on food, not on nutrient, availability.
ANSWER: Changed in the revised manuscript as follows: "To conclude, the distribution of *G. bulloides* seems to be limited by food availability, caused by stratification and consequent nutrient depletion of the surface water column, and increased sea surface temperatures."

REF. 1 COMMENT: L544: reword '… was found ubiquitous…'
ANSWER: We consider not rewording that sentence. See the details of that decision in the referee REF. 1 COMMENT about L547 below.

REF. 1 COMMENT: L547: there is an abundance peak of O. universa at station 16, so I don't understand this sentence.
ANSWER: *O. universa* is present at 19 of the 22 stations sampled, where it is present, the highest absolute abundance value is 0.468 ind·10 m$^{-3}$ (St. 16) and the lowest is 0.014 ind·10 m$^{-3}$ (St. 3), making a difference of 0.454 ind·10 m$^{-3}$; also the SD of all the absolute abundance values where *O. universa* is present makes: 0.123.
If you compare the highest and lowest values and the SD of the other main species you find higher values:
*G. ruber* (white) [*highest - lowest; SD*]: 1.681; 0.567
*T. sacculifer* (without sac): 1.3; 0.35
*G. bulloides*: 2.288; 0.552
*G. inflata*: 3.491; 1.053
For that reason we consider *O. universa* ubiquitous and its small difference in abundance detected in St. 16 is not considered a peak, taking into account the low number of species per towing (57 individuals in station 16, but others like station 16-18 consist of only 11 individuals) and the aliquots treatment that represent 52.33% of the sample, making more variable the results in stations with low numbers of individuals.

REF. 1 COMMENT: L552: 'quite ubiquitous' is very vague.
ANSWER: Regarding Cifelli (1974), *G. trilobus* (*T. sacculifer* (without sac)) is only absent at one station sampled and its relative abundance SD inside the Mediterranean (starting at Cifelli's station 49) is 5.52 %. "quite ubiquitous" is changed in the revised manuscript by the term "wide distribution".

REF. 1 COMMENT: L639: misnumbered? Fig. S4a.
ANSWER: No, Fig. S3a indicates the weight-long axis relation of *G. ruber* (white), which is the subject discussed in that paragraph. Fig. S4a compares the weight-area relation of the same species.

REF. 1 COMMENT: L640: suitable conditions?
ANSWER: Changed in the revised manuscript.

REF. 1 COMMENT: L652: 'effect of limited alimentation on calcification'. What is meant exactly? And I assume this only holds for bulloides?
ANSWER: We appreciate referee's comment here, as it was a confusing sentence. Now it is deleted in the revised manuscript.

REF. 1 COMMENT: L654-655: reword.
ANSWER: Added "is" between "carbonate chemistry" and "only partially affecting" in the revised manuscript.

REF. 1 COMMENT: L660: the samples were collected in May 2013
ANSWER: To avoid any confusion we add the word "collected" in the sentence as it follows: "…across the Mediterranean, collected in May 2013."

REF. 1 COMMENT: L663: that looks more like the average and standard deviation than like a range.
ANSWER: Changed in the revised manuscript as follows: "Average standing stocks in the upper 200 m of the water column are $1.42\pm1.43$ ind.$\cdot$10 m$^{-3}$"

REF. 1 COMMENT: L674: 'rather balanced' – what does that mean?
ANSWER: "Rather balanced" inside this manuscript means with similar absolute abundance values everywhere, without considerable peaks. See the answer of referee's REF. 1 COMMENT about L547 for further details. We delete "rather" in the revised manuscript to avoid confusion.

REF. 1 COMMENT: L679: 'trophic conditions and food availability' – what is the difference? And how is that clear from the analyses?
ANSWER: "tropic conditions" is deleted in the revised manuscript. The answer to that question can be found in the first referee comment of that report (Statistical analysis), specifically in Fig. R2.

In the section below, I give detailed comments (including line numbers) about very specific issues. However, in this section I already want to summarise some major points that are more relevant for the entire manuscript than at any specific place.

REF. 2 COMMENT: **1.** The work does still not normalize its data for the consistent differences in sampling employed by the other studies, with which comparisons of assemblages are anticipated. Cifelli (1974) sampled the upper 250m water depth, while Pujol and Vergnaud-Grazzini sampled the upper 350 m. This study uses mainly the association at 200m and partly an integrated column of the upper 200 m. Furthermore, mesh sizes have been different between most studies. In addition, the authors now state that their net had a diameter of only 40 cm (0.12m2 opening), in contrast to the 0.5m2 common with most plankton nets (e.g. Pujol and Vergnaud Grazzini 1995). While absolute abundances are certainly normalized for filtered water volume, this much smaller net opening means that the authors have much larger errors in their assemblage data than the compared studies, because of the much lower volume of filtered sea water. All this has already been criticised in my first review, but the authors did not change anything, although I for instance suggested already there to use equations provided by Berger (1969) to normalize all studies concerning mesh sizes. The authors try to argue that Cifelli (1974), who actually used a comparable mesh size, argue in favour of their interpretation of changing abundances due to changing environments. However, they totally ignore that Cifelli (1974) used another depth range in their studies, so certainly they found other abundances. In my opinion, the authors cannot succesfully show, that the assemblage differences they observe between studies with employing such different sampling techniques are not an artefact of the data, but a real trend.

ANSWER: The referee is clearly correct on the differences among the study approaches over the years. For this reason, we do our best to incorporate any attempts at normalization to plankton tow approaches, including that of Berger (1969) on the issue of mesh sizes. We have already incorporated new text on this in the previous revision, and we admittedly do not add new information on this issue here in this 2$^{nd}$ round. We do our best to compare results across studies as well as possible, always keeping in mind and stating very clearly the caveats to doing this more directly and satisfyingly. That said, given the admitted issues in direct comparison, instead of drawing firm and quantitative conclusions we make observations and derive sensible suggestions that result from them. The Editor and other reviewers did seem to appreciate more our efforts this way, including clear acknowledgment of limitation.

The Berger (1969) approach to normalizing for mesh sizes does so for major ocean basins, as opposed to smaller seas like the Mediterranean. Whether or not this technique would be suitable therefore is somewhat questionable for this Mediterranean plankton tow study.

REF. 2 COMMENT: **2.** The systematics are still not consistent. Why is quadrilobatus designated as belonging to the genus Globigerinoides? From André et al. (2013), which the authors cite themselves, it is very clear that the species genetically belongs to the trilobus–sacculifer plexus (at least as long as recent specimens are concerned). It makes absolutely no sense to not only treat it as a separate species from Trilobatus sacculifer, but even put it into another genus. It should instead be correctly categorized as another morphotype of T. sacculifer.

ANSWER: We agree with the referee's major comment and in accordance with André et al. (2013) plus Spezzaferri et al. (2015) we changed *Globigerinoides quadrilobatus* to *Trilobatus quadrilobatus* in the entire manuscript. New text is added on the methodology section inside the paragraph of L191-200 as follows: "*Trilobatus sacculifer* morphotype *quadrilobatus* was inferred from Spezzaferri et al. (2015) after André et al. (2013); this morphotype is referred as *T. quadrilobatus* in this study and is treated separately from *T. sacculifer* (without sac)."

REF. 2 COMMENT: **3.** The statistical analyses is still a huge problem. The authors state they applied a principal components analysis (PCA), which by the way is data visualization and no proper statistics (because it lacks any possibility to infer significance), and thus a step back from the faulty approach the authors applied in the first iteration of this paper. However, PCA does not include explanatory variables such as environmental parameters. So it is first not clear to me what have been done, i.e. what are Factors 1 and
in Fig. 7? Have samples (as it seems) been ordinated by environment, and then somehow overlain by assemblages? Or is it indeed a redundancy analysis that have been applied, and if so, constrained for which environmental parameters? Furthermore, since PCA is using euclidean distances for ordination, it is very unsuitable for abundance data, and other methods like principal coordinates analysis are much more suitable for comparing assemblages (Hammer and Harper, 2006; Legendre and Legendre, 2012). The authors also still do not use proper techniques to interpret their findings in relation to the hefty multicollinearity in their data. I suggested some techniques in my first review (e.g. GLM, GAM). The authors may also use any of the techniques applied by the Thunell-work group, who also do an excellent job in that (e.g. Marshall et al., 2013; Osborne et al., 2016). As it is now, however, the authors only visually interpret trends in the PCA by eye, which is no proper and robust method when reliable interpretations should be reached.

ANSWER: In the previous version of the manuscript as in this revised version, the PCA was conducted on the 9 environmental parameters considered (i.e., temperature, salinity, fluorescence, [PO4], [NO3], [O2], pH, pCO2 and [$CO_3^{2-}$]; figure 3 of the revised manuscript). *De facto*, the explanatory variables were included in this analysis. To avoid any confusion we made it clearer in the revised manuscript. The matrix used to perform the PCA was organized as follows: the lines correspond to the stations and the columns to the environmental parameters. As a result the two first factors of the PCA account for more than 77% of the total variance of all the environmental parameters taken together and we attributed the 1st factor to the temperature and the food availability (inferred here from nutrients concentrations and fluorescence), while we attributed the 2nd factor to the carbonate system. The 1st factor explains more than 55% of the variance and depicts well the general trend observed in the Mediterranean Sea with in general colder and more productive waters in the western basin and warmer and less productive waters in the eastern one (see Fig. 1c of the revised manuscript for the fluorescence). The 2nd factor accounts for about 22 % of the total variance. Once again this reflects the general trend observed within the Mediterranean Sea with in general lower pH/[$CO_3^{2-}$] in the western basin compared to the eastern basin (see Fig. 1d-e of the revised manuscript for the distribution of pH and [$CO_3^{2-}$]). We added a new table in the revised manuscript presenting the loadings of the environmental parameters in the PCA and additional Pearson correlation coefficients (r) for relationships between the environmental parameters, the PCA factors, the abundances of the selected species and the density area of the selected species (Table 2).

In figure 3 of the revised manuscript, as in the previous version, the scores of the environmental parameters are plotted according to the red axis and are depicted by the red vectors. On the same figure we plotted as well the stations scores (black axis) and for each station the total abundances (Figure 3a of the revised manuscript), the abundances of 5 selected species (Figure 3 b-f of the revised manuscript) and the density area of 3 selected species (Figure 3g-i of the revised manuscript) were overlaid using coloured circles (red for the eastern basin and blue for the western basin, the diameter of the circles being proportional to the abundances and the density area). Although the PCA approach does not strictly allow the distinction between the influence of covarying environmental parameters, the results presented in the figure 3 of the revised manuscript allows to visualize and understand in which conditions the lower/higher abundances and density area of the selected species were observed.

Below we provide the scatter plots combining both the PCA scores (for each factor) and the abundances or the density area (respectively fig R1 and R2).

From fig R1 and table 2, we can see that in general the total abundances are higher when the factor 1 is >0 and factor 2 <0 (fig R1 a and g) in other words when the temperature is lower, the food availability id higher and the pH lower such as in the western basin, with the exception of st. 15 and 16-18 (factor 1, fig R1a) and st. 10 and 12 (factor 2, fig R1 g). The same pattern is observed for *T. sacculifer* (without sac) (fig R1 c and i), *G. bulloides* (fig R1 e and k) and to a lesser degree for *G. inflata* (fig R1 b and h) as no significant correlations are found between the abundances of this species and the 2 factors (fig R1 b and h). The opposite trend is observed for *G. ruber* (white), with in general higher abundances observed when the temperature is higher, the food availability is lower and the pH higher such as in the eastern basin (fig R1 d and g) although no significant correlations are observed (Table 2). Finally no significant correlations were found between the abundances of *O. universa* and the 2 factors (fig R1 f and l; Table 2).

When we compare the density area to factor 1 and factor 2 (fig R2), it shows for *G. ruber* (white) that $\rho_A$ is higher when the temperatures are lower, the food availability higher and the pH lower (fig R2 a and d). The opposite occurs for *G. bulloides* and *O. universa* where the $\rho_A$ is higher when the temperature is higher, the food availability lower and the pH higher (fig R2, b and e).

For both *G. bulloides* and *G. ruber* (white) these observations taken together show that the two species have a higher $\rho_A$ when they are less abundant.

[Figure]

Fig R1: scatter plots between the PCA scores (factor 1 on the left and factor 2 on the right) and the total abundances (a and g), *G. inflata* (b and h), *T sacculifer* (without sac) (c and i), *G. ruber* (white) (d and j), *G. bulloides* (e and k) and *O. universa* (f and l). The red dotted lines show the zero line of each factor.

[Figure]

Fig R2: scatter plots between the PCA scores (factor 1 on the left and factor 2 on the right) and the density area of *G. ruber* (white) (a and d), *G. bulloides* (b and e) and *O. universa* (c and f). The red dotted lines show the zero line of each factor.

**2 Detailed comments**

REF. 2 COMMENT: Line 50, 'Pujol and Vergraud-Grazzini, 1995': This work is consistently misspelled. It should be Pujol and Vergnaud Grazzini, 1995!
ANSWER: Corrected in the revised manuscript by: "Pujol and Vergnaud-Grazzini, 1995". We changed the wrong "r" by the "n", but we maintain the hyphen, as it is cited with it also in Bárcena et al. (2004) or in Hernández-Almeida et al. (2011), for example.

REF. 2 COMMENT: Line 52, 'bottom sediments': Should be 'surface sediments'.
ANSWER: We disagree, as here we mean studies that cover longer time spans or study the more distant past than the ones working with surface sediments.

REF. 2 COMMENT: Line 63, 'prominent differenced': Should be 'prominently different'.
ANSWER: Changed in the revised manuscript deleting the adjective and leaving it as "different".

REF. 2 COMMENT: Line 65, 'retrieved in different sites': Should be 'retrieved from different sites'.
ANSWER: Changed in the revised manuscript.

REF. 2 COMMENT: Line 69, 'hydrographis': Should be 'hydrographic'.
ANSWER: Changed in the revised manuscript.

REF. 2 COMMENT: Line 79, 'study being carried out': Should be 'study have been carried out'.
ANSWER: Changed in the revised manuscript.

REF. 2 COMMENT: Lines 97f, 'For further studies that relate foraminiferal calcification with environmental parameters see Weinkauf et al. (2016); Table 7.': You should also cite Marshall et al. (2013) in this regard.
ANSWER: Marshall et al. (2013) is referred to within Table 7 of Weinkauf et al. (2016) together with other living plankton studies. For that reason, we do not consider to cite it here.

REF. 2 COMMENT: Lines 106f, 'In addition, few size-normalized weight (SNW) and area density (_A) studies from water column foraminifera are available in the literature': Area density is a form of size-normalized weight.
ANSWER: We appreciate the comment of the referee. We consider that nothing has to be changed regarding those lines in the revised manuscript.

REF. 2 COMMENT: Line 112, 'spring2013': Should be 'spring 2013'.
ANSWER: Changed in the revised manuscript.

REF. 2 COMMENT: Lines 120–122, 'In addition, empty tests are passive particles that ocean currents may displace horizontally, but that displacement is negligible due to their quick settling velocities (Caromel et al., 2014).': This is not always correct, and it might be good to show that drift distances in the Mediterranean are actually very low (van Sebille et al., 2015).
ANSWER: van Sebille et al. (2015) was considered in the first version of the manuscript with that sentence: "In addition, empty tests are passive particles that ocean currents may displace. On the other hand, average drift distances of foraminiferal tests are estimated to be less than 10 km in the Mediterranean (van Sebille et al., 2015)…". We reconsidered that reference after Referee #1 REF. 2 COMMENT about the topic: van Sebille et al. (2015) world scale results only represent six grid cells in the Mediterranean area on its Figure 5, making the results less reliable on that area. Despite Caromel et al. (2014) statement, we do not have a reliable proof that displacement is negligible in our study site, we delete also that sentence in L120-122. We consider Vergnaud-Grazzini et al. (1986) as proof enough for the location reliability of Thunell (1978) results.

REF. 2 COMMENT: Line 146, 'become': Should be 'becomes'.
ANSWER: Changed in the revised manuscript.

REF. 2 COMMENT: Line 166. 'primarily 200m depth': Should be 'primarily from 200m                                                                      depth'.
ANSWER: The whole sentence is modified on the revised manuscript to clarify better the sampling procedure as follows: "Twenty samples were collected with BONGO nets (mesh size 150 μm and 40 cm diameter, for further details see Posgay, 1980). Those nets sampled primarily 200 m depth, but also caught foraminifera during the net descent and ascent to the surface, which both involve negligible towing and capturing time compared to the sampling at 200 m depth (Table 1)."

REF. 2 COMMENT: Line 179, 'MODIS Aqua L2 satellite': Should be 'MODIS Aqua L2 satellite data'.
ANSWER: Changed in the revised manuscript.

REF. 2 COMMENT: Lines 186f, 'Samples were studied from the collecting bottles and the bottom collector, the latter representing 52.33% of the total sample were treated in aliquots of 1/2, 1/4, 1/6, until 1/8.': I do not understand this sentence.
ANSWER: We appreciate the referee comment here, we changed the sentence in the revised manuscript as follows: "When necessary, samples were split into aliquots of 1/4 and 1/6."

REF. 2 COMMENT: Line 188, '≥350–500 μm': Should be '350–500 μm'.
ANSWER: We agree. Changed in the revised manuscript.

REF. 2 COMMENT: Line 199, 'Globigerinella siphonifera/G. calida/G. radians plexus': Should be 'The Globigerinella siphonifera/G. calida/G. radians plexus'.
ANSWER: Changed in the revised manuscript.

REF. 2 COMMENT: Line 204f, 'the individuals were weighed together by triplicate with a Mettler Toledo XS3DU microbalance': Which means the authors were actually applying the mean area density approach as described in Weinkauf et al. (2013) instead of the more advanced area density approach as described by Marshall et al. (2013).
ANSWER: As we applied the mean weight only when more than one individual was available to weight we consider not to change the name of the approach (area density: $\rho_A$) to mean area density (MAD). But we include Weinkauf et al. (2013) as one of the references for our approach in the revised manuscript.

REF. 2 COMMENT: Lines 216, 'The PCA was performed on the environmental parameters:': So how to understand this? The samples were ordinated by environmental parameters? What then are the scores of the black axes, passively projected assemblage scores? Or is this indeed a redundancy analysis instead of PCA? Compare also general comments why PCA is unsuitable anyways.
ANSWER: See our reply of comment 3.

REF. 2 COMMENT: Line 218, '(Fig. 7)': What happened to Figs 3–6, which should be cited in the text before Fig. 7?
ANSWER: Figure order changed by order of appearance, in the text and the figure section of the manuscript: Fig. 7 turned to Fig. 3, Fig. 3 to Fig. 4, Fig. 4 to Fig. 5, Fig. 5 to Fig. 6, and Fig. 6 to Fig. 7.

REF. 2 COMMENT: Lines 218–228, 'The first factor exhibited positive loadings. . . are shown in Figure 7).': This entire passage belongs into the Results section.
ANSWER: Despite that paragraph comparing the results of the PCA, we consider that it should stay in the methodology section, as the results of our study are partly based on these results. We consider it necessary to clearly separate abundance and area density results and findings that come partly from the PCA, from the PCA results themselves.

REF. 2 COMMENT: Line 244, 'The exceptions are at Station 3. . . ': And what about stations 1 and 6?
ANSWER: We decided to delete that sentence in the revised manuscript, as we considered only extra information that can be checked by the interested readers in Appendix A.

REF. 2 COMMENT: Lines 246f, 'The 350–500-_m size fraction dominates in the western Mediterranean and is progressively reduced eastwards (Fig. 4)': I do not see this trend. This could be due to the bad layout of figs 3 and 4 (see below).
ANSWER: Changed in the revised manuscript as follows: "The 350-500-μm size fraction in the first leg dominates in the western Mediterranean and is progressively reduced eastwards (Fig. 5)". The sentence "mainly due to the contribution of small *G. inflata* from the 150-350 μm size fraction" is now deleted. Also is modified the following sentence of L248-249 in the revised manuscript as follows: "Higher percentages of individuals >500 μm in the first leg are found in the western part of the Mediterranean compared to the eastern part (Fig. 5)."

REF. 2 COMMENT: Line 272, 'G. quadrilobatus': Incorrect genus (see General Comments).
ANSWER: See the answer to the general comment 2.

REF. 2 COMMENT: Lines 274–276, 'The PCA performed on the environmental parameters and the sample scores on the two first components clearly shows a separation, regarding Factor 1, between the western and eastern Mediterranean stations (Fig. 7).': I do not understand how this 'PCA' was performed. Did it ordinate the samples on environmental data (as seems the case), then what are the black factors in fig. 7? Or is it indeed an RDA, then constrained for which environmental factors?
ANSWER: See our reply to comment 3.

REF. 2 COMMENT: Line 278, 'station 10 is an exception': But stations 1, 6, 20, 21, and 22 (all Western Mediterranean) all have low a abundances as well.
ANSWER: Notice that before "station 10 is an exception", goes the phrase "In the eastern basin", meaning that is an exception inside the eastern basin. The stations named by the referee are for the western basin and the Atlantic Ocean.

REF. 2 COMMENT: Line 279, 'Factor 2': Should be 'principal component 2' or 'PC 2'.
ANSWER: We consider it appropriate and understandable to name it Factor 2. In the revised manuscript we name it "PCA Factor 2".

REF. 2 COMMENT: Lines 283–285, 'Overall, the highest absolute abundance of all foraminifera seems related to food availability and only secondarily to the carbonate system (Fig. 7a).': While it makes the impression to be true, as it is this is eyeballing, because PCA cannot yield any significance but is only ordinating datapoints. Since many of your environmental factors show multicollinearity (as I already pointed out in my first revision) you need much more advanced, real statistical methods to say exactly whith which factors correlation is greatest. At the very least, you should use a more appropriate ordination method for abundances (probably constrained ordination, which at least delivers a significance for the overall correlation of data with environmental factors) than PCA, which uses euclidean distances.
ANSWER: See our reply to comment 3.

REF. 2 COMMENT: Lines 286–292, 'With the exception . . . path of Atlantic waters (Fig. 7b)': Where do you see this? Globigerinoides ruber (white) shows a peak (the richest sample) on the cold side of the ordination space, and G bulloides seems to be more correlated with pH. To convince me that those trends are true, you would have to show me something more robust than just a PCA impression (i.e. a compositional multiple regression as described by van den Boogart and Tolosana-Delgado (2013), as I also already suggested last time).
ANSWER: See our reply to comment 3.

REF. 2 COMMENT: Lines 298f, 'The Atlantic and the Ionian–Adriatic–Aegean grouping have similar proportions of species.': Except that from Atlantic to Ionian–Adriatic–Aegean grouping dominances are completely shifted: G. ruber becomes much more dominant, G. bulloides and T. sacculifer are strongly reduced in abundance, O. universa is much more prevalent, and G. inflata is hardly there anymore.
ANSWER: Sentence deleted in the revised manuscript.

REF. 2 COMMENT: Lines 313f, 'The high two-dimensional (silhouette) area-to-long axis correlation is best fitted by a power regression (Fig. S2).': Which, as I already argued in the first review, should be forced to have zero offset. The authors argued concerning this 'Size and mass of foraminifers relationship does not start at the origin (zero). The proloculus of planktic foraminifera measures between 15–30 _m in average, and has a certain calcite mass, which has so far not been determined (see Hemleben et al., 1989).'. This, however, only means that the model should stop short of zero. Especially when the authors argue that a zero-intercept model would not make sense because it would imply the existence of individuals with zero mass and size, is it not logical to them that non-zero-intercept model which allows a foraminifer to have mass at size zero or have a certain size without mass is even more problematic!
ANSWER: We agree about the fact that a growth curves should not go through origin since it is physically impossible. This indeed is not the case in the figure presented (Fig. S2). Above and below the regression line, the relation between the area and long axis would be certainly different, with a different slope of the line. In any case, the data points for Fig. S2 (also Fig. S4) are the result of image analysis by incident light microscope of foraminifera, and the graph just reflects the results obtained and the pattern that follow the data (a power regression in this case).

REF. 2 COMMENT: Lines 314f, 'The same growth pattern can be seen in G. ruber (white), G. bulloides, and O. universa': But this assumption is wrong at least in O. universa. There, size increase cannot be growth, because the spherical form is the terminal form and cannot grow considerably anymore.
ANSWER: We are aware that once the terminal chamber is reached, the individuals increase in size to a very small degree due to the incorporation of additional calcite layers (i.e. Spero et al., 2015). The data points together with the regression line show the general increasing size of the studied foraminifera. This suggests different foraminiferal ontogenetic stages such that smaller/younger ones and older/bigger ones are the end members. As we probably picked individuals in a wide range of ontogenetic stages, we can see a curve of growth. This is what we mean by "growth pattern" in the manuscript. We change "the same growth pattern" by "similar growth patterns" to avoid any confusion.

REF. 2 COMMENT: Lines 318f, 'The specimens of G. ruber (white) from the Atlantic have the largest size followed by individuals from the Tyrrhenian Sea, and those from the eastern Ionian Sea.': If this statement is made, I already requested a statistical proof in the last review, to which the authors responded 'We do not need a statistical test to know which is the smallest value'. Since this shows a complete lack of understanding for the nature of any quantitative analysis, here is a short Statistics 101 (I again refer the authors to basic introductory literature such as Hammer and Harper (2006) or Dytham (2011): When dealing with natural values, one value will always be larger than the other when measured accurately enough. The REF. 2 COMMENT you want to answer is not, is one value larger, to which you know the answer beforehand, but is one value significantly larger. This means, is the difference you observe between the values in two random samples large enough that, taking into account uncertainty from the fact that you only sampled a couple of randomly selected specimens from the population, you can be reasonably sure that the populations the samples were drawn from differ in this value. An easy example: I measure a difference of 0.3 cm between two samples. Do the populations from which those samples have been drawn differ in size? Well, when I use the variation in the samples to estimate the uncertainty in the estimate of the mean, I can tell with a certain probability. When the standard deviation in both samples (of, say, 100 specimens each) is 0.2 cm, then the 95% confidence interval is _0.02 cm, so the two populations do differ in size with a probability of more than 95 %. If the standard deviation is 5 cm, in contrast, the 95% confidence interval is _0.5 cm, so the two populations do not show a significant difference in size. This is, what statistics is for, and in this sense, yes, you do need statistics to know which value is smaller!
ANSWER: We appreciate the referee's comment. After applying a Student's t-test, we modified the sentence as follows in the revised manuscript: "The specimens of *G. ruber* (white) from the Atlantic have a significantly larger area than those from the Tyrrhenian Sea ($p \leq 0.003$), which in turn have significantly larger area than those from the East Ionian Sea grouping ($p \leq 0.001$)."

REF. 2 COMMENT: Line 337f, 'higher density area are related to slightly lower pH and higher food availability in the western Mediterranean and Atlantic stations': This must be proven, and from the PCA I doubt the pH relationship.
ANSWER: See our reply to comment 3.

REF. 2 COMMENT: Line 340, 'opposite trend as in G. ruber (white)': Should be 'opposite trend than G. ruber (white)'.
ANSWER: Changed in the revised manuscript.

REF. 2 COMMENT: Line 367f, 'Within the Mediterranean, a previous study with results comparable to ours, sampled the upper 350m (Pujol and Vergraud-Grazzini, 1995).': They also sampled with another mesh size, for which still no corrections have been applied.
ANSWER: See our reply to comment 1.

REF. 2 COMMENT: Line 401, 'smallesr': Should be 'smaller'.
ANSWER: Changed in the revised manuscript.

REF. 2 COMMENT: Lines 409–411, 'The lower absolute abundance of individuals in our study compared to Pujol and Vergraud-Grazzini (1995), together with low species diversity in the Mediterranean, may indicate a trend of changing conditions over the last decades, . . . ': I still believe that this has to do more with the different mesh-sizes. The size fraction between 120 _m and 150 _m in my experience contains a lot of the standing stock of foraminifers.
ANSWER: See our reply to comment 1.

REF. 2 COMMENT: Section Factors controlling the abundance of the main species: All trends described here are purely derived from the PCA by eye, without any appropriate test. While their explanation can be valuable, their interpretation should be toned down considerably.
ANSWER: See our reply to comment 3.

REF. 2 COMMENT: Lines 445f, 'The increasing dominance of G. ruber (white) from the western to the eastern Mediterranean Basin coincides with the eastward increasing salinity (Fig. 7d).': Or Temperature, or CO2. It is hard to say without proper analytical techniques under this degree of multicollinearity.
ANSWER: See our reply to comment 3.

REF. 2 COMMENT: Line 537: Remove second 'its'.
ANSWER: Changed in the revised manuscript.

REF. 2 COMMENT: Line 548, 'but abundances are slightly higher in the western basin to than the east.': I highly doubt that from the PCA alone. You could prove it though.
ANSWER: See our reply to comment 3.

REF. 2 COMMENT: Line 569f, 'In contrast, the density area of O. universa does not show any change between the western and eastern basins (Fig. 7i), and cannot be used to identify and quantify particular environmental effects.:' I also doubt that there is a difference between basins in G. bulloides, and since the authors still refuse to use proper quantitative techniques to prove it . . .
ANSWER: See our reply to comment 3.

REF. 2 COMMENT: Line 615, 'larger IQR indicates . . . ': This is only true, when the variation in the sample is normalized for expected value (i.e. mean). This means, calculating the coefficient of variation, which I already requested in the first review. The authors replied 'As also described above, in our comment to the reviewer comment about lines 480–482, we are unsure about what statistical method and/or calculation the reviewer is referring to here. Is there a distinct suggestion of some kind, with a reference? We are not sure how to calculate a "coefficient of variation" with regard to box plots and their statistics.'. No, I do not have a reference for it, because the coefficient of variation is such a basic and old method that its origins are lost in the mist of time, and you would not cite a reference as you would not cite a reference when calculating a mean value. Rather, the coefficient of variation is explained (and listed in the index) in every basic statistics book I suggested the authors to consult in my first review. It is also very easily found using Google and the search term 'coefficient of variation'. Again, in short, variation is always correlated to mean value, so variations of samples which mean value differs must be corrected for this stochastic effect. An example: Let's say you measured the length of twenty mice and found it to be 3_0.5 cm. You also measured the length of 20 elephants and found it to be 4_0.5 m. Which species has the higher variation? The absolute value is much larger for elephants (0.5 m) than for mice (0.5 cm), but when calculating the coefficient of variation you actually find mice to be more variable in size (0.166) than elephants (0.125). Since none of the IQRs in the manuscript are corrected (and I would recommend to use the standard deviation instead of the IQR anyways) all conclusions drawn by the authors concerning variation in their samples are invalid.

ANSWER: We added the Coefficient of Variation (CV) to Fig. 7. The CV is not influenced by the sample size (n) or the IQR. The CV of *G. ruber* between the Atlantic and the Tyrrhenian Sea is quite similar (0.04 of difference), showing little dispersion of our data between those locations. We decide to delete that sentence in the revised manuscript.

[Figure]

REF. 2 COMMENT: Line 624, 'variability in density area data increases with increasing absolute density area': Exactly as stochastically predicted. Calculate the coefficient of variation and compare again.

ANSWER: We observe that *G. ruber* CV ranges from 0.15 to 0.24, and for *G. bulloides* it goes from 0.04 to 0.2, showing little dispersion in our data between locations. We delete the following sentence in the revised manuscript: "At the same time, variability in $\rho_A$ data increases with increasing absolute $\rho_A$, which resembles the distribution of data in *G. ruber* (white) (Fig. 7)"

REF. 2 COMMENT: Line 633, 'retarded': Should be 'hampered'.
ANSWER: Changed in the revised manuscript.

REF. 2 COMMENT: Line 636, 'seems': Should be 'seem'.
ANSWER: Changed in the revised manuscript.

REF. 2 COMMENT: Line 640, 'suited conditions': Should be 'suitable conditions'.
ANSWER: Changed in the revised manuscript.

REF. 2 COMMENT: Line 648, ''heavier average': Should be 'steeper average', maybe.
ANSWER: We decided not to change "heavier" in the manuscript, as we considered the correct word for referring to more mass.

REF. 2 COMMENT: Line 651f, 'All of these findings support our idea of an effect of limited alimentation on calcification.': I do not understand this sentence.
ANSWER: That sentence is deleted in the revised manuscript.

REF. 2 COMMENT: Caption Fig 4 'Sample size is indicated by n below each station code.': This information is not present in the figure.
ANSWER: We modified the legend and the figure for the revised manuscript as follows: "**Fig. 5.** Percentage of each planktic foraminifera size fraction in each station from leg 1 (stations 1 to 13) and leg 2 (stations 22 to 14). Sample size is indicated in italics at the top of each station bar."

[Figure]

REF. 2 COMMENT: Figs 3 and 4: A lot of the interpretation by the authors in concerned with east-west trends. Then why are the graphs not ordered west–east, instead of by station number?

ANSWER: In the first manuscript version Figs. 3 and 4 (now re-ordered as 4 and 5) were ordered by station number. But in the second manuscript versions figures were already ordered west-east. It is not a strict west-east order, as we divide the two transects in the figures with a blank space (leg 1 and leg 2, see methodology section), but inside each transect they are ordered west-east. We consider them with an appropriate order now.

REF.3 COMMENT: Line 227: delete double parenthesis close to (without sac)). Substitute the parenthesis before G. ruber with bracket. Substitute the parenthesis after O. universa with a bracket. Delete the parenthesis after Figure 7.
ANSWER: We deleted the parenthesis after Figure 7. We consider a double parenthesis closing grammatically appropriate. With the referee's correction "area density" would be inside the category of "absolute abundances of foraminifera species", and is treated separately.
The sentence is now:
*'The sample scores of the first two factors with an overlay of absolute abundances of foraminifera species (G. ruber (white), G. bulloides, G. inflata, O. universa and T. sacculifer (without sac)) and area density (G. ruber (white), G. bulloides and O. universa) are shown in Figure 3.'*

REF.3 COMMENT: Line 248: please let space between words Overall,higher
ANSWER: Changed in the revised manuscript.

REF.3 COMMENT: Line 383: I would like only to make a small comment concerning the differences for the occurrence of G. elongatus. It is important to take in account that the time interval covered by the 2 cm of sediment analysed by Thunell (1978) is variable between some centuries to one or two millennia. This issue is important to consider because of recently Margaritelli et al (2016), in the central Tyrrhenian Sea fossil record, found G. elongatus over the last 4 millennia and the last specimens of this species are recorded in the last two centuries.
ANSWER: We appreciate the recent reference and the information regarding *G. elongatus* occurrence and the different type of information acquired in sediment versus plankton tows.

REF.3 COMMENT: Line 392: I would like to suggest to add more references concerning sedimentary cores. The manuscript is focused on living forams so that I would like to see references also concerning the last millennia (i.e., Margaritelli et al., or others).

ANSWER: We appreciate the reference provided. We added the following references concerning more recent sedimentary core studies with the presence of *G. quadrilobatus*: Margaritelli et al., 2016; Lirer et al., 2013.

REF.3 COMMENT: Line 401: Pujol & Vergraud-Grazzini 1995 and not 1998
ANSWER: Changed in the revised manuscript.

REF.3 COMMENT: Paragraph 4.2: In this paragraph, the authors discuss the Figure 6, but the previous paragraph is mainly focused on figure 7. Is it possible to find a solution? It is enough anomalous.

ANSWER: We agree with Referee's comment. Figure order changed by order of appearance, in the text and the figure section of the manuscript: Fig. 7 turned to Fig. 3, Fig. 3 to Fig. 4, Fig. 4 to Fig. 5, Fig. 5 to Fig. 6, and Fig. 6 to Fig. 7.

REF.3 COMMENT: Line 480: please let space between words Pujol & Vergraud-Grazzini (1995),In winter
ANSWER: Changed in the revised manuscript.

REF.3 COMMENT: Line 610: please let space between words G. ruber (white)is only
ANSWER: Changed in the revised manuscript.

**References**

[revised manuscript text omitted]

Atlantic (St. 1)

Strait of Gibraltar (St. 2)

Alboran Sea (St. 3)

SW Mediterranean (St. 5, 6)

Eastern Mediterranean (St. 9, 10, 11, 12, 13)

Ionian-Adriatic-Aegean (St. 14, 15, 17, 16, 16-18)

Tyrrhenian Sea (St. 19)

Legend:
- Others (gray)
- *G. elongatus* (pink)
- *T. sacculifer* (without sac) (light green)
- *O. universa* (orange)
- *G. bulloides* (dark green)
- *G. inflata* (yellow)
- *G. ruber* (white) (red)

**Figure 7**

[Figure]

---

## Author Response (AR3)

ICTA-ICP, Edifici Z, Carrer de les columnes s/n
Universitat Autònoma de Barcelona
E- 08193 Bellaterra, Barcelona, Spain
ICTA +34 93 586 8628
Direct +34 93 581 8974
patrizia.ziveri @uab.cat
*Patrizia Ziveri, Prof. Dr.*
*Research Professor*
*ICTA-UAB Scientific Director*

To Prof. Markus Kienast
Biogeosciences Editor

Ref: bg-2016-266

Low planktic foraminiferal diversity and abundance observed in a  spring 2013 West-East Mediterranean Sea plankton tow transect, Miguel Mallo et al.

March 7, 2017

Dear Editor,

In the following, please find our reply to the reviewer's comments. We have addressed all of R1's comments, and below explain why we cannot sufficiently account for reproduction and stratification.

The line numbers refer to the revised manuscript attached.

Referee 1:

We have included all the reviewer 1 comments. However, we cannot do any statistical analyses on the reproduction cycle, as explained below, and we have included a sentence in the Discussion section to clarify this point. The same is true for the analysis of water column stratification.

Reproduction: Concerning the effect of reproduction and ontogeny on test calcite mass, we cannot produce a statistically significant correlation. First, synodic lunar reproduction differs between species, and we do not have a sufficiently large number of individuals per species, which would allow statistically viable numbers. Second, although reproduction occurs at a monthly rate, it can be delayed for a month, or several months. This is particularly true under environmental conditions that do not favour reproduction as, for example, scarcity of the right kind of food. Under such conditions, tests may grow larger and heavier than under optimum conditions (see Mojtahid et al. 2015). This is a problem particularly in the oligotrophic eastern Mediterranean. As much as we would like to offer an explanation that includes reproduction cycles, we cannot possibly do so in a scientifically rigorous manner. Finally, reproduction is just another measure that may or may not affect calcite mass, among other effects we do not discuss because of the inherent scarcity of planktic foraminifers in some regions of the Mediterranean Sea. This should no longer impede the publication of data on planktic foraminifers, and information on modern planktic foraminifers for the Mediterranean is crucially needed.

We made modifications in the "4.2. Area density" 1st paragraph (lines 344-346): "Similar allometric developments can be seen in *G. ruber* (white), *G. bulloides*, and *O. universa* with that correlation, graphically represented by the shape of a power function (Fig. S2). The allometric developments of species result from increasing size of tests when adding chambers during the successive ontogenetic stages from juvenile to adult: planktic foraminifera grow "faster" when they are younger and smaller (steepest in the lower left part of the regression line) and "slower" when they are older and bigger (less steep in the upper right part of the regression line; Fig. S2)."

We have added a sentence on reproduction and ontogenetic development (growth) in lines 627-630: "Unfortunately, we cannot address the effects of reproduction (e.g. Bijma et al., 1994), and ontogenetic development on the distribution patterns and test calcite mass of species, because a lack data at the species level do not allow any such statistics."

Stratification: Temperature, which is accounted for in the PCA, is an expression of thermal stratification, admittedly not to 100 %, but good enough to produce interpretable results in our data set.

The new scatter plots are added in the supplementary material. PCA scores vs. abundance (Fig. S5), PCA scores vs. area density (Fig. S6).

We added the following sentence on the PCA results at the beginning of the discussion chapter 5.2.: "Abundance patterns of the five most frequent species in our samples possibly result from a combination of environmental conditions as, for example, food and temperature (Fig. 3)."

We agree with referee's comment about "Changing ecology (lines 446-447)" and we modified the sentence in the revised manuscript as follows: "These changing conditions could also imply changes in environmental conditions and distribution of planktic foraminifera,…"

Referee 2:

We appreciate the effort that reviewer no. 2, Manuel Weinkauf, has put into reviewing our manuscript. We appreciate some of his technical comments (typos etc.), which we have fixed in the new version of the manuscript.

However, we are disappointed that Weinkauf tries to impose his ideas upon our work. We can of course not give a full description of the planktic foraminifers of the Mediterranean, and have just sampled populations in a sound scientific manner. We have then interpreted our data in relation to the data earlier presented in other papers, despite the differences in data type, collection procedure, sample processing, etc. (Weinkauf's 4th paragraph).

We do not want to apply Weinkauf's 5th paragraph. In this manuscript, we apply standard methods, including standard mathematics. The first author, Miguel Mallo, is an MSc student, who is perhaps not expected to develop new methods. Consequently, our manuscript is an honest description and basic interpretation of some new and really important data (there are not many) of upper water column planktic foraminifers from the Mediterranean. We perceived some misconceptions from Weinkauf, for example, foraminifers grow faster as juveniles than adults (e.g., Brummer 1988; please see for reviews Hemleben et al. 1989, and Schiebel and Hemleben 2017); in his comment on the 'Lines 313-315', Weinkauf sadly disqualifies ("This is nonsense…"). This is neither scientifically correct nor does it reflect

[Figure]
 Institute of Environmental Science and Technology

[Figure]
 Universitat Autònoma de Barcelona

[Figure]
 EXCELENCIA MARÍA DE MAEZTU

appropriate conduct in the reviewing process. We have nonetheless added a sentence to unequivocally clarify our point, and to avoid misinterpretation. Having said all this, we believe that the reviewer is somewhat overambitious. However, having worked in the same field for more than 30 years, it is not a pleasure to read inappropriate and incorrect statements written in your own field of expertise.

We are well-aware of the work of Andreia Rebotim referred to by Weinkauf's new review (2nd paragraph in Weinkauf's 'General comments'), as one of us (Schiebel) is co-author of the paper, which was published in late February 2017, which is from a different region (N Atlantic), and which has its own problems. In some minor points, Weinkauf might be right; in others, he is not. For example, the depth habitat varies according to hydrography, and *T. sacculifer* and *G. ruber* dwell at different depths depending on the availability of food and light (e.g., Schiebel and Hemleben, 2017, Springer, and references therein).

In the end, the manuscript is our work and our responsibility. We feel that the reviewer is beyond the limits of good scientific practice by imposing his methods to other people's work. We do feel we have addressed all the relevant points in an honest and professional manner. We are looking forward to the editor's decision regarding this manuscript.

*Minor comments:*

Ref.1: The minor corrections have been addressed (lines 487, L496, L506, L511, L541-542, and L571-572 in the previous manuscript; see also new manuscript with track changes).

Ref.2: The minor corrections have been addressed (L40, L41f, L50f, L105f, L202, L434, L449, L453, L487, L506 in the previous manuscript; see also new manuscript with track changes).

To avoid any confusion, we changed the sentence from L166f to "Twenty samples were collected down to 200 m water depth with BONGO nets (Table 1)."

L188: the 1/6 aliquot is possible as we used a wet sample divider.

L288: we thank the referee for noticing this error. We change this to "where *p*H is lower".

Table 2 was corrected with the variables in italics, and the area density consistently written.

Figure 5 is corrected; we meant n=16.

**References:**

Bijma, J., Hemleben, C., and Wellnitz, K.: Lunar-influenced carbonate flux of the planktic foraminifer *Globigerinoides sacculifer* (Brady) from the central Red Sea, Deep-Sea Res. Part I, 41, 511-530, 1994.

Brummer, G. J. A., Hemleben, C., Spindler, M.: Planktonic foraminiferal ontogeny and new perspectives for micropalaeontology, Nature, 319, 50–52, doi:10.1038/319050a0, 1986.

Hemleben, C., Spindler, M., Anderson, O. R.: Modern planktonic Foraminifera, Springer, Berlin, 1989.

Mojtahid, M., Manceau, R., Schiebel, R., Hennekam, R., and de Lange, G.J.: Thirteen thousand years of

[Figure]

[Figure]

[Figure]

Institute of Environmental Science and Technology

southeastern Mediterranean climate variability inferred from an integrative planktic foraminiferal-based approach: Holocene climate in the SE Mediterranean, Paleoceanography, 30 (4), 402–422, doi: 10.1002/2014PA002705, 2015.

Schiebel, R., Hemleben, C.: Planktic Foraminifers in the Modern Ocean. Springer, Berlin, Heidelberg, 2017.

Patrizia Ziveri, Miki Mallo and co-authors

Patrizia Ziveri
ICREA Research Professor